# The Club Sandwich:
# Gapless Phases and Phase Transitions with Non-Invertible Symmetries

Lakshya Bhardwaj, Lea E. Bottini, Daniel Pajer, and Sakura Schäfer-Nameki

*Mathematical Institute, University of Oxford, Woodstock Road, Oxford, OX2 6GG, United Kingdom*

We provide a generalization of the Symmetry Topological Field Theory (SymTFT) framework to characterize phase transitions and gapless phases with categorical symmetries. The central tool is the club sandwich, which extends the SymTFT setup to include an interface between two topological orders: there is a symmetry boundary, which is gapped, and a physical boundary that may be gapless, but in addition, there is also a gapped interface in the middle. The club sandwich generalizes so-called Kennedy-Tasaki (KT) transformations. Building on the results in [1, 2] on gapped phases with categorical symmetries, we construct gapless theories describing phase transitions with non-invertible symmetries by applying suitable KT transformations on known phase transitions provided by the critical Ising model and the 3-state Potts model. We also describe in detail the order parameters in these gapless theories characterizing the phase transitions, which are generally mixtures of conventional and string-type order parameters mixed together by the action of categorical symmetries. Additionally, removing the physical boundary from the club sandwiches results in club quiches, which characterize all possible gapped boundary phases with (possibly non-invertible) symmetries that can arise on the boundary of a bulk gapped phase. We also provide a mathematical characterization of gapped boundary phases with symmetries as pivotal tensor functors whose targets are pivotal multi-fusion categories.

**CONTENTS**

## I. INTRODUCTION AND SUMMARY

One of the important questions in validating the recent flurry of activities on non-invertible or categorical symmetries – for some of the papers both in high energy and condensed matter physics, see [1–95], and reviews [96–101] – is to find concrete physical implications tied to the presence of these symmetries. Such "smoking gun" applications will not only substantiate the importance of these symmetries, but take their relevance from mere curiosities to genuinely fundamental concepts underlying the inner workings of quantum field theories (QFTs). The goal of this paper is to find such applications in the realm of phase transitions in the presence of categorical symmetries, and provide a characterization of the possible gapless phases at these critical points. This has applications in both high-energy physics and condensed matter alike, and should provide a unified framework to characterize phases in the presence of categorical symmetries in any dimension.

Characterizing phase transitions for standard group-like symmetries is of course well-understood and relies on a simple organizing principle: that of spontaneous symmetry breaking, going back to Landau's work. Many short-comings of this principle are known by now, but most pertinently it fails when the symmetry of the system is non-invertible or more generally categorical.

In this paper we build on the earlier works [1, 2], where we proposed a framework to characterize gapped phases with non-invertible symmetries, and study phase transitions between such gapped phases and provide a characterization of the gapless phases at the critical points. The classification of gapped phases is most succinctly formulated using the so-called Symmetry Topological Field Theory (SymTFT) or Symmetry Topological Order [102–105], often referred to as the "sandwich": a $d + 1$ dimensional TQFT sandwiched between two $d$-dimensional boundaries. We will provide a brief summary of the construction shortly. The SymTFT allows essentially the separation of symmetries and QFTs/phases, and most importantly contains the information about the generalized charges under non-invertible symmetries [65]. Gapped phases are characterized by inserting gapped boundary conditions both for the symmetry boundary and the physical boundary – the latter ensures that the resulting $d$-dimensional theory is topological, the former realizes the categorical symmetry. The generalized charges, encoded as topological operators of the SymTFT, become order parameters for the gapped phases [1, 2], see also [3, 4, 6, 9, 42] and in 2d [106–116].

To characterize phase transitions between gapped phases with categorical symmetries, we will need to extend this picture to include another layer to the sandwich, i.e. we will promote it to a **club-sandwich** [117]: two topological orders in $d + 1$ dimensions, separated by a $d$-dimensional interface, and book-ended on either side with $d$-dimensional boundary conditions.

The topological orders can for instance be SymTFTs $\mathfrak{Z}_{d+1}(\mathcal{S})$ and $\mathfrak{Z}_{d+1}(\mathcal{S}')$ for two symmetries $\mathcal{S}$ and $\mathcal{S}'$, with the interface providing a map between them. In this instance, $\mathcal{S}'$ is usually taken to be a reduced topological order:

$$
\begin{array}{ccccc}
\mathfrak{B}^{\text{sym}}_{\mathcal{S}} & \mathcal{I}_d & \mathfrak{B}^{\text{phys}}_d & \mathfrak{T}_d \circlearrowright \mathcal{S} \\
\boxed{\;\mathfrak{Z}_{d+1}(\mathcal{S}) \;\; \mathfrak{Z}_{d+1}(\mathcal{S}')\;} & = & \big| \\
\end{array}
\tag{I.1}
$$

Here $\mathfrak{B}^{\text{sym}}_{\mathcal{S}}$ is the symmetry boundary of the SymTFT for $\mathcal{S}$ and $\mathcal{I}_d$ the interface. The physical boundary $\mathfrak{B}^{\text{phys}}_d$ is a not-necessarily gapped boundary condition of the SymTFT for $\mathcal{S}'$, and thus in the standard sandwich construction would describe an $\mathcal{S}'$-symmetric theory after interval compactification. The key point here is that the collapse of the club sandwich yields a theory that is $\mathcal{S}$-symmetric. This map from $\mathcal{S}'$-symmetric theories to $\mathcal{S}$-symmetric theories may be thought of as non-invertible duality transformations [50, 118] generalizing the so-called Kennedy-Tasaki (KT) transformation [119, 120].

However, the topological orders do not necessarily have to be thought of as SymTFTs and this framework can be viewed more generally as a way to study $\mathcal{S}$-symmetric boundary conditions of a topological order $\mathfrak{Z}'$ (now not viewed as the SymTFT for another symmetry $\mathcal{S}'$).

We will be interested in phase transitions between two $\mathcal{S}$-symmetric gapped phases, which are characterized in terms of two gapped boundary conditions of $\mathfrak{Z}(\mathcal{S})$. This in turn corresponds mathematically to so-called Lagrangian algebras in the center $\mathcal{Z}(\mathcal{S})$. The interfaces $\mathcal{I}_d$ of interest are characterized by non-maximal (and thus non-Lagrangian) condensable algebras in $\mathcal{Z}(\mathcal{S})$. In the process we can also charaterize all the order parameters, which are manifestly encoded in the SymTFT and the club sandwich generalization.

Recent works that use SymTFT techniques to extend the study of gapless phases and phase-transitions – though applied to group-like symmetries – are [89–91, 121]. The setup in [121] is precisely the case of the club-sandwich, when the topological orders are both SymTFTs and the interface is the intersection of two Lagrangian algebras, applied in the context of group symmetries $\mathsf{Vec}_G$ in (1+1)d. More precisely: a phase transition between two gapped phases, which are defined by the Lagrangian algebras $\mathcal{L}_1$ and $\mathcal{L}_2$ respectively, is characterized by an algebra $\mathcal{A}_{1,2} = \mathcal{L}_1 \cap \mathcal{L}_2$, which simply means the set of topological lines that are both in $\mathcal{L}_1$

and $\mathcal{L}_2$. Higher order phase transitions can occur when three or more such gapped phases meet. As $\mathcal{A}_{1,2}$ is not maximal, it will lead to a reduced topological order, as described in the club quiche. As we will see not all con-

densable (non-Lagrangian) algebras are however of this type and the framework presented here should account for the most general, not necessarily group-like setup.

| $\mathcal{S}$ | $\mathcal{S}'$ | Club Quiche | Club Sandwich | Phase Transition | $\mathcal{S}$-symmetric CFT $\mathfrak{T}_C^{\mathcal{S}}$ |
|---|---|---|---|---|---|
| $\mathbb{Z}_4$ | $\mathbb{Z}_2$ | IV B 1 | V B | VI C | $\mathbb{Z}_2 \circlearrowright \text{Ising}_0 \oplus \text{Ising}_1 \circlearrowleft \mathbb{Z}_2$ with $\mathbb{Z}_4$ above and $\mathbb{Z}_4$ below |
| $\mathbb{Z}_4$ | $\mathbb{Z}_2$ | IV B 2 | V C | VI D | $\text{Ising} \circlearrowright \mathbb{Z}_4$ |
| $\mathbb{Z}_4$ | $\mathbb{Z}_2{}^\omega$ | IV B 3 | V D | - | $\text{SU}(2)_1 \circlearrowright \mathbb{Z}_4$ |
| $S_3$ | $\mathbb{Z}_3$ | IV C 1 | V E | VI E | $\mathbb{Z}_3 \circlearrowright \text{3-Potts}_0 \oplus \text{3-Potts}_1 \circlearrowleft \mathbb{Z}_3^{-1}$ with $\mathbb{Z}_2$ |
| $S_3$ | $\mathbb{Z}_2$ | IV C 2 | V F | VI F | $\mathbb{Z}_3$ over $\text{Ising}_0 \oplus \text{Ising}_1 \oplus \text{Ising}_2$ with $\mathbb{Z}_2$ and $\mathbb{Z}_2$ |
| $S_3$ | $\mathbb{Z}_2$ | IV C 3 | V G | VI G | $\text{Ising} \circlearrowright S_3$ |
| $\text{Rep}(S_3)$ | $\mathbb{Z}_2$ | IV D 1 | V H | VI H | $\text{Ising} \circlearrowright \text{Rep}(S_3)$ |
| $\text{Rep}(S_3)$ | $\mathbb{Z}_3$ | IV D 2 | V I | VI I | $\text{3-Potts} \circlearrowright \text{Rep}(S_3)$ |
| $\text{Rep}(S_3)$ | $\mathbb{Z}_2$ | IV D 3 | V J | VI J | $E \circlearrowright \text{Ising}_e \oplus (\text{Ising}_m)_{\sqrt{2}} \circlearrowleft 1_-$ with $E$ |
| $\text{Ising}$ | $\mathbb{Z}_2$ | IV E | V K | VI K | $P \circlearrowright \text{Ising}_e \oplus \text{Ising}_m \circlearrowleft P$ with $S$ |
| $\text{TY}(\mathbb{Z}_4)$ | $\mathbb{Z}_2$ | IV F 1 | V L | VI L | $S$ over $\text{Ising}_0^e \oplus (\text{Ising}_1^m)_{\sqrt{2}} \oplus (\text{Ising}_2^m)_{\sqrt{2}}$ with $A$ and $A$ |
| | $\mathbb{Z}_2$ | IV F 2 | - | - | - |
| | $\mathbb{Z}_2{}^\omega$ | IV F 3 | - | - | - |

TABLE I. List (and links to subsections) of symmetries $\mathcal{S}$ and $\mathcal{S}'$, associated club quiches, club sandwiches (i.e. KT transformations from $\mathcal{S}'$ to $\mathcal{S}$) and phase transitions. The last column indicates the schematic form of the phase transition, and the action of the $\mathcal{S}$ symmetry on it. For $\text{Rep}(S_3)$ the symmetry generators $1_-$ and $E$ are the sign and 2d irreducible representations. For the Ising category $P$ is the $\mathbb{Z}_2$ line and $S$ the Kramers-Wannier duality defect. Finally for $\text{TY}(\mathbb{Z}_4)$, $A$ generates $\mathbb{Z}_4$ and $S$ is the non-invertible duality defect with $S^2 = \oplus A^i$. $\mathbb{Z}_2^\omega$ denotes an anomalous $\mathbb{Z}_2$ symmetry.

### A. Summary of Setup

**Standard SymTFT Construction.** We begin with a theory $\mathfrak{T}$ in $d$ spacetime dimensions, with a fusion $(d-1)$-category symmetry $\mathcal{S}$. The first concept that we can associate to this is the SymTFT $\mathfrak{Z}(\mathcal{S})$, which gauges the symmetry $\mathcal{S}$ in $d+1$ dimensions. The topological defects of this SymTFT form the so-called Drinfeld center $\mathcal{Z}(\mathcal{S})$ of $\mathcal{S}$. The symmetry itself is realized as topological defects localized along a gapped boundary condition $\mathfrak{B}^{\text{sym}}_{\mathcal{S}}$ of the SymTFT. The $\mathcal{S}$ symmetric theory $\mathfrak{T}$ is then recovered as the interval compactification of $\mathfrak{Z}(\mathcal{S})$ upon selecting another boundary condition $\mathfrak{B}^{\text{phys}}_{\mathfrak{T}_d}$ on the right. This is the so-called "sandwich" construction, which we schematically depict as

$$\mathfrak{S}(\mathfrak{B}^{\text{sym}}_{\mathcal{S}}, \mathfrak{Z}(\mathcal{S}), \mathfrak{B}^{\text{phys}}_{\mathfrak{T}_d}) :=$$

$$(\text{I.2})$$

It is useful to also consider the "quiche"[122] associated to the symmetry $\mathcal{S}$, namely the combination of the SymTFT $\mathfrak{Z}(\mathcal{S})$ and the gapped boundary condition $\mathfrak{B}^{\text{sym}}_{\mathcal{S}}$ (but without the physical boundary condition), which we display as

$$\mathfrak{Q}_d = (\mathfrak{B}^{\text{sym}}_{\mathcal{S}}, \mathfrak{Z}(\mathcal{S})) =$$

$$(\text{I.3})$$

If $\mathfrak{B}^{\text{phys}}_{\mathfrak{T}}$ is also a gapped boundary condition of $\mathfrak{Z}(\mathcal{S})$, then the interval compactification leads to a gapped $\mathcal{S}$-symmetric phase [1, 2]. Such $\mathcal{S}$-symmetric gapped phases will be denoted by

$$\mathfrak{T}^{\mathcal{S}}_d = \mathfrak{S}(\mathfrak{B}^{\text{sym}}_{\mathcal{S}}, \mathfrak{Z}(\mathcal{S}), \mathfrak{B}^{\text{top}}), \qquad (\text{I.4})$$

where $\mathfrak{B}^{\text{top}}$ is another topological boundary condition of $\mathfrak{Z}(\mathcal{S})$. Gapped boundary conditions of $\mathfrak{Z}(\mathcal{S})$ are classified by Lagrangian algebras in $\mathcal{Z}(\mathcal{S})$. In $(1+1)$d these can be systematically classified given a SymTFT (see appendix B).

Note also that two symmetries $\mathcal{S}$ and $\tilde{\mathcal{S}}$ related by (possibly generalized) gaugings have the same SymTFT, but correspond to different symmetry boundaries of that SymTFT.

**Club Sandwich and Quiche.** In this paper we extend this set-up to allow for gapless phases with $\mathcal{S}$ symmetry.

The first step is to generalize the construction of $\mathcal{S}$-symmetric gapped phases to one for $\mathcal{S}$-symmetric gapped boundaries. This requires the introduction of what we call a "**club-quiche**" (i.e. an open club sandwich). Concretely, this is realized by coupling a $(d+1)$-dimensional TQFT $\mathfrak{Z}'_{d+1}$ via a topological interface $\mathcal{I}_d$ to the SymTFT $\mathfrak{Z}(\mathcal{S})$ with topological boundary condition $\mathfrak{B}^{\text{sym}}_{\mathcal{S}}$. This club quiche will be depicted by

$$\mathfrak{Q}^C(\mathfrak{B}^{\text{sym}}_{\mathcal{S}}, \mathfrak{Z}(\mathcal{S}), \mathcal{I}, \mathfrak{Z}') :=$$

$$(\text{I.5})$$

and realizes an $\mathcal{S}$-symmetric boundary $\mathfrak{B}'_d$ of $\mathfrak{Z}'_{d+1}$ upon collapsing the interval between $\mathfrak{B}^{\text{sym}}_{\mathcal{S}}$ and $\mathcal{I}_d$.

In practice, in the cases of interest, the interface $\mathcal{I}_d$ is specified by a **condensable** algebra $\mathcal{A}$ (generically non-Lagrangian) in $\mathfrak{Z}_{d+1}(\mathcal{S})$, and therefore we can express $\mathfrak{Z}'_{d+1}$ as a reduced topological order for a symmetry $\mathcal{S}'$

$$\mathfrak{Z}'_{d+1} = \mathfrak{Z}_{d+1}(\mathcal{S})/\mathcal{A} = \mathfrak{Z}_{d+1}(\mathcal{S}'). \qquad (\text{I.6})$$

The symmetry $\mathcal{S}'$ is generically smaller than (or equal to, in the case of trivial $\mathcal{A}$) $\mathcal{S}$. See appendix B for necessary conditions to determine condensable algebras.

We can complete this club quiche to a **club sandwich** by slotting in a physical boundary condition onto the right hand side of $\mathfrak{Z}'$. This results in a theory $\mathfrak{T}_d$ with symmetry $\mathcal{S}$

$$(\text{I.7})$$

One may ask now how this is different from the construction of the standard SymTFT (I.2).

**Phase Transitions.** One central application is that of phase transitions and construction of gapless phases with $\mathcal{S}$ symmetry. Say we start with a phase transition for a symmetry $\mathcal{S}'$, e.g. $\mathbb{Z}_2$ which in 2d is modeled by the Ising CFT. We can then consruct using the KT transformations (realized as club quiches) from $\mathcal{S}'$ to the larger symmetry $\mathcal{S}$ a phase transition that is $\mathcal{S}$-symmetric, i.e. a gapless phase that is $\mathcal{S}$-symmetric.

### B. Overview of Results

In sections II and III we summarize the construction of the SymTFT quiches and sandwiches, respectively.

We then extend this to the club-quiches in section IV, where we provide the general $d$-dimensional construction, and give explicit examples for $d = 2$ (i.e. 3d topological orders separated by the topological interface $\mathcal{I}_2$), for abelian and non-abelian groups, non-invertible symmetries such as $\mathsf{Rep}(S_3)$, $\mathsf{Ising}$ and more general Tambara-Yamagami $\mathsf{TY}(\mathbb{Z}_4)$ fusion categories. The club-quiches are completed to club-sandwiches in section IV – these are labeled by two symmetry categories $\mathcal{S}$ and $\mathcal{S}'$, and the club sandwich provies a KT transformation between these symmetries. Finally in section VI we explain how this can be applied to construct phase transitions with categorical symmetries $\mathcal{S}$. We then provide examples using input phase transitions for $\mathbb{Z}_2$ (Ising) and $\mathbb{Z}_3$ (3-state Potts model) to construct new phase transitions. In appendix A we provide a mathematical perspective on the club sandwich construction.

We summarized all the examples and the relevant sections where they are discussed in table I.

## II. SYMTFT QUICHE

As defined in [105], a quiche $\mathcal{Q}$ is a pair

$$\mathcal{Q}_d = (\mathfrak{B}_d, \mathfrak{Z}_{d+1}) \tag{II.1}$$

of a $(d + 1)$-dimensional [123] (oriented, unitary) TQFT $\mathfrak{Z}_{d+1}$ and a $d$-dimensional topological boundary condition $\mathfrak{B}_d$ of $\mathfrak{Z}_{d+1}$. We will sometimes call such a quiche as a $d$-dimensional quiche for convenience, though it should be kept in mind that a $d$-dimensional quiche has a $d$-dimensional component, and a $(d + 1)$-dimensional component. We display a quiche as

$$\mathfrak{B}_d$$

$$\mathfrak{Z}_{d+1}$$

$$\tag{II.2}$$

Our focus will mainly be on $d = 2$ in this work. An irreducible $d$-dimensional quiche is a quiche that cannot be expressed as a sum of other $d$-dimensional quiches. Practically, this means that the space of topological local operators of $\mathfrak{Z}_{d+1}$ is one-dimensional (comprising of scalar multiples of the identity operator in the bulk), and the space of topological local operators of $\mathfrak{B}_d$ is also one-dimensional (comprising of scalar multiples of the identity operator on the boundary).

### A. Gapped Boundary Phases

Physically, quiches are important objects to study for understanding gapped boundary phases, which we now define.

First we review the definition of gapped (bulk) phases. A $d$-dimensional gapped phase is defined as an equivalence class of $d$-dimensional gapped systems, in which two gapped systems are regarded to be equivalent if one of the systems can be deformed into the other without closing the gap (even at infinite volume). One can identify a gapped $d$-dimensional phase $[\mathfrak{Z}_d]$ as a deformation class of $d$-dimensional TQFTs [124]. An irreducible gapped phase is defined to be a gapped phase that cannot be expressed as a sum of other gapped phases. In other words, an irreducible gapped phase has a single vacuum (at infinite volume) [125]. Practically, the space of topological local operators of a TQFT $\mathfrak{Z}_d$ lying in an irreducible gapped phase $[\mathfrak{Z}_d]$ is one-dimensional (comprising of scalar multiples of the identity operator), or in other words such a TQFT $\mathfrak{Z}_d$ is an irreducible TQFT.

Consider now gapped systems that are comprised of a $(d + 1)$-dimensional bulk and a $d$-dimensional boundary. A priori, a gapped bulk may have a gapless boundary, but we require here the boundary to be gapped as well. We define a $d$-dimensional gapped boundary phase $[\mathcal{Q}_d]$ to be an equivalence class of gapped bulk+boundary systems, in which two systems are regarded to be equivalent if and only if one of them can be deformed into the other without closing the gap in the bulk or on the boundary (even when both bulk and boundary have infinite volume). A couple of comments are in order:

- Note that a $d$-dimensional gapped boundary phase $[\mathcal{Q}_d]$ comes associated to a $(d + 1)$-dimensional gapped bulk phase $[\mathfrak{Z}_{d+1}]$, which is the gapped phase the bulk system is in, away from the boundary. We will denote this by

$$[\mathcal{Q}_d] \mapsto [\mathfrak{Z}_{d+1}]. \tag{II.3}$$

However, there may be multiple different $d$-dimensional gapped boundary phases $[\mathcal{Q}_d]_i$, for which the $(d+1)$-dimensional gapped bulk phase is the same as $[\mathfrak{Z}_{d+1}]$

$$[\mathcal{Q}_d]_i \mapsto [\mathfrak{Z}_{d+1}], \qquad \forall \, i. \tag{II.4}$$

Thus, the presence of a boundary may add additional components to the phase diagram.

- Gapped $d$-dimensional boundary phases $[\mathcal{Q}_d]$ whose associated gapped $(d + 1)$-dimensional bulk phase $[\mathfrak{Z}_{d+1}]$ is trivial are the same as gapped $d$-dimensional phases discussed at the beginning of this subsection.

One can identify a gapped $d$-dimensional boundary phase $[\mathcal{Q}_d]$ as a deformation class of $d$-dimensional quiches

$$[\mathcal{Q}_d] = [(\mathfrak{B}_d, \mathfrak{Z}_{d+1})]. \tag{II.5}$$

An irreducible gapped $d$-dimensional boundary phase is one that cannot expressed as a sum of other gapped

$d$-dimensional boundary phases. Practically, any $d$-dimensional quiche $\mathcal{Q}_d$ lying in an irreducible gapped $d$-dimensional boundary phase $[\mathcal{Q}_d]$ is an irreducible quiche, as defined above.

Let us restrict to $d = 2$ from this point on. In this case, an irreducible gapped boundary phase comprises of a pair

$$[\mathcal{Q}] = ([\mathfrak{B}], \mathfrak{Z}) \tag{II.6}$$

where $\mathfrak{Z}$ is an irreducible 3d TQFT and $[\mathfrak{B}]$ is a deformation class of irreducible topological boundary conditions of $\mathfrak{Z}$. Such a 3d TQFT $\mathfrak{Z}$ is described by the data of its topological line operators (also referred to as anyons), which form a (unitary) modular tensor category (MTC) $\mathcal{Z}$. This cannot be an arbitrary MTC as it must admit a topological boundary condition, which becomes the requirement that it must be possible to express the MTC $\mathcal{Z}$ as the Drinfeld center $\mathcal{Z}(\mathcal{C})$ of a (unitary) fusion category $\mathcal{C}$.

We can characterize a deformation class of irreducible topological boundary conditions of $\mathfrak{Z}$ by a Lagrangian algebra in the MTC $\mathcal{Z}$. We refer to the deformation class corresponding to a Lagrangian algebra $\mathcal{L}$ as $[\mathfrak{B}](\mathcal{L})$. The deformation class comprises of a (real) one-parameter family of irreducible topological boundary conditions

$$\mathfrak{B}_\lambda(\mathcal{L}), \qquad \lambda \in \mathbb{R}. \tag{II.7}$$

These boundaries are related as

$$\mathfrak{B}_\lambda(\mathcal{L}) = \mathfrak{T}_\lambda \boxtimes \mathfrak{B}_0(\mathcal{L}) \tag{II.8}$$

i.e. the boundary $\mathfrak{B}_\lambda(\mathcal{L})$ can be obtained from the boundary $\mathfrak{B}_0(\mathcal{L})$ by stacking it with an invertible 2d TQFT $\mathfrak{T}_\lambda$, known as the Euler term [126]. More generally, we have

$$\mathfrak{B}_{\lambda_2}(\mathcal{L}) = \mathfrak{T}_{\lambda_2 - \lambda_1} \boxtimes \mathfrak{B}_{\lambda_1}(\mathcal{L}). \tag{II.9}$$

The Lagrangian algebra $\mathcal{L}$ can be expressed as

$$\mathcal{L} = \bigoplus_a n_a \mathbf{Q}_a \tag{II.10}$$

where the sum is over the simple bulk anyons $\mathbf{Q}_a$ in $\mathcal{Z}$. Physically, the presence of a term $n_a \mathbf{Q}_a$ in $\mathcal{L}$ means that there is an $n_a$-dimensional vector space of topological local operators along any corresponding topological boundary $\mathfrak{B}_\lambda(\mathcal{L})$ at which the line $\mathbf{Q}_a$ can end

$$\tag{II.11}$$

As an example, let the gapped bulk phase $\mathfrak{Z}$ be the Toric Code, which is described by the well-known modular tensor category

$$\mathcal{Z} = \{1, e, m, f\} \tag{II.12}$$

where $e$ and $m$ are electric and magnetic bosonic lines, and $f$ is a dyonic fermionic line. There are two (1+1)d gapped boundary phases with associated (2+1)d gapped bulk phase being the toric code. One of them corresponds to the Lagrangian algebra

$$\mathcal{L}_e = 1 \oplus e \tag{II.13}$$

in which the electric line (anyon) can end along the boundary, and the other is described by Lagrangian algebra

$$\mathcal{L}_m = 1 \oplus m \tag{II.14}$$

in which the magnetic line (anyon) can end along the boundary.

## B. Symmetry TFT

Let $\mathcal{S}$ be a symmetry that can arise in $d$-dimensional systems. The class of symmetries that we are interested in are described by (spherical) fusion $(d-1)$-categories. This includes as special cases finite symmetry groups with/without 't Hooft anomalies and finite higher-form and higher-group symmetries with/without 't Hooft anomalies. However, there are also more general fusion $(d-1)$-categories that are not of this type. Such categorical symmetries are also referred to as non-invertible symmetries, as they typically include symmetry elements that do not admit an inverse.

Given such a symmetry $\mathcal{S}$, we can associate to it a $(d+1)$-dimensional TQFT $\mathfrak{Z}_{d+1}(\mathcal{S})$, known as the Symmetry TFT (or SymTFT). The symmetry TFT has the property that it admits at least one (but usually multiple) irreducible topological boundary condition $\mathfrak{B}_\mathcal{S}^{\mathrm{sym}}$ such that the topological defects living on $\mathfrak{B}_\mathcal{S}^{\mathrm{sym}}$ form the $(d-1)$-category $\mathcal{S}$. Thus, the setup of SymTFT involves quiches of the form

$$\mathcal{Q}_d = (\mathfrak{B}_\mathcal{S}^{\mathrm{sym}}, \mathfrak{Z}_{d+1}(\mathcal{S})) \tag{II.15}$$

which we display as

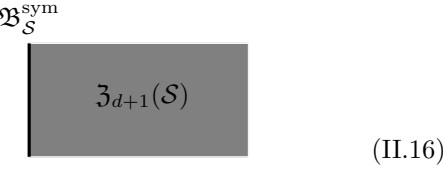

$$\tag{II.16}$$

Such a boundary $\mathfrak{B}_\mathcal{S}^{\mathrm{sym}}$ is known as a symmetry boundary of the SymTFT $\mathfrak{Z}_{d+1}(\mathcal{S})$.

Conversely, any irreducible quiche $\mathcal{Q}_d = (\mathfrak{B}_d, \mathfrak{Z}_{d+1})$ provides an example of a SymTFT setup, with the symmetry being

$$\mathcal{S}(\mathcal{Q}_d) = \mathcal{C}_{\mathfrak{B}_d} \tag{II.17}$$

where $\mathcal{C}_{\mathfrak{B}_d}$ is the fusion $(d-1)$-category formed by topological defects living on the boundary $\mathfrak{B}_d$.

For $d = 2$, the MTC associated to the 3d SymTFT $\mathfrak{Z}(\mathcal{S})$ can be identified with the Drinfeld center $\mathcal{Z}(\mathcal{S})$ associated to the fusion $(d-1)$-category $\mathcal{S}$. There is a canonical Lagrangian algebra $\mathcal{L}_{\mathcal{S}}^{\mathrm{sym}}$ in $\mathcal{Z}(\mathcal{S})$ that describes a symmetry boundary $\mathfrak{B}_{\mathcal{S}}^{\mathrm{sym}}$

$$\mathcal{L}_{\mathcal{S}}^{\mathrm{sym}} = \bigoplus_a n_a^{\mathrm{sym}} \mathbf{Q}_a, \qquad n_a^{\mathrm{sym}} \in \mathbb{Z}_{\geq 0} \qquad \text{(II.18)}$$

where $\mathbf{Q}_a$ are simple bulk anyons and $n_a$ is the dimension of morphism space $\mathrm{Hom}_{\mathcal{S}}(F(\mathbf{Q}_a), 1)$ in $\mathcal{S}$ between the object $F(\mathbf{Q}_a) \in \mathcal{S}$ obtained by applying the forgetful functor $F: \mathcal{Z}(\mathcal{S}) \to \mathcal{S}$ and the identity object $1 \in \mathcal{S}$.

## III. SYMTFT SANDWICH

A sandwich $\mathfrak{S}_d$ is obtained by supplying a quiche $\mathcal{Q}_d = (\mathfrak{B}_d, \mathfrak{Z}_{d+1})$ with a (possibly non-topological) boundary condition $\mathfrak{B}_d^{\mathrm{phys}}$, known as the physical boundary, on the right

$$\mathfrak{S}_d = (\mathfrak{B}_d, \mathfrak{Z}_{d+1}, \mathfrak{B}_d^{\mathrm{phys}}). \qquad \text{(III.1)}$$

In the context of SymTFT, we express the sandwich as

$$\mathfrak{S}_d = \left( \mathfrak{B}_{\mathcal{S}}^{\mathrm{sym}}, \mathfrak{Z}_{d+1}(\mathcal{S}), \mathfrak{B}_{\mathfrak{T}_d}^{\mathrm{phys}} \right) \qquad \text{(III.2)}$$

which can be depicted as

$$
\begin{array}{ccc}
\mathfrak{B}_{\mathcal{S}}^{\mathrm{sym}} & \mathfrak{B}_{\mathfrak{T}_d}^{\mathrm{phys}} & \mathfrak{T}_d \curvearrowright \mathcal{S} \\
\boxed{\mathfrak{Z}_{d+1}(\mathcal{S})} & = & \Big| 
\end{array}
\qquad \text{(III.3)}
$$

whose interval compactification produces a $d$-dimensional QFT $\mathfrak{T}_d$ with symmetry $\mathcal{S}$. Conversely, any $d$-dimensional QFT $\mathfrak{T}_d$ with symmetry $\mathcal{S}$ can be expressed as such a sandwich. We have a one-to-one correspondence

$$\{\mathcal{S}\text{-symmetric QFTs}\}$$
$$\updownarrow$$
$$\{\text{Physical boundaries of the SymTFT } \mathfrak{Z}_{d+1}(\mathcal{S})\}$$
$$\text{(III.4)}$$

### A. Generalized Charges

As discussed in detail in [65], the topological $(q + 1)$-dimensional operators of the SymTFT $\mathfrak{Z}_{d+1}(\mathcal{S})$ capture the charges of $q$-dimensional operators appearing in an $\mathcal{S}$-symmetric $d$-dimensional QFT.

Let us briefly review how this comes about. Any (possibly non-topological) $q$-dimensional operator $\mathcal{O}_q$ in a $d$-dimensional $\mathcal{S}$-symmetric QFT $\mathfrak{T}_d$ is charged under the

symmetry only if it somehow interacts with the symmetry boundary. This is possible only if the sandwich construction of $\mathcal{O}_q$ involves a bulk topological $(q+1)$-dimensional operator $\mathbf{Q}_{q+1}$ ending on the physical boundary $\mathfrak{B}_{\mathfrak{T}_d}^{\mathrm{phys}}$ along a (possibly non-topological) $q$-dimensional operator $\mathcal{M}_q$ as shown below

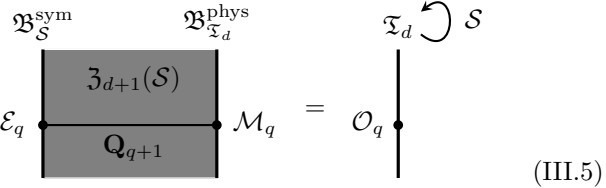

$$\text{(III.5)}$$

The end $\mathcal{E}_q$ of $\mathbf{Q}_{q+1}$ is a topological $q$-dimensional operator along the symmetry boundary $\mathfrak{B}_{\mathcal{S}}^{\mathrm{sym}}$, which may be attached to topological operators living on the boundary $\mathfrak{B}_{\mathcal{S}}^{\mathrm{sym}}$ (in which case $\mathcal{O}_q$ is also attached to topological operators in $\mathfrak{T}_d$ generating the symmetry $\mathcal{S}$ and hence lives in twisted sector for the symmetry). The action of the symmetry $\mathcal{S}$ on $\mathcal{O}_q$ is captured in how the bulk operator $\mathbf{Q}_{q+1}$ interacts with the symmetry boundary $\mathfrak{B}_{\mathcal{S}}^{\mathrm{sym}}$ (via the end $\mathcal{E}_q$), and hence $\mathbf{Q}_{q+1}$ captures the charge of $\mathcal{O}_q$ under $\mathcal{S}$. Note that if there is no end $\mathcal{M}_q$ of a bulk topological operator $\mathbf{Q}_{q+1}$ on a physical boundary $\mathfrak{B}_{\mathfrak{T}_d}^{\mathrm{phys}}$, then there is no $q$-dimensional operator in the theory $\mathfrak{T}_d$ carrying the generalized charge $\mathbf{Q}_{q+1}$.

### B. Gapped Phases with Non-Invertible Symmetries

Consider a categorical symmetry $\mathcal{S}$. One can define an $\mathcal{S}$-symmetric $d$-dimensional gapped phase as an equivalence class of $\mathcal{S}$-symmetric $d$-dimensional gapped systems, in which two such systems are regarded to be equivalent if one of the systems can be deformed into the other without closing the gap (even at infinite volume) and without losing $\mathcal{S}$-symmetry at any point along the deformation path. Note that an $\mathcal{S}$-symmetric gapped phase $\left[ \mathfrak{Z}_d^{\mathcal{S}} \right]$ has an underlying (non-symmetric) gapped phase $\left[ \mathfrak{Z}_d \right]$

$$\left[ \mathfrak{Z}_d^{\mathcal{S}} \right] \mapsto \left[ \mathfrak{Z}_d \right], \qquad \text{(III.6)}$$

where $[\mathfrak{Z}_d]$ is the gapped phase occupied by any $\mathcal{S}$-symmetric system $\mathfrak{Z}_d$ lying in the $\mathcal{S}$-symmetric gapped phase $\left[ \mathfrak{Z}_d^{\mathcal{S}} \right]$. Multiple $\mathcal{S}$-symmetric gapped phases $\left[ \mathfrak{Z}_d^{\mathcal{S}} \right]_i$ may have the same underlying gapped phase $[\mathfrak{Z}_d]$

$$\left[ \mathfrak{Z}_d^{\mathcal{S}} \right]_i \mapsto [\mathfrak{Z}_d], \qquad \forall\, i. \qquad \text{(III.7)}$$

One can identify an $\mathcal{S}$-symmetric gapped $d$-dimensional phase as a deformation class of $\mathcal{S}$-symmetric $d$-dimensional TQFTs. An irreducible $\mathcal{S}$-symmetric gapped phase is one that cannot be expressed as a sum of other $\mathcal{S}$-symmetric gapped phases. Equivalently, the only local operators left invariant by $\mathcal{S}$ symmetry in an irreducible $\mathcal{S}$-symmetric phase are multiples of

the identity operator. It should be noted that the non-symmetric gapped phase underlying an irreducible $\mathcal{S}$-symmetric gapped phase may be reducible.

In terms of the SymTFT sandwich, an irreducible $\mathcal{S}$-symmetric TQFT $\mathfrak{Z}_d^{\mathcal{S}}$ is obtained by supplying an irreducible topological boundary condition $\mathfrak{B}_{\mathfrak{Z}_d^{\mathcal{S}}}^{\mathrm{phys}}$ of $\mathfrak{Z}_{d+1}(\mathcal{S})$ as the physical boundary. Thus, an irreducible $\mathcal{S}$-symmetric gapped phase is a deformation class of SymTFT sandwiches

$$\left[\mathfrak{S}_d^{\mathcal{S}}\right] = \left(\mathfrak{B}_{\mathcal{S}}^{\mathrm{sym}}, \mathfrak{Z}_{d+1}(\mathcal{S}), \left[\mathfrak{B}^{\mathrm{phys}}\right]\right), \qquad \text{(III.8)}$$

where $\left[\mathfrak{B}^{\mathrm{phys}}\right]$ is a deformation class of irreducible topological boundary conditions $\mathfrak{B}^{\mathrm{phys}}$ of $\mathfrak{Z}(\mathcal{S})$ that are used as physical boundary conditions.

For $d = 2$, as discussed previously, such a deformation class is described by a Lagrangian algebra $\mathcal{L}^{\mathrm{phys}}$ in the Drinfeld center $\mathcal{Z}(\mathcal{S})$. We thus have the one-to-one correspondence

$$\{(1+1)\text{d irreducible } \mathcal{S}\text{-symmetric gapped phases}\}$$

$$\updownarrow$$

$$\{\text{Lagrangian algebras } \mathcal{L}^{\mathrm{phys}} \text{ in } \mathcal{Z}(\mathcal{S})\}$$

$$\text{(III.9)}$$

As discussed in the previous subsection, the possible generalized charges appearing in an $\mathcal{S}$-symmetric theory $\mathfrak{T}$ correspond to the bulk topological operators that can end along the physical boundary $\mathfrak{B}_{\mathfrak{T}}^{\mathrm{phys}}$. For an irreducible $(1+1)$d $\mathcal{S}$-symmetric gapped phase $\left[\mathfrak{Z}^{\mathcal{S}}\right](\mathcal{L}^{\mathrm{phys}})$, these are captured by the corresponding Lagrangian algebra $\mathcal{L}^{\mathrm{phys}}$: the simple anyons $\mathbf{Q}_a$ appearing in $\mathcal{L}^{\mathrm{phys}}$ are the generalized charges carried by local operators appearing in $\left[\mathfrak{Z}^{\mathcal{S}}\right](\mathcal{L}^{\mathrm{phys}})$. In other words, these are the generalized charges carried by **order parameters** for the $\mathcal{S}$-symmetric gapped phase $\left[\mathfrak{Z}^{\mathcal{S}}\right](\mathcal{L}^{\mathrm{phys}})$, i.e. the operators having non-trivial vacuum expectation value (vev) in the phase $\left[\mathfrak{Z}^{\mathcal{S}}\right](\mathcal{L}^{\mathrm{phys}})$.

Mathematically, the classification of $\mathcal{S}$-symmetric $(1+1)$d gapped phases is the classification of deformation classes of certain pivotal 2-functors. See appendix A for more details.

## IV. CLUB QUICHE

A $d$-dimensional club quiche $\mathcal{Q}_d^C$ is a tuple

$$\mathcal{Q}_d^C = \left(\mathfrak{B}_d, \mathfrak{Z}_{d+1}, \mathcal{I}_d, \mathfrak{Z}'_{d+1}\right), \qquad \text{(IV.1)}$$

where $\mathfrak{Z}_{d+1}$ and $\mathfrak{Z}'_{d+1}$ are $(d+1)$-dimensional TQFTs, $\mathfrak{B}_d$ is a topological boundary condition of $\mathfrak{Z}_{d+1}$ and $\mathcal{I}_d$ is a topological interface from $\mathfrak{Z}_{d+1}$ to $\mathfrak{Z}'_{d+1}$. We will often use it in the context of SymTFT, where the club quiche takes the form

$$\mathcal{Q}_d^C = \left(\mathfrak{B}_{\mathcal{S}}^{\mathrm{sym}}, \mathfrak{Z}_{d+1}(\mathcal{S}), \mathcal{I}_d, \mathfrak{Z}'_{d+1}\right) \qquad \text{(IV.2)}$$

and we depict it as

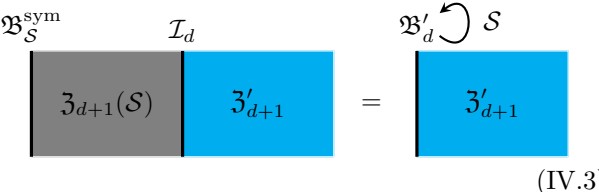

$$\text{(IV.3)}$$

where compactifying the interval occupied by the SymTFT $\mathfrak{Z}_{d+1}(\mathcal{S})$ leads to an $\mathcal{S}$-symmetric quiche

$$\mathcal{Q}_d^{\mathcal{S}} = (\mathfrak{B}'_d, \mathfrak{Z}'_{d+1}) \qquad \text{(IV.4)}$$

with the $\mathcal{S}$-symmetry being realized on a topological boundary $\mathfrak{B}'_d$ of the TQFT $\mathfrak{Z}'_{d+1}$. Conversely, any topological boundary $\mathfrak{B}'_d$ with symmetry $\mathcal{S}$ of $\mathfrak{Z}'_{d+1}$ can be expressed as such a club quiche. We have a one-to-one correspondence

$$\{\mathcal{S}\text{-symmetric Topological Boundaries of } \mathfrak{Z}'_{d+1}\}$$

$$\updownarrow$$

$$\{\text{Topological Interfaces from SymTFT } \mathfrak{Z}_{d+1}(\mathcal{S}) \text{ to } \mathfrak{Z}'_{d+1}\}$$

$$\text{(IV.5)}$$

Such topological interfaces are the same as topological boundary conditions of the folded $(d + 1)$-dimensional TQFT $\mathfrak{Z}_{d+1}(\mathcal{S}) \boxtimes \overline{\mathfrak{Z}}'_{d+1}$, where $\overline{\mathfrak{Z}}'_{d+1}$ is the $(d + 1)$-dimensional TQFT obtained by reflecting $\mathfrak{Z}'_{d+1}$, and the boxtimes operation $\boxtimes$ denotes the stacking of TQFTs.

### A. Gapped Boundary Phases with Non-Invertible Symmetries

Physically, such a club quiche can be understood as characterizing gapped boundary phases with $\mathcal{S}$-symmetry, where the symmetry is localized completely along the boundary. We do not incorporate any symmetry in the bulk, but will consider situations where both bulk and boundaries are symmetric in an upcoming work [127].

An $\mathcal{S}$-symmetric $d$-dimensional gapped boundary phase is defined as an equivalence class of $\mathcal{S}$-symmetric gapped bulk+boundary systems (where the bulk is $(d + 1)$-dimensional), in which two such systems are regarded to be equivalent if one of them can be deformed into the other without closing the gap in the bulk or on the boundary (even when both bulk and boundary have infinite volume) and without losing $\mathcal{S}$-symmetry at any point along the deformation path. One can identify an $\mathcal{S}$-symmetric gapped $d$-dimensional boundary phase as a deformation class of $\mathcal{S}$-symmetric $d$-dimensional quiches, and so we denote an $\mathcal{S}$-symmetric gapped $d$-dimensional boundary phase as $[\mathcal{Q}_d^{\mathcal{S}}]$.

Note that $[\mathcal{Q}_d^{\mathcal{S}}]$ has an underlying (non-symmetric) gapped boundary phase $[\mathcal{Q}_d]$

$$[\mathcal{Q}_d^{\mathcal{S}}] \mapsto [\mathcal{Q}_d], \qquad \text{(IV.6)}$$

where $[\mathcal{Q}_d]$ is the gapped boundary phase occupied by any $\mathcal{S}$-symmetric bulk+boundary system lying in the $\mathcal{S}$-symmetric gapped boundary phase $[\mathcal{Q}_d^{\mathcal{S}}]$.

An irreducible $\mathcal{S}$-symmetric gapped boundary phase is one that cannot be expressed as a sum of other $\mathcal{S}$-symmetric gapped boundary phases. Equivalently, the only local operators living on the boundary left invariant by $\mathcal{S}$ symmetry in an irreducible $\mathcal{S}$-symmetric boundary phase are multiples of the identity operator. It should be noted that the non-symmetric gapped boundary phase underlying an irreducible $\mathcal{S}$-symmetric gapped phase may be reducible.

For $d = 2$, one can characterize an $\mathcal{S}$-symmetric gapped boundary phase as

$$[\mathcal{Q}^{\mathcal{S}}] = (\mathfrak{B}_{\mathcal{S}}^{\text{sym}}, \mathfrak{Z}(\mathcal{S}), [\mathcal{I}], \mathfrak{Z}'), \qquad (\text{IV.7})$$

where $[\mathcal{I}]$ is a deformation class of topological interfaces from the 3d SymTFT $\mathfrak{Z}(\mathcal{S})$ to a 3d TQFT $\mathfrak{Z}'$. An irreducible $\mathcal{S}$-symmetric gapped boundary phase is then characterized by a deformation class of irreducible topological interfaces from $\mathfrak{Z}(\mathcal{S})$ to an irreducible 3d TQFT $\mathfrak{Z}'$, which by folding is the same as a deformation class of irreducible topological boundary conditions of the irreducible 3d TQFT $\mathfrak{Z}(\mathcal{S}) \boxtimes \overline{\mathfrak{Z}}'$. As we discussed earlier, such a deformation class of boundary conditions is characterized by a Lagrangian algebra in the MTC $\mathcal{Z}(\mathcal{S}) \boxtimes \overline{\mathcal{Z}}'$, where $\mathcal{Z}(\mathcal{S})$ is the Drinfeld center of $\mathcal{S}$ and $\overline{\mathcal{Z}}'$ is the MTC formed by topological lines of $\overline{\mathfrak{Z}}'$. We are thus led to one-to-one correspondence

$$\{\mathcal{S}\text{-symmetric gapped boundary phases w/ bulk phase } \mathfrak{Z}'\}$$

$$\updownarrow$$

$$\{\text{Lagrangian algebras in } \mathcal{Z}(\mathcal{S}) \boxtimes \overline{\mathcal{Z}}'\}$$

$$(\text{IV.8})$$

Consider a Lagrangian algebra $\mathcal{L}_{\mathcal{I}}$ of $\mathcal{Z}(\mathcal{S}) \boxtimes \overline{\mathcal{Z}}'$ characterizing an irreducible $\mathcal{S}$-symmetric gapped boundary phase $[\mathcal{Q}_{\mathcal{I}}^{\mathcal{S}}]$. It can be expressed as

$$\mathcal{L}_{\mathcal{I}} = \bigoplus_{a,a'} n_{a,a'} \left( \mathbf{Q}_a, \overline{\mathbf{Q}}'_{a'} \right), \qquad (\text{IV.9})$$

where $\mathbf{Q}_a$ are simple anyons in $\mathfrak{Z}(\mathcal{S})$ and $\mathbf{Q}'_{a'}$ are simple anyons in $\mathfrak{Z}'$. Let $\mathcal{I}$ be an irreducible topological interface in the deformation class $[\mathcal{I}]$. The presence of a term $n_{a,a'} \left( \mathbf{Q}_a, \overline{\mathbf{Q}}'_{a'} \right)$ in $\mathcal{L}_{\mathcal{I}}$ means that there is an $n_{a,a'}$-dimensional vector space of topological local operators along $\mathcal{I}$ acting as line changing operators from the line $\mathbf{Q}_a$ to the line $(\mathbf{Q}'_{a'})^*$, which is the dual of $\mathbf{Q}'_{a'}$ in the MTC $\mathcal{Z}'$ or in other words the orientation reversed ver-

sion of it

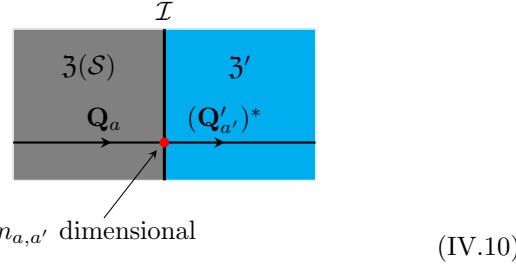

$$(\text{IV.10})$$

After contracting $\mathfrak{Z}(\mathcal{S})$, such an operator descends to a topological local operator along the resulting $\mathcal{S}$-symmetric boundary $\mathfrak{B}'$ of $\mathfrak{Z}'$, which is attached to the bulk anyon $\mathbf{Q}'_{a'}$, and carries a generalized charge $\mathbf{Q}_a$ under the symmetry $\mathcal{S}$ acting on $\mathfrak{B}'$. This may be regarded as an order parameter in the $\mathbf{Q}_a$-anyon sector for the resulting $\mathcal{S}$-symmetric $(1+1)$d gapped boundary phase $[\mathcal{Q}_{\mathcal{I}}^{\mathcal{S}}]$. Thus, the Lagrangian algebra $\mathcal{L}_{\mathcal{I}}$ captures the order parameters for the associated $\mathcal{S}$-symmetric gapped boundary phase.

Using the Lagrangian algebra $\mathcal{L}_{\mathcal{I}}$, one can also deduce the underlying non-symmetric gapped boundary phase

$$[\mathcal{Q}_{\mathcal{I}}] = ([\mathfrak{B}'], \mathfrak{Z}') . \qquad (\text{IV.11})$$

A topological boundary condition $\mathfrak{B}'$ in the deformation class $[\mathfrak{B}']$ is in general reducible

$$\mathfrak{B}' = \bigoplus_i \mathfrak{B}'_i, \qquad (\text{IV.12})$$

where $\mathfrak{B}'_i$ are irreducible topological boundary conditions of $\mathfrak{Z}'$. We can characterize $[\mathfrak{B}']$ in terms of a "Lagrangian algebra" $\mathcal{L}_{[\mathfrak{B}']}$

$$\mathcal{L}_{[\mathfrak{B}']} := \bigoplus_i \mathcal{L}_{[\mathfrak{B}'_i]}, \qquad (\text{IV.13})$$

where $\mathcal{L}_{[\mathfrak{B}'_i]}$ are Lagrangian algebras associated to deformation classes $[\mathfrak{B}'_i]$ of irreducible topological boundary conditions $\mathfrak{B}'_i$. This is related to $\mathcal{L}_{\mathcal{I}}$ via

$$\mathcal{L}_{[\mathfrak{B}']} = \bigoplus_{a,a'} n_{a*}^{\text{sym}} n_{a,a'} \mathbf{Q}'_{a'}, \qquad (\text{IV.14})$$

where $n_{a,a'}$ are the coefficients appearing in the Lagrangian algebra (IV.9) associated to the interface and $n_{a*}^{\text{sym}}$ is the coefficient for $\mathbf{Q}_a^*$ appearing in the Lagrangian algebra (II.18) associated to the symmetry boundary $\mathfrak{B}_{\mathcal{S}}^{\text{sym}}$.

We will further restrict the studies in this paper to $\mathcal{S}$-symmetric **minimal** $(1+1)$d gapped boundary phases, which are special types of $\mathcal{S}$-symmetric irreducible $(1+1)$d gapped boundary phases. In order to define such phases, note that the Lagrangian algebra $\mathcal{L}_{\mathcal{I}}$ specifies a non-Lagrangian condensable algebra $\mathcal{A}_{\mathcal{Z}(\mathcal{S})}(\mathcal{L}_{\mathcal{I}}) \in \mathcal{Z}(\mathcal{S})$ and a non-Lagrangian condensable algebra $\mathcal{A}_{\mathcal{Z}'}(\mathcal{L}_{\mathcal{I}}) \in$

$\mathcal{Z}'$, which are obtained respectively by restricting $\mathcal{L}_\mathcal{I}$ to $\overline{\mathbf{Q}'}_{a'} = 1$ or to $\mathbf{Q}_a = 1$, i.e.

$$\mathcal{A}_{\mathcal{Z}(\mathcal{S})}(\mathcal{L}_\mathcal{I}) = \bigoplus_a n_{a,1} \mathbf{Q}_a$$
$$\mathcal{A}_{\mathcal{Z}'}(\mathcal{L}_\mathcal{I}) = \bigoplus_{a'} n_{1,a'} \mathbf{Q}'_{a'} \,. \qquad \text{(IV.15)}$$

These condensable algebras capture the possible ends along a topological interface $\mathcal{I}$ in class $[\mathcal{I}]$ of topological bulk lines from left and right

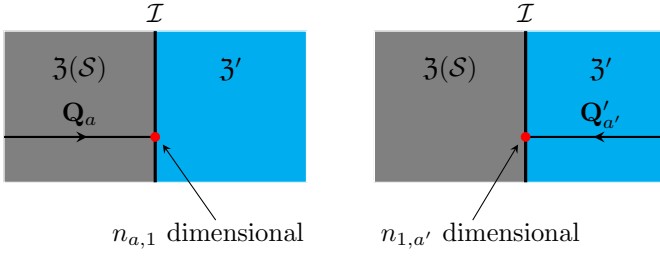

$$n_{a,1} \text{ dimensional} \qquad n_{1,a'} \text{ dimensional}$$

$$\text{(IV.16)}$$

Note that, if the algebra $\mathcal{A}_{\mathcal{Z}'}(\mathcal{L}_\mathcal{I})$ is non-trivial, then we have non-identity topological local operators in the irreducible $\mathcal{S}$-symmetric gapped boundary phase $[\mathcal{Q}_\mathcal{I}^\mathcal{S}]$ that are completely uncharged under $\mathcal{S}$. This is because the $\mathcal{S}$ symmetry is captured by topological lines living on the symmetry boundary $\mathfrak{B}_\mathcal{S}^{\mathrm{sym}}$ while a topological operator of the type appearing on the right hand side of (IV.16) does not interact with the symmetry boundary. This means that there is physical information in such a phase that is disconnected from the symmetry $\mathcal{S}$, e.g. some of the order parameters for such a phase carry trivial generalized charges under $\mathcal{S}$. We are thus led to define a **minimal** irreducible $\mathcal{S}$-symmetric gapped boundary phase $[\mathcal{Q}_\mathcal{I}^\mathcal{S}]$ to be a phase specified by a Lagrangian algebra $\mathcal{L}_\mathcal{I} \in \mathcal{Z}(\mathcal{S}) \boxtimes \overline{\mathcal{Z}}'$ whose associated condensable algebra $\mathcal{A}_{\mathcal{Z}'}(\mathcal{L}_\mathcal{I}) \in \mathcal{Z}'$ is trivial, i.e. $\mathcal{A}_{\mathcal{Z}'}(\mathcal{L}_\mathcal{I}) = 1$. Correspondingly we call such a Lagrangian $\mathcal{L}_\mathcal{I}$ as a **minimal Lagrangian**.

One can also define minimal club quiches and minimal $\mathcal{S}$-symmetric gapped phases even in higher spacetime dimensions by requiring that no topological line operators of $\mathfrak{Z}'_{d+1}$ can end on the topological interface $\mathcal{I}$.

The minimal $\mathcal{S}$-symmetric (1+1)d gapped boundary phases are classified (upto equivalences) by condensable algebras in $\mathcal{Z}(\mathcal{S})$. Pick a condensable algebra $\mathcal{A} \in \mathcal{Z}(\mathcal{S})$, then the associated gapped bulk phase $\mathfrak{Z}'$ is determined by computing local $\mathcal{A}$-modules in $\mathcal{Z}(\mathcal{S})$ [128], which form a "smaller" modular tensor category $\mathcal{Z}(\mathcal{S})/\mathcal{A}$

$$\mathcal{Z} = \mathcal{Z}(\mathcal{S})/\mathcal{A}. \qquad \text{(IV.17)}$$

The deformation class $[\mathcal{I}]$ of topological interfaces is then determined by picking a Lagrangian algebra

$$\mathcal{L}_\mathcal{I} \in \mathcal{Z}(\mathcal{S}) \boxtimes \overline{\mathcal{Z}(\mathcal{S})/\mathcal{A}} \qquad \text{(IV.18)}$$

such that the condensable algebra $\mathcal{A}_{\mathcal{Z}(\mathcal{S})}(\mathcal{L}_\mathcal{I}) \in \mathcal{Z}(\mathcal{S})$ associated to $\mathcal{L}_\mathcal{I}$ is the same as $\mathcal{A}$

$$\mathcal{A}_{\mathcal{Z}(\mathcal{S})}(\mathcal{L}_\mathcal{I}) = \mathcal{A}. \qquad \text{(IV.19)}$$

There may be multiple possibilities for $\mathcal{L}_\mathcal{I}$ satisfying the above condition, but they are all related by the action of some 0-form symmetry of $\mathfrak{Z}'$, which are auto-equivalences that do not change the physical properties of the resulting minimal $\mathcal{S}$-symmetric (1+1)d gapped boundary phase.

There are a couple of special choices for the condensable algebra $\mathcal{A}$:

- First of all, we can choose the trivial algebra

$$\mathcal{A} = 1, \qquad \text{(IV.20)}$$

which corresponds to the deformation class containing the identity interface $\mathcal{I}$. The corresponding minimal $\mathcal{S}$-symmetric gapped boundary phase is simply the phase in which the symmetry boundary $\mathfrak{B}_\mathcal{S}^{\mathrm{sym}}$ lies. The associated gapped bulk phase is

$$\mathfrak{Z}' = \mathfrak{Z}(\mathcal{S}). \qquad \text{(IV.21)}$$

- We can also choose a Lagrangian algebra

$$\mathcal{A} = \mathcal{L} \in \mathcal{Z}(\mathcal{S}). \qquad \text{(IV.22)}$$

In particular, for any $\mathcal{S}$ we have at least the choice $\mathcal{A} = \mathcal{L}_\mathcal{S}^{\mathrm{sym}}$ where $\mathcal{L}_\mathcal{S}^{\mathrm{sym}}$ is the Lagrangian algebra for the symmetry boundary $\mathfrak{B}_\mathcal{S}^{\mathrm{sym}}$. In this case the minimal $\mathcal{S}$-symmetric (1+1)d gapped boundary phase is simply an irreducible $\mathcal{S}$-symmetric (1+1)d gapped phase, i.e. the associated (2+1)d gapped bulk phase is trivial

$$\mathfrak{Z}' = 1. \qquad \text{(IV.23)}$$

The $\mathcal{S}$-symmetric (1+1)d gapped phase is the one obtained by performing a sandwich compactification of the SymTFT $\mathfrak{Z}(\mathcal{S})$ with the physical boundary specified by the Lagrangian algebra

$$\mathcal{L}^{\mathrm{phys}} = \mathcal{L}. \qquad \text{(IV.24)}$$

More details about such phases can be found in [2].

### B. $\mathbb{Z}_4$ Symmetry

Let us find (1+1)d $\mathbb{Z}_4$-symmetric minimal gapped boundary phases. In this case the symmetry is

$$\mathcal{S} = \mathsf{Vec}_{\mathbb{Z}_4} = \{1, P, P^2, P^3\}. \qquad \text{(IV.25)}$$

and $\mathcal{Z}(\mathcal{S})$ is the 3d $\mathbb{Z}_4$ Dijkgraaf-Witten gauge theory without twist (sometimes also called the $\mathbb{Z}_4$ toric code). Let us label the anyons in this theory by

$$e^i m^j, \qquad i, j \in \{0, 1, 2, 3\}. \qquad \text{(IV.26)}$$

The bosons are the electric lines $e^i$, the magnetic lines $m^j$, and the dyonic line $e^2 m^2$. We are interested in non-trivial condensable algebras that are non-Lagrangian. There are three possibilities

$$\mathcal{A}_e = 1 \oplus e^2, \quad \mathcal{A}_m = 1 \oplus m^2, \quad \mathcal{A}_{em} = 1 \oplus e^2 m^2. \qquad \text{(IV.27)}$$

We discuss these three possibilities in turn. The symmetry boundary is taken to be specified by the Lagrangian algebra

$$\mathcal{L}_{\mathsf{Vec}_{\mathbb{Z}_4}}^{\mathrm{sym}} = \bigoplus_{i=0}^{3} e^i. \tag{IV.28}$$

### 1. Condensable Algebra $\mathcal{A}_e$

In this case, one can compute that the bulk phase $\mathfrak{Z}'$ is the toric code

$$\mathcal{Z}' = \mathcal{Z}(\mathsf{Vec}_{\mathbb{Z}_4})/\mathcal{A}_e = \{1, e', m', e'm'\} = \mathcal{Z}(\mathsf{Vec}_{\mathbb{Z}_2}) \tag{IV.29}$$

This can be equivalently seen by noting that the Lagrangian algebra

$$\begin{aligned} \mathcal{L}_{\mathcal{I}} =& 1 \oplus e\,\overline{e}' \oplus e^2 \oplus e^3\,\overline{e}' \oplus m^2\,\overline{m}' \oplus em^2\,\overline{e}'\overline{m}' \\ &\oplus e^2 m^2\,\overline{m}' \oplus e^3 m^2\,\overline{e}'\overline{m}' \in \mathcal{Z}(\mathsf{Vec}_{\mathbb{Z}_4}) \boxtimes \overline{\mathcal{Z}}', \end{aligned} \tag{IV.30}$$

is minimal, and its associated condensable algebra in $\mathcal{Z}(\mathsf{Vec}_{\mathbb{Z}_4})$ matches $\mathcal{A}_e$

$$\mathcal{A}_{\mathcal{Z}(\mathsf{Vec}_{\mathbb{Z}_4})}(\mathcal{L}_{\mathcal{I}}) = \mathcal{A}_e. \tag{IV.31}$$

We can also compute the Lagrangian algebra in $\mathcal{Z}'$ corresponding to the underlying non-symmetric gapped boundary phase to be

$$\mathcal{L}_{[\mathfrak{B}']} = 2(1 \oplus e'), \tag{IV.32}$$

which means that the topological boundary condition $\mathfrak{B}'$ produced by colliding $\mathfrak{B}_{\mathcal{S}}^{\mathrm{sym}}$ with $\mathcal{I}$ is reducible, and can be expressed as

$$\mathfrak{B}' = \mathfrak{B}_0^e \oplus \mathfrak{B}_1^e, \tag{IV.33}$$

where $\mathfrak{B}_i^e$ is an irreducible topological boundary condition lying in the deformation class $[\mathfrak{B}](\mathcal{L}_e)$ associated to Lagrangian algebra

$$\mathcal{L}_e = 1 \oplus e' \in \mathcal{Z}'. \tag{IV.34}$$

The simple topological line operators on $\mathfrak{B}'$ are

$$1_{ij}, \quad P_{ij}, \qquad i, j \in \{0, 1\}, \tag{IV.35}$$

where $1_{ii}$ is the identity line on $\mathfrak{B}_i^e$; $P_{ii}$ is the line generating $\mathbb{Z}_2$ symmetry of $\mathfrak{B}_i^e$; $1_{01}, 1_{10}$ are boundary changing line operators satisfying

$$1_{ij} \otimes 1_{jk} = 1_{ik} \tag{IV.36}$$

and $P_{01}, P_{10}$ are another set of boundary changing line operators obtained by fusing with $P_{ii}$ lines

$$P_{ii} \otimes 1_{ij} = 1_{ij} \otimes P_{jj} = P_{ij}. \tag{IV.37}$$

We can recognize various boundary operators in terms of the operators descending from the club quiche:

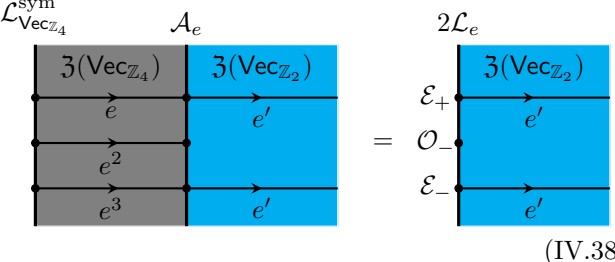

$$\tag{IV.38}$$

The products of these operators are

$$\mathcal{E}_s^2 = \mathcal{O}_-, \qquad \mathcal{E}_s \mathcal{O}_- = \mathcal{E}_{-s}, \qquad \mathcal{O}_-^2 = 1 \tag{IV.39}$$

for $s \in \{+, -\}$, which we can relate as

$$\begin{aligned} v_0 &= \frac{1 + \mathcal{O}_-}{2}, & v_1 &= \frac{1 - \mathcal{O}_-}{2} \\ \mathcal{E}_0 &= \frac{\mathcal{E}_+ + \mathcal{E}_-}{2}, & \mathcal{E}_1 &= i\frac{\mathcal{E}_+ - \mathcal{E}_-}{2}, \end{aligned} \tag{IV.40}$$

where $v_i$ are the identity local operators along the irreducible boundaries $\mathfrak{B}_i^e$, and $\mathcal{E}_i$ are the ends of $e'$ along boundaries $\mathfrak{B}_i^e$. Note that the overlap between the operators on two different boundaries is indeed trivial, as required by consistency

$$v_0 v_1 = 0, \quad \mathcal{E}_0 v_1 = 0, \quad \mathcal{E}_1 v_0 = 0, \quad \mathcal{E}_0 \mathcal{E}_1 = 0 \tag{IV.41}$$

and the $v_i$ are indeed identity along the boundaries as we have

$$v_i v_i = v_i, \qquad \mathcal{E}_i v_i = \mathcal{E}_i, \qquad \mathcal{E}_i \mathcal{E}_i = v_i. \tag{IV.42}$$

The linking action of $\mathbb{Z}_4$ symmetry is known on the ends of $\mathcal{Z}(\mathsf{Vec}_{\mathbb{Z}_4})$ anyons, which descends to the following linking action

$$P: \ \mathcal{E}_s \to is\mathcal{E}_s, \quad \mathcal{O}_- \to -\mathcal{O}_-, \quad 1 \to 1, \tag{IV.43}$$

implying that we have

$$P: \ v_0 \leftrightarrow v_1, \qquad \mathcal{E}_0 \to \mathcal{E}_1, \qquad \mathcal{E}_1 \to -\mathcal{E}_0. \tag{IV.44}$$

A topological line operator having such an action exists on the boundary $\mathfrak{B}'$ only if its components $\mathfrak{B}_i^e$ are the same boundaries, i.e. there is no relative Euler term between $\mathfrak{B}_0^e$ and $\mathfrak{B}_1^e$

$$\mathfrak{B}_0^e = \mathfrak{B}_1^e = \mathfrak{B}_\lambda(\mathcal{L}_e) \tag{IV.45}$$

for some $\lambda \in \mathbb{R}$. This parameter $\lambda$ drops out at the level of boundary phases where only deformation classes of boundary conditions matter. Given (IV.45), the linking action of lines on $v_i$ of lines living on $\mathfrak{B}'$ is

$$X_{ij}: \ v_i \to v_j, \qquad X \in \{1, P\} \tag{IV.46}$$

and the linking action on $\mathcal{E}_i$ is

$$\begin{aligned} 1_{ij} &: \ \mathcal{E}_i \to \mathcal{E}_j \\ P_{ij} &: \ \mathcal{E}_i \to -\mathcal{E}_j. \end{aligned} \tag{IV.47}$$

We can then identify the line operator on $\mathfrak{B}'$ generating the $\mathbb{Z}_4$ symmetry as

$$\phi(P) = 1_{01} \oplus P_{10}. \qquad \text{(IV.48)}$$

The $\mathbb{Z}_2$ subgroup of $\mathbb{Z}_4$ is generated by

$$\phi(P^2) = [\phi(P)]^2 = P_{00} \oplus P_{11} \qquad \text{(IV.49)}$$

and the consistency condition

$$\phi(P^4) = \phi(1) = 1 \qquad \text{(IV.50)}$$

is satisfied as

$$\phi(P^4) = [\phi(P)]^4 = 1_{00} \oplus 1_{11} = 1. \qquad \text{(IV.51)}$$

Note that the identity line on $\mathfrak{B}'$ is the sum $1_{00} \oplus 1_{11}$ on identity lines on $\mathfrak{B}_0^e$ and $\mathfrak{B}_1^e$.

Mathematically, we have provided information of a pivotal tensor functor

$$\phi: \ \mathsf{Vec}_{\mathbb{Z}_4} \to \mathrm{Mat}_2(\mathsf{Vec}_{\mathbb{Z}_2}) \qquad \text{(IV.52)}$$

where $\mathrm{Mat}_2(\mathsf{Vec}_{\mathbb{Z}_2})$ is the multi-fusion category formed by $2 \times 2$ matrices in $\mathsf{Vec}_{\mathbb{Z}_2}$, and is physically the category formed by topological line operators living on the boundary $\mathfrak{B}'$. This functor describes a choice of lines on $\mathfrak{B}'$ generating $\mathbb{Z}_4$ symmetry, and is the mathematical characterization of the minimal $\mathbb{Z}_4$-symmetric (1+1)d gapped boundary phase under study. See appendix A for more details.

Note that, at the beginning of this analysis, we could have picked another Lagrangian algebra

$$\mathcal{L}_{[\mathcal{I}']} = 1 \oplus e\,\overline{m}' \oplus e^2 \oplus e^3\,\overline{m}' \oplus m^2\,\overline{e}' \oplus em^2\,\overline{m}'\overline{e}'$$
$$\oplus\, e^2 m^2\,\overline{e}' \oplus e^3 m^2\,\overline{m}'\overline{e}' \in \mathcal{Z}(\mathsf{Vec}_{\mathbb{Z}_4}) \boxtimes \overline{\mathcal{Z}}', \qquad \text{(IV.53)}$$

whose associated condensable algebra $\mathcal{A}_{\mathcal{Z}(\mathsf{Vec}_{\mathbb{Z}_4})}(\mathcal{L}_{[\mathcal{I}']})$ is also $\mathcal{A}_e$. This choice leads to the $\mathbb{Z}_4$-symmetry being realized on

$$\mathfrak{B}' = \mathfrak{B}_0^m \oplus \mathfrak{B}_1^m \qquad \text{(IV.54)}$$

where

$$\mathfrak{B}_0^m = \mathfrak{B}_1^m = \mathfrak{B}_\lambda(\mathcal{L}_m) \qquad \text{(IV.55)}$$

and $\mathfrak{B}_\lambda(\mathcal{L}_m)$ is an irreducible topological boundary of the toric code associated to the Lagrangian algebra $1 \oplus m' \in \mathcal{Z}'$. The lines living on such a $\mathfrak{B}_0^m \oplus \mathfrak{B}_1^m$ have identical properties as the lines living on $\mathfrak{B}_0^e \oplus \mathfrak{B}_1^e$, and hence we obtain an equivalent minimal $\mathbb{Z}_4$-symmetric gapped boundary phase. This is related to the fact that the irreducible boundaries $\mathfrak{B}_\lambda(\mathcal{L}_e)$ and $\mathfrak{B}_\lambda(\mathcal{L}_m)$ are exchanged by the action of a 0-form symmetry of the toric code, namely the electric-magnetic symmetry exchanging the electric and magnetic lines $e' \leftrightarrow m'$.

### 2. Condensable Algebra $\mathcal{A}_m$

In this case, $\mathfrak{Z}'$ is again the toric code and we can take

$$\mathcal{L}_{\mathcal{I}} = 1 \oplus m\,\overline{m}' \oplus m^2 \oplus m^3\,\overline{m}' \oplus e^2\,\overline{e}' \oplus e^2 m\,\overline{e}'\overline{m}'$$
$$\oplus\, e^2 m^2\,\overline{e}' \oplus e^2 m^3\,\overline{e}'\overline{m}' \in \mathcal{Z}(\mathsf{Vec}_{\mathbb{Z}_4}) \boxtimes \overline{\mathcal{Z}}'. \qquad \text{(IV.56)}$$

The Lagrangian algebra associated to the underlying non-symmetric gapped boundary phase is computed to be

$$\mathcal{L}_{[\mathfrak{B}']} = 1 \oplus e' \qquad \text{(IV.57)}$$

which means that $\mathfrak{B}'$ is an irreducible topological boundary on which the electric anyon $e'$ is condensed, i.e.

$$\mathfrak{B}' = \mathfrak{B}_\lambda(\mathcal{L}_e) \qquad \text{(IV.58)}$$

for some $\lambda \in \mathbb{R}$. Again, the parameter $\lambda$ will not appear at the level of phases. As discussed earlier, the topological lines on the boundary are

$$1, \qquad P' \qquad \text{(IV.59)}$$

where 1 is the identity line and $P'$ generates $\mathbb{Z}_2$ symmetry localized on the boundary.

The operators descending from the club quiche are

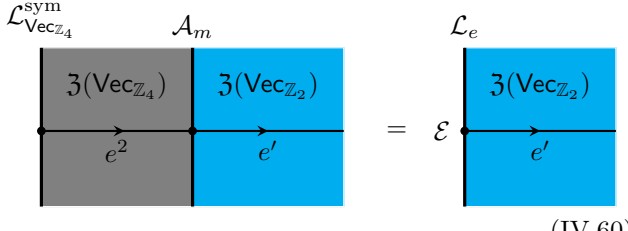

$$\text{(IV.60)}$$

with the action of $\mathbb{Z}_4$ being

$$P: \ \mathcal{E} \to -\mathcal{E}. \qquad \text{(IV.61)}$$

We can thus recognize the $\mathbb{Z}_4$ symmetry generator on $\mathfrak{B}'$ as

$$\phi(P) = P' \qquad \text{(IV.62)}$$

which indeed satisfies the consistency condition

$$\phi(P^4) = 1 \qquad \text{(IV.63)}$$

because $[\phi(P)]^4 = 1$.

Mathematically, we have provided a pivotal tensor functor

$$\phi: \ \mathsf{Vec}_{\mathbb{Z}_4} \to \mathsf{Vec}_{\mathbb{Z}_2} \qquad \text{(IV.64)}$$

where $\mathsf{Vec}_{\mathbb{Z}_2}$ is the fusion category formed by topological lines living on the boundary $\mathfrak{B}'$. All the data of this functor simply descends from the non-trivial homomorphism $\mathbb{Z}_4 \to \mathbb{Z}_2$.

### 3. Condensable Algebra $\mathcal{A}_{em}$

For the condensable algebra $\mathcal{A}_{em} = 1 \oplus e^2 m^2$, $\mathfrak{Z}'$ is the double semion, which is also the SymTFT $\mathfrak{Z}(\mathsf{Vec}^\omega_{\mathbb{Z}_2})$ for $\mathbb{Z}_2$ symmetry with a non-trivial 't Hooft anomaly

$$\omega \neq 0 \in H^3(\mathbb{Z}_2, U(1)) = \mathbb{Z}_2. \tag{IV.65}$$

That is, the anyons of $\mathfrak{Z}'$ are

$$\mathcal{Z}' = \mathcal{Z}(\mathsf{Vec}_{\mathbb{Z}_4})/\mathcal{A}_{em} = \{1, s, \overline{s}, s\overline{s}\} = \mathcal{Z}(\mathsf{Vec}^\omega_{\mathbb{Z}_2}), \tag{IV.66}$$

where $s$ is a semion, $\overline{s}$ is an anti-semion and $s\overline{s}$ is a boson.

We can take the Lagrangian algebra completion of $\mathcal{A}_{em}$ to be

$$\begin{aligned}\mathcal{L}_\mathcal{I} =&\, 1 \oplus em\,\overline{s} \oplus e^2 m^2 \oplus e^3 m^3\,\overline{s} \oplus em^3\,s \oplus e^2\,s\overline{s} \\ &\oplus e^3 m\,s \oplus m^2\,s\overline{s} \in \mathcal{Z}(\mathsf{Vec}_{\mathbb{Z}_4}) \boxtimes \overline{\mathcal{Z}}'\end{aligned} \tag{IV.67}$$

from which we find the Lagrangian algebra for the underlying non-symmetric gapped boundary phase to be

$$\mathcal{L}_{[\mathfrak{B}']} = \mathcal{L}_{s\overline{s}} = 1 \oplus s\overline{s}, \tag{IV.68}$$

which corresponds to an irreducible topological boundary condition of the double semion on which the anomalous $\mathbb{Z}_2$ symmetry is realized, that is

$$\mathfrak{B}' = \mathfrak{B}_\lambda(\mathcal{L}_{s\overline{s}}) \tag{IV.69}$$

for some $\lambda \in \mathbb{R}$. The line operators living on $\mathfrak{B}'$ are

$$1, \qquad P', \tag{IV.70}$$

where $P'$ generates the anomalous $\mathbb{Z}_2$ symmetry, with the anomaly encoded in the relation

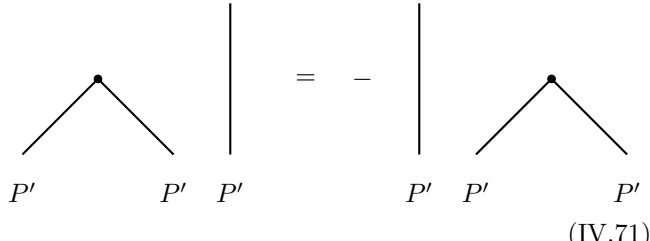

$$\tag{IV.71}$$

obeyed by the topological line $P'$.

The local operators in the club quiche compactification are

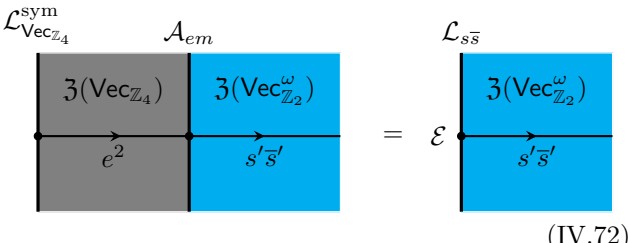

$$\tag{IV.72}$$

with the action of $\mathbb{Z}_4$ being

$$P : \ \mathcal{E} \to -\mathcal{E} \tag{IV.73}$$

which determines the line operator on $\mathfrak{B}'$ generating the $\mathbb{Z}_4$ symmetry to be

$$\phi(P) = P'. \tag{IV.74}$$

Note that the $\mathbb{Z}_4$ symmetry is taken to be non-anomalous, which means that we should be able to find topological local operators $\mathcal{O}_{ij}$ at the junctions of line operators generating $\mathbb{Z}_4$ symmetry

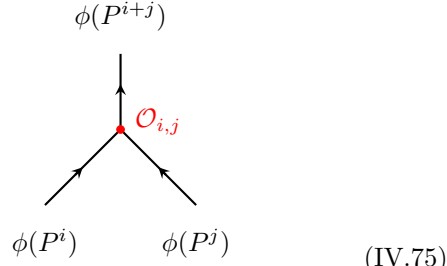

$$\tag{IV.75}$$

obeying relationship

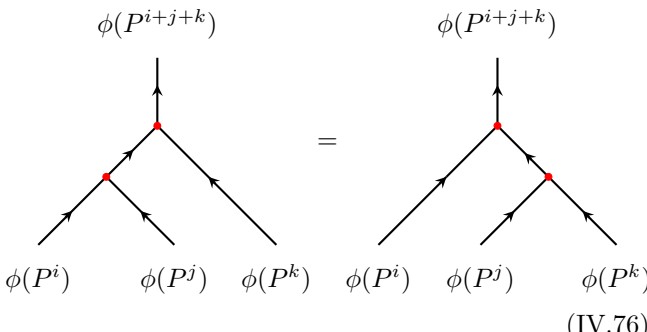

$$\tag{IV.76}$$

We can reduce a choice of such operators to a choice of elements $\beta_{i,j} \in \mathbb{C}^\times$ via

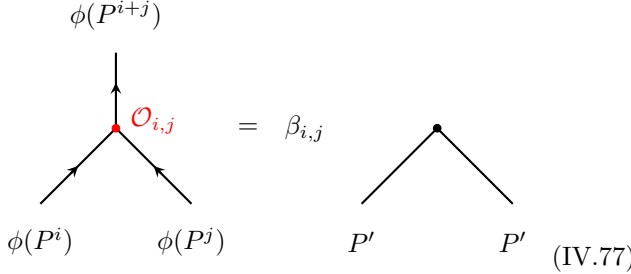

$$\tag{IV.77}$$

for $i, j$ both odd, where a black dot denotes the operator appearing in (IV.71),

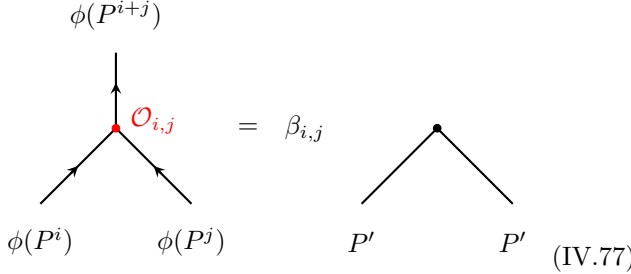

$$\tag{IV.78}$$

for one of $i, j$ being odd and the other being even, and

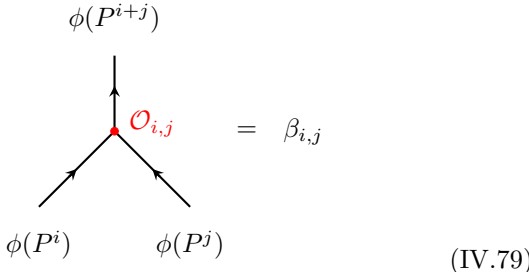

$$\text{(IV.79)}$$

for $i, j$ both even. Using (IV.71), the reader can verify that a choice obeying (IV.76) is

$$\beta_{i,0} = \beta_{0,i} = 1$$
$$\beta_{1,1} = \beta_{2,1} = \beta_{3,1} = \beta_{2,2} = \beta_{2,3} = 1 \qquad \text{(IV.80)}$$
$$\beta_{1,2} = \beta_{1,3} = \beta_{3,2} = \beta_{3,3} = -1 \,.$$

Mathematically, we have provided a pivotal tensor functor

$$\phi : \ \mathsf{Vec}_{\mathbb{Z}_4} \to \mathsf{Vec}_{\mathbb{Z}_2}^{\omega} \,. \qquad \text{(IV.81)}$$

### C. $S_3$ Symmetry

Let us now discuss the non-trivial, minimal (1+1)d $S_3$-symmetric gapped boundary phases. Here the symmetry is

$$\mathcal{S} = \mathsf{Vec}_{S_3} = \{1, a, a^2, b, ab, a^2 b\} \,, \qquad \text{(IV.82)}$$

where

$$b^2 = 1 \,, \qquad a^3 = 1 \,, \qquad ab = ba^2 \,. \qquad \text{(IV.83)}$$

The SymTFT $\mathfrak{Z}(\mathcal{S})$ for this symmetry is then the 3d $S_3$ Dijkgraaf-Witten gauge theory without a twist (which can be obtained from the 3d $\mathbb{Z}_3$ DW gauge theory by gauging the $\mathbb{Z}_2$ automorphism symmetry, which exchanges the pairs $(e, m) \leftrightarrow (e^2, m^2)$). The anyons of $\mathfrak{Z}(\mathsf{Vec}_{S_3})$ are labeled by conjugacy classes of $S_3$ and the irreducible representations of the centralizers of these conjugacy classes:

$$\mathfrak{Z}(\mathsf{Vec}_{S_3}) = \{1, 1_-, E, a_1, a_\omega, a_{\omega^2}, b_+, b_-\} \,, \quad \text{(IV.84)}$$

where $1_-$ is the sign representation of $S_3$, $E$ the 2d irreducible representation of $S_3$, $\omega = e^{\pm 2\pi i/3}$ is a $\mathbb{Z}_3$ irreducible representation, and $+, -$ denote the $\mathbb{Z}_2$ irreducible representations. The bosonic lines in this case are

$$1, 1_-, E, a_1, b_+ \,, \qquad \text{(IV.85)}$$

from which one can construct the condensable algebras of $\mathfrak{Z}(\mathsf{Vec}_{S_3})$. There are three condensable, non-Lagrangian, algebras

$$\mathcal{A}_{1_-} = 1 \oplus 1_- \,, \quad \mathcal{A}_E = 1 \oplus E \,, \quad \mathcal{A}_{a_1} = 1 \oplus a_1 \,, \quad \text{(IV.86)}$$

which we now study case by case. Note that the algebra $1 \oplus b_+$ is not a condensable algebra, as it does not satisfy the condition (CA3) in appendix B.

For the symmetry boundary we fix a Lagrangian algebra which corresponds to the $S_3$ symmetry,

$$\mathcal{L}_{\mathsf{Vec}_{S_3}} = 1 \oplus 1_- \oplus 2E \,. \qquad \text{(IV.87)}$$

#### 1. Condensable Algebra $\mathcal{A}_{1_-}$

In this case, we have that $\mathfrak{Z}'$ is the $\mathbb{Z}_3$ DW theory with anyon content

$$\mathcal{Z}' = \mathcal{Z}/\mathcal{A}_{1_-} = \mathcal{Z}(\mathsf{Vec}_{\mathbb{Z}_3})$$
$$= \{1, e, e^2, m, m^2, em, e^2 m, em^2, e^2 m^2\} \,. \qquad \text{(IV.88)}$$

A Lagrangian algebra in $\mathcal{Z}(S_3) \boxtimes \overline{\mathcal{Z}}'$ which completes $\mathcal{A}_{1_-}$ is

$$\mathcal{L}_{\mathcal{I}} = 1 \oplus 1_- \oplus a_1 \overline{m} \oplus a_1 \overline{m}^2 \oplus a_{\omega^2} \overline{em} \oplus$$
$$\oplus a_{\omega^2} \overline{e}^2 \overline{m}^2 \oplus E\overline{e} \oplus E\overline{e}^2 \oplus a_\omega \overline{em}^2 \oplus a_\omega \overline{e}^2 \overline{m} \,. \qquad \text{(IV.89)}$$

The Lagrangian algebra for the underlying non-symmetric gapped boundary phase is obtained by following the $\mathcal{L}_{\mathsf{Vec}_{S_3}}$ lines and seeing what they transform into via $\mathcal{L}_{\mathcal{I}}$; this gives

$$\mathcal{L}_{\mathfrak{B}'} = 2(1 \oplus e \oplus e^2) \,. \qquad \text{(IV.90)}$$

This implies that the underlying boundary of the $\mathbb{Z}_3$ DW theory is reducible and given by

$$\mathfrak{B}' = \mathfrak{B}_0^e \oplus \mathfrak{B}_1^e \,, \qquad \text{(IV.91)}$$

where $\mathfrak{B}_i^e$ is an irreducible topological boundary condition lying in the deformation class $[\mathfrak{B}](\mathcal{L}_e)$ associated to Lagrangian algebra

$$\mathcal{L}_e = 1 \oplus e \oplus e^2 \in \mathcal{Z}' \,. \qquad \text{(IV.92)}$$

The simple topological lines on the boundary $\mathfrak{B}'$ are

$$1_{ij}, \quad P_{ij}, \quad P_{ij}^2, \qquad i, j \in \{0, 1\} \,. \qquad \text{(IV.93)}$$

where $1_{ii}$ is the identity line on $\mathfrak{B}_i^e$; $P_{ii}$ is a line generating $\mathbb{Z}_3$ symmetry of $\mathfrak{B}_i^e$; $P_{ii}^2$ is the square of $P_{ii}$; $1_{01}, 1_{10}$ are boundary changing line operators satisfying

$$1_{ij} \otimes 1_{jk} = 1_{ik} \,. \qquad \text{(IV.94)}$$

$P_{01}, P_{10}$ are another set of boundary changing line operators obtained by fusing with $P_{ii}$ lines

$$P_{ii} \otimes 1_{ij} = 1_{ij} \otimes P_{jj} = P_{ij} \qquad \text{(IV.95)}$$

and $P_{01}^2, P_{10}^2$ are boundary changing line operators obtained by fusing with $P_{ii}^2$ lines

$$P_{ii}^2 \otimes 1_{ij} = 1_{ij} \otimes P_{jj}^2 = P_{ij}^2 \,. \qquad \text{(IV.96)}$$

We can recognize various boundary operators in terms of operators descending from the club quiche

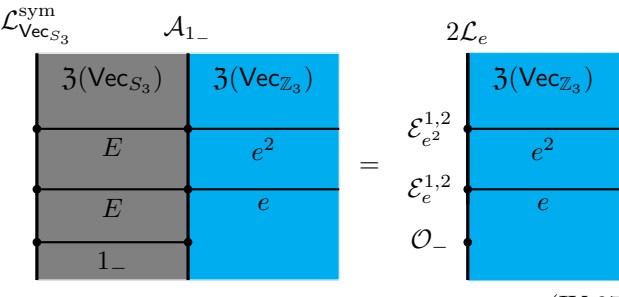

$$(\text{IV.97})$$

where we obtain two operators $\mathcal{E}^i_{e^2}$ and two operators $\mathcal{E}^i_{e^2}$ because there are two possible ends of the $E$ line along the symmetry boundary denoted by $\mathcal{L}_{\mathsf{Vec}_{S_3}}$. The products of these operators are determined to be

$$\mathcal{O}_-^2 = 1, \quad \mathcal{O}_-\mathcal{E}^1_e = \mathcal{E}^1_e, \quad \mathcal{O}_-\mathcal{E}^2_e = -\mathcal{E}^2_e$$
$$\mathcal{E}^1_e\mathcal{E}^1_e = \mathcal{E}^2_{e^2}, \quad \mathcal{E}^2_e\mathcal{E}^2_e = \mathcal{E}^1_{e^2}, \quad \mathcal{E}^1_e\mathcal{E}^2_e = 0, \quad (\text{IV.98})$$

which implies we can just focus on $\mathcal{E}^1_e$ and $\mathcal{E}^2_e$ as $\mathcal{E}^i_{e^2}$ is related to the square of $\mathcal{E}^j_e$. These products are determined in a similar way as in [2], by demanding that

- their product is consistent with the fusions of attached bulk anyons,

- their product is associative,

- their product is consistent with the action of topological lines living on the boundaries, and

- performing rescalings of these operators to put the product in a neat form.

The identity local operators along the irreducible boundaries $\mathfrak{B}^e_0$ and $\mathfrak{B}^e_1$ are

$$v_0 = \frac{1+\mathcal{O}_-}{2} \quad , \quad v_1 = \frac{1-\mathcal{O}_-}{2}. \quad (\text{IV.99})$$

$\mathcal{E}^1_e$ is then the end of $e$ along $\mathfrak{B}^e_0$ and $\mathcal{E}^2_e$ is the end of $e$ along $\mathfrak{B}^e_1$. This is due to the products

$$v_iv_j = \delta_{ij}v_j, \quad \mathcal{E}^1_ev_0 = \mathcal{E}^1_e, \quad \mathcal{E}^1_ev_1 = 0$$
$$\mathcal{E}^2_ev_0 = 0, \quad \mathcal{E}^2_ev_1 = \mathcal{E}^2_e, \quad \mathcal{E}^1_e\mathcal{E}^2_e = 0. \quad (\text{IV.100})$$

The linking action of the $S_3$ symmetry generators are

$$b: \begin{array}{c} \mathcal{E}^1_{e^i} \leftrightarrow \mathcal{E}^2_{e^i} \\ \mathcal{O}_- \to -\mathcal{O}_- \end{array} \qquad a: \begin{array}{c} \mathcal{E}^1_{e^i} \to \omega\mathcal{E}^1_{e^i} \\ \mathcal{E}^2_{e^i} \to \omega^2\mathcal{E}^2_{e^i} \\ \mathcal{O}_- \to \mathcal{O}_-. \end{array} \quad (\text{IV.101})$$

which identifies the $S_3$ generators to be the following line operators on $\mathfrak{B}'$

$$\phi(a) = P_{00} \oplus P^2_{11}$$
$$\phi(b) = 1_{01} \oplus 1_{10} \quad (\text{IV.102})$$

and constrains the relative Euler term between the boundaries $\mathfrak{B}^e_0$ and $\mathfrak{B}^e_0$ to be trivial

$$\mathfrak{B}^e_0 = \mathfrak{B}^e_1 = \mathfrak{B}_\lambda(\mathcal{L}_e) \quad (\text{IV.103})$$

for some $\lambda \in \mathbb{R}$. The reader can check that the lines $\phi(a), \phi(b)$ satisfy all the $S_3$ relations.

Mathematically, we have provided information of a pivotal tensor functor

$$\phi: \ \mathsf{Vec}_{S_3} \to \text{Mat}_2(\mathsf{Vec}_{\mathbb{Z}_3}) \quad (\text{IV.104})$$

where physically $\text{Mat}_2(\mathsf{Vec}_{\mathbb{Z}_3})$ is the multi-fusion the category formed by topological line operators living on the boundary $\mathfrak{B}'$. This functor describes a choice of lines on $\mathfrak{B}'$ generating $S_3$ symmetry, and is the mathematical characterization of the minimal $S_3$-symmetric (1+1)d gapped boundary phase under study. See appendix A for more details.

### 2. Condensable Algebra $\mathcal{A}_E$

In this case, we have that $\mathfrak{Z}'$ is the $\mathbb{Z}_2$ DW theory (i.e. toric code)

$$\mathcal{Z}' = \mathcal{Z}/\mathcal{A}_E = \mathcal{Z}(\mathsf{Vec}_{\mathbb{Z}_2}) = \{1, e, m, f\}. \quad (\text{IV.105})$$

A Lagrangian algebra in $\mathcal{Z}(S_3) \boxtimes \overline{\mathcal{Z}}'$ which completes $\mathcal{A}_E$ is

$$\mathcal{L}_I = 1 \oplus E \oplus 1_-\overline{e} \oplus E\overline{e} \oplus b_+\overline{m} \oplus b_-\overline{f}. \quad (\text{IV.106})$$

The Lagrangian algebra for the underlying non-symmetric gapped boundary phase is obtained by following the $\mathcal{L}_{\mathsf{Vec}_{S_3}}$ lines and seeing what they transform into via $\mathcal{L}_\mathcal{I}$; this gives

$$\mathcal{L}_{\mathfrak{B}'} = 3(1 \oplus e). \quad (\text{IV.107})$$

This implies that the underlying boundary of the $\mathbb{Z}_2$ DW theory is reducible and given by

$$\mathfrak{B}' = \mathfrak{B}^e_0 \oplus \mathfrak{B}^e_1 \oplus \mathfrak{B}^e_2, \quad (\text{IV.108})$$

where $\mathfrak{B}^e_i$ is an irreducible topological boundary condition lying in the deformation class $[\mathfrak{B}](\mathcal{L}_e)$ associated to Lagrangian algebra $\mathcal{L}_e = 1 \oplus e$.

The topological lines on the boundary $\mathfrak{B}'$ are

$$1_{ij}, \quad P_{ij}, \quad i, j \in \{0, 1\}. \quad (\text{IV.109})$$

where $1_{ii}$ is the identity line on $\mathfrak{B}^e_i$; $P_{ii}$ is the line generating $\mathbb{Z}_2$ symmetry of $\mathfrak{B}^e_i$; $1_{ij}$ for $i \neq j$ are boundary changing line operators satisfying

$$1_{ij} \otimes 1_{jk} = 1_{ik} \quad (\text{IV.110})$$

and $P_{ij}$ for $i \neq j$ are another set of boundary changing line operators obtained by fusing with $P_{ii}$ lines

$$P_{ii} \otimes 1_{ij} = 1_{ij} \otimes P_{jj} = P_{ij}. \quad (\text{IV.111})$$

The various boundary operators in terms of operators descending from the club quiche are

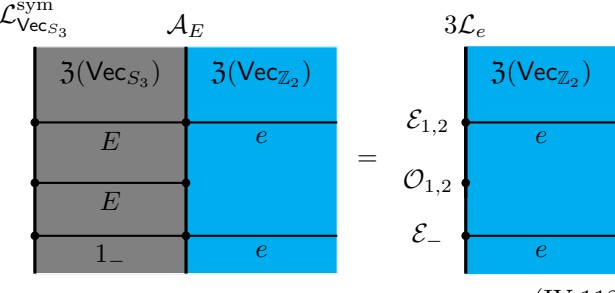

$$\text{(IV.112)}$$

The products of these operators are determined to be

$$
\begin{array}{lll}
\mathcal{E}_-\mathcal{E}_- = 1, & \mathcal{E}_-\mathcal{O}_1 = \mathcal{E}_1, & \mathcal{E}_1\mathcal{E}_1 = \mathcal{O}_2, \\
\mathcal{O}_1\mathcal{O}_1 = \mathcal{O}_2, & \mathcal{E}_-\mathcal{O}_2 = -\mathcal{E}_2, & \mathcal{E}_2\mathcal{E}_2 = \mathcal{O}_1, \\
\mathcal{O}_2\mathcal{O}_2 = \mathcal{O}_1, & \mathcal{E}_-\mathcal{E}_1 = \mathcal{O}_1, & \mathcal{E}_1\mathcal{E}_2 = -1, \\
\mathcal{O}_1\mathcal{O}_2 = 1, & \mathcal{E}_-\mathcal{E}_2 = -\mathcal{O}_2 & \mathcal{E}_1\mathcal{O}_1 = -\mathcal{E}_2, \\
\mathcal{E}_2\mathcal{O}_2 = -\mathcal{E}_1, & \mathcal{E}_1\mathcal{O}_2 = \mathcal{E}_-, & \mathcal{E}_2\mathcal{O}_1 = -\mathcal{E}_-\,.
\end{array}
$$

$$\text{(IV.113)}$$

From these we construct the operators

$$
v_i = \frac{1 + \omega^i \mathcal{O}_1 + \omega^{2i} \mathcal{O}_2}{3}
$$
$$
\widehat{\mathcal{E}}_i = \frac{\mathcal{E}_- + \omega^i \mathcal{E}_1 - \omega^{2i} \mathcal{E}_2}{3} = \mathcal{E}_- v_i\,,
$$

$$\text{(IV.114)}$$

where $i \in \{0, 1, 2\}$, with $v_i$ being identity local operators along $\mathfrak{B}_i^e$, and $\widehat{\mathcal{E}}_i$ being the ends of $e$ along $\mathfrak{B}_i^e$. This can be seen from the products

$$
v_i v_j = \delta_{ij} v_j, \qquad \widehat{\mathcal{E}}_i v_j = \delta_{ij}\widehat{\mathcal{E}}_j, \qquad \widehat{\mathcal{E}}_i\widehat{\mathcal{E}}_j = \delta_{ij} v_j\,.
$$

$$\text{(IV.115)}$$

derived using the above product rules.

The linking actions of the $S_3$ symmetry generators are

$$
\begin{array}{ll}
& \mathcal{O}_1 \leftrightarrow \mathcal{O}_2 \qquad\qquad \mathcal{O}_k \to \omega^k \mathcal{O}_k \\
b: & \mathcal{E}_1 \leftrightarrow \mathcal{E}_2 \qquad a: \quad \mathcal{E}_k \to \omega^k \mathcal{E}_k \\
& \mathcal{E}_- \to -\mathcal{E}_- \qquad\qquad \mathcal{E}_- \to \mathcal{E}_-\,,
\end{array}
$$

$$\text{(IV.116)}$$

for $k \in \{1, 2\}$, from which we find the linking actions

$$
\begin{array}{ll}
& v_0 \to v_0 \qquad\qquad v_0 \to v_1 \\
& v_1 \to v_2 \qquad\qquad v_1 \to v_2 \\
& v_2 \to v_1 \qquad\qquad v_2 \to v_0 \\
b: & \widehat{\mathcal{E}}_0 \to -\widehat{\mathcal{E}}_0 \qquad a: \quad \widehat{\mathcal{E}}_0 \to \widehat{\mathcal{E}}_1 \\
& \widehat{\mathcal{E}}_1 \to -\widehat{\mathcal{E}}_2 \qquad\qquad \widehat{\mathcal{E}}_1 \to \widehat{\mathcal{E}}_2 \\
& \widehat{\mathcal{E}}_2 \to -\widehat{\mathcal{E}}_1 \qquad\qquad \widehat{\mathcal{E}}_2 \to \widehat{\mathcal{E}}_0\,.
\end{array}
$$

$$\text{(IV.117)}$$

which identifies the $S_3$ generators to be the following line operators on $\mathfrak{B}'$

$$
\phi(a) = 1_{01} \oplus 1_{12} \oplus 1_{20}
$$
$$
\phi(b) = P_{00} \oplus P_{12} \oplus P_{21}
$$

$$\text{(IV.118)}$$

and constrains the relative Euler term between the boundaries $\mathfrak{B}_i^e$ to be trivial

$$
\mathfrak{B}_0^e = \mathfrak{B}_1^e = \mathfrak{B}_2^e = \mathfrak{B}_\lambda(\mathcal{L}_e)
$$

$$\text{(IV.119)}$$

for some $\lambda \in \mathbb{R}$. The reader can check that the lines $\phi(a), \phi(b)$ satisfy all the $S_3$ relations.

Mathematically, we have provided information of a pivotal tensor functor

$$
\phi: \; \mathsf{Vec}_{S_3} \to \mathrm{Mat}_3(\mathsf{Vec}_{\mathbb{Z}_2})\,,
$$

$$\text{(IV.120)}$$

where $\mathrm{Mat}_3(\mathsf{Vec}_{\mathbb{Z}_2})$ is the multi-fusion category formed by $3 \times 3$ matrices in $\mathsf{Vec}_{\mathbb{Z}_2}$, and is physically the category formed by topological line operators living on the boundary $\mathfrak{B}'$.

### 3. Condensable Algebra $\mathcal{A}_{a_1}$

In this case, $\mathcal{Z}'$ is again the toric code

$$
\mathcal{Z}' = \mathcal{Z}/\mathcal{A}_{a_1} = \mathcal{Z}(\mathsf{Vec}_{\mathbb{Z}_2}) = \{1, e, m, f\}\,.
$$

$$\text{(IV.121)}$$

A Lagrangian algebra in $\mathcal{Z}(S_3) \boxtimes \overline{\mathcal{Z}}'$ which completes $\mathcal{A}_E$ is

$$
\mathcal{L}_I = 1 \oplus a_1 \oplus 1_-\overline{e} \oplus a_1\overline{e} \oplus b_+\overline{m} \oplus b_-\overline{f}\,.
$$

$$\text{(IV.122)}$$

The Lagrangian algebra for the underlying non-symmetric gapped boundary phase is obtained easily to be

$$
\mathcal{L}_{\mathfrak{B}'} = 1 \oplus e\,.
$$

$$\text{(IV.123)}$$

This implies that the underlying boundary of the $\mathbb{Z}_2$ DW theory is irreducible

$$
\mathfrak{B}' = \mathfrak{B}^e\,,
$$

$$\text{(IV.124)}$$

and associated to Lagrangian algebra $\mathcal{L}_e = 1 \oplus e$. The topological lines on the boundary are

$$
1, \quad P\,,
$$

$$\text{(IV.125)}$$

where $P$ generates the $\mathbb{Z}_2$ symmetry localized on the boundary.

Here the club quiche is simply

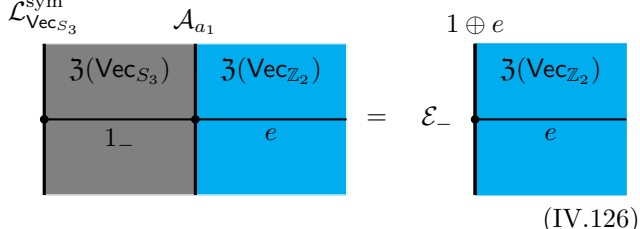

$$\text{(IV.126)}$$

with the action of $S_3$ being

$$
a: \; \mathcal{E}_- \to \mathcal{E}_-, \qquad b: \; \mathcal{E}_- \to -\mathcal{E}_-
$$

$$\text{(IV.127)}$$

from which we identify the lines

$$\phi(b) = P, \qquad \phi(a) = 1 \qquad (\text{IV.128})$$

to be the generators of the $S_3$ symmetry along $\mathfrak{B}'$.

Mathematically, we have provided a pivotal tensor functor

$$\phi: \ \mathsf{Vec}_{S_3} \to \mathsf{Vec}_{\mathbb{Z}_2} \qquad (\text{IV.129})$$

where $\mathsf{Vec}_{\mathbb{Z}_2}$ is the fusion category formed by topological lines living on the boundary $\mathfrak{B}'$. All the data of this functor simply descends from the non-trivial homomorphism $S_3 \to \mathbb{Z}_2$ that measures the (parity of the) number of transpositions involved in a permutation of 3 objects.

### D.  Rep($S_3$) Symmetry

We now consider minimal $(1+1)$d Rep($S_3$)-symmetric gapped boundary phases. In this case we have

$$\mathcal{S} = \mathsf{Rep}(S_3) = \{1, 1_-, E\}, \qquad (\text{IV.130})$$

where $1_-$ denotes the 1d sign irreducible representation of $S_3$ and $E$ denotes the 2d irreducible representation of $S_3$. The fusion rules are

$$1_- \otimes E = E \otimes 1_- = E, \qquad E^2 = 1 \oplus 1_- \oplus E. \ (\text{IV.131})$$

Recall that Rep($S_3$) can be obtained by gauging $S_3$ symmetry, which implies the two symmetries have the same SymTFT

$$\mathfrak{Z}(\mathsf{Rep}(S_3)) = \mathfrak{Z}(\mathsf{Vec}_{S_3}). \qquad (\text{IV.132})$$

The difference lies in the choice of the symmetry boundary $\mathfrak{B}^{\text{sym}}$, which for $\mathcal{S} = \mathsf{Rep}(S_3)$ we take to be specified by the Lagrangian algebra

$$\mathcal{L}^{\text{sym}}_{\mathsf{Rep}(S_3)} = 1 \oplus a_1 \oplus b_+. \qquad (\text{IV.133})$$

Since the SymTFTs are the same, the non-trivial condensable algebras remain the same $\mathcal{A}_E$, $\mathcal{A}_{1_-}$ and $\mathcal{A}_{a_1}$ we discussed above. Each of these choices defines a Rep($S_3$)-symmetric gapped boundary, analogously to the $\mathsf{Vec}_{S_3}$ case. However, the fact that the symmetry boundary is now associated to $\mathcal{L}^{\text{sym}}_{\mathsf{Rep}(S_3)}$ implies that the results are different to what we obtained before. We now discuss them in turn.

#### 1.  Condensable algebra $\mathcal{A}_E$

In this case, as we discussed earlier, we have that $\mathfrak{Z}'$ is the toric code, and a Lagrangian algebra in $\mathcal{Z}(S_3) \boxtimes \overline{\mathcal{Z}'}$ which completes $\mathcal{A}_E$ was provided in (IV.106). Here, we instead work with a different but equivalent Lagrangian algebra

$$\mathcal{L}_{\mathcal{I}} = 1 \oplus E \oplus 1_- \overline{m} \oplus E\overline{m} \oplus b_- \overline{f} \oplus b_+ \overline{e}. \qquad (\text{IV.134})$$

obtained from (IV.106) by applying the *em*-duality symmetry of the toric code [129].

The Lagrangian algebra associated to the underlying non-symmetric gapped boundary phase is

$$\mathcal{L}_{\mathcal{B}'} = 1 \oplus e = \mathcal{L}_e, \qquad (\text{IV.135})$$

which means that $\mathfrak{B}'$ is an irreducible topological boundary associated to electric Lagrangian algebra $\mathcal{L}_e$. The topological lines on $\mathfrak{B}'$ are as in (IV.125).

The club quiche is:

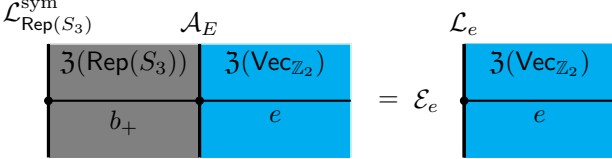

$$(\text{IV.136})$$

The Rep($S_3$) generators linking action descends from the corresponding action on the ends of the $\mathcal{Z}(\mathsf{Vec}_{S_3})$ anyons and is given by

$$1_-: \begin{array}{l} \mathcal{E}_e \to -\mathcal{E}_e, \\ \mathrm{id} \to \mathrm{id}, \end{array} \qquad E: \begin{array}{l} \mathcal{E}_e \to 0, \\ \mathrm{id} \to 2\,\mathrm{id}. \end{array} \qquad (\text{IV.137})$$

where id denotes the identity local operator along $\mathfrak{B}'$. See [2] for more details. Comparing with the action of lines living on $\mathfrak{B}'$, we find that the Rep($S_3$) symmetry is realized on $\mathfrak{B}'$ by lines

$$\phi(1_-) = P \ , \quad \phi(E) = 1 \oplus P. \qquad (\text{IV.138})$$

It is easy to verify that the Rep($S_3$) fusion rules are satisfied

$$\phi(1_-)\phi(E) = \phi(E)\phi(1_-) = P \otimes (1 \oplus P) = 1 \oplus P = \phi(E),$$
$$[\phi(E)]^2 = (1 \oplus P)^2 = 2(1 \oplus P) = \phi(1) \oplus \phi(1_-) \oplus \phi(E).$$
$$(\text{IV.139})$$

Mathematically, we have provided a pivotal tensor functor

$$\phi: \ \mathsf{Rep}(S_3) \to \mathsf{Vec}_{\mathbb{Z}_2} \qquad (\text{IV.140})$$

where $\mathsf{Vec}_{\mathbb{Z}_2}$ is the fusion category formed by topological lines living on the boundary $\mathfrak{B}'$.

#### 2.  Condensable algebra $\mathcal{A}_{1_-}$

In this case, as we discussed earlier, we have that $\mathfrak{Z}'$ is the $\mathbb{Z}_3$ DW theory, and a Lagrangian algebra completing $\mathcal{A}_{1_-}$ was provided in (IV.89). Here, we instead work with a different but equivalent Lagrangian algebra

$$\mathcal{L}_{\mathcal{I}} = 1 \oplus 1_- \oplus a_1 \overline{e} \oplus a_1 \overline{e}^2 \oplus a_{\omega^2} \overline{em} \oplus$$
$$\oplus a_{\omega^2} \overline{e}^2 \overline{m}^2 \oplus E\overline{m} \oplus E\overline{m}^2 \oplus a_\omega \overline{e}^2 \overline{m} \oplus a_\omega \overline{em}^2.$$
$$(\text{IV.141})$$

obtained from (IV.89) by applying the *em* exchange duality.

From this we see that the Lagrangian algebra associated to the underlying non-symmetric gapped boundary phase is

$$\mathcal{L}_{\mathfrak{B}'} = 1 \oplus e \oplus e^2 = \mathcal{L}_e \,, \tag{IV.142}$$

which means that $\mathfrak{B}'$ is an irreducible topological boundary associated to electric Lagrangian algebra $\mathcal{L}_e$. The topological lines on $\mathfrak{B}'$ are

$$1, \qquad P, \qquad P^2 \,, \tag{IV.143}$$

where $P$ is the generators of the $\mathbb{Z}_3$ symmetry localized along $\mathfrak{B}'$.

The operators coming from the club quiche are:

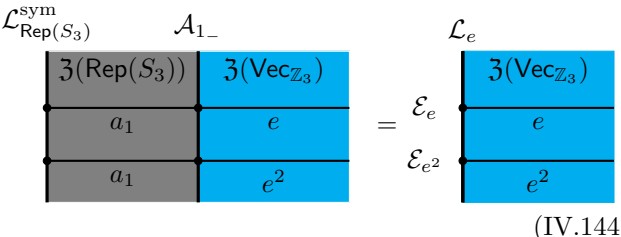

$$\text{(IV.144)}$$

The $\mathsf{Rep}(S_3)$ generators linking action is given by

$$
1_- : \begin{array}{c} \text{id} \to \text{id} \\ \mathcal{E}_e \to \mathcal{E}_e \\ \mathcal{E}_{e^2} \to \mathcal{E}_{e^2} \end{array} \qquad
E : \begin{array}{c} \text{id} \to 2\,\text{id} \\ \mathcal{E}_e \to -\mathcal{E}_e \\ \mathcal{E}_{e^2} \to -\mathcal{E}_{e^2} \,. \end{array} \tag{IV.145}
$$

which implies the lines implementing $\mathsf{Rep}(S_3)$ symmetry on $\mathfrak{B}'$ are

$$\phi(1_-) = 1 \quad, \quad \phi(E) = P \oplus P^2 \,. \tag{IV.146}$$

The reader can easily check that the $\mathsf{Rep}(S_3)$ fusions are satisfied by these line operators.

Mathematically, we have provided a pivotal tensor functor

$$\phi : \ \mathsf{Rep}(S_3) \to \mathsf{Vec}_{\mathbb{Z}_3} \,, \tag{IV.147}$$

where $\mathsf{Vec}_{\mathbb{Z}_3}$ is the fusion category formed by topological lines living on the boundary $\mathfrak{B}'$.

### 3. Condensable algebra $\mathcal{A}_{a_1}$

In this case, we know that $\mathfrak{Z}'$ is again the toric code. The Lagrangian algebra in $\mathcal{Z}(S_3) \boxtimes \overline{\mathcal{Z}}'$ which completes $\mathcal{A}_{a_1}$ is described in (IV.122).

From this we see that the Lagrangian algebra associated to the underlying non-symmetric gapped boundary is

$$\mathcal{L}_{\mathfrak{B}'} = (1 \oplus e) \oplus (1 \oplus m) = \mathcal{L}_e \oplus \mathcal{L}_m \,, \tag{IV.148}$$

i.e. $\mathfrak{B}'$ is a reducible boundary of the form

$$\mathfrak{B}' = \mathfrak{B}_e \oplus \mathfrak{B}_m \,. \tag{IV.149}$$

where $\mathfrak{B}_e$ is an irreducible topological boundary with associated Lagrangian algebra $\mathcal{L}_e = 1 \oplus e$ and $\mathfrak{B}_m$ is an irreducible topological boundary with associated Lagrangian algebra $\mathcal{L}_m = 1 \oplus m$.

The topological line operators on $\mathfrak{B}'$ are

$$1_{ii}, \qquad P_{ii}, \qquad S_{ij} \,, \tag{IV.150}$$

where $i, j \in \{e, m\}$ and $i \neq j$. See the end of appendix A for more details. The line $1_{ii}$ is the identity line on boundary $\mathfrak{B}_i$, the line $P_{ii}$ is the generator of the $\mathbb{Z}_2$ symmetry localized along the boundary $\mathfrak{B}_i$, and the line $S_{ij}$ changes the boundary $\mathfrak{B}_i$ to the boundary $\mathfrak{B}_j$, with fusion rules

$$
\begin{aligned}
P_{ii} \otimes S_{ij} &= S_{ij} \otimes P_{jj} = S_{ij} \\
S_{ij} \otimes S_{ji} &= 1_{ii} \oplus P_{ii} \,.
\end{aligned} \tag{IV.151}
$$

Note the non-invertibility of the boundary changing line operators $S_{ij}$. Note also that the general linking actions of boundary changing lines are

$$
\begin{aligned}
S_{em} : \ v_e \to \sqrt{2} e^{-\lambda} v_m, \qquad \mathcal{E}_e \to 0 \\
S_{me} : \ v_m \to \sqrt{2} e^{\lambda} v_e, \qquad \mathcal{E}_m \to 0
\end{aligned} \tag{IV.152}
$$

for some $\lambda \in \mathbb{R}$ which captures the relative Euler term between the two boundaries $\mathfrak{B}_e$ and $\mathfrak{B}_m$.

The operators coming from the club quiche are

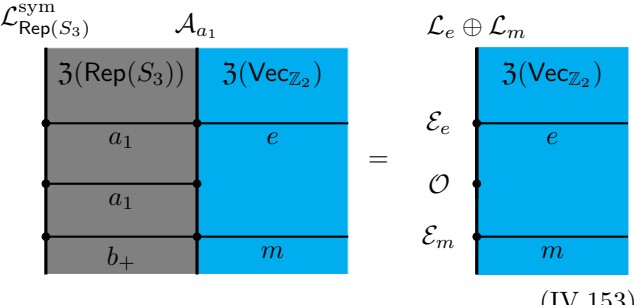

$$\text{(IV.153)}$$

The products of these operators are fixed to be

$$
\begin{aligned}
\mathcal{O}^2 = 2 - \mathcal{O}, \qquad \mathcal{O}\mathcal{E}_e = \mathcal{E}_e, \qquad \mathcal{E}_e^2 = (2 + \mathcal{O})/3 \,, \\
\mathcal{E}_e\mathcal{E}_m = 0, \qquad \mathcal{O}\mathcal{E}_m = -2\mathcal{E}_m, \qquad \mathcal{E}_m^2 = (1 - \mathcal{O})/3 \,.
\end{aligned} \tag{IV.154}
$$

The computation is identical to the one described in section 5.3.4 of [2] where the $\mathsf{Rep}(S_3)$ SSB phase is discussed. The identity operators along the two boundaries $\mathfrak{B}_e$ and $\mathfrak{B}_m$ can be identified respectively as

$$v_e = \frac{1}{3}(2 + \mathcal{O}), \qquad v_m = \frac{1}{3}(1 - \mathcal{O}) \tag{IV.155}$$

The linking actions of $\mathsf{Rep}(S_3)$ symmetry generators are

$$
\begin{aligned}
1_- : \ & 1 \to 1, \qquad \mathcal{O} \to \mathcal{O}, \qquad \mathcal{E}_e \to \mathcal{E}_e, \qquad \mathcal{E}_m \to -\mathcal{E}_m, \\
E : \ & 1 \to 2, \qquad \mathcal{O} \to -\mathcal{O}, \qquad \mathcal{E}_e \to -\mathcal{E}_e, \qquad \mathcal{E}_m \to 0 \,.
\end{aligned} \tag{IV.156}
$$

where $1 = v_e + v_m$ denotes the identity operator along the reducible boundary $\mathfrak{B}'$. This implies the following linking actions

$$1_- : \ v_e \to v_e, \qquad v_m \to v_m$$
$$E : \ v_e \to 2v_m + v_e, \qquad v_m \to v_e. \qquad \text{(IV.157)}$$

Therefore the lines realizing the $\mathsf{Rep}(S_3)$ symmetry along $\mathfrak{B}'$ are

$$\phi(1_-) = 1_{ee} \oplus P_{mm}, \quad \phi(E) = S_{em} \oplus S_{me} \oplus P_{ee} \quad \text{(IV.158)}$$

with the relative Euler term between $\mathfrak{B}_e$ and $\mathfrak{B}_m$ fixed such that the linking actions of boundary changing lines are

$$S_{em} : \ v_e \to 2v_m, \qquad S_{me} : \ v_m \to v_e. \qquad \text{(IV.159)}$$

The reader can check that these lines satisfy the $\mathsf{Rep}(S_3)$ fusion rules.

Mathematically, we have provided information about a pivotal tensor functor

$$\phi : \ \mathsf{Rep}(S_3) \to \mathsf{Ising}_{2\times 2}^{\sqrt{2}}, \qquad \text{(IV.160)}$$

where $\mathsf{Ising}_{2\times 2}^{e^{-\lambda}}$ is what we call the pivotal multi-fusion category formed by topological lines (IV.150) with the quantum dimensions of $S_{ij}$ lines given by their linking actions on $v_i$ described in (IV.152).

### E.   $\mathsf{Ising}$ Symmetry

We now study $(1+1)$d $\mathsf{Ising}$-symmetric minimal gapped boundary phases. This is the smallest Tambara-Yamagami symmetry category $\mathsf{TY}(\mathbb{Z}_2)$

$$\mathsf{Ising} = \mathsf{TY}(\mathbb{Z}_2) = \{1, P, S\} \qquad \text{(IV.161)}$$

with fusion rules

$$P^2 = 1, \ P \otimes S = S \otimes P = S, \ S^2 = 1 \oplus P. \quad \text{(IV.162)}$$

The SymTFT $\mathfrak{Z}(\mathsf{Ising})$ is the doubled Ising theory, whose bulk anyons are

$$\mathcal{Z}(\mathsf{Ising}) = \mathsf{Ising} \boxtimes \overline{\mathsf{Ising}} = \{1, P, \overline{P}, P\overline{P}, S, \overline{S}, P\overline{S}, S\overline{P}, S\overline{S}\}. \qquad \text{(IV.163)}$$

$\mathfrak{Z}(\mathsf{Ising})$ admits only one topological boundary condition corresponding to the Lagrangian algebra

$$\mathcal{L}_{\mathsf{Ising}}^{\text{sym}} = 1 \oplus P\overline{P} \oplus S\overline{S}. \qquad \text{(IV.164)}$$

Solving the conditions for a condensable algebra, we find that there exist only one non-trivial non-Lagrangian condensable algebra given by

$$\mathcal{A} = 1 \oplus P\overline{P}. \qquad \text{(IV.165)}$$

The bulk phase $\mathfrak{Z}'$ is the underlined toric code with the Lagrangian algebra $\mathcal{L}_\mathcal{I} \in \mathcal{Z}(\mathsf{Ising}) \boxtimes \overline{\mathcal{Z}(\mathsf{Vec}_{\mathbb{Z}_2})}$ completing $\mathcal{A}$ being

$$\mathcal{L}_\mathcal{I} = 1 \oplus P\overline{P} \oplus P\overline{f} \oplus \overline{P}f \oplus S\overline{S}\,\overline{e} \oplus S\overline{S}\,\overline{m}. \quad \text{(IV.166)}$$

From this, we compute the Lagrangian algebra associated to the underlying non-symmetric gapped boundary is

$$\mathcal{L}_{\mathfrak{B}'} = (1 \oplus e) \oplus (1 \oplus m) = \mathcal{L}_e \oplus \mathcal{L}_m, \qquad \text{(IV.167)}$$

i.e. $\mathfrak{B}'$ is a reducible boundary of the form

$$\mathfrak{B}' = \mathfrak{B}_e \oplus \mathfrak{B}_m, \qquad \text{(IV.168)}$$

which is the same as in (IV.149). The topological line operators on $\mathfrak{B}'$ are discussed in (IV.150), with fusion (IV.151), and linking action (IV.152) for some $\lambda \in \mathbb{R}$ capturing the relative Euler term between $\mathfrak{B}_e$ and $\mathfrak{B}_m$, which will turn out to be different in this case.

The operators we obtain from the club quiche are

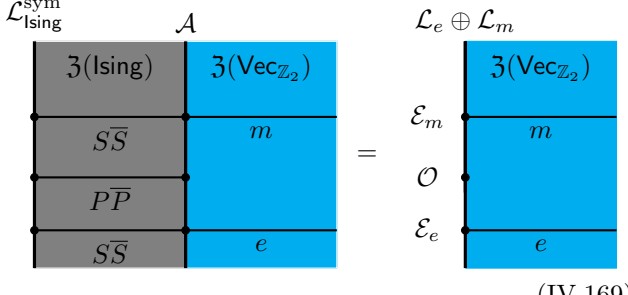

$$\text{(IV.169)}$$

The products of these operators are determined to be

$$\mathcal{O}^2 = 1, \quad \mathcal{O}\mathcal{E}_e = \mathcal{E}_e, \quad \mathcal{E}_e^2 = (1 + \mathcal{O})/2,$$
$$\mathcal{E}_e\mathcal{E}_m = 0, \quad \mathcal{O}\mathcal{E}_m = -\mathcal{E}_m, \quad \mathcal{E}_m^2 = (1 - \mathcal{O})/2. \qquad \text{(IV.170)}$$

The identity operators along the two boundaries $\mathfrak{B}_e$ and $\mathfrak{B}_m$ can be identified respectively as

$$v_e = \frac{1}{2}(1 + \mathcal{O}), \qquad v_m = \frac{1}{2}(1 - \mathcal{O}). \qquad \text{(IV.171)}$$

The linking actions of the $\mathsf{Ising}$ symmetry generators are given by

$$P : \ 1 \to 1, \quad \mathcal{O} \to \mathcal{O}, \quad \mathcal{E}_e \to -\mathcal{E}_e, \quad \mathcal{E}_m \to -\mathcal{E}_m,$$
$$S : \ 1 \to \sqrt{2}, \quad \mathcal{O} \to -\sqrt{2}\mathcal{O}, \quad \mathcal{E}_e \to 0, \quad \mathcal{E}_m \to 0. \qquad \text{(IV.172)}$$

This implies the following linking actions

$$P : \ v_e \to v_e, \qquad v_m \to v_m$$
$$S : \ v_e \to \sqrt{2}v_m, \qquad v_m \to \sqrt{2}v_e. \qquad \text{(IV.173)}$$

We can therefore identity the line operators on $\mathfrak{B}'$ realizing the $\mathsf{Ising}$ symmetry as

$$\phi(P) = P_{ee} \oplus P_{mm}$$
$$\phi(S) = S_{em} \oplus S_{me}. \qquad \text{(IV.174)}$$

with the relative Euler term between $\mathfrak{B}_e$ and $\mathfrak{B}_m$ fixed such that the linking actions of boundary changing lines are

$$S_{em} : \ v_e \to \sqrt{2}v_m, \qquad S_{me} : \ v_m \to \sqrt{2}v_e. \quad \text{(IV.175)}$$

The reader can check that these lines satisfy the Ising fusion rules.

Mathematically, we have provided information about a pivotal tensor functor

$$\phi : \; \mathsf{Ising} \to \mathsf{Ising}^1_{2\times 2} \,, \qquad (\text{IV.176})$$

where $\mathsf{Ising}^{e^{-\lambda}=1}_{2\times 2}$ is the pivotal multi-fusion category formed by topological lines living on the underlying boundary $\mathfrak{B}'$.

### F.  $\mathsf{TY}(\mathbb{Z}_N)$ Symmetry

The Tambara-Yamagami $\mathbb{Z}_N$ categories are a generalization of the Ising symmetry, denoted by

$$\mathcal{S} = \mathsf{TY}(\mathbb{Z}_N) \,. \qquad (\text{IV.177})$$

The simple lines are $A^i$ and $S$ with

$$A^N \cong 1 \,, \quad A \otimes S \cong S \otimes A \cong S \,, \quad S \otimes S \cong \bigoplus_{i=1}^{N} A^i \,. \tag{IV.178}$$

The SymTFT $\mathfrak{Z}(\mathsf{TY}(Z_N))$ is constructed by gauging a $\mathbb{Z}_2$ outer automorphism of the 3d DW theory for $\mathbb{Z}_N$ with anyons $L_{e,m}$ such that $e,m \in \{0,\cdots,N-1\}$. The topological lines of the gauged theory, i.e. of the SymTFT of $\mathsf{TY}(\mathbb{Z}_N)$, are

$$L_e^{\pm} \,, \qquad L_{e,m} \,, \qquad \Sigma_e^{\pm} \,. \qquad (\text{IV.179})$$

such that $e > m$ for $L_{e,m}$. For more details on the fusions and S-matrices see [2, 25]. Let us focus on $N = 4$ for concreteness. The Lagrangian algebras were determined in [2] to be

$$\begin{aligned}
\mathcal{L}_1 &= L_0^+ + L_0^- + L_{1,0} + L_{2,0} + L_{3,0} \\
\mathcal{L}_2 &= L_0^+ + L_0^- + L_2^+ + L_2^- + 2L_{2,0} \\
\mathcal{L}_3 &= L_0^+ + L_2^- + L_{2,0} + \Sigma_1^+ + \Sigma_3^- \,.
\end{aligned} \qquad (\text{IV.180})$$

The symmetry boundary is provided by

$$\mathcal{L}^{\text{sym}}_{\mathsf{TY}(\mathbb{Z}_4)} = \mathcal{L}_1 \,. \qquad (\text{IV.181})$$

We can solve the necessary condition for condensable algebras in appendix B and find

| dim | $\mathcal{A}$ | Algebra |
|---|---|---|
| 4 | $\mathcal{A}_{1,2}$ | $L_0^+ + L_0^- + L_{2,0}$ |
| 4 | $\mathcal{A}_{2,3}$ | $L_0^+ + L_2^- + L_{2,0}$ |
| 4 | $\mathcal{A}_2$ | $L_0^+ + L_0^- + L_2^+ + L_2^-$ |
| 4 | $\mathcal{A}_2^{(2)}$ | $L_0^+ + L_2^+ + L_{2,0}$ |
| 2 | $\mathcal{A}'_{1,2}$ | $L_0^+ + L_0^-$ |
| 2 | $\mathcal{A}_2^{(3)}$ | $L_0^+ + L_2^+$ |
| 2 | $\mathcal{A}'_{2,3}$ | $L_0^+ + L_2^-$ |
| 1 | $\mathcal{A}_{1,2,3}$ | $L_0^+$ |

$$(\text{IV.182})$$

We indicate with the subscripts the Lagrangian algebras in which these algebras are contained in and dim is the dimension of the algebra (which are all non-maximal, $\dim < 8$).

Not all of these correspond to condensable algebras, as we will see. The conditions we solve are only necessary, not sufficient. In order to determine the condensable algebras among these, we check whether there is a consistent reduced topological order. This shows that $\mathcal{A}_2^{(2)}$ for instance is not consistent, while $\mathcal{A}_2$, $\mathcal{A}_{2,3}$ and $\mathcal{A}_{1,2}$ are.

#### 1.  Condensable Algebra $\mathcal{A}_{1,2}$

Consider the interface characterized by the algebra $\mathcal{A}_{1,2} = L_0^+ + L_0^- + L_{2,0}$. First we need to determine the reduced topological order $\mathfrak{Z}(\mathcal{S}')$. Given the dimension of $\mathfrak{Z}(\mathcal{S})$ equals 8 and the algebra is of dimension 4, the reduced topological order has dimension 2, and so it is either the toric code or the double-semion. We now show that it is the former

$$\mathfrak{Z}(\mathcal{S}') = \mathfrak{Z}(\mathsf{Vec}_{\mathbb{Z}_2}) \,. \qquad (\text{IV.183})$$

The Lagrangian algebra of $\mathcal{Z}(\mathsf{TY}(\mathbb{Z}_4)) \boxtimes \overline{\mathcal{Z}(\mathsf{Vec}_{\mathbb{Z}_2})}$ that completes the algebra $\mathcal{A}_{1,2}$ is

$$\begin{aligned}
\mathcal{L}_{1,2} =& L_0^+ + L_0^- + L_{2,0} + L_2^+ \overline{e} + L_2^- \overline{e} + L_{2,0}\overline{e} \\
& + L_{1,0}\overline{m} + L_{3,0}\overline{m} + L_{2,1}\overline{f} + L_{3,2}\overline{f} \,.
\end{aligned} \qquad (\text{IV.184})$$

From this, one finds the Lagrangian algebra associated to the underlying non-symmetric gapped boundary

$$\mathcal{L}_{\mathfrak{B}'} = \mathcal{L}_e \oplus 2\mathcal{L}_m \,, \qquad (\text{IV.185})$$

which implies that the underlying boundary of the $\mathbb{Z}_2$ DW theory is reducible and given by

$$\mathfrak{B}' = \mathfrak{B}_0^e \oplus \mathfrak{B}_1^m \oplus \mathfrak{B}_2^m \,, \qquad (\text{IV.186})$$

where $\mathfrak{B}$'s are irreducible topological boundaries with $\mathfrak{B}_e^0$ corresponding to the Lagrangian algebra $\mathcal{L}_e = 1 \oplus e$ and $\mathfrak{B}_m^1$ and $\mathfrak{B}_m^2$ corresponding to the Lagrangian algebra $\mathcal{L}_e = 1 \oplus m$. The boundary changing lines from $\mathfrak{B}_i^m$ to $\mathfrak{B}_j^m$ for $i \neq j$ are the invertible lines $1_{ij}, P_{ij}$, while the boundary changing lines between $\mathfrak{B}_i^m$ and $\mathfrak{B}_j^e$ are the non-invertible lines $S_{ij}, S_{ji}$. The relative Euler terms between these boundaries are fixed later.

The club quiche and the resulting operators can be

depicted as follows:

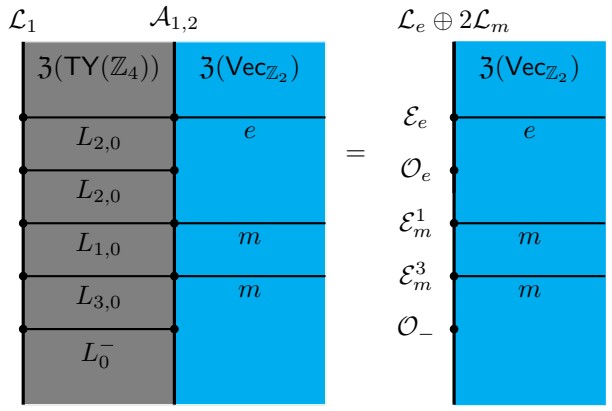

(IV.187)

The algebra of these operators can be derived to be

$$
\begin{aligned}
&\mathcal{O}_-\mathcal{O}_- = 1, && \mathcal{O}_e\mathcal{O}_e = (1+\mathcal{O}_-), \\
&\mathcal{O}_-\mathcal{E}_e = -\mathcal{E}_e, && \mathcal{E}_e\mathcal{E}_e = (1-\mathcal{O}_-), \\
&\mathcal{O}_-\mathcal{O}_e = \mathcal{O}_e, \\
&\mathcal{O}_-\mathcal{E}_m^1 = \mathcal{E}_m^1, && \mathcal{E}_e\mathcal{E}_m^1 = 0, \\
&\mathcal{O}_-\mathcal{E}_m^3 = \mathcal{E}_m^3, && \mathcal{E}_e\mathcal{E}_m^3 = 0, \\
&\mathcal{O}_e\mathcal{E}_e = 0, && \mathcal{E}_m^1\mathcal{E}_m^1 = \sqrt{2}\mathcal{O}_e, \\
&\mathcal{O}_e\mathcal{E}_m^1 = \sqrt{2}\mathcal{E}_m^3, && \mathcal{E}_m^3\mathcal{E}_m^3 = \sqrt{2}\mathcal{O}_e, \\
&\mathcal{O}_e\mathcal{E}_m^3 = \sqrt{2}\mathcal{E}_m^1, && \mathcal{E}_m^1\mathcal{E}_m^3 = (1+\mathcal{O}_-).
\end{aligned}
\tag{IV.188}
$$

Subsequently, it is again useful to define the combinations

$$
\begin{aligned}
v_0 &= \frac{1}{2}(1-\mathcal{O}_-), && \tilde{\mathcal{E}}_0 = \frac{1}{\sqrt{2}}\mathcal{E}_e, \\
v_1 &= \frac{1}{4}(1+\mathcal{O}_- + \sqrt{2}\mathcal{O}_e), && \tilde{\mathcal{E}}_1 = \frac{1}{2\sqrt{2}}(\mathcal{E}_m^1 + \mathcal{E}_m^3), \\
v_2 &= \frac{1}{4}(1+\mathcal{O}_- - \sqrt{2}\mathcal{O}_e), && \tilde{\mathcal{E}}_2 = \frac{i}{2\sqrt{2}}(\mathcal{E}_m^1 - \mathcal{E}_m^3),
\end{aligned}
\tag{IV.189}
$$

where one can also notice that

$$
\tilde{\mathcal{E}}_0 = \frac{\mathcal{E}_e v_0}{\sqrt{2}}, \quad \tilde{\mathcal{E}}_1 = \frac{\mathcal{E}_m^1 v_1}{\sqrt{2}}, \quad \tilde{\mathcal{E}}_2 = -\frac{i\mathcal{E}_m^3 v_2}{\sqrt{2}}. \tag{IV.190}
$$

Here $v_0$ is the identity local operator along the irreducible boundary $\mathfrak{B}_0^e$ while $v_1$ and $v_2$ are the identity local operators along the irreducible boundaries $\mathfrak{B}_1^m$ and $\mathfrak{B}_2^m$ respectively. Furthermore, $\tilde{\mathcal{E}}_0$ is the end of $e$ along $\mathfrak{B}_0^e$ and $\tilde{\mathcal{E}}_1$ and $\tilde{\mathcal{E}}_2$ are the ends of $m$ along $\mathfrak{B}_1^m$ and $\mathfrak{B}_2^m$ respectively. This follows from the products

$$
v_i v_j = \delta_{ij} v_j, \quad \tilde{\mathcal{E}}_i v_j = \delta_{ij}\tilde{\mathcal{E}}_j, \quad \tilde{\mathcal{E}}_i\tilde{\mathcal{E}}_j = \delta_{ij} v_j, \tag{IV.191}
$$

for $i,j \in \{0,1,2\}$.

The linking actions of the $\mathsf{TY}(\mathbb{Z}_4)$ symmetry generators are given by

$$
\begin{aligned}
A^k: \quad & 1 \to 1, \quad \mathcal{O}_- \to \mathcal{O}_-, \quad \mathcal{O}_e \to (-1)^k\mathcal{O}_e, \\
& \mathcal{E}_e \to (-1)^k\mathcal{E}_e, \quad \mathcal{E}_m^1 \to i^k\mathcal{E}_m^1, \quad \mathcal{E}_m^3 \to (-i)^k\mathcal{E}_m^3, \\
S: \quad & 1 \to 2, \quad \mathcal{O}_- \to -2\mathcal{O}_-, \quad \mathcal{O}_e, \mathcal{E}_e, \mathcal{E}_m^1, \mathcal{E}_m^3 \to 0,
\end{aligned}
\tag{IV.192}
$$

which is also consistent with the fusion rules (IV.188). The symmetry action also implies the linking actions

$$
\begin{aligned}
A: \quad & v_0 \to v_0, \quad v_1 \leftrightarrow v_2 \\
& \tilde{\mathcal{E}}_0 \to -\tilde{\mathcal{E}}_0, \quad \tilde{\mathcal{E}}_1 \to \tilde{\mathcal{E}}_2, \quad \tilde{\mathcal{E}}_2 \to -\tilde{\mathcal{E}}_1, \\
S: \quad & v_0 \to 2(v_1+v_2), \quad v_1, v_2 \to v_0 \\
& \tilde{\mathcal{E}}_0, \tilde{\mathcal{E}}_1, \tilde{\mathcal{E}}_2 \to 0.
\end{aligned}
\tag{IV.193}
$$

We can therefore identify the line operators on $\mathfrak{B}'$ realizing the $\mathsf{TY}(\mathbb{Z}_4)$ symmetry as

$$
\begin{aligned}
\phi(A) &= P_{00} \oplus 1_{12} \oplus P_{21} \\
\phi(S) &= S_{01} \oplus S_{02} \oplus S_{10} \oplus S_{20},
\end{aligned}
\tag{IV.194}
$$

where

$$
S_{ij} \otimes S_{jk} = 1_{ik} \oplus P_{ik}. \tag{IV.195}
$$

One can then easily check that indeed

$$
\phi(S) \otimes \phi(S) = \bigoplus_{i=0}^{3} \phi(A^i). \tag{IV.196}
$$

The relative Euler terms are fixed as follows. Since the linking action of $A$ on $v_1$ produces $v_2$, the linking action of $1_{12}$ on $v_1$ must be

$$
1_{12}: \quad v_1 \to v_2, \tag{IV.197}
$$

which means that there are no relative Euler terms between the magnetic boundaries $\mathfrak{B}_1^m$ and $\mathfrak{B}_2^m$. On the other hand, to recover the linking action of $S$ on $v_0$, we must have the following linking action of $S_{01}$

$$
S_{01}: \quad v_0 \to 2v_1, \tag{IV.198}
$$

which fixes the relative Euler term between electric and magnetic boundaries $\mathfrak{B}_0^e$ and $\mathfrak{B}_1^m$.

Mathematically, this provides us with necessary information to describe a pivotal tensor functor

$$
\phi: \mathsf{TY}(\mathbb{Z}_4) \to \mathcal{S}(\mathfrak{B}'), \tag{IV.199}
$$

where $\mathcal{S}(\mathfrak{B}')$ is the pivotal multi-fusion category formed by the line operators living on $\mathfrak{B}'$ also accounting for the above determined relative Euler terms.

### 2. Condensable Algebra $\mathcal{A}_{2,3}$

For the algebra $\mathcal{A}_{2,3}$ we find again that the reduced topological order is

$$
\mathfrak{Z}(\mathcal{S}') = \mathfrak{Z}(\mathsf{Vec}_{\mathbb{Z}_2}). \tag{IV.200}
$$

The Lagrangian algebra of $\mathcal{Z}(\mathsf{TY}(\mathbb{Z}_4)) \boxtimes \overline{\mathcal{Z}(\mathsf{Vec}_{\mathbb{Z}_2})}$ that completes $\mathcal{A}_{2,3}$ in this case is

$$
\begin{aligned}
\mathcal{L}_{2,3} =& L_0^+ + L_2^- + L_{2,0} + L_{2,0}\bar{e} + L_2^+\bar{e} + L_0^-\bar{e} \\
& + \Sigma_1^+\overline{m} + \Sigma_1^-\overline{f} + \Sigma_3^+\overline{f} + \Sigma_3^-\,\overline{m}.
\end{aligned}
\tag{IV.201}
$$

From this, we see that the Lagrangian algebra associated to the underlying non-symmetric gapped boundary is

$$\mathcal{L}_{\mathfrak{B}'} = 2(1 \oplus e) = 2\mathcal{L}_e \qquad \text{(IV.202)}$$

which corresponds to a reducible topological boundary condition of the toric code

$$\mathfrak{B}' = \mathfrak{B}_0^e \oplus \mathfrak{B}_1^e. \qquad \text{(IV.203)}$$

The topological line operators on $\mathfrak{B}'$ are the same ones discussed in (IV.109), with fusions given by (IV.110) and (IV.111).

The operators we obtain from the club quiche are:

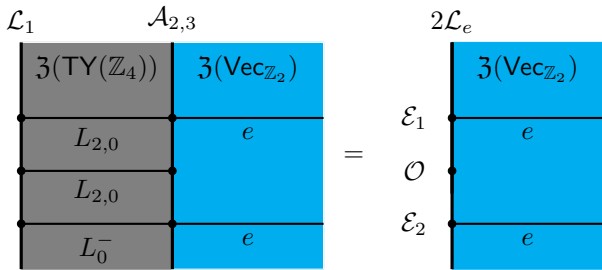

$$\text{(IV.204)}$$

The products of these operators are determined to be

$$\mathcal{O}^2 = 1, \quad \mathcal{O}\mathcal{E}_1 = \mathcal{E}_2, \quad \mathcal{O}\mathcal{E}_2 = \mathcal{E}_1$$
$$\mathcal{E}_1^2 = 1 \quad \mathcal{E}_2^2 = 1 \quad \mathcal{E}_1\mathcal{E}_2 = \mathcal{O}. \qquad \text{(IV.205)}$$

We can now define the combinations

$$v_0 = (1 + \mathcal{O})/2, \quad v_1 = (1 - \mathcal{O})/2,$$
$$\tilde{\mathcal{E}}_0 = (\mathcal{E}_1 + \mathcal{E}_2)/2, \quad \tilde{\mathcal{E}}_1 = (\mathcal{E}_1 - \mathcal{E}_2)/2, \qquad \text{(IV.206)}$$

where one can also notice that

$$\tilde{\mathcal{E}}_0 = \mathcal{E}_1 v_0 = \mathcal{E}_2 v_0, \qquad \tilde{\mathcal{E}}_1 = \mathcal{E}_1 v_1 = -\mathcal{E}_2 v_1. \qquad \text{(IV.207)}$$

Here $v_0$ and $v_1$ are the identity local operators along the two irreducible boundaries $\mathfrak{B}_0^e$ and $\mathfrak{B}_1^e$ respectively, while $\tilde{\mathcal{E}}_0$ and $\tilde{\mathcal{E}}_1$ are the end of $e$ along the respective boundaries. This follows from the products

$$v_i v_j = \delta_{ij} v_j, \quad \tilde{\mathcal{E}}_0^2 = v_0, \quad \tilde{\mathcal{E}}_1^2 = v_1, \quad \tilde{\mathcal{E}}_0 \tilde{\mathcal{E}}_1 = 0,$$
$$\tilde{\mathcal{E}}_0 v_0 = \tilde{\mathcal{E}}_0, \quad \tilde{\mathcal{E}}_0 v_1 = 0, \quad \tilde{\mathcal{E}}_1 v_1 = \tilde{\mathcal{E}}_1, \quad \tilde{\mathcal{E}}_1 v_0 = 0. \qquad \text{(IV.208)}$$

The linking actions of the $\mathsf{TY}(\mathbb{Z}_4)$ symmetry generators are given by

$$A : 1 \to 1 \quad \mathcal{O} \to -\mathcal{O} \quad \mathcal{E}_1 \to -\mathcal{E}_1 \quad \mathcal{E}_2 \to \mathcal{E}_2$$
$$S : 1 \to 2 \quad \mathcal{O} \to 0 \quad \mathcal{E}_1 \to 0 \quad \mathcal{E}_2 \to -2\mathcal{E}_2, \qquad \text{(IV.209)}$$

which implies the following linking actions

$$A : v_0 \leftrightarrow v_1, \quad \tilde{\mathcal{E}}_0 \to -\tilde{\mathcal{E}}_1, \quad \tilde{\mathcal{E}}_1 \to -\tilde{\mathcal{E}}_0$$
$$S : v_0, v_1 \to 1 = v_0 + v_1, \quad \tilde{\mathcal{E}}_0 \to \tilde{\mathcal{E}}_1 - \tilde{\mathcal{E}}_0, \quad \tilde{\mathcal{E}}_1 \to \tilde{\mathcal{E}}_0 - \tilde{\mathcal{E}}_1. \qquad \text{(IV.210)}$$

We can therefore identify the line operators on $\mathfrak{B}'$ realizing the $\mathsf{TY}(\mathbb{Z}_4)$ symmetry as

$$\phi(A) = P_{01} \oplus P_{10}$$
$$\phi(S) = 1_{01} \oplus P_{00} \oplus 1_{10} \oplus P_{11}. \qquad \text{(IV.211)}$$

One can easily check that these satisfy the $\mathsf{TY}(\mathbb{Z}_4)$ fusion rules, e.g.

$$[\phi(S)]^2 = 2\,1_{00} \oplus 2\,1_{11} \oplus 2\,P_{01} \oplus 2\,P_{10} = \bigoplus_{i=0}^{3} \phi(A^i). \qquad \text{(IV.212)}$$

From the linking actions one finds that there is no relative Euler term between $\mathfrak{B}_0^e$ and $\mathfrak{B}_1^e$.

Mathematically, this provides information about a pivotal tensor functor

$$\phi : \mathsf{TY}(\mathbb{Z}_4) \to \mathrm{Mat}_2(\mathsf{Vec}_{\mathbb{Z}_2}) \qquad \text{(IV.213)}$$

where $\mathrm{Mat}_2(\mathsf{Vec}_{\mathbb{Z}_2})$ is the multi-fusion category formed by the topological line operators living on $\mathfrak{B}'$.

### 3. Condensable algebra $\mathcal{A}_2$

In this case, the bulk phase $\mathfrak{Z}'$ is the double semion, with the Lagrangian algebra $\mathcal{L}_\mathcal{I} \in \mathcal{Z}(\mathsf{TY}(\mathbb{Z}_4)) \boxtimes \overline{\mathcal{Z}(\mathsf{Vec}_{\mathbb{Z}_2}^\omega)}$ given by

$$\mathcal{L}_\mathcal{I} = L_0^+ \oplus L_0^- \oplus L_2^+ \oplus L_2^- \oplus L_1^+ s \oplus L_1^- s$$
$$\oplus L_3^+ s \oplus L_3^- s \oplus 2L_{3,1}\bar{s} \oplus 2L_{2,0} s\bar{s}. \qquad \text{(IV.214)}$$

From this, we see that the Lagrangian algebra associated to the underlying non-symmetric gapped boundary is

$$\mathcal{L}_{\mathfrak{B}'} = 2(1 \oplus s\bar{s}) = 2\mathcal{L}_{s\bar{s}}, \qquad \text{(IV.215)}$$

which corresponds to a reducible topological boundary condition of the double semion

$$\mathfrak{B}' = \mathfrak{B}_0^{s\bar{s}} \oplus \mathfrak{B}_1^{s\bar{s}}. \qquad \text{(IV.216)}$$

The topological line operators on $\mathfrak{B}'$ are the same ones discussed in (IV.109), with fusions given by (IV.110) and (IV.111).

The operators we obtain from the club quiche are:

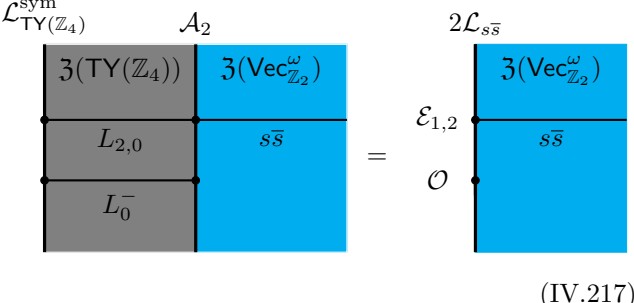

$$\text{(IV.217)}$$

The products of these operators are determined to be

$$\mathcal{O}^2 = 1, \quad \mathcal{O}\mathcal{E}_1 = \mathcal{E}_1, \quad \mathcal{O}\mathcal{E}_2 = -\mathcal{E}_2$$
$$\mathcal{E}_1^2 = (1+\mathcal{O})/2, \quad \mathcal{E}_2^2 = (1-\mathcal{O})/2, \quad \mathcal{E}_1\mathcal{E}_2 = 0\,.$$
$$\text{(IV.218)}$$

We can now define the combinations

$$v_0 = (1+\mathcal{O})/2, \quad v_1 = (1-\mathcal{O})/2$$
$$\tilde{\mathcal{E}}_0 = \mathcal{E}_1, \qquad \tilde{\mathcal{E}}_1 = \mathcal{E}_2\,.$$
$$\text{(IV.219)}$$

Here $v_0$ and $v_1$ are the identity local operators along the two irreducible boundaries $\mathfrak{B}_0^{s\bar{s}}$ and $\mathfrak{B}_1^{s\bar{s}}$ respectively, while $\tilde{\mathcal{E}}_0$ and $\tilde{\mathcal{E}}_1$ are the end of $s\bar{s}$ along the respective boundaries. This follows from the products

$$v_i v_j = \delta_{ij} v_j \quad \tilde{\mathcal{E}}_0^2 = v_0 \quad \tilde{\mathcal{E}}_1^2 = v_1 \quad \tilde{\mathcal{E}}_0\tilde{\mathcal{E}}_1 = 0$$
$$\tilde{\mathcal{E}}_0 v_0 = \tilde{\mathcal{E}}_0 \quad \tilde{\mathcal{E}}_0 v_1 = 0 \quad \tilde{\mathcal{E}}_1 v_1 = \tilde{\mathcal{E}}_1 \quad \tilde{\mathcal{E}}_1 v_0 = 0\,.$$
$$\text{(IV.220)}$$

The linking actions of the $\mathsf{TY}(\mathbb{Z}_4)$ symmetry generators are given by

$$A: 1 \to 1 \quad \mathcal{O} \to \mathcal{O} \quad \mathcal{E}_1 \to -\mathcal{E}_1 \quad \mathcal{E}_2 \to -\mathcal{E}_2$$
$$S: 1 \to 2 \quad \mathcal{O} \to -2\mathcal{O} \quad \mathcal{E}_1 \to 0 \quad \mathcal{E}_2 \to 0\,,$$
$$\text{(IV.221)}$$

which implies the following linking actions

$$A: v_0 \to v_0 \quad v_1 \to v_1 \quad \tilde{\mathcal{E}}_0 \to -\tilde{\mathcal{E}}_0 \quad \tilde{\mathcal{E}}_1 \to -\tilde{\mathcal{E}}_1$$
$$S: v_0 \to 2v_1 \quad v_1 \to 2v_0 \quad \tilde{\mathcal{E}}_0 \to 0 \quad \tilde{\mathcal{E}}_1 \to 0\,.$$
$$\text{(IV.222)}$$

We can therefore identify the line operators on $\mathfrak{B}'$ realizing the $\mathsf{TY}(\mathbb{Z}_4)$ symmetry as

$$\phi(A) = P_{00} \oplus P_{11}$$
$$\phi(S) = 1_{01} \oplus P_{01} \oplus 1_{10} \oplus P_{10}\,.$$
$$\text{(IV.223)}$$

One can easily check that these satisfy the $\mathsf{TY}(\mathbb{Z}_4)$ fusion rules, e.g.

$$[\phi(S)]^2 = 2\,1_{00} \oplus 2\,1_{11} \oplus 2\,P_{00} \oplus 2\,P_{11} = \bigoplus_{i=0}^{3} \phi(A^i)\,.$$
$$\text{(IV.224)}$$

From the linking actions one finds that there is no relative Euler term between $\mathfrak{B}_0^{s\bar{s}}$ and $\mathfrak{B}_1^{s\bar{s}}$. Also note that the junction operators between the $\mathsf{TY}(\mathbb{Z}_4)$ line operators have to be chosen judiciously as in section IV B 3, in order to be consistent with the 't Hooft anomaly.

Mathematically, this provides information about a pivotal tensor functor

$$\phi: \mathsf{TY}(\mathbb{Z}_4) \to \mathrm{Mat}_2(\mathsf{Vec}_{\mathbb{Z}_2}^{\omega})\,, \tag{IV.225}$$

where $\mathrm{Mat}_2(\mathsf{Vec}_{\mathbb{Z}_2}^{\omega})$ is the multi-fusion category formed by the line operators living on $\mathfrak{B}'$.

## V. CLUB SANDWICH

Just like closing a quiche leads to a sandwich, closing a club quiche leads to a club sandwich. More precisely, a $d$-dimensional club sandwich $\mathfrak{S}_d^C$ is obtained by supplying

a $d$-dimensional club quiche (IV.1) on the right with a (possibly non-topological) boundary condition $\mathfrak{B}_d^{\mathrm{phys}}$ of the $(d+1)$-dimensional TQFT $\mathfrak{Z}_{d+1}'$

$$\mathfrak{S}_d^C = (\mathfrak{B}_d, \mathfrak{Z}_{d+1}, \mathcal{I}_d, \mathfrak{Z}_{d+1}', \mathfrak{B}_d^{\mathrm{phys}})\,. \tag{V.1}$$

Applied to $\mathfrak{Z} = \mathfrak{Z}(\mathcal{S})$, i.e. in the context of the SymTFT, the club sandwich becomes

$$\mathfrak{S}_d^C = (\mathfrak{B}_\mathcal{S}^{\mathrm{sym}}, \mathfrak{Z}_{d+1}(\mathcal{S}), \mathcal{I}_d, \mathfrak{Z}_{d+1}', \mathfrak{B}_d^{\mathrm{phys}})\,, \tag{V.2}$$

which we depict as

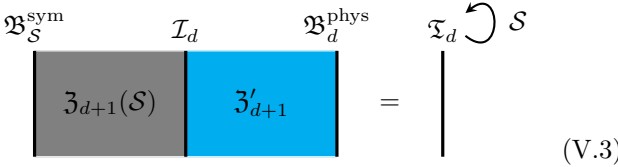

$$\text{(V.3)}$$

The full interval compactification shown on the right hand side involves collapsing both intervals occupied by $\mathfrak{Z}_{d+1}(\mathcal{S})$ and $\mathfrak{Z}_{d+1}'$, respectively. The result is an $\mathcal{S}$-symmetric QFT $\mathfrak{T}_d$. Thus, a club quiche $\mathcal{Q}_d^C$ can be viewed as a machine mapping (possibly non-topological) boundary conditions of a $(d+1)$-dimensional TQFT $\mathfrak{Z}_{d+1}'$ to $\mathcal{S}$-symmetric $d$-dimensional QFTs

$$\mathcal{Q}_d^C : \{\text{Boundaries of TQFT } \mathfrak{Z}_{d+1}'\}$$
$$\to \{\mathcal{S}\text{ -symmetric QFTs}\}\,. \tag{V.4}$$

The standard SymTFT sandwich in this context is obtained by specializing to $\mathfrak{Z}_{d+1}'$ to be the trivial topological order.

### A. Kennedy-Tasaki (KT) Transformations

The club sandwich realizes what are known as the Kennedy-Tasaki (KT) transformations and generalizations thereof. For this, specialize $\mathfrak{Z}_{d+1}'$ to be the SymTFT associated to a symmetry $\mathcal{S}'$

$$\mathfrak{Z}_{d+1}' = \mathfrak{Z}_{d+1}(\mathcal{S}')\,. \tag{V.5}$$

Then the club sandwich construction provides a map

$$\mathcal{K}_{\mathcal{I}_d}^{\mathcal{S},\mathcal{S}'} : \{\mathcal{S}'\text{-symmetric QFTs}\} \to \{\mathcal{S}\text{-symmetric QFTs}\} \tag{V.6}$$

We begin with an $\mathcal{S}'$-symmetric $d$-dimensional QFT $\mathfrak{T}_d^{\mathcal{S}'}$, which by the sandwich construction provides a generally non-topological boundary condition $\mathfrak{B}_{\mathfrak{T}_d^{\mathcal{S}'}}^{\mathrm{phys}}$ of the SymTFT $\mathfrak{Z}_{\mathcal{S}'}'$

$$\begin{array}{ccc} \mathfrak{B}_{\mathcal{S}'}^{\mathrm{sym}} & \mathfrak{B}_{\mathfrak{T}_d^{\mathcal{S}'}}^{\mathrm{phys}} & \mathfrak{T}_d^{\mathcal{S}'} \\ \boxed{\mathfrak{Z}_{d+1}(\mathcal{S}')} & = & | \end{array}$$
$$\text{(V.7)}$$

Inserting this boundary on the right hand side of a club sandwich, results upon interval compactification to an $\mathcal{S}$-symmetric QFT $\mathfrak{T}_d^{\mathcal{S}}$

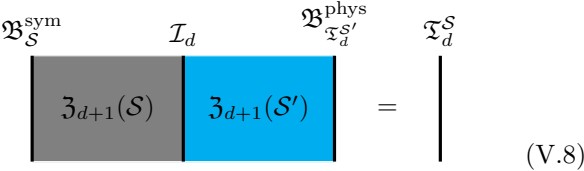

$$\text{(V.8)}$$

The map (V.6) is then

$$\mathcal{K}_{\mathcal{I}}^{\mathcal{S},\mathcal{S}'} :\ \mathfrak{T}_d^{\mathcal{S}'} \mapsto \mathfrak{T}_d^{\mathcal{S}} \qquad \text{(V.9)}$$

We refer to such a transformation as a **KT transformation** (short for Kennedy-Tasaki). The physical $\mathfrak{B}_{\mathfrak{T}_d^{\mathcal{S}}}^{\text{phys}}$ associated to the $\mathcal{S}$-symmetric QFT $\mathfrak{T}_d^{\mathcal{S}}$ appearing in its sandwich construction

$$\text{(V.10)}$$

is obtained simply by compactifying the interval occupied by $\mathfrak{Z}_{d+1}(\mathcal{S}')$ in the club sandwich (V.8) leading to the composition of $\mathcal{I}_d$ and $\mathfrak{B}_{\mathfrak{T}_d^{\mathcal{S}'}}^{\text{phys}}$, i.e. we have

$$\text{(V.11)}$$

Let us restrict to minimal club quiche $\mathcal{Q}_d^C$. Then, the above map $\mathcal{K}_{\mathcal{I}}^{\mathcal{S},\mathcal{S}'}$ respects irreducibility:

$$\mathcal{K}_{\mathcal{I}}^{\mathcal{S},\mathcal{S}'}: \quad \begin{array}{c} \{\text{Irreducible } \mathcal{S}'\text{-symmetric } d\text{-dimensional QFTs}\} \\ \downarrow \\ \{\text{Irreducible } \mathcal{S}\text{-symmetric } d\text{-dimensional QFTs}\} \end{array}$$

$$\text{(V.12)}$$

In order to see this, notice that irreducibility means that the physical boundary $\mathfrak{B}_{\mathfrak{T}_d^{\mathcal{S}'}}^{\text{phys}}$ carries no topological local operators that are not proportional to identity, or more succinctly no non-identity topological local operators. Minimality of the club quiche means that no topological bulk line of $\mathfrak{Z}_{d+1}(\mathcal{S}')$ can end along the topological interface $\mathcal{I}_d$. Consequently, no new topological local operators are produced by compactification of bulk lines in the compactification shown in (V.11). Thus, the physical boundary $\mathfrak{B}_{\mathfrak{T}_d^{\mathcal{S}}}^{\text{phys}}$ carries no non-identity topological local operators and hence $\mathfrak{T}_d^{\mathcal{S}}$ is irreducible. We refer to

KT transformations arising from minimal club quiches as **minimal KT transformations**. Note that, although the theory $\mathfrak{T}_d^{\mathcal{S}}$ is irreducible as an $\mathcal{S}$-symmetric QFT, it may be reducible when we forget the $\mathcal{S}$ symmetry, which happens whenever the boundary $\mathfrak{B}_d'$ of $\mathfrak{Z}_{d+1}(\mathcal{S}')$ produced by compactifying the interval occupied by $\mathfrak{Z}_{d+1}(\mathcal{S})$

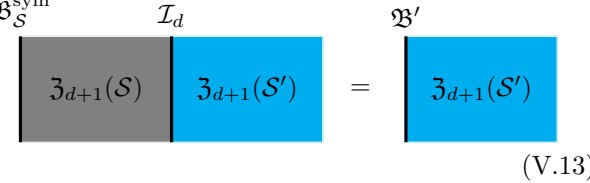

$$\text{(V.13)}$$

is reducible.

Let $\mathfrak{T}^{\mathcal{S}'}$ be a 2d TQFT lying in an irreducible $\mathcal{S}'$-symmetric (1+1)d gapped phase described by a Lagrangian algebra

$$\mathcal{L}_{\mathfrak{T}^{\mathcal{S}'}}^{\text{phys}} = \bigoplus_{a'} n_{a'}^{\text{phys}} (\mathbf{Q}'_{a'})^* . \qquad \text{(V.14)}$$

Applying a minimal KT transformation $\mathcal{K}_{\mathcal{I}}^{\mathcal{S},\mathcal{S}'}$ we find a 2d TQFT $\mathfrak{T}^{\mathcal{S}}$ lying in an irreducible $\mathcal{S}$-symmetric (1+1)d gapped phase described by the Lagrangian algebra

$$\mathcal{L}_{\mathfrak{T}^{\mathcal{S}}}^{\text{phys}} = \bigoplus_{a} n_{a,a'} n_{a'}^{\text{phys}} \mathbf{Q}_a . \qquad \text{(V.15)}$$

**Computational realization.** At the level of generalized charges, a minimal 2d KT transformation works as follows. A property of a minimal Lagrangian algebra $\mathcal{L}_{\mathcal{I}} \in \mathcal{Z}(\mathcal{S}) \boxtimes \overline{\mathcal{Z}}(\mathcal{S}')$ is that given a simple anyon $\overline{\mathbf{Q}}'_{a'} \in \overline{\mathcal{Z}}(\mathcal{S}')$, there always exists at least one term in $\mathcal{L}_{\mathcal{I}}$ of the form

$$n_{aa'} \mathbf{Q}_a \overline{\mathbf{Q}}'_{a'} \in \mathcal{L}_{\mathcal{I}} \qquad \text{(V.16)}$$

for some simple anyon $\mathbf{Q}_a \in \mathcal{Z}(\mathcal{S})$ and $n_{aa'} > 0$. This means that a simple anyon $\mathbf{Q}'_{a'}$ of $\mathcal{Z}(\mathcal{S}')$ can always be converted into some anyon of $\mathcal{Z}(\mathcal{S})$ as it passes through the interface $\mathcal{I}$ from the right to the left. That is, a minimal interface $\mathcal{I}$ converts each irreducible generalized charge of $\mathcal{S}'$ into a (possibly reducible) generalized charge of $\mathcal{S}$. This map between generalized charges is mathematically encoded in a functor

$$F_{\mathcal{I}} :\ \mathcal{Z}(\mathcal{S}') \to \mathcal{Z}(\mathcal{S}) , \qquad \text{(V.17)}$$

which takes the form

$$F_{\mathcal{I}}(\mathbf{Q}'_{a'}) = \bigoplus_{a} n_{aa'} \mathbf{Q}_a^* , \qquad \text{(V.18)}$$

where the coefficients $n_{aa'}$ appear in the Lagrangian algebra $\mathcal{L}_{\mathcal{I}}$ as in (IV.9).

Let's discuss the KT transformations induced by the minimal club quiches studied in the previous section in turn.

## B. KT from $\mathbb{Z}_2$ to $\mathbb{Z}_4$ Symmetry I

First consider the club quiche (IV.38) associated to the algebra $\mathcal{A}_e = 1 \oplus e^2$, which was studied in section IV B 1.

The symmetry before the KT transformation is $\mathbb{Z}_2$

$$\mathcal{S}' = \mathsf{Vec}_{\mathbb{Z}_2} \qquad (V.19)$$

and the symmetry obtained after the KT transformation is $\mathbb{Z}_4$

$$\mathcal{S} = \mathsf{Vec}_{\mathbb{Z}_4} . \qquad (V.20)$$

Let $\mathfrak{T}^{\mathcal{S}'}$ be a $\mathbb{Z}_2$-symmetric 2d QFT. We consider the Club Sandwich:

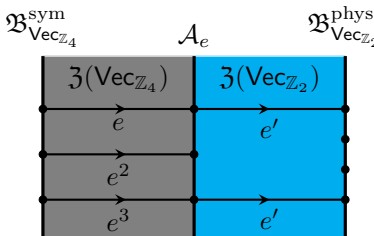

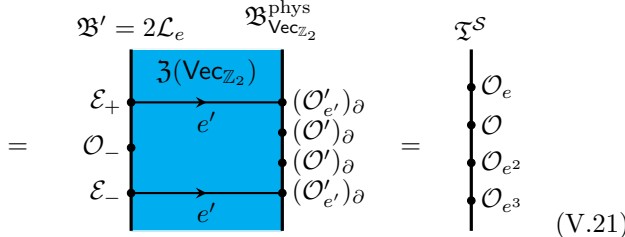

$$\qquad (V.21)$$

We will discuss the various operators shortly.

The KT transformation $\mathfrak{T}^{\mathcal{S}}$ is a $\mathbb{Z}_4$-symmetric 2d QFT, whose underlying non-symmetric QFT is easily obtained by noting from (IV.45) that the boundary $\mathfrak{B}'$ obtained by compactifying the interval occupied by $\mathfrak{Z}(\mathcal{S})$ is

$$\mathfrak{B}' = \mathfrak{B}_{\mathcal{S}'}^{\mathrm{sym}} \oplus \mathfrak{B}_{\mathcal{S}'}^{\mathrm{sym}} . \qquad (V.22)$$

Thus, the QFT $\mathfrak{T}^{\mathcal{S}}$ involves two copies of the sandwich construction for the QFT $\mathfrak{T}^{\mathcal{S}'}$, and hence we can write

$$\mathfrak{T}^{\mathcal{S}} = (\mathfrak{T}^{\mathcal{S}'})_0 \oplus (\mathfrak{T}^{\mathcal{S}'})_1 , \qquad (V.23)$$

where $(\mathfrak{T}^{\mathcal{S}'})_i$ is a copy of the QFT $\mathfrak{T}^{\mathcal{S}'}$. One may say that $\mathfrak{T}^{\mathcal{S}}$ is a QFT with two **universes**, such that each universe comprises of a copy of QFT $\mathfrak{T}^{\mathcal{S}'}$. Note that the relative Euler term between the two $\mathfrak{T}^{\mathcal{S}'}$ universes is trivial, which is a consequence of the fact that the relative Euler term between the two irreducible boundaries comprising $\mathfrak{B}'$ is trivial, as displayed in (IV.45).

Under the full club sandwich compactification, the lines of the boundary $\mathfrak{B}'$ descend to lines of $\mathfrak{T}^{\mathcal{S}}$. The line operators living on $\mathfrak{B}'$ are labeled as in (IV.35). The line $1_{ii}$ descends to the identity line on $(\mathfrak{T}^{\mathcal{S}'})_i$, and the

line $P_{ii}$ descends to the $\mathbb{Z}_2$ symmetry of $(\mathfrak{T}^{\mathcal{S}'})_i$. The lines $1_{01}$ and $1_{10}$ descend to lines exchanging the two copies of the QFT $\mathfrak{T}^{\mathcal{S}'}$. These two lines $1_{01}$ and $1_{10}$ have trivial quantum dimensions, correlated to the fact that there is no relative Euler term between the two $\mathfrak{T}^{\mathcal{S}'}$ universes. Finally the lines $P_{10}$ and $P_{01}$ are combinations shown in (IV.37).

From (IV.48), we learn that the generator $P$ of the $\mathbb{Z}_4$ symmetry $\mathcal{S}$ of $\mathfrak{T}^{\mathcal{S}}$ is realized by the line operator

$$\phi(P) = 1_{01} \oplus P_{10} . \qquad (V.24)$$

Note that the $\mathbb{Z}_2$ subgroup of $\mathbb{Z}_4$ is identified as the diagonal $\mathbb{Z}_2$ of the two $\mathbb{Z}_2$ symmetries acting on the individual $\mathfrak{T}^{\mathcal{S}'}$ universes

$$\phi(P^2) = P_{00} \oplus P_{11} . \qquad (V.25)$$

That is, the $\mathbb{Z}_4$ symmetry exchanges the two universes, but the $\mathbb{Z}_2$ subgroup of $\mathbb{Z}_4$ is completely localized within individual universes. We may thus express the $\mathbb{Z}_4$-symmetric theory $\mathfrak{T}^{\mathcal{S}}$ as

$$\mathfrak{T}^{\mathcal{S}} \quad = \quad \mathbb{Z}_2 \overset{\mathbb{Z}_4}{\underset{\mathbb{Z}_4}{\raisebox{0pt}{$\circlearrowleft$}}} (\mathfrak{T}^{\mathcal{S}'})_0 \oplus (\mathfrak{T}^{\mathcal{S}'})_1 \, \mathbb{Z}_2$$

$$\qquad (V.26)$$

From the expression for the Lagrangian algebra $\mathcal{L}_{\mathcal{I}}$ in (IV.30) we observe that the map

$$\mathcal{Z}(\mathsf{Vec}_{\mathbb{Z}_2}) \to \mathcal{Z}(\mathsf{Vec}_{\mathbb{Z}_4}) \qquad (V.27)$$

of generalized charges is

$$\begin{aligned} 1 &\to 1 \oplus e^2, & m' &\to m^2 \oplus e^2 m^2 \\ e' &\to e \oplus e^3, & e'm' &\to em^2 \oplus e^3 m^2 . \end{aligned} \qquad (V.28)$$

According to this map we can identify the operators as follows – also depicted in (V.21):

- An untwisted operator $\mathcal{O}'$ of $\mathfrak{T}^{\mathcal{S}'}$ uncharged under $\mathbb{Z}_2$ descends to two operators of $\mathfrak{T}^{\mathcal{S}}$: one of them being an untwisted operator $\mathcal{O}$ that is uncharged under $\mathbb{Z}_4$, and the other being an untwisted operator $\mathcal{O}_{e^2}$ having charge 2 under $\mathbb{Z}_4$. We can recognize these operators as

$$\mathcal{O} = \mathcal{O}'_0 + \mathcal{O}'_1 , \qquad \mathcal{O}_{e^2} = \mathcal{O}'_0 - \mathcal{O}'_1 , \qquad (V.29)$$

where $\mathcal{O}'_i$ is a copy of $\mathcal{O}'$ in the universe $(\mathfrak{T}^{\mathcal{S}'})_i$.

- An untwisted operator $\mathcal{O}'_{e'}$ of $\mathfrak{T}^{\mathcal{S}'}$ charged under $\mathbb{Z}_2$ descends to two operators of $\mathfrak{T}^{\mathcal{S}}$: one of them being an untwisted operator $\mathcal{O}_e$ that has charge 1 under $\mathbb{Z}_4$, and the other being an untwisted operator $\mathcal{O}_{e^3}$ that has charge 3 under $\mathbb{Z}_4$. We can recognize these operators as

$$\begin{aligned} \mathcal{O}_e &= \mathcal{O}'_{e',0} - i\mathcal{O}'_{e',1} \\ \mathcal{O}_{e^3} &= \mathcal{O}'_{e',0} + i\mathcal{O}'_{e',1} , \end{aligned} \qquad (V.30)$$

where $\mathcal{O}'_{e',i}$ is a copy of $\mathcal{O}'_{e'}$ in the universe $(\mathfrak{T}^{\mathcal{S}'})_i$.

- An operator $\mathcal{O}'_{m'}$ in $P'$-twisted sector of $\mathfrak{T}^{\mathcal{S}'}$, where $P'$ is the generator of $\mathbb{Z}_2$ symmetry, and uncharged under $\mathbb{Z}_2$ descends to two operators of $\mathfrak{T}^{\mathcal{S}}$: one of them being an operator $\mathcal{O}_{m^2}$ in $P^2$-twisted sector, where $P$ is the generator of $\mathbb{Z}_4$, that is uncharged under $\mathbb{Z}_4$, and the other being an operator $\mathcal{O}_{e^2 m^2}$ in $P^2$-twisted sector having charge 2 under $\mathbb{Z}_4$. We can recognize these operators as

$$
\begin{aligned}
\mathcal{O}_{m^2} &= \mathcal{O}'_{m',0} + \mathcal{O}'_{m',1} \\
\mathcal{O}_{e^2 m^2} &= \mathcal{O}'_{m',0} - \mathcal{O}'_{m',1} \,,
\end{aligned}
\tag{V.31}
$$

where $\mathcal{O}'_{m',i}$ is a copy of $\mathcal{O}'_{m'}$ in the universe $(\mathfrak{T}^{\mathcal{S}'})_i$.

- An operator $\mathcal{O}'_{e'm'}$ in $P'$-twisted sector of $\mathfrak{T}^{\mathcal{S}'}$ and charged under $\mathbb{Z}_2$ descends to two operators of $\mathfrak{T}^{\mathcal{S}}$: one of them being an operator $\mathcal{O}_{em^2}$ in $P^2$-twisted sector that has charge 1 under $\mathbb{Z}_4$, and the other being an operator $\mathcal{O}_{e^3 m^2}$ in $P^2$-twisted sector having charge 3 under $\mathbb{Z}_4$. We can recognize these operators as

$$
\begin{aligned}
\mathcal{O}_{em^2} &= \mathcal{O}'_{e'm',0} - i\mathcal{O}'_{e'm',1} \\
\mathcal{O}_{e^3 m^2} &= \mathcal{O}'_{e'm',0} + i\mathcal{O}'_{e'm',1} \,,
\end{aligned}
\tag{V.32}
$$

where $\mathcal{O}'_{e'm',i}$ is a copy of $\mathcal{O}'_{e'm'}$ in the universe $(\mathfrak{T}^{\mathcal{S}'})_i$.

## C. KT from $\mathbb{Z}_2$ to $\mathbb{Z}_4$ Symmetry II

Now consider the club quiche (IV.60) associated to the algebra $\mathcal{A}_m = 1 \oplus m^2$, which was studied in section IV B 2.

As in the previous example, the symmetry before the KT transformation is $\mathbb{Z}_2$

$$
\mathcal{S}' = \mathsf{Vec}_{\mathbb{Z}_2}
\tag{V.33}
$$

and the symmetry obtained after the KT transformation is $\mathbb{Z}_4$

$$
\mathcal{S} = \mathsf{Vec}_{\mathbb{Z}_4} \,.
\tag{V.34}
$$

Again let $\mathfrak{T}^{\mathcal{S}'}$ be a $\mathbb{Z}_2$-symmetric 2d QFT, and we want to understand its KT transformation $\mathfrak{T}^{\mathcal{S}}$, which is a $\mathbb{Z}_4$-symmetric 2d QFT.

The boundary $\mathfrak{B}'$ obtained by compactifying the interval occupied by $\mathfrak{Z}(\mathcal{S})$ is

$$
\mathfrak{B}' = \mathfrak{B}^{\mathrm{sym}}_{\mathcal{S}'}
\tag{V.35}
$$

and hence the QFT underlying $\mathfrak{T}^{\mathcal{S}}$ is exactly the same as the QFT underlying $\mathfrak{T}^{\mathcal{S}'}$. From (IV.62), we know that the generator $P$ of the $\mathbb{Z}_4$ symmetry is realized as the generator $P'$ of the $\mathbb{Z}_2$ symmetry, and hence we can express $\mathfrak{T}^{\mathcal{S}}$ as

$$
\mathfrak{T}^{\mathcal{S}} = \mathfrak{T}^{\mathcal{S}'} \overset{\curvearrowleft}{\curvearrowright} \mathbb{Z}_4
\tag{V.36}
$$

In other words, we have used the non-trivial homomorphism $\mathbb{Z}_4 \to \mathbb{Z}_2$ to regard a $\mathbb{Z}_2$-symmetric QFT as a $\mathbb{Z}_4$-symmetric QFT.

From the expression (IV.56) for $\mathcal{L}_{\mathcal{I}}$ we observe that the map

$$
\mathcal{Z}(\mathsf{Vec}_{\mathbb{Z}_2}) \to \mathcal{Z}(\mathsf{Vec}_{\mathbb{Z}_4})
\tag{V.37}
$$

of generalized charges is

$$
\begin{aligned}
1 &\to 1 \oplus m^2, & m' &\to m \oplus m^3 \\
e' &\to e^2 \oplus e^2 m^2, & e'm' &\to e^2 m \oplus e^2 m^3
\end{aligned}
\tag{V.38}
$$

According to this map the operators are identified as follows:

- An untwisted operator $\mathcal{O}'$ of $\mathfrak{T}^{\mathcal{S}'}$ uncharged under $\mathbb{Z}_2$ descends to two operators of $\mathfrak{T}^{\mathcal{S}}$: one of them being an untwisted operator $\mathcal{O}$ that is uncharged under $\mathbb{Z}_4$, and the other being a $P^2$-twisted sector operator $\mathcal{O}_{m^2}$ uncharged under $\mathbb{Z}_4$. We can recognize these operators as

$$
\mathcal{O} = \mathcal{O}' \,, \qquad \mathcal{O}_{m^2} = \mathcal{O}' \,,
\tag{V.39}
$$

where $\mathcal{O}_{m^2}$ is the operator $\mathcal{O}'$ viewed as sitting at the end of line $\phi(P^2)$.

- An untwisted operator $\mathcal{O}'_{e'}$ of $\mathfrak{T}^{\mathcal{S}'}$ charged under $\mathbb{Z}_2$ descends to two operators of $\mathfrak{T}^{\mathcal{S}}$: one of them being an untwisted operator $\mathcal{O}_{e^2}$ that has charge 2 under $\mathbb{Z}_4$, and the other being a $P^2$-twisted sector operator $\mathcal{O}_{e^2 m^2}$ that has charge 2 under $\mathbb{Z}_4$. We can recognize these operators as

$$
\mathcal{O}_{e^2} = \mathcal{O}'_{e'} \qquad \mathcal{O}_{e^2 m^2} = \mathcal{O}'_{e'} \,,
\tag{V.40}
$$

where $\mathcal{O}_{e^2 m^2}$ is the operator $\mathcal{O}'_{e'}$ viewed as sitting at the end of line $\phi(P^2)$.

- An operator $\mathcal{O}'_{m'}$ in $P'$-twisted sector of $\mathfrak{T}^{\mathcal{S}'}$ and uncharged under $\mathbb{Z}_2$ descends to two operators of $\mathfrak{T}^{\mathcal{S}}$: one of them being an operator $\mathcal{O}_m$ in $P$-twisted sector that is uncharged under $\mathbb{Z}_4$, and the other being an operator $\mathcal{O}_{m^3}$ in $P^3$-twisted sector that is also uncharged under $\mathbb{Z}_4$. We can recognize these operators as

$$
\mathcal{O}_m = \mathcal{O}'_{m'} \qquad \mathcal{O}_{m^3} = \mathcal{O}'_{m'} \,,
\tag{V.41}
$$

where $\mathcal{O}_m$ is the operator $\mathcal{O}'_{m'}$ viewed as sitting at the end of line $\phi(P)$ and $\mathcal{O}_{m^3}$ is the operator $\mathcal{O}'_{m'}$ viewed as sitting at the end of line $\phi(P^3)$.

- An operator $\mathcal{O}'_{e'm'}$ in $P'$-twisted sector of $\mathfrak{T}^{\mathcal{S}'}$ and charged under $\mathbb{Z}_2$ descends to two operators of $\mathfrak{T}^{\mathcal{S}}$: one of them being an operator $\mathcal{O}_{e^2 m}$ in $P$-twisted sector that has charge 2 under $\mathbb{Z}_4$, and the other being an operator $\mathcal{O}_{e^2 m^3}$ in $P^3$-twisted sector having charge 2 under $\mathbb{Z}_4$. We can recognize these operators as

$$
\mathcal{O}_{e^2 m} = \mathcal{O}'_{e'm'} \,, \quad \mathcal{O}_{e^2 m^3} = \mathcal{O}'_{e'm'} \,,
\tag{V.42}
$$

where $\mathcal{O}_{e^2m}$ is the operator $\mathcal{O}'_{e'm'}$ viewed as sitting at the end of line $\phi(P)$ and $\mathcal{O}_{e^2m^3}$ is the operator $\mathcal{O}'_{e'm'}$ viewed as sitting at the end of line $\phi(P^3)$.

### D. KT from Anomalous $\mathbb{Z}_2$ to Non-anomalous $\mathbb{Z}_4$ Symmetry

Now consider the club quiche (IV.72) associated to the algebra $\mathcal{A}_{em}$, which was studied in section IV B 3.

The symmetry before the KT transformation is $\mathbb{Z}_2$ with non-trivial 't Hooft anomaly

$$\mathcal{S}' = \mathsf{Vec}_{\mathbb{Z}_2}^{\omega} \qquad (\text{V.43})$$

and the symmetry obtained after the KT transformation is a non-anomalous $\mathbb{Z}_4$

$$\mathcal{S} = \mathsf{Vec}_{\mathbb{Z}_4} \,. \qquad (\text{V.44})$$

Let $\mathfrak{T}^{\mathcal{S}'}$ be a 2d QFT with anomalous $\mathbb{Z}_2$-symmetry. We want to understand its KT transformation $\mathfrak{T}^{\mathcal{S}}$ which is a 2d QFT with non-anomalous $\mathbb{Z}_4$-symmetry. For this purpose, first note that the boundary $\mathfrak{B}'$ obtained by compactifying the interval occupied by $\mathfrak{Z}(\mathcal{S})$ is

$$\mathfrak{B}' = \mathfrak{B}_{\mathcal{S}'}^{\mathrm{sym}} \,. \qquad (\text{V.45})$$

and hence the QFT underlying $\mathfrak{T}^{\mathcal{S}}$ is exactly the same as the QFT underlying $\mathfrak{T}^{\mathcal{S}'}$. From (IV.74), we know that the generator $P$ of the $\mathbb{Z}_4$ symmetry is realized as the generator $P'$ of the $\mathbb{Z}_2$ symmetry, and hence we can express $\mathfrak{T}^{\mathcal{S}}$ as

$$\mathfrak{T}^{\mathcal{S}} = \mathfrak{T}^{\mathcal{S}'} \, \overset{\curvearrowleft}{\curvearrowright} \, \mathbb{Z}_4 \qquad (\text{V.46})$$

Even though the map at the level of topological lines is simply a homomorphism $\mathbb{Z}_4 \to \mathbb{Z}_2$

$$(1, P, P^2, P^3) \to (1, P', 1, P') \qquad (\text{V.47})$$

the junctions of $\mathbb{Z}_4$ symmetry lines are chosen in a fashion that does not follow from the above homomorphism. See section IV B 3 for a detailed discussion.

From the expression (IV.67) for $\mathcal{L}_{\mathcal{I}}$ we observe that the map

$$\mathcal{Z}(\mathsf{Vec}_{\mathbb{Z}_2}^{\omega}) \to \mathcal{Z}(\mathsf{Vec}_{\mathbb{Z}_4}) \qquad (\text{V.48})$$

of generalized charges is

$$
\begin{aligned}
1 &\to 1 \oplus e^2 m^2, & \bar{s} &\to em^3 \oplus e^3 m \\
s &\to em \oplus e^3 m^3, & s\bar{s} &\to e^2 \oplus m^2 \,.
\end{aligned}
\qquad (\text{V.49})
$$

According to this map:

- An untwisted operator $\mathcal{O}'$ of $\mathfrak{T}^{\mathcal{S}'}$ uncharged under $\mathbb{Z}_2$ descends to two operators of $\mathfrak{T}^{\mathcal{S}}$: one of them being an untwisted operator $\mathcal{O}$ that is uncharged

under $\mathbb{Z}_4$, and the other being a $P^2$-twisted sector operator $\mathcal{O}_{e^2m^2}$ carrying charge 2 under $\mathbb{Z}_4$. We can recognize these operators as

$$\mathcal{O} = \mathcal{O}' \,, \qquad \mathcal{O}_{e^2m^2} = \mathcal{O}' \qquad (\text{V.50})$$

where $\mathcal{O}_{e^2m^2}$ is the operator $\mathcal{O}'$ viewed as sitting at the end of line $P^2$. The charge 2 can be observed from the property (IV.80) of the local operators living at the junctions of $\mathbb{Z}_4$ lines:

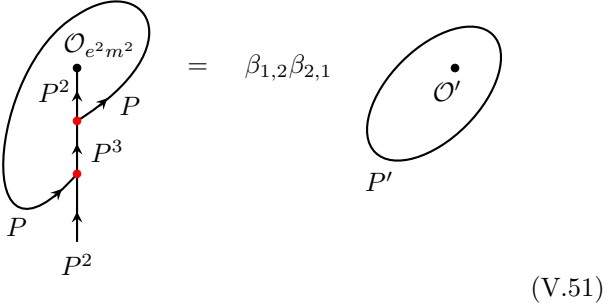

$$(\text{V.51})$$

which equals $(-1)$ times the operator $\mathcal{O}_{e^2m^2}$.

- An untwisted operator $\mathcal{O}'_{s\bar{s}}$ of $\mathfrak{T}^{\mathcal{S}'}$ charged under $\mathbb{Z}_2$ descends to two operators of $\mathfrak{T}^{\mathcal{S}}$: one of them being an untwisted operator $\mathcal{O}_{e^2}$ that has charge 2 under $\mathbb{Z}_4$, and the other being a $P^2$-twisted sector operator $\mathcal{O}_{m^2}$ that is uncharged under $\mathbb{Z}_4$. We can recognize these operators as

$$\mathcal{O}_{e^2} = \mathcal{O}'_{s\bar{s}} \qquad \mathcal{O}_{m^2} = \mathcal{O}'_{s\bar{s}} \,, \qquad (\text{V.52})$$

where $\mathcal{O}_{m^2}$ is the operator $\mathcal{O}'_{s\bar{s}}$ viewed as sitting at the end of line $P^2$. The fact that $\mathcal{O}_{m^2}$ is uncharged is seen by an argument similar to the one shown in (V.51), where now the linking action of $P'$ provides an extra factor of $(-1)$.

- An operator $\mathcal{O}'_s$ in $P'$-twisted sector of $\mathfrak{T}^{\mathcal{S}'}$ and having charge-$1/2$ under $\mathbb{Z}_2$ [130], with linking action of $P'$ on $\mathcal{O}'_s$ being $i$, descends to two operators of $\mathfrak{T}^{\mathcal{S}}$: one of them being an operator $\mathcal{O}_{em}$ in $P$-twisted sector that has charge 1 under $\mathbb{Z}_4$, and the other being an operator $\mathcal{O}_{e^3m^3}$ in $P^3$-twisted sector that has charge 3 under $\mathbb{Z}_4$. We can recognize these operators as

$$\mathcal{O}_{em} = \mathcal{O}'_s \qquad \mathcal{O}_{e^3m^3} = \mathcal{O}'_s \qquad (\text{V.53})$$

where $\mathcal{O}_{em}$ is the operator $\mathcal{O}'_s$ viewed as sitting at the end of line $P$ and $\mathcal{O}_{e^3m^3}$ is the operator $\mathcal{O}'_s$ viewed as sitting at the end of line $P^3$. The $\mathbb{Z}_4$

charge of $\mathcal{O}_{em}$ is observed via

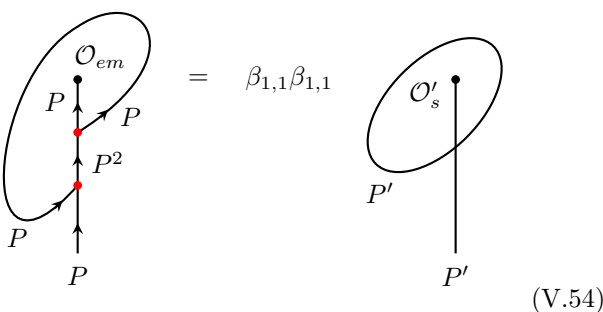

$$(V.54)$$

where the only contribution is the $P'$ linking action $i$. Similarly, the $\mathbb{Z}_4$ charge of $\mathcal{O}_{e^3 m^3}$ is observed via

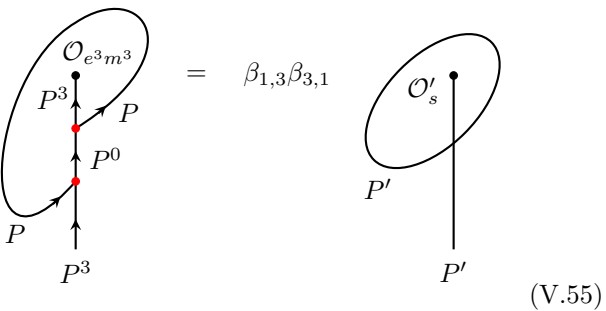

$$(V.55)$$

where along with the contribution from the $P'$ linking action, we have an extra $(-1)$ factor from $\beta_{1,3}\beta_{3,1}$.

- An operator $\mathcal{O}'_{\overline{s}}$ in $P'$-twisted sector of $\mathfrak{T}^{\mathcal{S}'}$ and having charge-$1/2$ under $\mathbb{Z}_2$, with linking action of $P'$ on $\mathcal{O}'_{\overline{s}}$ being $-i$, descends to two operators of $\mathfrak{T}^{\mathcal{S}}$: one of them being an operator $\mathcal{O}_{e^3 m}$ in $P$-twisted sector that has charge 3 under $\mathbb{Z}_4$, and the other being an operator $\mathcal{O}_{em^3}$ in $P^3$-twisted sector having charge 1 under $\mathbb{Z}_4$. We can recognize these operators as

$$\mathcal{O}_{e^3 m} = \mathcal{O}'_{\overline{s}} \qquad \mathcal{O}_{em^3} = \mathcal{O}'_{\overline{s}} \qquad (V.56)$$

where $\mathcal{O}_{e^3 m}$ is the operator $\mathcal{O}'_{\overline{s}}$ viewed as sitting at the end of line $P$ and $\mathcal{O}_{em^3}$ is the operator $\mathcal{O}'_{\overline{s}}$ viewed as sitting at the end of line $P^3$. The charges of the two operators can be observed in a similar fashion as in (V.54) and (V.55)

### E. KT from $\mathbb{Z}_3$ to $S_3$ Symmetry

Consider the club quiche (IV.97) associated to the algebra $\mathcal{A}_{1_-}$, which was studied in section IV C 1.

The symmetry before the KT transformation maps a $\mathbb{Z}_3$ to an $S_3$ symmetry

$$\mathcal{S}' = \mathsf{Vec}_{\mathbb{Z}_3} \to \mathcal{S} = \mathsf{Vec}_{S_3}. \qquad (V.57)$$

Let $\mathfrak{T}^{\mathcal{S}'}$ be a $\mathbb{Z}_3$-symmetric 2d QFT. Its KT transformation $\mathfrak{T}^{\mathcal{S}}$ is an $S_3$-symmetric 2d QFT.

The boundary $\mathfrak{B}'$ obtained by compactifying the interval occupied by $\mathfrak{Z}(\mathcal{S})$ is

$$\mathfrak{B}' = \mathfrak{B}^{\mathrm{sym}}_{\mathcal{S}'} \oplus \mathfrak{B}^{\mathrm{sym}}_{\mathcal{S}'}. \qquad (V.58)$$

Thus, the QFT $\mathfrak{T}^{\mathcal{S}}$ involves two copies of the sandwich construction for the QFT $\mathfrak{T}^{\mathcal{S}'}$, and hence we can write

$$\mathfrak{T}^{\mathcal{S}} = (\mathfrak{T}^{\mathcal{S}'})_0 \oplus (\mathfrak{T}^{\mathcal{S}'})_1, \qquad (V.59)$$

where $(\mathfrak{T}^{\mathcal{S}'})_i$ is a copy of the QFT $\mathfrak{T}^{\mathcal{S}'}$. Note that the relative Euler term between the two $\mathfrak{T}^{\mathcal{S}'}$ universes is trivial, which is a consequence of the fact that the relative Euler term between the two irreducible boundaries comprising $\mathfrak{B}'$ is trivial, as displayed in (IV.103).

The lines on $\mathfrak{B}'$ descend to lines on $\mathfrak{T}^{\mathcal{S}}$, which we label in the same way. $1_{ii}$ is the identity of $(\mathfrak{T}^{\mathcal{S}'})_i$, $P_{ii}$ is the generator of $\mathbb{Z}_3$ symmetry of $(\mathfrak{T}^{\mathcal{S}'})_i$ and $P_{ii}^2$ is its square. The rest of the lines are universe changing. Equation (IV.102) describes how the $S_3$ symmetry is then realized on $\mathfrak{T}^{\mathcal{S}}$, which we schematically depict as

$$\mathfrak{T}^{\mathcal{S}} = \mathbb{Z}_3 \underset{\underbrace{\hspace{2cm}}_{\mathbb{Z}_2}}{\overset{\curvearrowright}{\bigcirc}} (\mathfrak{T}^{\mathcal{S}'})_0 \oplus (\mathfrak{T}^{\mathcal{S}'})_1 \overset{\curvearrowleft}{\bigcirc} \mathbb{Z}_3^{-1}$$

$$(V.60)$$

displaying the actions of $\mathbb{Z}_3$ and $\mathbb{Z}_2$ subgroups of $S_3$ generated respectively by $a$ and $b$, and the notation $\mathbb{Z}_3^{-1}$ captures the fact that the generator $\phi(a)$ of $\mathbb{Z}_3$ inside $S_3$ when acting on the universe $(\mathfrak{T}^{\mathcal{S}'})_1$ is the inverse $P_{11}^2$ of the generator $P_{11}$ of the original $\mathbb{Z}_3$ symmetry of $(\mathfrak{T}^{\mathcal{S}'})_1$.

From the expression for the Lagrangian algebra $\mathcal{L}_{\mathcal{I}}$ in (IV.89) we observe that the map

$$\mathcal{Z}(\mathsf{Vec}_{\mathbb{Z}_3}) \to \mathcal{Z}(\mathsf{Vec}_{S_3}) \qquad (V.61)$$

of generalized charges is

$$\begin{aligned} 1 &\to 1 \oplus 1_-, & e &\to E, & e^2 &\to E \\ m &\to a_1, & em &\to a_\omega, & e^2 m &\to a_{\omega^2} \quad (V.62) \\ m^2 &\to a_1, & em^2 &\to a_{\omega^2}, & e^2 m^2 &\to a_\omega. \end{aligned}$$

According to this map:

- An untwisted operator $\mathcal{O}'$ of $\mathfrak{T}^{\mathcal{S}'}$ uncharged under $\mathbb{Z}_3$ descends to two operators of $\mathfrak{T}^{\mathcal{S}}$: one of them being an untwisted operator $\mathcal{O}$ that is uncharged under $S_3$, and the other being an untwisted operator $\mathcal{O}_{1_-}$ charged under $b$. We can recognize these operators as

$$\mathcal{O} = \mathcal{O}'_0 + \mathcal{O}'_1, \qquad \mathcal{O}_{1_-} = \mathcal{O}'_0 - \mathcal{O}'_1 \qquad (V.63)$$

where $\mathcal{O}'_i$ is a copy of $\mathcal{O}'$ in the universe $(\mathfrak{T}^{\mathcal{S}'})_i$.

- An untwisted operator $\mathcal{O}'_e$ of $\mathfrak{T}^{\mathcal{S}'}$ having charge 1 under $\mathbb{Z}_3$ descends to two untwisted operators $\mathcal{O}^{(1)}_{E,1}, \mathcal{O}^{(1)}_{E,2}$ of $\mathfrak{T}^{\mathcal{S}}$. These operators live in representation $E$ of $S_3$. We can recognize these operators as

$$\mathcal{O}^{(1)}_{E,1} = \mathcal{O}'_{e,0}, \qquad \mathcal{O}^{(1)}_{E,2} = \mathcal{O}'_{e,1} \qquad (V.64)$$

where $\mathcal{O}'_{e,i}$ is a copy of $\mathcal{O}'_e$ in the universe $(\mathfrak{T}^{\mathcal{S}'})_i$.

- Similarly, an untwisted operator $\mathcal{O}'_{e^2}$ of $\mathfrak{T}^{\mathcal{S}'}$ having charge 2 under $\mathbb{Z}_3$ descends to two untwisted operators $\mathcal{O}^{(2)}_{E,1}, \mathcal{O}^{(2)}_{E,2}$ of $\mathfrak{T}^{\mathcal{S}}$. These operators live in representation $E$ of $S_3$. We can recognize these operators as

$$\mathcal{O}^{(2)}_{E,1} = \mathcal{O}'_{e^2,1}, \qquad \mathcal{O}^{(2)}_{E,2} = \mathcal{O}'_{e^2,0} \qquad (V.65)$$

where $\mathcal{O}'_{e^2,i}$ is a copy of $\mathcal{O}'_{e^2}$ in the universe $(\mathfrak{T}^{\mathcal{S}'})_i$.

- An operator $\mathcal{O}'_{e^k m}$ in $P$-twisted sector of $\mathfrak{T}^{\mathcal{S}'}$, where $P$ is the generator of $\mathbb{Z}_3$ symmetry, and having charge $k$ under $\mathbb{Z}_3$ descends to two operators of $\mathfrak{T}^{\mathcal{S}}$: $\mathcal{O}^{(1)}_{a,\omega^k}$ in $a$-twisted sector having charge $k$ under $a$ and $\mathcal{O}^{(1)}_{a^2,\omega^{2k}}$ in $a^2$-twisted sector having charge $2k$ under $a$. The two operators are exchanged by the action of $b$. We can recognize these operators as

$$\mathcal{O}^{(1)}_{a,\omega^k} = \mathcal{O}'_{e^k m,0}, \quad \mathcal{O}^{(1)}_{a^2,\omega^{2k}} = \mathcal{O}'_{e^k m,1} \qquad (V.66)$$

where $\mathcal{O}'_{e^k m,i}$ is a copy of $\mathcal{O}'_{e^k m}$ in the universe $(\mathfrak{T}^{\mathcal{S}'})_i$.

- An operator $\mathcal{O}'_{e^k m^2}$ in $P^2$-twisted sector of $\mathfrak{T}^{\mathcal{S}'}$, and having charge $k$ under $\mathbb{Z}_3$ descends to two operators of $\mathfrak{T}^{\mathcal{S}}$: $\mathcal{O}^{(2)}_{a,\omega^{2k}}$ in $a$-twisted sector having charge $2k$ under $a$ and $\mathcal{O}^{(2)}_{a^2,\omega^k}$ in $a^2$-twisted sector having charge $k$ under $a$. The two operators are exchanged by the action of $b$. We can recognize these operators as

$$\mathcal{O}^{(2)}_{a,\omega^{2k}} = \mathcal{O}'_{e^k m^2,1}, \quad \mathcal{O}^{(2)}_{a^2,\omega^k} = \mathcal{O}'_{e^k m^2,0} \qquad (V.67)$$

where $\mathcal{O}'_{e^k m^2,i}$ is a copy of $\mathcal{O}'_{e^k m^2}$ in the universe $(\mathfrak{T}^{\mathcal{S}'})_i$.

### F.   KT from $\mathbb{Z}_2$ to $S_3$ Symmetry I

Consider the club quiche (IV.112) associated to the algebra $\mathcal{A}_E$, which was studied in section IV C 2. The associated KT transformation converts $\mathbb{Z}_2$ symmetry into $S_3$ symmetry

$$\mathcal{S}' = \mathsf{Vec}_{\mathbb{Z}_2} \quad \longrightarrow \quad \mathcal{S} = \mathsf{Vec}_{S_3}. \qquad (V.68)$$

Let $\mathfrak{T}^{\mathcal{S}'}$ be a $\mathbb{Z}_2$-symmetric 2d QFT. Its KT transformation $\mathfrak{T}^{\mathcal{S}}$ is an $S_3$-symmetric 2d QFT which can be expressed as

$$\mathfrak{T}^{\mathcal{S}} = (\mathfrak{T}^{\mathcal{S}'})_0 \oplus (\mathfrak{T}^{\mathcal{S}'})_1 \oplus (\mathfrak{T}^{\mathcal{S}'})_2, \qquad (V.69)$$

i.e. it comprises of three universes with each universe containing a copy of $\mathfrak{T}^{\mathcal{S}'}$. Note also that the relative Euler terms between any two universes is trivial.

The $S_3$ symmetry acts on the universes such that

$$(V.70)$$

where the $\mathbb{Z}_2$ symmetry is taken to be generated by $b$ and the $\mathbb{Z}_3$ symmetry generated by $a$, whose precise actions are encoded in their identification (IV.118).

From the expression for the Lagrangian algebra $\mathcal{L}_{\mathcal{I}}$ in (IV.106) we observe that the map

$$\mathcal{Z}(\mathsf{Vec}_{\mathbb{Z}_2}) \to \mathcal{Z}(\mathsf{Vec}_{S_3}) \qquad (V.71)$$

of generalized charges is

$$1 \to 1 \oplus E, \quad e \to 1_- \oplus E, \quad m \to b_+, \quad em \to b_-. \qquad (V.72)$$

According to this map the operators are identified as follows:

- An untwisted operator $\mathcal{O}'$ of $\mathfrak{T}^{\mathcal{S}'}$ uncharged under $\mathbb{Z}_2$ descends to three operators of $\mathfrak{T}^{\mathcal{S}}$: one of them being an untwisted operator $\mathcal{O}$ that is uncharged under $S_3$, and the other two $\mathcal{O}^{(1)}_{E,1}, \mathcal{O}^{(1)}_{E,2}$ being untwisted operators transforming in representation $E$ of $S_3$. We can recognize these operators as

$$\begin{aligned} \mathcal{O} &= \mathcal{O}'_0 + \mathcal{O}'_1 + \mathcal{O}'_2 \\ \mathcal{O}^{(1)}_{E,1} &= \mathcal{O}'_0 + \omega^2 \mathcal{O}'_1 + \omega \mathcal{O}'_2 \\ \mathcal{O}^{(1)}_{E,2} &= \mathcal{O}'_0 + \omega \mathcal{O}'_1 + \omega^2 \mathcal{O}'_2, \end{aligned} \qquad (V.73)$$

where $\mathcal{O}'_i$ is a copy of $\mathcal{O}'$ in the universe $(\mathfrak{T}^{\mathcal{S}'})_i$.

- An untwisted operator $\mathcal{O}'_e$ of $\mathfrak{T}^{\mathcal{S}'}$ charged under $\mathbb{Z}_2$ descends to three untwisted operators: one of them $\mathcal{O}_{1_-}$ transforming in representation $1_-$ of $S_3$ and the other two $\mathcal{O}^{(2)}_{E,1}, \mathcal{O}^{(2)}_{E,2}$ transforming in representation $E$ of $S_3$. We can recognize these operators as

$$\begin{aligned} \mathcal{O}_{1_-} &= \mathcal{O}'_{e,0} + \mathcal{O}'_{e,1} + \mathcal{O}'_{e,2} \\ \mathcal{O}^{(2)}_{E,1} &= \mathcal{O}'_{e,0} + \omega^2 \mathcal{O}'_{e,1} + \omega \mathcal{O}'_{e,2} \\ \mathcal{O}^{(2)}_{E,2} &= -\mathcal{O}'_{e,0} - \omega \mathcal{O}'_{e,1} - \omega^2 \mathcal{O}'_{e,2}, \end{aligned} \qquad (V.74)$$

where $\mathcal{O}'_{e,i}$ is a copy of $\mathcal{O}'_e$ in the universe $(\mathfrak{T}^{\mathcal{S}'})_i$.

- An operator $\mathcal{O}'_{e^k m}$ in $P$-twisted sector of $\mathfrak{T}^{\mathcal{S}'}$, where $P$ is the generator of $\mathbb{Z}_2$ symmetry, and having charge $k$ under $\mathbb{Z}_2$ descends to three operators of $\mathfrak{T}^{\mathcal{S}}$: $\mathcal{O}^{(k)}_{a^i b}$ in $a^i b$-twisted sector. The action of $b$ on these operators is

$$b: \quad \mathcal{O}^{(k)}_b \to (-1)^k \mathcal{O}^{(k)}_b, \quad \mathcal{O}^{(k)}_{ab} \leftrightarrow (-1)^k \mathcal{O}^{(k)}_{a^2 b}, \quad \text{(V.75)}$$

We can recognize these operators as

$$\mathcal{O}^{(k)}_{a^i b} = \mathcal{O}'_{e^k m, i}, \qquad \text{(V.76)}$$

where $\mathcal{O}'_{e^k m, i}$ is a copy of $\mathcal{O}'_{e^k m}$ in the universe $(\mathfrak{T}^{\mathcal{S}'})_i$.

### G.  KT from $\mathbb{Z}_2$ to $S_3$ Symmetry II

Consider the club quiche (IV.126) associated to the algebra $\mathcal{A}_{a_1}$, which was studied in section IV C 3. The associated KT transformation converts $\mathbb{Z}_2$ symmetry into $S_3$ symmetry

$$\mathcal{S}' = \mathsf{Vec}_{\mathbb{Z}_2} \quad \longrightarrow \quad \mathcal{S} = \mathsf{Vec}_{S_3}. \qquad \text{(V.77)}$$

Let $\mathfrak{T}^{\mathcal{S}'}$ be a $\mathbb{Z}_2$-symmetric 2d QFT. Its KT transformation $\mathfrak{T}^{\mathcal{S}}$ is an $S_3$-symmetric 2d QFT which can be expressed as

$$\mathfrak{T}^{\mathcal{S}} = \mathfrak{T}^{\mathcal{S}'} \,\overset{\curvearrowleft}{\phantom{.}}\, S_3 \qquad \text{(V.78)}$$

i.e. we regard the $\mathbb{Z}_2$-symmetric theory $\mathfrak{T}^{\mathcal{S}'}$ as an $S_3$-symmetric theory simply by pulling back symmetries using the non-trivial homomorphism $S_3 \to \mathbb{Z}_2$.

From the expression for the Lagrangian algebra $\mathcal{L}_{\mathcal{I}}$ in (IV.122) we observe that the map

$$\mathcal{Z}(\mathsf{Vec}_{\mathbb{Z}_2}) \to \mathcal{Z}(\mathsf{Vec}_{S_3}) \qquad \text{(V.79)}$$

of generalized charges is

$$1 \to 1 \oplus a_1, \quad e \to 1_- \oplus a_1, \quad m \to b_+, \quad em \to b_-. \qquad \text{(V.80)}$$

According to this map:

- An untwisted operator $\mathcal{O}'$ of $\mathfrak{T}^{\mathcal{S}'}$ uncharged under $\mathbb{Z}_2$ descends to three operators of $\mathfrak{T}^{\mathcal{S}}$: one of them being an untwisted operator $\mathcal{O}$ that is uncharged under $S_3$, and the other two $\mathcal{O}^{(1)}_a, \mathcal{O}^{(1)}_{a^2}$ being in $a, a^2$-twisted sectors which are uncharged under $a$ but exchanged by $b$. We can recognize these operators as

$$\mathcal{O} = \mathcal{O}', \qquad \mathcal{O}^{(1)}_{a^k} = \mathcal{O}', \qquad \text{(V.81)}$$

where $\mathcal{O}^{(1)}_{a^k}$ is the operator $\mathcal{O}'$ viewed as lying in the $a^k$-twisted sector.

- An untwisted operator $\mathcal{O}'_e$ of $\mathfrak{T}^{\mathcal{S}'}$ charged under $\mathbb{Z}_2$ descends to three operators of $\mathfrak{T}^{\mathcal{S}}$: one of them being an untwisted operator $\mathcal{O}_{1_-}$ in representation $1_-$ of $S_3$, and the other two $\mathcal{O}^{(2)}_a, \mathcal{O}^{(2)}_{a^2}$ being in $a, a^2$-twisted sectors which are uncharged under $a$ but exchanged by $b$. We can recognize these operators as

$$\mathcal{O}_{1_-} = \mathcal{O}'_e \qquad \mathcal{O}^{(2)}_{a^k} = (-1)^k \mathcal{O}'_e, \qquad \text{(V.82)}$$

where $\mathcal{O}^{(2)}_{a^k}$ is the operator $\mathcal{O}'_e$ viewed as lying in the $a^k$-twisted sector.

- An operator $\mathcal{O}'_{e^k m}$ in $P$-twisted sector of $\mathfrak{T}^{\mathcal{S}'}$, where $P$ is the generator of $\mathbb{Z}_2$ symmetry before KT transformation, and having charge $k$ under $\mathbb{Z}_2$ descends to three operators of $\mathfrak{T}^{\mathcal{S}}$: $\mathcal{O}^{(k)}_{a^i b}$ in $a^i b$-twisted sector. The action of $b$ on these operators is described in (V.75). We can recognize these operators as

$$\mathcal{O}^{(k)}_{a^i b} = \mathcal{O}'_{e^k m}, \qquad \text{(V.83)}$$

where $\mathcal{O}^{(k)}_{a^i b}$ is the operator $\mathcal{O}'_{e^k m}$ viewed as lying in the $a^i b$-twisted sector.

### H.  KT from $\mathbb{Z}_3$ to $\mathsf{Rep}(S_3)$ Symmetry

Consider the $\mathsf{Rep}(S_3)$ club quiche (IV.144) associated to the algebra $\mathcal{A}_{1_-}$ studied in section IV D 2. The associated KT transformation converts $\mathbb{Z}_3$ symmetry into $\mathsf{Rep}(S_3)$ symmetry

$$\mathcal{S}' = \mathsf{Vec}_{\mathbb{Z}_3} \quad \longrightarrow \quad \mathcal{S} = \mathsf{Rep}(S_3). \qquad \text{(V.84)}$$

Let $\mathfrak{T}^{\mathcal{S}'}$ be a $\mathbb{Z}_3$-symmetric 2d QFT. Its KT transformation $\mathfrak{T}^{\mathcal{S}}$ is a $\mathsf{Rep}(S_3)$-symmetric 2d QFT which can be expressed as

$$\mathfrak{T}^{\mathcal{S}} = \mathfrak{T}^{\mathcal{S}'} \,\overset{\curvearrowleft}{\phantom{.}}\, \mathsf{Rep}(S_3) \qquad \text{(V.85)}$$

i.e. the underlying 2d QFT for $\mathfrak{T}^{\mathcal{S}}$ is the same as for $\mathfrak{T}^{\mathcal{S}'}$, with the $\mathsf{Rep}(S_3)$ symmetry realized in terms of the $\mathbb{Z}_3$ symmetry as described in (IV.146).

From the expression for the Lagrangian algebra $\mathcal{L}_{\mathcal{I}}$ in (IV.141) we observe that the map

$$\mathcal{Z}(\mathsf{Vec}_{\mathbb{Z}_3}) \to \mathcal{Z}(\mathsf{Rep}(S_3)) \qquad \text{(V.86)}$$

of generalized charges is

$$\begin{aligned}
1 &\to 1 \oplus 1_-, & m &\to E, & m^2 &\to E \\
e &\to a_1, & em &\to a_\omega, & em^2 &\to a_{\omega^2} \\
e^2 &\to a_1, & e^2 m &\to a_{\omega^2}, & e^2 m^2 &\to a_\omega
\end{aligned} \qquad \text{(V.87)}$$

According to this map:

- An untwisted operator $\mathcal{O}'$ of $\mathfrak{T}^{\mathcal{S}'}$ uncharged under $\mathbb{Z}_3$ descends to two operators of $\mathfrak{T}^{\mathcal{S}}$: one of them being an untwisted operator $\mathcal{O}$ that is uncharged under $\mathsf{Rep}(S_3)$, and the other being an operator $\mathcal{O}_{1_-}$ in the $1_-$-twisted sector that is uncharged under $\mathsf{Rep}(S_3)$. We call such an operator $\mathcal{O}_{1_-}$ as an operator living in $1_-$-multiplet of $\mathsf{Rep}(S_3)$ symmetry. We can recognize these operators as

$$\mathcal{O} = \mathcal{O}', \qquad \mathcal{O}_{1_-} = \mathcal{O}', \qquad (\text{V.88})$$

where $\mathcal{O}_{1_-}$ is the operator $\mathcal{O}'$ viewed as lying in the $1_-$-twisted sector.

- An untwisted operator $\mathcal{O}'_{e^k}$ of $\mathfrak{T}^{\mathcal{S}'}$ having charge $k = 1, 2$ under $\mathbb{Z}_3$ descends to two operators of $\mathfrak{T}^{\mathcal{S}}$: one of them being an untwisted operator $\mathcal{O}^{(k)}_{a_1, u}$ and the other operator $\mathcal{O}^{(k)}_{a_1, t}$ in $1_-$-twisted sector. These operators are uncharged under $1_-$, but mixed together by the action of $E$ as described in section 5.2.2 of [2]. These mixings imply that the linking action of $E$ on these operators is $-1$. We say that the two operators lie in an $a_1$-multiplet. We can recognize these operators as

$$\mathcal{O}^{(k)}_{a_1, u} = \mathcal{O}'_{e^k} \qquad \mathcal{O}^{(k)}_{a_1, t} = \mathcal{O}'_{e^k}, \qquad (\text{V.89})$$

where $\mathcal{O}^{(k)}_{a_1, t}$ is the operator $\mathcal{O}'_{e^k}$ viewed as lying in the $1_-$-twisted sector. The linking action of $E = P \oplus P^2$ is $-1$ because $\omega + \omega^2 = -1$.

- An operator $\mathcal{O}'_{m^k}$, for $k = 1, 2$, in $P^k$-twisted sector of $\mathfrak{T}^{\mathcal{S}'}$, where $P$ is the generator of $\mathbb{Z}_3$ symmetry before KT transformation, and uncharged under $\mathbb{Z}_3$ descends to two operators of $\mathfrak{T}^{\mathcal{S}}$: $\mathcal{O}^{(k)}_{E, +}$ in $E$-twisted sector and $\mathcal{O}^{(k)}_{E, -}$ sitting between the lines $E$ and $1_-$. We say that the two operators lie in an $E$-multiplet. These operators are uncharged under $1_-$, but mixed together by the action of $E$ as described in section 5.2.3 of [2]. We can recognize these operators as

$$\mathcal{O}^{(k)}_{E, +} = \mathcal{O}'_{m^k} \qquad \mathcal{O}^{(k)}_{E, -} = \mathcal{O}'_{m^k}, \qquad (\text{V.90})$$

where $\mathcal{O}^{(k)}_{E, \pm}$ is the operator $\mathcal{O}'_{m^k}$ viewed as lying at the end of $E = P \oplus P^2$ line, since it lies at the end of one of these lines, and the operator $\mathcal{O}^{(k)}_{E, -}$ is additionally regarded as attached also the $1_-$ line.

- An operator $\mathcal{O}'_{e^i m^k}$, for $i, k \in \{1, 2\}$, in $P^k$-twisted sector of $\mathfrak{T}^{\mathcal{S}'}$ and having charge $i$ under $\mathbb{Z}_3$ descends to two operators of $\mathfrak{T}^{\mathcal{S}}$: $\mathcal{O}^{(k)}_{a_{\omega^{ki}}, +}$ in $E$-twisted sector and $\mathcal{O}^{(k)}_{a_{\omega^{ki}}, -}$ sitting between the lines $E$ and $1_-$. These operators are uncharged under $1_-$, but mixed together by the action of $E$. We say that the two operators lie in an $a_{\omega^{ki}}$-multiplet. We can recognize these operators as

$$\mathcal{O}^{(k)}_{a_{\omega^{ki}}, +} = \mathcal{O}'_{e^i m^k} \qquad \mathcal{O}^{(k)}_{a_{\omega^{ki}}, +} = \mathcal{O}'_{e^i m^k}. \qquad (\text{V.91})$$

## I. KT from $\mathbb{Z}_2$ to $\mathsf{Rep}(S_3)$ Symmetry I

Consider the $\mathsf{Rep}(S_3)$ club quiche (IV.136) associated to the algebra $\mathcal{A}_E$ studied in section IV D 1. The associated KT transformation converts a $\mathbb{Z}_2$ symmetry into a $\mathsf{Rep}(S_3)$ symmetry

$$\mathcal{S}' = \mathsf{Vec}_{\mathbb{Z}_2} \quad \longrightarrow \quad \mathcal{S} = \mathsf{Rep}(S_3). \qquad (\text{V.92})$$

Let $\mathfrak{T}^{\mathcal{S}'}$ be a $\mathbb{Z}_2$-symmetric 2d QFT. Its KT transformation $\mathfrak{T}^{\mathcal{S}}$ is a $\mathsf{Rep}(S_3)$-symmetric 2d QFT which can be expressed as

$$\mathfrak{T}^{\mathcal{S}} = \mathfrak{T}^{\mathcal{S}'} \overset{\curvearrowleft}{} \mathsf{Rep}(S_3) \tag{V.93}$$

i.e. the underlying 2d QFT for $\mathfrak{T}^{\mathcal{S}}$ is the same as for $\mathfrak{T}^{\mathcal{S}'}$, with the $\mathsf{Rep}(S_3)$ symmetry realized in terms of the $\mathbb{Z}_2$ symmetry as described in (IV.139).

From the expression for the Lagrangian algebra $\mathcal{L}_{\mathcal{I}}$ in (IV.134) we observe that the map

$$\mathcal{Z}(\mathsf{Vec}_{\mathbb{Z}_2}) \to \mathcal{Z}(\mathsf{Rep}(S_3)) \qquad (\text{V.94})$$

of generalized charges is

$$1 \to 1 \oplus E, \quad m \to 1_- \oplus E, \quad e \to b_+, \quad em \to b_-. \tag{V.95}$$

According to this map:

- An untwisted operator $\mathcal{O}'$ of $\mathfrak{T}^{\mathcal{S}'}$ uncharged under $\mathbb{Z}_2$ descends to three operators of $\mathfrak{T}^{\mathcal{S}}$: one of them being an untwisted operator $\mathcal{O}$ that is uncharged under $\mathsf{Rep}(S_3)$, and the other two being operators $\mathcal{O}^{(0)}_{E, +}, \mathcal{O}^{(0)}_{E, -}$ lying in an $E$-multiplet of $\mathsf{Rep}(S_3)$ symmetry discussed above. We can recognize these operators as

$$\mathcal{O} = \mathcal{O}', \quad \mathcal{O}^{(0)}_{E, +} = \mathcal{O}', \quad \mathcal{O}^{(0)}_{E, -} = \mathcal{O}' \otimes \mathrm{id}_P, \quad (\text{V.96})$$

where $\mathcal{O}^{(0)}_{E, +}$ is the operator $\mathcal{O}'$ viewed as lying at the end of $E = 1 \oplus P$ line, since the identity line is a part of $E$, and the operator $\mathcal{O}^{(0)}_{E, -}$ is obtained by fusing $\mathcal{O}'$ with the identity operator $\mathrm{id}_P$ living along the topological line $P$ generating the $\mathbb{Z}_2$ symmetry before KT transformation. This operator $\mathcal{O}' \otimes \mathrm{id}_P$ can then be regarded as transitioning between $E$ and $1_-$ as both these lines involve the line $P$.

- A $P$-twisted sector operator $\mathcal{O}'_m$ of $\mathfrak{T}^{\mathcal{S}'}$ uncharged under $\mathbb{Z}_2$ descends to three operators of $\mathfrak{T}^{\mathcal{S}}$: $\mathcal{O}_{1_-}$ living in $1_-$-multiplet, and $\mathcal{O}^{(1)}_{E, +}, \mathcal{O}^{(1)}_{E, -}$ in $E$-multiplet of $\mathsf{Rep}(S_3)$ symmetry. We can recognize these operators as

$$\mathcal{O}_{1_-} = \mathcal{O}'_m, \quad \mathcal{O}^{(1)}_{E, +} = \mathcal{O}'_m, \quad \mathcal{O}^{(1)}_{E, -} = \mathcal{O}'_m. \quad (\text{V.97})$$

- An untwisted operator $\mathcal{O}'_e$ of $\mathfrak{T}^{\mathcal{S}'}$ charged under $\mathbb{Z}_2$ descends to three operators of $\mathfrak{T}^{\mathcal{S}}$: $\mathcal{O}_{b_+}$ which is untwisted, $\mathcal{O}_{b_+,+}$ lying in $E$-twisted sector, and $\mathcal{O}_{b_+,-}$ lying between $E$ and $1_-$ lines. The three operators are charged under $1_-$ and mixed together by $E$ as described in section 5.2.4 of [2]. We say that the three operators lie in a $b_+$-multiplet of $\mathsf{Rep}(S_3)$ symmetry. We can recognize these operators as

$$\mathcal{O}_{b_+} = \mathcal{O}'_e, \quad \mathcal{O}_{b_+,+} = \mathcal{O}'_e, \quad \mathcal{O}_{b_+,-} = \mathcal{O}'_e \otimes \mathrm{id}_P. \quad \text{(V.98)}$$

- A $P$-twisted sector operator $\mathcal{O}'_{em}$ of $\mathfrak{T}^{\mathcal{S}'}$ charged under $\mathbb{Z}_2$ descends to three operators of $\mathfrak{T}^{\mathcal{S}}$: $\mathcal{O}_{b_-}$ lying in $1_-$-twisted sector, $\mathcal{O}_{b_-,+}$ lying in $E$-twisted sector, and $\mathcal{O}_{b_-,-}$ lying between $E$ and $1_-$ lines. The three operators are charged under $1_-$ and mixed together by $E$. We say that the three operators lie in a $b_-$-multiplet of $\mathsf{Rep}(S_3)$ symmetry. We can recognize these operators as

$$\mathcal{O}_{b_-} = \mathcal{O}'_{em}, \quad \mathcal{O}_{b_-,+} = \mathcal{O}'_{em}, \quad \mathcal{O}_{b_-,-} = \mathcal{O}'_{em}. \quad \text{(V.99)}$$

### J. KT from $\mathbb{Z}_2$ to $\mathsf{Rep}(S_3)$ Symmetry II

Consider the $\mathsf{Rep}(S_3)$ club quiche (IV.153) associated to the algebra $\mathcal{A}_{a_1}$ studied in section IV D 3. The associated KT transformation converts a $\mathbb{Z}_2$ symmetry into a $\mathsf{Rep}(S_3)$ symmetry

$$\mathcal{S}' = \mathsf{Vec}_{\mathbb{Z}_2} \quad \longrightarrow \quad \mathcal{S} = \mathsf{Rep}(S_3). \quad \text{(V.100)}$$

Let $\mathfrak{T}^{\mathcal{S}'}$ be a $\mathbb{Z}_2$-symmetric 2d QFT. Let's understand its KT transformation $\mathfrak{T}^{\mathcal{S}}$ which is a $\mathsf{Rep}(S_3)$-symmetric 2d QFT.

The boundary $\mathfrak{B}'$ obtained after compactifying the interval occupied by $\mathfrak{Z}(\mathcal{S})$ is

$$\mathfrak{B}' = \mathfrak{B}^{\mathrm{sym}}_{\mathcal{S}'} \oplus (\mathfrak{B}^{\mathrm{sym}}_{\mathcal{S}'}/\mathbb{Z}_2)_{\sqrt{2}} \quad \text{(V.101)}$$

as in (IV.149), where $\mathfrak{B}^{\mathrm{sym}}_{\mathcal{S}'}/\mathbb{Z}_2$ is the boundary of $\mathfrak{Z}(\mathcal{S}')$ obtained by gauging the $\mathbb{Z}_2$ symmetry of the boundary $\mathfrak{B}^{\mathrm{sym}}_{\mathcal{S}'}$ and the subscript $\sqrt{2}$ describes that we additionally stack $\mathfrak{B}^{\mathrm{sym}}_{\mathcal{S}'}/\mathbb{Z}_2$ with a non-trivial Euler term. The 2d QFT underlying $\mathfrak{T}^{\mathcal{S}}$ is thus

$$\mathfrak{T}^{\mathcal{S}} = \mathfrak{T}^{\mathcal{S}'} \oplus (\mathfrak{T}^{\mathcal{S}'}/\mathbb{Z}_2)_{\sqrt{2}}, \quad \text{(V.102)}$$

where $(\mathfrak{T}^{\mathcal{S}'}/\mathbb{Z}_2)_{\sqrt{2}}$ is obtained by gauging the $\mathcal{S}' = \mathbb{Z}_2$ symmetry of $\mathfrak{T}^{\mathcal{S}'}$ and stacking an Euler term. For further proceedings, let us define

$$\mathfrak{T}_e := \mathfrak{T}^{\mathcal{S}'}, \qquad \mathfrak{T}_m := (\mathfrak{T}^{\mathcal{S}'}/\mathbb{Z}_2)_{\sqrt{2}}. \quad \text{(V.103)}$$

In other words, $\mathfrak{T}^{\mathcal{S}}$ comprises of two universes described respectively by theories $\mathfrak{T}_e$ and $\mathfrak{T}_m$.

The lines living on $\mathfrak{B}'$ are described in (IV.150). These lines descend to line operators of $\mathfrak{T}^{\mathcal{S}}$ as follows. The line $1_{ii}$ becomes the identity line in $\mathfrak{T}_i$, while the line $P_{ii}$

becomes the $\mathbb{Z}_2$ symmetry generator of $\mathfrak{T}_i$. Here $P_{ee}$ is the generator of the original $\mathbb{Z}_2$ symmetry of $\mathfrak{T}^{\mathcal{S}'}$, while $P_{mm}$ is the generator of the dual $\mathbb{Z}_2$ symmetry of $\mathfrak{T}^{\mathcal{S}'}/\mathbb{Z}_2$ obtained after gauging the $\mathbb{Z}_2$ symmetry of $\mathfrak{T}^{\mathcal{S}'}$. The lines $S_{em}$ and $S_{me}$ are universe changing lines. Their linking actions on identity local operators $\mathrm{id}_e$ and $\mathrm{id}_m$ of $\mathfrak{T}_e$ and $\mathfrak{T}_m$ respectively are

$$S_{em}: \ \mathrm{id}_e \to 2\mathrm{id}_m \qquad S_{me}: \ \mathrm{id}_m \to \mathrm{id}_e, \quad \text{(V.104)}$$

which follows from (IV.159).

The $\mathsf{Rep}(S_3)$ symmetry generators are identified in terms of these lines as in (IV.158), and thus we can schematically depict the $\mathsf{Rep}(S_3)$-symmetric theory $\mathfrak{T}^{\mathcal{S}}$ as

$$\mathfrak{T}^{\mathcal{S}} \quad = \quad E \ \underset{E}{\underbrace{\circlearrowright \ \mathfrak{T}^{\mathcal{S}'} \oplus (\mathfrak{T}^{\mathcal{S}'}/\mathbb{Z}_2)_{\sqrt{2}} \ \circlearrowleft}} \ 1_-$$
$$\text{(V.105)}$$

Here $E$ acts as $P_{ee}$ as well as between the two summands as described in (IV.158). From the expression for the Lagrangian algebra $\mathcal{L}_{\mathcal{I}}$ in (IV.122) we observe that the map

$$\mathcal{Z}(\mathsf{Vec}_{\mathbb{Z}_2}) \to \mathcal{Z}(\mathsf{Rep}(S_3)) \quad \text{(V.106)}$$

of generalized charges is

$$1 \to 1 \oplus a_1, \quad e \to 1_- \oplus a_1, \quad m \to b_+, \quad em \to b_-. \quad \text{(V.107)}$$

According to this map:

- An untwisted operator $\mathcal{O}'$ of $\mathfrak{T}^{\mathcal{S}'}$ uncharged under $\mathbb{Z}_2$ descends to three operators of $\mathfrak{T}^{\mathcal{S}}$: one of them being an untwisted operator $\mathcal{O}_1$ that is uncharged under $\mathsf{Rep}(S_3)$, and the other two being operators $\mathcal{O}^{(0)}_{a_1,u}, \mathcal{O}^{(0)}_{a_1,t}$ lying in an $a_1$-multiplet of $\mathsf{Rep}(S_3)$ symmetry discussed above. We can recognize these operators as

$$\mathcal{O}_1 = (\mathcal{O}')_e + (\mathcal{O}')_m$$
$$\mathcal{O}^{(0)}_{a_1,u} = (\mathcal{O}')_e - 2(\mathcal{O}')_m \quad \text{(V.108)}$$
$$\mathcal{O}^{(0)}_{a_1,t} = -\frac{3(\mathcal{O}')_e}{1+2\omega},$$

where $\omega$ is a third root of unity. Here $(\mathcal{O}')_e$ is a copy of $\mathcal{O}'$ in $\mathfrak{T}_e$ and $(\mathcal{O}')_m$ is the operator in $\mathfrak{T}_m$ obtained after gauging $\mathbb{Z}_2$. Both $(\mathcal{O}')_e$ and $(\mathcal{O}')_m$ are uncharged under the $\mathbb{Z}_2$ symmetries of $\mathfrak{T}_e$ and $\mathfrak{T}_m$ respectively. The form of $\mathcal{O}^{(0)}_{a_1,u}$ follows from the identification of operator $\mathcal{O}$ in terms of $v_e$ and $v_m$ in (IV.155) since we have that $\mathcal{O}^{(0)}_{a_1,u}$ is the fusion $\mathcal{O} \otimes \mathcal{O}_1$. The form of the operator $\mathcal{O}^{(0)}_{a_1,t}$ results from a particular choice of junction local operators between $\mathsf{Rep}(S_3)$ lines and was derived in eqn. (5.94) of [2]. Note that the operator $\mathcal{O}^{(0)}_{a_1,t}$ can be regarded as living in the $1_-$-twisted sector because $1_-$ contains the identity line $1_{ee}$ for $\mathfrak{T}_e$.

- An untwisted operator $\mathcal{O}'_e$ of $\mathfrak{T}^{\mathcal{S}'}$ charged under $\mathbb{Z}_2$ descends to three operators of $\mathfrak{T}^{\mathcal{S}}$: one of them $\mathcal{O}_{1_-}$ living in $1_-$-multiplet of $\mathsf{Rep}(S_3)$, and the other two being operators $\mathcal{O}^{(1)}_{a_1,u}, \mathcal{O}^{(1)}_{a_1,t}$ lying in an $a_1$-multiplet of $\mathsf{Rep}(S_3)$ symmetry discussed above. We can recognize these operators as

$$\mathcal{O}_{1_-} = (\mathcal{O}'_e)_e + (\mathcal{O}'_e)_m$$
$$\mathcal{O}^{(1)}_{a_1,u} = -\frac{3(\mathcal{O}'_e)_e}{1 + 2\omega} \qquad (V.109)$$
$$\mathcal{O}^{(1)}_{a_1,t} = (\mathcal{O}'_e)_e - 2(\mathcal{O}'_e)_m \,.$$

  Here $(\mathcal{O}'_e)_e$ is a copy of $\mathcal{O}'_e$ in $\mathfrak{T}_e$ and $(\mathcal{O}'_e)_m$ is the operator in $\mathfrak{T}_m$ obtained after gauging $\mathbb{Z}_2$, which is thus in $P_{mm}$ twisted sector and uncharged under $P_{mm}$. These operators are uncharged under $1_-$ as it involves only $P_{mm}$ but $(\mathcal{O}'_e)_m$ is uncharged under $P_{mm}$. Note that the twisted and untwisted components of the $a_1$-multiplet $\{\mathcal{O}^{(1)}_{a_1,u}, \mathcal{O}^{(1)}_{a_1,t}\}$ are "reversed" as compared to the twisted and untwisted components of the $a_1$-multiplet $\{\mathcal{O}^{(0)}_{a_1,u}, \mathcal{O}^{(0)}_{a_1,t}\}$ described in (V.108). This is because we now have $\mathcal{O}^{(1)}_{a_1,t} = \mathcal{O} \otimes \mathcal{O}_{1_-}$ where $\mathcal{O}$ is the topological local operator appearing in (IV.155), as opposed to $\mathcal{O}^{(0)}_{a_1,t} = \mathcal{O} \otimes \mathcal{O}_1$ previously.

- A $P$-twisted sector operator $\mathcal{O}'_m$ of $\mathfrak{T}^{\mathcal{S}'}$, where $P$ is the generator of $\mathbb{Z}_2$ symmetry of $\mathfrak{T}^{\mathcal{S}'}$, and uncharged under $\mathbb{Z}_2$ descends to three operators of $\mathfrak{T}^{\mathcal{S}}$: $\mathcal{O}_{b_+}, \mathcal{O}_{b_+,+}, \mathcal{O}_{b_+,-}$ transforming in $b_+$-multiplet of $\mathsf{Rep}(S_3)$ symmetry discussed above. We can recognize these operators as

$$\mathcal{O}_{b_+} = (\mathcal{O}'_m)_m\,, \quad \mathcal{O}_{b_+,+} = (\mathcal{O}'_m)_e\,, \quad \mathcal{O}_{b_+,-} = (\mathcal{O}'_m)_e\,. \qquad (V.110)$$

  Here $(\mathcal{O}'_m)_e$ is a copy of $\mathcal{O}'_m$ in $\mathfrak{T}_e$ and $(\mathcal{O}'_m)_m$ is the operator in $\mathfrak{T}_m$ obtained after gauging $\mathbb{Z}_2$, which is thus an untwisted operator in $\mathfrak{T}_m$ charged under $P_{mm}$.

- A $P$-twisted sector operator $\mathcal{O}'_{em}$ of $\mathfrak{T}^{\mathcal{S}'}$ charged under $\mathbb{Z}_2$ descends to three operators of $\mathfrak{T}^{\mathcal{S}}$: $\mathcal{O}_{b_-}, \mathcal{O}_{b_-,+}, \mathcal{O}_{b_-,-}$ transforming in $b_-$-multiplet of $\mathsf{Rep}(S_3)$ symmetry discussed above. We can recognize these operators as

$$\mathcal{O}_{b_-} = (\mathcal{O}'_{em})_m\,, \quad \mathcal{O}_{b_+,+} = (\mathcal{O}'_{em})_e\,, \quad \mathcal{O}_{b_+,-} = (\mathcal{O}'_{em})_e \qquad (V.111)$$

  Here $(\mathcal{O}'_{em})_e$ is a copy of $\mathcal{O}'_{em}$ in $\mathfrak{T}_e$ and $(\mathcal{O}'_{em})_m$ is the operator in $\mathfrak{T}_m$ obtained after gauging $\mathbb{Z}_2$, which is thus a $P_{mm}$-twisted operator in $\mathfrak{T}_m$ charged under $P_{mm}$.

### K. KT from $\mathbb{Z}_2$ to Ising Symmetry

Consider the Ising club quiche (IV.169) associated to the algebra $\mathcal{A}$ studied in section IV E. The associated KT transformation converts $\mathbb{Z}_2$ symmetry into Ising symmetry

$$\mathcal{S}' = \mathsf{Vec}_{\mathbb{Z}_2} \quad \longrightarrow \quad \mathcal{S} = \mathsf{Ising}\,. \qquad (V.112)$$

Let $\mathfrak{T}^{\mathcal{S}'}$ be a $\mathbb{Z}_2$-symmetric 2d QFT. Let's understand its KT transformation $\mathfrak{T}^{\mathcal{S}}$ which is an Ising-symmetric 2d QFT.

The boundary $\mathfrak{B}'$ obtained after compactifying the interval occupied by $\mathfrak{Z}(\mathcal{S})$ is

$$\mathfrak{B}' = \mathfrak{B}^{\mathrm{sym}}_{\mathcal{S}'} \oplus (\mathfrak{B}^{\mathrm{sym}}_{\mathcal{S}'}/\mathbb{Z}_2) \qquad (V.113)$$

as in (IV.168), where $\mathfrak{B}^{\mathrm{sym}}_{\mathcal{S}'}/\mathbb{Z}_2$ is the boundary of $\mathfrak{Z}(\mathcal{S}')$ obtained by gauging the $\mathbb{Z}_2$ symmetry of the boundary $\mathfrak{B}^{\mathrm{sym}}_{\mathcal{S}'}$. The 2d QFT underlying $\mathfrak{T}^{\mathcal{S}}$ is thus

$$\mathfrak{T}^{\mathcal{S}} = \mathfrak{T}^{\mathcal{S}'} \oplus (\mathfrak{T}^{\mathcal{S}'}/\mathbb{Z}_2)\,, \qquad (V.114)$$

where $\mathfrak{T}^{\mathcal{S}'}/\mathbb{Z}_2$ is obtained by gauging the $\mathcal{S}' = \mathbb{Z}_2$ symmetry of $\mathfrak{T}^{\mathcal{S}'}$. For further proceedings, let us define

$$\mathfrak{T}_e := \mathfrak{T}^{\mathcal{S}'}\,, \qquad \mathfrak{T}_m := \mathfrak{T}^{\mathcal{S}'}/\mathbb{Z}_2\,. \qquad (V.115)$$

In other words, $\mathfrak{T}^{\mathcal{S}}$ comprises of two universes described respectively by theories $\mathfrak{T}_e$ and $\mathfrak{T}_m$.

The lines living on $\mathfrak{B}'$ and their map to lines of $\mathfrak{T}^{\mathcal{S}}$ are the same as in section V J. However, the linking actions on identity local operators $\mathrm{id}_e$ and $\mathrm{id}_m$ of $\mathfrak{T}_e$ and $\mathfrak{T}_m$ respectively of the universe changing lines are now

$$S_{em}: \ \mathrm{id}_e \to \sqrt{2}\mathrm{id}_m \qquad S_{me}: \ \mathrm{id}_m \to \sqrt{2}\mathrm{id}_e\,, \quad (V.116)$$

which follows from (IV.175).

The Ising symmetry generators are identified in terms of these lines as in (IV.174), and thus we can schematically depict the Ising-symmetric theory $\mathfrak{T}^{\mathcal{S}}$ as

$$\mathfrak{T}^{\mathcal{S}} \quad = \quad P \ \substack{\circlearrowleft} \ \underbrace{\mathfrak{T}^{\mathcal{S}'} \oplus (\mathfrak{T}^{\mathcal{S}'}/\mathbb{Z}_2)}_{S} \ \substack{\circlearrowright} \ P \qquad (V.117)$$

From the expression for the Lagrangian algebra $\mathcal{L}_{\mathcal{I}}$ in (IV.166) we observe that the map

$$\mathcal{Z}(\mathsf{Vec}_{\mathbb{Z}_2}) \to \mathcal{Z}(\mathsf{Ising}) \qquad (V.118)$$

of generalized charges is

$$1 \to 1 \oplus P\overline{P}, \quad e \to S\overline{S}, \quad m \to S\overline{S}, \quad em \to P \oplus \overline{P}\,. \qquad (V.119)$$

According to this map:

- An untwisted operator $\mathcal{O}'$ of $\mathfrak{T}^{\mathcal{S}'}$ uncharged under $\mathbb{Z}_2$ descends to two operators of $\mathfrak{T}^{\mathcal{S}}$: one of them being an untwisted operator $\mathcal{O}$ that is uncharged under Ising, and the other being an operator $\mathcal{O}_{P\overline{P}}$ which transforms trivially under $P$ but transforms

by a sign whenever the line $S$ is moved past it. We can recognize these operators as

$$\mathcal{O} = (\mathcal{O}')_e + (\mathcal{O}')_m, \quad \mathcal{O}_{P\overline{P}} = (\mathcal{O}')_e - (\mathcal{O}')_m. \quad \text{(V.120)}$$

Here $(\mathcal{O}')_e$ is a copy of $\mathcal{O}'$ in $\mathfrak{T}_e$ and $(\mathcal{O}')_m$ is the operator in $\mathfrak{T}_m$ obtained after gauging $\mathbb{Z}_2$. Both $(\mathcal{O}')_e$ and $(\mathcal{O}')_m$ are uncharged under the $\mathbb{Z}_2$ symmetries of $\mathfrak{T}_e$ and $\mathfrak{T}_m$ respectively.

- An untwisted operator $\mathcal{O}'_e$ of $\mathfrak{T}^{\mathcal{S}'}$ charged under $\mathbb{Z}_2$ descends to two operators of $\mathfrak{T}^{\mathcal{S}}$: one of them being an untwisted operator $\mathcal{O}^{(e)}_{S\overline{S},u}$ that is charged under $P$, and the other being $P$-twisted sector operator $\mathcal{O}^{(e)}_{S\overline{S},t}$ uncharged under $P$. The two operators are exchanged by the action of $S$. We say that the two operators lie in an $S\overline{S}$-multiplet of Ising symmetry. We can recognize these operators as

$$\mathcal{O}^{(e)}_{S\overline{S},u} = (\mathcal{O}'_e)_e, \quad \mathcal{O}^{(e)}_{S\overline{S},t} = (\mathcal{O}'_e)_m. \quad \text{(V.121)}$$

Here $(\mathcal{O}'_e)_e$ is a copy of $\mathcal{O}'_e$ in $\mathfrak{T}_e$ and $(\mathcal{O}'_e)_m$ is the operator in $\mathfrak{T}_m$ obtained after gauging $\mathbb{Z}_2$, which is thus in $P_{mm}$ twisted sector and uncharged under $P_{mm}$.

- A $P'$-twisted operator $\mathcal{O}'_m$ of $\mathfrak{T}^{\mathcal{S}'}$, where $P'$ is the generator of $\mathbb{Z}_2$ symmetry of $\mathfrak{T}^{\mathcal{S}'}$ before KT transformation, and which is uncharged under $\mathbb{Z}_2$, descends to two operators of $\mathfrak{T}^{\mathcal{S}}$: $\mathcal{O}^{(m)}_{S\overline{S},u}, \mathcal{O}^{(m)}_{S\overline{S},t}$ lying in $S\overline{S}$-multiplet of Ising symmetry. We can recognize these operators as

$$\mathcal{O}^{(m)}_{S\overline{S},u} = (\mathcal{O}'_m)_m, \quad \mathcal{O}^{(m)}_{S\overline{S},t} = (\mathcal{O}'_m)_e. \quad \text{(V.122)}$$

Here $(\mathcal{O}'_m)_e$ is a copy of $\mathcal{O}'_m$ in $\mathfrak{T}_e$ and $(\mathcal{O}'_m)_m$ is the operator in $\mathfrak{T}_m$ obtained after gauging $\mathbb{Z}_2$, which is thus an untwisted operator in $\mathfrak{T}_m$ charged under $P_{mm}$.

- A $P'$-twisted operator $\mathcal{O}'_{em}$ of $\mathfrak{T}^{\mathcal{S}'}$ charged under $\mathbb{Z}_2$ descends to two operators of $\mathfrak{T}^{\mathcal{S}}$: a $P$-twisted sector operator $\mathcal{O}_P$ charged under $P$ and transforming by $i$ whenever the line $S$ is moved past it, and a $P$-twisted sector operator $\mathcal{O}_{\overline{P}}$ charged under $P$ and transforming by $-i$ whenever the line $S$ is moved past it. We can recognize these operators as

$$\mathcal{O}_P = (\mathcal{O}'_{em})_e - i(\mathcal{O}'_{em})_m, \quad \mathcal{O}_{\overline{P}} = (\mathcal{O}'_{em})_e + i(\mathcal{O}'_{em})_m. \quad \text{(V.123)}$$

Here $(\mathcal{O}'_{em})_e$ is a copy of $\mathcal{O}'_{em}$ in $\mathfrak{T}_e$ and $(\mathcal{O}'_{em})_m$ is the operator in $\mathfrak{T}_m$ obtained after gauging $\mathbb{Z}_2$, which is thus a $P_{mm}$-twisted operator in $\mathfrak{T}_m$ charged under $P_{mm}$. The action of $S$ on these operators follows from the facts that moving $S_{em}$ past $(\mathcal{O}'_{em})_e$ converts it into $(\mathcal{O}'_{em})_m$, and moving $S_{me}$ past $(\mathcal{O}'_{em})_m$ converts it into $-(\mathcal{O}'_{em})_e$.

## L. KT from $\mathbb{Z}_2$ to $\mathsf{TY}(\mathbb{Z}_4)$ Symmetry

Consider the club quiche (IV.187) for the algebra $\mathcal{A}_{1,2}$ discussed in section IV F 1. The associated KT transformation converts $\mathbb{Z}_2$ symmetry into $\mathsf{TY}(\mathbb{Z}_4)$ symmetry

$$\mathcal{S}' = \mathsf{Vec}_{\mathbb{Z}_2} \to \mathcal{S} = \mathsf{TY}(\mathbb{Z}_4). \quad \text{(V.124)}$$

Let $\mathfrak{T}^{\mathcal{S}'}$ be a $\mathbb{Z}_2$-symmetric 2d QFT. We want to understand the properties of its KT transformation $\mathfrak{T}^{\mathcal{S}}$ which is a $\mathsf{TY}(\mathbb{Z}_4)$-symmetric 2d QFT.

The boundary $\mathfrak{B}'$ after interval compactification of $\mathfrak{Z}(\mathcal{S})$ is

$$\mathfrak{B}' = \mathfrak{B}^{\text{sym}}_{\mathcal{S}'} \oplus (\mathfrak{B}^{\text{sym}}_{\mathcal{S}'}/\mathbb{Z}_2)_{\sqrt{2}} \oplus (\mathfrak{B}^{\text{sym}}_{\mathcal{S}'}/\mathbb{Z}_2)_{\sqrt{2}} \quad \text{(V.125)}$$

and the resulting theory $\mathfrak{T}^{\mathcal{S}}$ decomposes as

$$\mathfrak{T}^{\mathcal{S}} = \mathfrak{T}^e_0 \oplus \mathfrak{T}^m_1 \oplus \mathfrak{T}^m_2, \quad \text{(V.126)}$$

where

$$\mathfrak{T}^e_0 = \mathfrak{T}^{\mathcal{S}'}, \qquad \mathfrak{T}^m_1 = \mathfrak{T}^m_2 = (\mathfrak{T}^{\mathcal{S}'}/\mathbb{Z}_2)_{\sqrt{2}}, \quad \text{(V.127)}$$

where we remind the reader that $\sqrt{2}$ subscripts capture the presence of additional Euler terms.

The action of the symmetry $\mathcal{S} = \mathsf{TY}(\mathbb{Z}_4)$ follows from (IV.194) and we can schematically depict $\mathfrak{T}^{\mathcal{S}}$ with $\mathsf{TY}(\mathbb{Z}_4)$ action as

$$\mathfrak{T}^{\mathcal{S}} = \mathfrak{T}^e_0 \oplus \mathfrak{T}^m_1 \oplus \mathfrak{T}^m_2$$

with $S$ above and $A$, $A$ below. $\quad \text{(V.128)}$

From the Lagrangian $\mathcal{L}_{1,2}$ we observe that the map

$$\mathcal{Z}(\mathsf{Vec}_{\mathbb{Z}_2}) \to \mathcal{Z}(\mathsf{TY}(\mathbb{Z}_4)) \quad \text{(V.129)}$$

of generalized charges is

$$\begin{aligned} 1 &\to L^+_0 + L^-_0 + L_{2,0} \\ e &\to L^+_2 + L^-_2 + L_{2,0} \\ m &\to L_{1,0} + L_{3,0} \\ em &\to L_{2,1} + L_{3,2}. \end{aligned} \quad \text{(V.130)}$$

These imply the following maps on operators:

- An untwisted operator $\mathcal{O}'$ of $\mathfrak{T}^{\mathcal{S}'}$ uncharged under $\mathbb{Z}_2$ maps to four operators of $\mathfrak{T}^{\mathcal{S}}$. One of them is an untwisted operator $\mathcal{O}$ that is uncharged under $\mathsf{TY}(\mathbb{Z}_4)$. Another is an untwisted operator $\mathcal{O}^-_0$ with generalized charge $L^-_0$ which is uncharged under $\mathbb{Z}_4$ subsymmetry but picks up a sign when moved past the duality defect $S$. We call such an operator as living in the $L^-_0$ multiplet. The remaining two operators $\{\mathcal{O}^{u,(1)}_{2,0}, \mathcal{O}^{t,(1)}_{2,0}\}$ lie in an $L_{2,0}$-multiplet.

In more detail, $\mathcal{O}_{2,0}^{u,(1)}$ is an untwisted operator of charge 2 under the $\mathbb{Z}_4$ subsymmetry, and $\mathcal{O}_{2,0}^{t,(1)}$ is an $A^2$-twisted sector operator uncharged under $\mathbb{Z}_4$, such that the two operators are exchanged by the action of $S$. See sections 7.1.1. and 7.1.3. in [2] for further details. To conclude, the action of the $\mathsf{TY}(\mathbb{Z}_4)$ symmetry on these operators is

$$A^k : \mathcal{O}_0^- \to \mathcal{O}_0^-, \quad \mathcal{O}_{2,0}^{u,(1)} \to (-1)^k \mathcal{O}_{2,0}^{u,(1)}, \quad \mathcal{O}_{2,0}^{t,(1)} \to \mathcal{O}_{2,0}^{t,(1)}$$
$$S : \mathcal{O}_0^- \to -\mathcal{O}_0^-, \quad \mathcal{O}_{2,0}^{u,(1)} \leftrightarrow \mathcal{O}_{2,0}^{t,(1)}. \tag{V.131}$$

This is simply the action upon moving past the lines, which is in general different from the linking action of lines on these operators.

We can recognize these operators as

$$\mathcal{O} = (\mathcal{O}')_0^e + (\mathcal{O}')_1^m + (\mathcal{O}')_2^m$$
$$\mathcal{O}_0^- = (\mathcal{O}')_0^e - (\mathcal{O}')_1^m - (\mathcal{O}')_2^m$$
$$\mathcal{O}_{2,0}^{u,(1)} = (\mathcal{O}')_1^m - (\mathcal{O}')_2^m \tag{V.132}$$
$$\mathcal{O}_{2,0}^{t,(1)} = (\mathcal{O}')_0^e,$$

where $(\mathcal{O}')_0^e$ is a copy of $\mathcal{O}'$ in $\mathfrak{T}_0^e$ and $(\mathcal{O}')_i^m$ are $(\mathcal{O}')^m$ in $\mathfrak{T}_i^m$, where $(\mathcal{O}')^m$ is the operator obtained from $\mathcal{O}'$ after gauging $\mathbb{Z}_2$, which is an untwisted operator uncharged under the dual $\mathbb{Z}_2$ symmetry. The sign in $\mathcal{O}_{2,0}^{u,(1)}$ is a result of the presence of non-trivial junction local operators between symmetry generators.

- An untwisted operator $\mathcal{O}'_e$ of $\mathfrak{T}^{\mathcal{S}'}$ charged under $\mathbb{Z}_2$ maps to two operators $\mathcal{O}_2^\pm$ in the $L_2^\pm$ multiplet of $\mathsf{TY}(\mathbb{Z}_4)$ and two operators $\mathcal{O}_{2,0}^{u,(2)}, \mathcal{O}_{2,0}^{t,(2)}$ in $L_{2,0}$ multiplet. The operators $\mathcal{O}_2^\pm$ are in $A^2$-twisted sector and transform as

$$A^k : \mathcal{O}_2^\pm \to (-1)^k \mathcal{O}_2^\pm$$
$$S : \mathcal{O}_2^\pm \to \pm \mathcal{O}_2^\pm, \tag{V.133}$$

when these lines are moved past them. We can recognize these operators as

$$\mathcal{O}_2^+ = (\mathcal{O}'_e)_0^e + (\mathcal{O}'_e)_1^m - (\mathcal{O}'_e)_2^m$$
$$\mathcal{O}_2^- = (\mathcal{O}'_e)_0^e - (\mathcal{O}'_e)_1^m + (\mathcal{O}'_e)_2^m$$
$$\mathcal{O}_{2,0}^{u,(2)} = (\mathcal{O}'_e)_0^e \tag{V.134}$$
$$\mathcal{O}_{2,0}^{t,(2)} = (\mathcal{O}'_e)_1^m + (\mathcal{O}'_e)_2^m,$$

where $(\mathcal{O}'_e)_0^e$ is a copy of $\mathcal{O}'_e$ in $\mathfrak{T}_0^e$ and $(\mathcal{O}'_e)_i^m$ are $(\mathcal{O}'_e)^m$ in $\mathfrak{T}_i^m$, where $(\mathcal{O}'_e)^m$ is the operator obtained from $\mathcal{O}'_e$ after gauging $\mathbb{Z}_2$, which is an operator in twisted sector for the dual $\mathbb{Z}_2$ symmetry and uncharged under the dual $\mathbb{Z}_2$ symmetry.

- A $P'$-twisted sector operator $\mathcal{O}'_m$ of $\mathfrak{T}^{\mathcal{S}'}$, where $P'$ is the generator of $\mathbb{Z}_2$ symmetry before KT transformation, that is uncharged under $\mathbb{Z}_2$, maps to two

operators $\mathcal{O}_{1,0}^i$ and two operators $\mathcal{O}_{3,0}^i$ for $i \in \{u, t\}$ of $\mathfrak{T}^{\mathcal{S}}$, transforming in the $L_{1,0}$ and $L_{3,0}$ multiplets, respectively. Here $\mathcal{O}_{e,0}^u$ are untwisted operators and $\mathcal{O}_{e,0}^t$ are in $A^e$-twisted sector such that the action of the $\mathcal{S} = \mathsf{TY}(\mathbb{Z}_4)$ is [2]

$$A^k : \mathcal{O}_{e,0}^u \to e^{i\pi ek/2} \mathcal{O}_{e,0}^u, \quad \mathcal{O}_{e,0}^t \to \mathcal{O}_{e,0}^t,$$
$$S : \mathcal{O}_{e,0}^u \leftrightarrow \mathcal{O}_{e,0}^t, \tag{V.135}$$

when these lines are moved past the local operators. We can recognize these local operators as

$$\mathcal{O}_{1,0}^u = (\mathcal{O}'_m)_1^m - i(\mathcal{O}'_m)_2^m$$
$$\mathcal{O}_{1,0}^t = (\mathcal{O}'_m)_0^e$$
$$\mathcal{O}_{3,0}^u = (\mathcal{O}'_m)_1^m + i(\mathcal{O}'_m)_2^m \tag{V.136}$$
$$\mathcal{O}_{3,0}^t = (\mathcal{O}'_m)_0^e,$$

where $(\mathcal{O}'_m)_0^e$ is a copy of $\mathcal{O}'_m$ in $\mathfrak{T}_0^e$ and $(\mathcal{O}'_m)_i^m$ are $(\mathcal{O}'_m)^m$ in $\mathfrak{T}_i^m$, where $(\mathcal{O}'_m)^m$ is the operator obtained from $\mathcal{O}'_m$ after gauging $\mathbb{Z}_2$, which is an operator in untwisted sector for the dual $\mathbb{Z}_2$ symmetry and charged under the dual $\mathbb{Z}_2$ symmetry.

- Finally, a $P'$-twisted sector operator $\mathcal{O}'_{em}$ of $\mathfrak{T}^{\mathcal{S}'}$ that is charged under $\mathbb{Z}_2$ maps to two operators $\mathcal{O}_{2,1}^i$ and two operators $\mathcal{O}_{3,2}^i$ for $i \in \{1, 2\}$ of $\mathfrak{T}^{\mathcal{S}}$, transforming in the $L_{2,1}$ and $L_{3,2}$ multiplets, respectively. Here $\mathcal{O}_{e,m}^1$ are operators $A^m$-twisted sectors and $\mathcal{O}_{e,m}^2$ are operators in $A^e$-twisted sectors such that the action of the $\mathcal{S} = \mathsf{TY}(\mathbb{Z}_4)$ is [2]

$$A^k : \mathcal{O}_{e,m}^1 \to e^{2\pi iek/4} \mathcal{O}_{e,m}^1, \quad \mathcal{O}_{e,m}^2 \to e^{2\pi imk/4} \mathcal{O}_{e,m}^2$$
$$S : \mathcal{O}_{e,m}^1 \leftrightarrow \mathcal{O}_{e,m}^2, \tag{V.137}$$

when these lines are moved past the local operators. We can recognize these local operators as

$$\mathcal{O}_{2,1}^1 = (\mathcal{O}'_{em})_0^e$$
$$\mathcal{O}_{2,1}^2 = (\mathcal{O}'_{em})_1^m - i(\mathcal{O}'_m)_2^m$$
$$\mathcal{O}_{3,2}^1 = (\mathcal{O}'_{em})_1^m + i(\mathcal{O}'_{em})_2^m \tag{V.138}$$
$$\mathcal{O}_{3,2}^2 = (\mathcal{O}'_{em})_0^e,$$

where $(\mathcal{O}'_{em})_0^e$ is a copy of $\mathcal{O}'_{em}$ in $\mathfrak{T}_0^e$ and $(\mathcal{O}'_{em})_i^m$ are $(\mathcal{O}'_{em})^m$ in $\mathfrak{T}_i^m$, where $(\mathcal{O}'_{em})^m$ is the operator obtained from $\mathcal{O}'_{em}$ after gauging $\mathbb{Z}_2$, which is an operator in twisted sector for the dual $\mathbb{Z}_2$ symmetry and charged under the dual $\mathbb{Z}_2$ symmetry.

## VI. PHASE TRANSITIONS: NEW FROM OLD

One of the key applications of the club sandwich constructions, i.e. the KT transformations, is the study of phase transitions between gapped phases with categorical symmetries. This comprises a central aspect of the categorical Landau paradigm that was developed in [1, 2]. In

there the main result was the characterization of gapped phases with categorical symmetries and their associated order parameters using the SymTFT.

For group-like symmetries this was discussed in [42] using a similar SymTFT description. Here we extend this to include any categorical symmetry, including non-invertible symmetries such as $\mathsf{Rep}(S_3)$ and $\mathsf{TY}(\mathbb{Z}_N)$.

### A. General Setup

Suppose that we know an irreducible $\mathcal{S}'$-symmetric $d$-dimensional CFT $\mathfrak{T}_C^{\mathcal{S}'}$ (note that unlike the earlier parts of the paper, the subscripts will not refer to space-time dimensions anymore) admitting a relevant operator $\mathcal{O}'$ that is uncharged under $\mathcal{S}'$ (sometimes also referred to as an $\mathcal{S}'$-symmetric local operator), such that deforming the CFT with $+\mathcal{O}'$ leads to an irreducible $\mathcal{S}'$-symmetric $d$-dimensional TQFT $\mathfrak{T}_1^{\mathcal{S}'}$, and deforming the CFT with $-\mathcal{O}'$ leads to another irreducible $\mathcal{S}'$-symmetric $d$-dimensional TQFT $\mathfrak{T}_2^{\mathcal{S}'}$

$$\mathfrak{T}_2^{\mathcal{S}'} \xleftarrow{\quad -\mathcal{O}' \quad} \mathfrak{T}_C^{\mathcal{S}'} \xrightarrow{\quad +\mathcal{O}' \quad} \mathfrak{T}_1^{\mathcal{S}'} \tag{VI.1}$$

In such a situation, $\mathfrak{T}_C^{\mathcal{S}'}$ is referred to as a **phase transition** between the $\mathcal{S}'$-symmetric gapped phases $[\mathfrak{T}_1^{\mathcal{S}'}]$ and $[\mathfrak{T}_2^{\mathcal{S}'}]$.

By applying a minimal KT transformation $\mathcal{K}_{\mathcal{I}}^{\mathcal{S},\mathcal{S}'}$, we obtain an irreducible $\mathcal{S}$-symmetric $d$-dimensional CFT $\mathfrak{T}_C^{\mathcal{S}}$ acting as a phase transition between two irreducible $\mathcal{S}$-symmetric gapped phases $[\mathfrak{T}_1^{\mathcal{S}}]$ and $[\mathfrak{T}_2^{\mathcal{S}}]$:

$$\mathfrak{T}_2^{\mathcal{S}} \xleftarrow{\quad -\mathcal{O} \quad} \mathfrak{T}_C^{\mathcal{S}} \xrightarrow{\quad +\mathcal{O} \quad} \mathfrak{T}_1^{\mathcal{S}} \tag{VI.2}$$

Here $\mathfrak{T}_i^{\mathcal{S}}$ is obtained from $\mathfrak{T}_i^{\mathcal{S}'}$, for $i \in \{1,2,C\}$, by applying the KT transformation $\mathcal{K}_{\mathcal{I}}^{\mathcal{S},\mathcal{S}'}$. In order to see this, note that since $\mathcal{O}'$ is uncharged, it arises in the $\mathcal{S}'$-sandwich construction from an operator $\mathcal{O}'_\partial$ completely localized along the physical boundary: $\mathfrak{B}_{\mathfrak{T}_C^{\mathcal{S}'}}^{\text{phys}}$

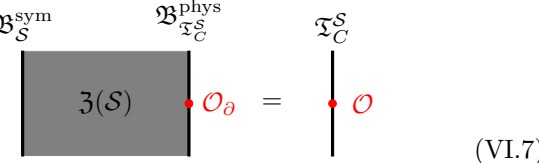

$$\tag{VI.3}$$

Deforming the boundary $\mathfrak{B}_{\mathfrak{T}_C^{\mathcal{S}'}}^{\text{phys}}$ by $\pm\mathcal{O}'_\partial$ leads to topological physical boundaries $\mathfrak{B}_{\mathfrak{T}_{1,2}^{\mathcal{S}'}}^{\text{phys}}$ for $\mathfrak{T}_{1,2}^{\mathcal{S}'}$

$$\mathfrak{B}_{\mathfrak{T}_2^{\mathcal{S}'}}^{\text{phys}} \xleftarrow{\quad -\mathcal{O}'_\partial \quad} \mathfrak{B}_{\mathfrak{T}_C^{\mathcal{S}'}}^{\text{phys}} \xrightarrow{\quad +\mathcal{O}'_\partial \quad} \mathfrak{B}_{\mathfrak{T}_1^{\mathcal{S}'}}^{\text{phys}} \tag{VI.4}$$

since the sandwich construction with these physical boundaries describes the phase transition (VI.1). This can be thought of as a boundary phase transition. We can now feed in this boundary phase transition into a club sandwich to obtain an $\mathcal{S}$-symmetric phase transition. First compactifying the interval occupied by $\mathfrak{Z}(\mathcal{S}')$, we obtain an operator $\mathcal{O}_\partial$ localized along $\mathfrak{B}_{\mathfrak{T}_C^{\mathcal{S}}}^{\text{phys}}$

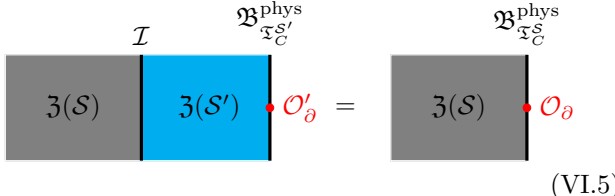

$$\tag{VI.5}$$

that generates a boundary phase transition

$$\mathfrak{B}_{\mathfrak{T}_2^{\mathcal{S}}}^{\text{phys}} \xleftarrow{\quad -\mathcal{O}_\partial \quad} \mathfrak{B}_{\mathfrak{T}_C^{\mathcal{S}}}^{\text{phys}} \xrightarrow{\quad +\mathcal{O}_\partial \quad} \mathfrak{B}_{\mathfrak{T}_1^{\mathcal{S}}}^{\text{phys}} \tag{VI.6}$$

After the full club sandwich compactification, $\mathcal{O}_\partial$ descends to an operator $\mathcal{O}$ in $\mathfrak{T}_C^{\mathcal{S}}$ uncharged under the symmetry $\mathcal{S}$

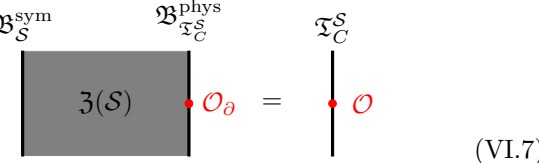

$$\tag{VI.7}$$

which is responsible for the desired $\mathcal{S}$-symmetric phase transition (VI.2).

In conclusion, one can obtain new phase transitions from known phase transitions by applying minimal KT transformations. This is quite useful as a minimal KT transformation maps $\mathcal{S}'$-symmetric theories to $\mathcal{S}$-symmetric theories where $\mathcal{S}$ is morally larger than $\mathcal{S}'$

$$\text{``}\mathcal{S} > \mathcal{S}'\text{''}. \tag{VI.8}$$

One can then begin with a small enough $\mathcal{S}'$ like $\mathbb{Z}_2$ for which a transition is well-known, and iteratively keep applying KT transformations to generate new phase transitions for larger and larger symmetries, which may not be invertible in general.

### B. Input Phase Transitions

Let us now discuss a couple of known phase transitions that we will use to construct new phase transitions by applying KT transformations to them.

#### 1. The Critical Ising Model

The first one is a $\mathbb{Z}_2$-symmetric transition provided by the 2d Ising CFT. The $\mathbb{Z}_2$ symmetry is the spin flip symmetry that we label as $\eta$. We will focus on three special

local operators in the Ising CFT, namely the order/spin operator $\sigma$, the disorder operator $\mu$, and the energy operator $\epsilon$. The order operator $\sigma$ is an untwisted sector local operator (i.e. a local operator unattached to any line operators) that is charged non-trivially under the $\mathbb{Z}_2$ symmetry $\eta$

$$\eta: \ \sigma \to -\sigma \qquad (\text{VI.9})$$

On the other hand, the disorder operator $\mu$ is an $\eta$-twisted sector local operator, i.e. it is attached to the line $\eta$ generating the $\mathbb{Z}_2$ symmetry. Additionally it carries trivial $\mathbb{Z}_2$ charge

$$\eta: \ \mu \to \mu \qquad (\text{VI.10})$$

Finally, the energy operator $\epsilon$ is also an untwisted operator, but is uncharged under the $\mathbb{Z}_2$ symmetry

$$\eta: \ \epsilon \to \epsilon \qquad (\text{VI.11})$$

These three operators correspond to three different generalized charges for the $\mathbb{Z}_2$ symmetry. As we discussed in section III A, the generalized charges of a symmetry are captured by the anyons of its associated SymTFT. In the case of non-anomalous $\mathbb{Z}_2$ symmetry, the SymTFT $\mathfrak{Z}(\mathsf{Vec}_{\mathbb{Z}_2})$ is the toric code, whose anyons are labeled as in (IV.105). Furthermore, the symmetry boundary $\mathfrak{B}^{\text{sym}}_{\mathsf{Vec}_{\mathbb{Z}_2}}$ is taken to be the one corresponding to the Lagrangian algebra

$$\mathcal{L}^{\text{sym}}_{\mathsf{Vec}_{\mathbb{Z}_2}} = 1 \oplus e \qquad (\text{VI.12})$$

Then the generalized charges for the above three operators are

$$q(\sigma) = e, \qquad q(\mu) = m, \qquad q(\epsilon) = 1 \qquad (\text{VI.13})$$

The sandwich construction of an operator involves compactification of the bulk line capturing its generalized charge. In the current case, the sandwich construction of the three operators is

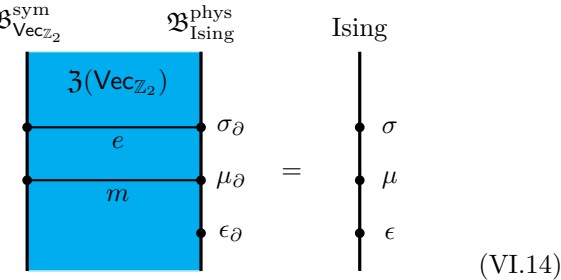

$$(\text{VI.14})$$

in terms of local operators $\sigma_\partial$, $\mu_\partial$ and $\epsilon_\partial$ on the physical boundary $\mathfrak{B}^{\text{phys}}_{\text{Ising}}$. Note that even though the $e$ line completely ends along the symmetry boundary $\mathfrak{B}^{\text{sym}}_{\mathsf{Vec}_{\mathbb{Z}_2}}$, the $m$ line does not end but becomes the line $P$ living on $\mathfrak{B}^{\text{sym}}_{\mathsf{Vec}_{\mathbb{Z}_2}}$, which is suppressed in the above figure. The boundary line $P$ becomes the line $\eta$ after the interval compactification, as a consequence of which after the compactification

the local operator $\mu$ is attached to the $\eta$ line. The charges under $\mathbb{Z}_2$ arise from the fact that the end of line $e$ along $\mathfrak{B}^{\text{sym}}_{\mathsf{Vec}_{\mathbb{Z}_2}}$ is charged under the boundary line $P$, while the end of line $m$ along $\mathfrak{B}^{\text{sym}}_{\mathsf{Vec}_{\mathbb{Z}_2}}$ is uncharged under the boundary line $P$.

Deforming the Ising CFT by the energy operator $\epsilon$ leads to $\mathbb{Z}_2$-symmetric gapped phases in the IR, which are different depending on the sign of the deformation. The two gapped phases are

$$[\mathfrak{T}^{\mathsf{Vec}_{\mathbb{Z}_2}}_1] = \mathbb{Z}_2 \text{ SSB phase for } \mathbb{Z}_2 \text{ symmetry}$$
$$[\mathfrak{T}^{\mathsf{Vec}_{\mathbb{Z}_2}}_2] = \text{Trivial gapped phase for } \mathbb{Z}_2 \text{ symmetry}$$
$$(\text{VI.15})$$

Upto an overall Euler term, the TQFT $\mathfrak{T}^{\mathsf{Vec}_{\mathbb{Z}_2}}_1$ comprises of two universes, with both universes occupied by the trivial theory

$$\mathfrak{T}^{\mathsf{Vec}_{\mathbb{Z}_2}}_1 = \text{Trivial}_0 \oplus \text{Trivial}_1 \qquad (\text{VI.16})$$

where the two copies of the trivial theory are distinguished by subscript labels $0, 1$. The identity local operators $v_0, v_1$ of the two trivial theories satisfy product rules

$$v_0^2 = v_0, \qquad v_1^2 = v_1, \qquad v_0 v_1 = 0 \qquad (\text{VI.17})$$

and are also referred to as vacua. The topological line operators of $\mathfrak{T}^{\mathsf{Vec}_{\mathbb{Z}_2}}_1$ are

$$1_{ij}, \qquad i, j \in \{0, 1\} \qquad (\text{VI.18})$$

where $1_{ii}$ is the identity line in $\text{Trivial}_i$, and $1_{01}, 1_{10}$ exchange the two copies of trivial theory. There is no relative Euler term, and hence the linking actions of $1_{01}, 1_{10}$ on the vacua are

$$1_{01}: \ v_0 \to v_1, \qquad 1_{10}: \ v_1 \to v_0 \qquad (\text{VI.19})$$

The fusion rules of the lines are

$$1_{ij} \otimes 1_{jk} = 1_{ik}. \qquad (\text{VI.20})$$

The $\mathbb{Z}_2$ symmetry $\eta$ is identified in the IR as

$$\eta \equiv 1_{01} \oplus 1_{10}, \qquad (\text{VI.21})$$

which exchanges the two vacua

$$\eta: \ v_0 \leftrightarrow v_1. \qquad (\text{VI.22})$$

As a $\mathbb{Z}_2$-symmetric theory, we may represent $\mathfrak{T}^{\mathsf{Vec}_{\mathbb{Z}_2}}_1$ as

$$\mathfrak{T}^{\mathsf{Vec}_{\mathbb{Z}_2}}_1 = \text{Trivial}_0 \oplus \text{Trivial}_1$$
$$\mathbb{Z}_2 \qquad\qquad (\text{VI.23})$$

The spin operator $\sigma$ of the Ising CFT acquires a non-zero vacuum expectation value (vev) in the two vacua, and can be identified as the operator

$$\sigma \equiv v_0 - v_1, \qquad (\text{VI.24})$$

where the coefficients of the vacua reflect the vev of $\sigma$ and the action of $\eta$ indeed respects equation (VI.9). The disorder operator $\mu$ has zero vev and does not appear in the IR theory $\mathfrak{T}_1^{\mathsf{Vec}_{\mathbb{Z}_2}}$. Consequently, there are no operators that arise at the end of $\eta$ line in the IR, and it is not possible to end the lines $1_{01}, 1_{10}$.

One says that the operator $\sigma$ carrying generalized charge $e$ is an order parameter for the $\mathbb{Z}_2$-symmetric gapped phase $[\mathfrak{T}_1^{\mathsf{Vec}_{\mathbb{Z}_2}}]$. The generalized charges of the order parameters for a symmetric gapped phase appear in the Lagrangian algebra for the corresponding physical boundary employed in the sandwich construction. Indeed, the Lagrangian algebra for the boundary $\mathfrak{B}_{\mathfrak{T}_1^{\mathsf{Vec}_{\mathbb{Z}_2}}}^{\mathrm{phys}}$ is

$$\mathcal{L}_{\mathfrak{T}_1^{\mathsf{Vec}_{\mathbb{Z}_2}}}^{\mathrm{phys}} = 1 \oplus e \,. \qquad \text{(VI.25)}$$

On the other hand, upto an Euler term, the TQFT $\mathfrak{T}_2^{\mathsf{Vec}_{\mathbb{Z}_2}}$ is the trivial theory on which the $\mathbb{Z}_2$ symmetry is realized trivially by the identity line operator

$$\eta \equiv 1 \qquad \text{(VI.26)}$$

As a $\mathbb{Z}_2$-symmetric theory, we may represent $\mathfrak{T}_2^{\mathsf{Vec}_{\mathbb{Z}_2}}$ as

$$\mathfrak{T}_2^{\mathsf{Vec}_{\mathbb{Z}_2}} = \mathrm{Trivial} \, \overset{\curvearrowright}{\phantom{.}} \, \mathbb{Z}_2 \tag{VI.27}$$

Under the RG flow to $\mathfrak{T}_2^{\mathsf{Vec}_{\mathbb{Z}_2}}$, the spin operator $\sigma$ of the Ising CFT does not acquire a non-zero vacuum expectation value (vev), and hence $\mathfrak{T}_2^{\mathsf{Vec}_{\mathbb{Z}_2}}$ does not have any local operators charged non-trivially under the $\mathbb{Z}_2$ symmetry. On the other hand, the disorder operator $\mu$ acquires a non-zero vev appearing in the IR theory as the identity operator viewed as living at the end of the identity line $\eta$. For this reason, one says that the operator $\mu$ carrying generalized charge $m$ is an order parameter for the $\mathbb{Z}_2$-symmetric gapped phase $[\mathfrak{T}_2^{\mathsf{Vec}_{\mathbb{Z}_2}}]$. An order parameter in a non-trivial twisted sector is also referred to as a **string order parameter**. Thus, the phase $[\mathfrak{T}_1^{\mathsf{Vec}_{\mathbb{Z}_2}}]$ is described by a conventional order parameter, while the phase $[\mathfrak{T}_2^{\mathsf{Vec}_{\mathbb{Z}_2}}]$ is described by a string order parameter. Correspondingly, the Lagrangian algebra for the boundary $\mathfrak{B}_{\mathfrak{T}_2^{\mathsf{Vec}_{\mathbb{Z}_2}}}^{\mathrm{phys}}$ is

$$\mathcal{L}_{\mathfrak{T}_2^{\mathsf{Vec}_{\mathbb{Z}_2}}}^{\mathrm{phys}} = 1 \oplus m \,. \qquad \text{(VI.28)}$$

The boundary phase transition is implemented by deforming the conformal boundary $\mathfrak{B}_{\mathrm{Ising}}^{\mathrm{phys}}$ by the operator $\epsilon_\partial$. On one side of the transition, the operator $\sigma_\partial$ survives while the operator $\mu_\partial$ does not, and hence the gapped boundary arising in the IR is described by the Lagrangian algebra (VI.25). On the other side of the transition, the operator $\sigma_\partial$ does not survive while the operator $\mu_\partial$ survives, and hence the gapped boundary arising in the IR is described by the Lagrangian algebra (VI.28).

### 2. 3-State Potts Model

We also consider a $\mathbb{Z}_3$-symmetric phase transition, which is given by the 2d RCFT called the three-state Potts model with $c = 4/5$. Our focus will only be on the $\mathbb{Z}_3$ symmetry of the CFT, but more generally it is well known that the CFT admits a fusion category symmetry with 12 simple objects. See [131] for more details about this.

We will call the $\mathbb{Z}_3$ symmetry generating line $\eta$ and focus on five special local operators in the CFT (that are all primary fields)

$$\sigma, \quad \sigma^*, \quad \mu, \quad \mu^*, \quad \epsilon, \qquad \text{(VI.29)}$$

where $\sigma$ is an untwisted local operator with charge 1 under $\mathbb{Z}_3$, $\sigma^*$ is an untwisted local operator with charge 2 under $\mathbb{Z}_3$, $\mu$ is an $\eta$-twisted sector operator uncharged under $\mathbb{Z}_3$, $\mu^*$ is an $\eta$-twisted sector operator uncharged under $\mathbb{Z}_3$, and $\epsilon$ is an untwisted relevant operator uncharged under $\mathbb{Z}_3$. Collecting together, the action of $\eta$ on these operators is

$$\eta: \ \sigma \to \omega\sigma, \quad \sigma^* \to \omega^2\sigma^*, \quad (\mu, \mu^*, \epsilon) \to (\mu, \mu^*, \epsilon) \,. \tag{VI.30}$$

These operators correspond to different generalized charges for the $\mathbb{Z}_3$ symmetry. As we discussed in section III A, the generalized charges of a symmetry are captured by the anyons of its associated SymTFT. In the case of non-anomalous $\mathbb{Z}_3$ symmetry, the SymTFT $\mathfrak{Z}(\mathsf{Vec}_{\mathbb{Z}_3})$ is the 3d Dijkgraaf-Witten discrete gauge theory based on the gauge group $\mathbb{Z}_3$ without any twist, whose anyons are labeled as in (IV.88). Furthermore, the symmetry boundary $\mathfrak{B}_{\mathsf{Vec}_{\mathbb{Z}_3}}^{\mathrm{sym}}$ is taken to be the one corresponding to the Lagrangian algebra

$$\mathcal{L}_{\mathsf{Vec}_{\mathbb{Z}_3}}^{\mathrm{sym}} = 1 \oplus e \oplus e^2 \,. \qquad \text{(VI.31)}$$

Then the generalized charges for the above operators are

$$q(\sigma) = e, \ \ q(\sigma^*) = e^2, \ \ q(\mu) = m, \ \ q(\mu^*) = m^2, \ \ q(\epsilon) = 1 \,. \tag{VI.32}$$

Deforming the CFT by the relevant operator $\epsilon$ leads to $\mathbb{Z}_3$-symmetric gapped phases in the IR, which are different depending on the sign $\pm\epsilon$ of the deformation. The two gapped phases are [132]

$[\mathfrak{T}_1^{\mathsf{Vec}_{\mathbb{Z}_3}}] = \mathbb{Z}_3$ SSB phase for $\mathbb{Z}_3$ symmetry

$[\mathfrak{T}_2^{\mathsf{Vec}_{\mathbb{Z}_3}}] = $ Trivial gapped phase for $\mathbb{Z}_3$ symmetry

$$\text{(VI.33)}$$

Upto an overall Euler term, the TQFT $\mathfrak{T}_1^{\mathsf{Vec}_{\mathbb{Z}_3}}$ comprises of three universes, all occupied by the trivial theory

$$\mathfrak{T}_1^{\mathsf{Vec}_{\mathbb{Z}_3}} = \mathrm{Trivial}_0 \oplus \mathrm{Trivial}_1 \oplus \mathrm{Trivial}_2 \,, \qquad \text{(VI.34)}$$

where the copies of the trivial theory are distinguished by subscript labels. The identity local operators $v_i$ of the trivial theories satisfy product rules

$$v_i v_j = \delta_{ij} v_i \,, \qquad \text{(VI.35)}$$

and are also referred to as vacua. The topological line operators of $\mathfrak{T}_1^{\mathsf{Vec}_{\mathbb{Z}_3}}$ are

$$1_{ij}, \qquad i,j \in \{0,1,2\}, \qquad (\text{VI.36})$$

where $1_{ii}$ is the identity line in $\mathrm{Trivial}_i$, and $1_{ij}$ for $i \neq j$ changes $\mathrm{Trivial}_i$ to $\mathrm{Trivial}_j$. There is no relative Euler term, and hence the linking actions of $1_{ij}$ on the vacua are

$$1_{ij} : \ v_i \to v_j \,. \qquad (\text{VI.37})$$

The fusion rules of the lines are

$$1_{ij} \otimes 1_{jk} = 1_{ik} \,. \qquad (\text{VI.38})$$

The $\mathbb{Z}_3$ symmetry $\eta$ is identified in the IR as

$$\eta \equiv 1_{01} \oplus 1_{12} \oplus 1_{20} \,, \qquad (\text{VI.39})$$

which cyclically permutes the three vacua

$$\eta : \ v_0 \to v_1 \to v_2 \to v_0 \,. \qquad (\text{VI.40})$$

As a $\mathbb{Z}_3$-symmetric theory, we may represent $\mathfrak{T}_1^{\mathsf{Vec}_{\mathbb{Z}_3}}$ as

$$\mathbb{Z}_3$$

$$\mathfrak{T}_1^{\mathsf{Vec}_{\mathbb{Z}_3}} = \mathrm{Trivial}_0 \oplus \mathrm{Trivial}_1 \oplus \mathrm{Trivial}_2 \qquad (\text{VI.41})$$

The operators $\sigma$ and $\sigma^*$ of the CFT acquire non-zero vevs in the three vacua, and can be identified as the operators

$$\sigma \equiv v_0 + \omega^2 v_1 + \omega v_2, \quad \sigma^* \equiv v_0 + \omega v_1 + \omega^2 v_2 \,, \ (\text{VI.42})$$

where the coefficients of the vacua reflect the vevs of the operators and the action of $\eta$ indeed respects equation (VI.30). The operators $\mu, \mu^*$ have zero vevs and do not appear in the IR theory $\mathfrak{T}_1^{\mathsf{Vec}_{\mathbb{Z}_3}}$. Consequently, there are no operators that arise at the end of $\eta$ and $\eta^2$ lines in the IR, and it is not possible to end the lines $1_{ij}$ for $i \neq j$.

One says that the operators $\sigma, \sigma^*$ carrying generalized charges $e, e^2$ are order parameters for the $\mathbb{Z}_3$-symmetric gapped phase $[\mathfrak{T}_1^{\mathsf{Vec}_{\mathbb{Z}_3}}]$. The generalized charges of the order parameters for a symmetric gapped phase appear in the Lagrangian algebra for the corresponding physical boundary employed in the sandwich construction. Indeed, the Lagrangian algebra for the boundary $\mathfrak{B}_{\mathfrak{T}_1^{\mathsf{Vec}_{\mathbb{Z}_3}}}^{\mathrm{phys}}$ is

$$\mathcal{L}_{\mathfrak{T}_1^{\mathsf{Vec}_{\mathbb{Z}_3}}}^{\mathrm{phys}} = 1 \oplus e \oplus e^2 \,. \qquad (\text{VI.43})$$

On the other hand, up to an Euler term, the TQFT $\mathfrak{T}_2^{\mathsf{Vec}_{\mathbb{Z}_3}}$ is the trivial theory on which the $\mathbb{Z}_3$ symmetry is realized trivially by the identity line operator

$$\eta \equiv 1 \,. \qquad (\text{VI.44})$$

As a $\mathbb{Z}_3$-symmetric theory, we may represent $\mathfrak{T}_2^{\mathsf{Vec}_{\mathbb{Z}_3}}$ as

$$\mathfrak{T}_2^{\mathsf{Vec}_{\mathbb{Z}_3}} = \mathrm{Trivial} \, \circlearrowleft \, \mathbb{Z}_3$$
$$(\text{VI.45})$$

Under the RG flow to $\mathfrak{T}_2^{\mathsf{Vec}_{\mathbb{Z}_2}}$, the operators $\sigma, \sigma^*$ of the CFT do not acquire a non-zero vacuum expectation value (vev), and hence $\mathfrak{T}_2^{\mathsf{Vec}_{\mathbb{Z}_3}}$ does not have any local operators charged non-trivially under the $\mathbb{Z}_3$ symmetry. On the other hand, the disorder operators $\mu, \mu^*$ acquire non-zero vevs, appearing in the IR theory as identity operators viewed as living at the ends of the lines $\eta, \eta^2$. For this reason, one says that the operators $\mu, \mu^*$ carrying generalized charges $m, m^2$ are order parameters for the $\mathbb{Z}_3$-symmetric gapped phase $[\mathfrak{T}_2^{\mathsf{Vec}_{\mathbb{Z}_3}}]$. Both of these are string order parameters. Correspondingly, the Lagrangian algebra for the boundary $\mathfrak{B}_{\mathfrak{T}_2^{\mathsf{Vec}_{\mathbb{Z}_3}}}^{\mathrm{phys}}$ is

$$\mathcal{L}_{\mathfrak{T}_2^{\mathsf{Vec}_{\mathbb{Z}_3}}}^{\mathrm{phys}} = 1 \oplus m \oplus m^2 \,. \qquad (\text{VI.46})$$

## C. Phase Transition between $\mathbb{Z}_4$ SSB and $\mathbb{Z}_2$ SSB Phases for $\mathbb{Z}_4$ Symmetry

Let us now discuss the phase transitions obtained by applying KT transformations discussed in the previous section. First consider the KT transformation studied in section V B.

This KT transformation converts $\mathbb{Z}_2$ symmetry into $\mathbb{Z}_4$ symmetry. Thus, we can input the $\mathbb{Z}_2$-symmetric phase transition provided by the 2d Ising CFT to obtain a $\mathbb{Z}_4$-symmetric phase transition.

The $\mathbb{Z}_4$-symmetric gapped phases lying on the two sides of the transition are obtained as KT transformations of the SSB and Trivial phases for $\mathbb{Z}_2$ symmetry discussed in section VI B 1. These can be quickly deduced by composing Lagrangian algebras as described in (V.15). This amounts to applying the map (V.28) to Lagrangian algebras $\mathcal{L}_{\mathfrak{T}_{1,2}^{\mathcal{S}'}}^{\mathrm{phys}}$. We find Lagrangian algebras associated to the boundaries $\mathfrak{B}_{\mathfrak{T}_{1,2}^{\mathcal{S}}}^{\mathrm{phys}}$

$$\mathcal{L}_{\mathfrak{T}_1^{\mathcal{S}}}^{\mathrm{phys}} = 1 \oplus e \oplus e^2 \oplus e^3$$
$$\mathcal{L}_{\mathfrak{T}_2^{\mathcal{S}}}^{\mathrm{phys}} = 1 \oplus e^2 \oplus m^2 \oplus e^2 m^2 \qquad (\text{VI.47})$$

Performing the $\mathbb{Z}_4$ sandwich construction with these physical boundaries, we can identify the KT transformed gapped phases as

$$[\mathfrak{T}_1^{\mathcal{S}}] = \mathbb{Z}_4 \text{ SSB phase for } \mathbb{Z}_4 \text{ symmetry}$$
$$[\mathfrak{T}_2^{\mathcal{S}}] = \mathbb{Z}_2 \text{ SSB phase for } \mathbb{Z}_4 \text{ symmetry} \qquad (\text{VI.48})$$

Note that in the phase $[\mathfrak{T}_2^{\mathcal{S}}]$ the $\mathbb{Z}_2$ subgroup of $\mathbb{Z}_4$ remains spontaneously unbroken. See [2] for more details.

Another equivalent way to deduce these $\mathbb{Z}_4$-symmetric gapped phases is to apply the results of section V B.

Applying them to the theory $\mathfrak{T}^{\mathcal{S}'} = \mathfrak{T}_2^{\mathcal{S}'}$ appearing in (VI.27), we learn that the KT transformed theory $\mathfrak{T}^{\mathcal{S}} = \mathfrak{T}_2^{\mathcal{S}}$ takes the form

$$\mathfrak{T}_2^{\mathcal{S}} = \mathbb{Z}_2 \overset{\mathbb{Z}_4}{\underset{\mathbb{Z}_4}{\curvearrowright \mathrm{Trivial}_0 \oplus \mathrm{Trivial}_1 \curvearrowleft}} \mathbb{Z}_2 \qquad \text{(VI.49)}$$

with the $\mathbb{Z}_4$ symmetry generated by

$$\phi(P) = 1_{01} \oplus P_{10} = 1_{01} \oplus (P_{11} \otimes 1_{10}) \equiv 1_{01} \oplus 1_{10}\,, \quad \text{(VI.50)}$$

where we have used the fact that the $\mathbb{Z}_2$ symmetry in the trivial theory $\mathrm{Trivial}_1$ is generated by the identity line $1_{11}$. Moreover the $\mathbb{Z}_2$ subgroup of the $\mathbb{Z}_4$ symmetry is generated by the identity line operator as

$$\phi(P^2) = P_{00} \oplus P_{11} \equiv 1_{00} \oplus 1_{11}\,. \qquad \text{(VI.51)}$$

Thus $\mathfrak{T}_2^{\mathcal{S}}$ is a 2d TQFT with two vacua on which the $\mathbb{Z}_2$ subgroup of $\mathbb{Z}_4$ acts and the generators of $\mathbb{Z}_4$ act by exchanging the two vacua. This is precisely the $\mathbb{Z}_2$ SSB phase for $\mathbb{Z}_4$ symmetry.

Now let us consider the KT transformation of the theory $\mathfrak{T}^{\mathcal{S}'} = \mathfrak{T}_1^{\mathcal{S}'}$ appearing in (VI.23). We know that the KT transformed theory $\mathfrak{T}^{\mathcal{S}} = \mathfrak{T}_1^{\mathcal{S}}$ has two universes of $\mathfrak{T}_1^{\mathcal{S}'}$, i.e.

$$\mathfrak{T}_1^{\mathcal{S}} = (\mathfrak{T}_1^{\mathcal{S}'})_0 \oplus (\mathfrak{T}_1^{\mathcal{S}'})_1\,, \qquad \text{(VI.52)}$$

where $(\mathfrak{T}_1^{\mathcal{S}'})_i$ denote two universes of $\mathfrak{T}_1^{\mathcal{S}'}$. Each universe $(\mathfrak{T}_1^{\mathcal{S}'})_i$ is further comprised of two vacua

$$\begin{aligned}(\mathfrak{T}_1^{\mathcal{S}'})_0 &= \mathrm{Trivial}_0 \oplus \mathrm{Trivial}_2 \\ (\mathfrak{T}_1^{\mathcal{S}'})_1 &= \mathrm{Trivial}_1 \oplus \mathrm{Trivial}_3\,.\end{aligned} \qquad \text{(VI.53)}$$

In total, $\mathfrak{T}_1^{\mathcal{S}}$ is a 2d TQFT comprising of four vacua $\mathrm{Trivial}_i$, where $i \in \{0,1,2,3\}$. The line operators in such a theory are $1_{ij}$ where $i,j \in \{0,1,2,3\}$ with fusions

$$1_{ij} \otimes 1_{jk} = 1_{ik}\,. \qquad \text{(VI.54)}$$

The generator of the $\mathbb{Z}_2$ subgroup of the $\mathbb{Z}_4$ symmetry acts by exchanging the vacua within each universe and is hence realized by line

$$\phi(P^2) \equiv 1_{02} \oplus 1_{20} \oplus 1_{13} \oplus 1_{31}\,. \qquad \text{(VI.55)}$$

The generator of $\mathbb{Z}_4$ symmetry is realized by line

$$\phi(P) \equiv 1_{01} \oplus 1_{12} \oplus 1_{23} \oplus 1_{30}\,, \qquad \text{(VI.56)}$$

which cyclically permutes the four vacua. This is precisely the $\mathbb{Z}_4$ SSB phase for $\mathbb{Z}_4$ symmetry.

The KT transformation of the Ising CFT $\mathfrak{T}_C^{\mathcal{S}'}$ is thus a $\mathbb{Z}_4$-symmetric CFT $\mathfrak{T}_C^{\mathcal{S}}$ providing a transition between the $\mathbb{Z}_4$ and $\mathbb{Z}_2$ SSB phases for $\mathbb{Z}_4$ symmetry, which can be expressed as

$$\mathfrak{T}_C^{\mathcal{S}} = \mathbb{Z}_2 \overset{\mathbb{Z}_4}{\underset{\mathbb{Z}_4}{\curvearrowright \mathrm{Ising}_0 \oplus \mathrm{Ising}_1 \curvearrowleft}} \mathbb{Z}_2 \qquad \text{(VI.57)}$$

where $\mathrm{Ising}_i$ is a copy of the Ising CFT. One may say that $\mathfrak{T}_C^{\mathcal{S}}$ is a gapless theory with two universes, such that each universe comprises of a copy of Ising CFT. Note that the relative Euler term between the two Ising universes is trivial.

The relevant topological line operators of $\mathfrak{T}_C^{\mathcal{S}}$ are

$$1_{ij}\,, \quad \eta_{ij}\,, \qquad \text{(VI.58)}$$

where $1_{ii}$ is the identity line of $\mathrm{Ising}_i$, $\eta_{ii}$ is the $\mathbb{Z}_2$ spin flip symmetry of $\mathrm{Ising}_i$, the lines $1_{01}, 1_{10}$ exchange the two universes $\mathrm{Ising}_0$ and $\mathrm{Ising}_1$, and the lines $\eta_{01}, \eta_{10}$ are obtained as

$$\begin{aligned}\eta_{01} &:= 1_{01} \otimes \eta_{11} = \eta_{00} \otimes 1_{01} \\ \eta_{10} &:= 1_{10} \otimes \eta_{00} = \eta_{11} \otimes 1_{10}\,.\end{aligned} \qquad \text{(VI.59)}$$

The generator of $\mathbb{Z}_4$ symmetry is realized as the line operator

$$\phi(P) = 1_{01} \oplus \eta_{10}\,. \qquad \text{(VI.60)}$$

The $\mathbb{Z}_2$ subgroup of $\mathbb{Z}_4$ is identified as the diagonal $\mathbb{Z}_2$ of the two $\mathbb{Z}_2$ spin flip symmetries acting on the two Ising universes, i.e.

$$\phi(P^2) = \eta_{00} \oplus \eta_{11}\,. \qquad \text{(VI.61)}$$

The action of the $\mathbb{Z}_4$ symmetry on the local operators living in the two Ising universes is

$$P: \begin{aligned}\sigma_0 &\to \sigma_1, & \sigma_1 &\to -\sigma_0 \\ \mu_0 &\to \mu_1, & \mu_1 &\to \mu_0 \\ \epsilon_0 &\to \epsilon_1, & \epsilon_1 &\to \epsilon_0\,.\end{aligned} \qquad \text{(VI.62)}$$

The relevant operator responsible for the $\mathbb{Z}_4$-symmetric transition is $\epsilon_0 + \epsilon_1$. Indeed, one side of the deformation sends $\mathrm{Ising}_i$ to the trivial phase $\mathrm{Trivial}_i$ for $\mathbb{Z}_2$ symmetry $\eta_{ii}$, thus in total realizing the $\mathbb{Z}_2$ SSB phase $[\mathfrak{T}_2^{\mathcal{S}}]$ for $\mathbb{Z}_4$ symmetry; while the other side of the deformation sends $\mathrm{Ising}_i$ to the $\mathbb{Z}_2$ SSB phase for $\mathbb{Z}_2$ symmetry $\eta_{ii}$, thus in total realizing the $\mathbb{Z}_4$ SSB phase $[\mathfrak{T}_1^{\mathcal{S}}]$ for $\mathbb{Z}_4$ symmetry.

From equations (V.29) and (V.30), we learn that the order parameters for the $\mathbb{Z}_4$ SSB phase $[\mathfrak{T}_1^{\mathcal{S}}]$ are realized by local operators

$$\mathcal{O}_e = \sigma_0 - i\sigma_1, \quad \mathcal{O}_{e^2} = \mathrm{id}_0 - \mathrm{id}_1, \quad \mathcal{O}_{e^3} = \sigma_0 + i\sigma_1\,, \qquad \text{(VI.63)}$$

where $\mathrm{id}_i$ is the identity local operator of $\mathsf{Ising}_i$. The generalized charge carried by operator $\mathcal{O}_{e^p}$ is $e^p$, which

means that it is an untwisted sector local operator with charge $p$ under the $\mathbb{Z}_4$ symmetry. After the RG flow, these operators are realized in the IR as

$$\begin{aligned} \mathrm{id}_0 &\equiv v_0 + v_2, & \sigma_0 &\equiv v_0 - v_2 \\ \mathrm{id}_1 &\equiv v_1 + v_3, & \sigma_1 &\equiv v_1 - v_3 \,, \end{aligned} \qquad (\mathrm{VI.64})$$

where $v_i$ are the vacua or in other words the identity local operators of the IR theories $\mathrm{Trivial}_i$. This follows from (VI.24). From this, we learn that the IR images of the above order parameters is

$$\begin{aligned} \mathcal{O}_e &\equiv v_0 - iv_1 - v_2 + iv_3 \\ \mathcal{O}_{e^2} &\equiv v_0 - v_1 + v_2 - v_3 \\ \mathcal{O}_{e^3} &\equiv v_0 + iv_1 - v_2 - iv_3 \,. \end{aligned} \qquad (\mathrm{VI.65})$$

Similarly, from equations (V.29) and (V.31), the order parameters for the $\mathbb{Z}_2$ SSB phase $[\mathfrak{T}_2^{\mathcal{S}}]$ for $\mathbb{Z}_4$ symmetry are realized by local operators

$$\mathcal{O}_{m^2} = \mu_0 + \mu_1, \quad \mathcal{O}_{e^2} = \mathrm{id}_0 - \mathrm{id}_1, \quad \mathcal{O}_{e^2 m^2} = \mu_0 - \mu_1 \,. \qquad (\mathrm{VI.66})$$

The subscripts of the operators describe their generalized charges. Here $\mathcal{O}_{m^2}$ and $\mathcal{O}_{e^2 m^2}$ are in the $P^2$-twisted sector and hence are string order parameters, coming attached to line $\eta_{00} \oplus \eta_{11}$. After the RG flow, these operators are realized in the IR as

$$\mathrm{id}_0 \equiv v_0, \quad \mathrm{id}_1 \equiv v_1, \quad \mu_0 \equiv v_0, \quad \mu_1 \equiv v_1 \,, \qquad (\mathrm{VI.67})$$

where $v_i$ are the vacua of the IR theories $\mathrm{Trivial}_i$. From this, we learn that the IR images of the above order parameters is

$$\mathcal{O}_{m^2} \equiv v_0 + v_1, \quad \mathcal{O}_{e^2} \equiv v_0 - v_1, \quad \mathcal{O}_{e^2 m^2} \equiv v_0 - v_1 \,, \qquad (\mathrm{VI.68})$$

where $\mathcal{O}_{m^2}$ and $\mathcal{O}_{e^2 m^2}$ are viewed as local operators living at the ends of the line $\phi(P^2)$ whose IR image is the identity line $1_{00} \oplus 1_{11}$.

### D. Phase Transition between $\mathbb{Z}_2$ SSB and Trivial Phases for $\mathbb{Z}_4$ Symmetry

Now consider the KT transformation studied in section V B.

This KT transformation also converts $\mathbb{Z}_2$ symmetry to $\mathbb{Z}_4$ symmetry, but this time it is done using pullback via the non-trivial homomorphism $\mathbb{Z}_4 \to \mathbb{Z}_2$. We again use the Ising CFT to obtain a $\mathbb{Z}_4$-symmetric phase transition described below.

This KT transformation simply amounts to regarding the generator of $\mathbb{Z}_2$ symmetry as the generator of a new $\mathbb{Z}_4$ symmetry. Thus the $\mathbb{Z}_2$ SSB phase for $\mathbb{Z}_2$ symmetry becomes the $\mathbb{Z}_2$ SSB phase for $\mathbb{Z}_4$ symmetry, which is the phase in which the generator of $\mathbb{Z}_4$ is spontaneously broken but the $\mathbb{Z}_2$ subgroup of $\mathbb{Z}_4$ is preserved. On the other hand, the trivial phase for $\mathbb{Z}_2$ becomes the trivial

phase for $\mathbb{Z}_4$. That is the gapped phases on the two sides of the $\mathbb{Z}_4$-symmetric transition are

$$\begin{aligned} [\mathfrak{T}_1^{\mathcal{S}}] &= \mathbb{Z}_2 \text{ SSB phase for } \mathbb{Z}_4 \text{ symmetry} \\ [\mathfrak{T}_2^{\mathcal{S}}] &= \text{Trivial phase for } \mathbb{Z}_4 \text{ symmetry} \end{aligned} \qquad (\mathrm{VI.69})$$

This can also be seen by composing with the Lagrangian $\mathcal{L}_{\mathcal{I}}$ shown in (IV.56), after which we obtain the Lagrangian algebras

$$\begin{aligned} \mathcal{L}_{\mathfrak{T}_1^{\mathcal{S}}}^{\mathrm{phys}} &= 1 \oplus e^2 \oplus m^2 \oplus e^2 m^2 \\ \mathcal{L}_{\mathfrak{T}_2^{\mathcal{S}}}^{\mathrm{phys}} &= 1 \oplus m \oplus m^2 \oplus m^3 \,, \end{aligned} \qquad (\mathrm{VI.70})$$

which precisely correspond to the above quoted gapped phases.

The $\mathbb{Z}_4$-symmetric phase transition is simply the Ising CFT regarded as a $\mathbb{Z}_4$ symmetric theory

$$\mathfrak{T}_C^{\mathcal{S}} = \mathrm{Ising} \,\circlearrowleft\, \mathbb{Z}_4 \qquad (\mathrm{VI.71})$$

with the generator of $\mathbb{Z}_4$ being realized by the $\eta$ line

$$\phi(P) = \eta \,. \qquad (\mathrm{VI.72})$$

The relevant operator responsible for the $\mathbb{Z}_4$-symmetric transition is $\epsilon$. From equations (V.39) and (V.40), we learn that the order parameters for the $\mathbb{Z}_2$ SSB phase $[\mathfrak{T}_1^{\mathcal{S}}]$ are realized by local operators

$$\mathcal{O}_{e^2} = \sigma, \quad \mathcal{O}_{m^2} = \mathrm{id}, \quad \mathcal{O}_{e^2 m^2} = \sigma \,, \qquad (\mathrm{VI.73})$$

where $\mathcal{O}_{m^2}$ is the identity local operator viewed as lying in the $P^2$-twisted sector, and $\mathcal{O}_{e^2 m^2}$ is the $\sigma$ operator viewed as lying in the $P^2$-twisted sector. Using the IR image (VI.24) of $\sigma$, the IR images of above operators are

$$\mathcal{O}_{e^2} = v_0 - v_1, \quad \mathcal{O}_{m^2} = 1 = v_0 + v_1, \quad \mathcal{O}_{e^2 m^2} = v_0 - v_1 \,. \qquad (\mathrm{VI.74})$$

Similarly, from equations (V.39) and (V.41), the order parameters for the trivial phase $[\mathfrak{T}_2^{\mathcal{S}}]$ for $\mathbb{Z}_4$ symmetry are realized by local operators

$$\mathcal{O}_m = \mu, \quad \mathcal{O}_{m^2} = \mathrm{id}, \quad \mathcal{O}_{m^3} = \mu \,, \qquad (\mathrm{VI.75})$$

where $\mathcal{O}_m, \mathcal{O}_{m^3}$ are $\mu$ operators viewed as lying in the $P, P^3$-twisted sectors respectively. The IR images of the above operators are

$$\mathcal{O}_m \equiv \mathcal{O}_{m^2} \equiv \mathcal{O}_{m^3} \equiv 1 \,. \qquad (\mathrm{VI.76})$$

The identity local operator can be viewed to be living in any $P^k$-twisted sector because the generator of $\mathbb{Z}_4$ symmetry in the trivial phase is realized by the identity line.

### E. Phase Transition between $S_3$ SSB and $\mathbb{Z}_2$ SSB Phases for $S_3$ Symmetry

Now consider the KT transformation studied in section V E. This KT transformation converts $\mathbb{Z}_3$ symmetry to $S_3$

symmetry. Thus, we can input the $\mathbb{Z}_3$-symmetric phase transition provided by the 3-State Potts Model to obtain an $S_3$-symmetric phase transition.

The $S_3$-symmetric gapped phases lying on the two sides of the transition are obtained as KT transformations of the SSB and Trivial phases for $\mathbb{Z}_3$ symmetry discussed in section VI B 2. These can be quickly deduced by composing Lagrangian algebras as described in (V.15). This amounts to applying the map (V.62) to Lagrangian algebras $\mathcal{L}^{\mathrm{phys}}_{\mathfrak{T}^{\mathcal{S}'}_{1,2}}$. We find Lagrangian algebras associated to the boundaries $\mathfrak{B}^{\mathrm{phys}}_{\mathfrak{T}^{\mathcal{S}}_{1,2}}$ to be

$$\begin{aligned} \mathcal{L}^{\mathrm{phys}}_{\mathfrak{T}^{\mathcal{S}}_1} &= 1 \oplus 1_- \oplus 2E \\ \mathcal{L}^{\mathrm{phys}}_{\mathfrak{T}^{\mathcal{S}}_2} &= 1 \oplus 1_- \oplus 2a_1 \,. \end{aligned} \tag{VI.77}$$

Performing the $S_3$ sandwich construction with these physical boundaries, we can identify the KT transformed gapped phases as

$$\begin{aligned} [\mathfrak{T}^{\mathcal{S}}_1] &= S_3 \text{ SSB phase for } S_3 \text{ symmetry} \\ [\mathfrak{T}^{\mathcal{S}}_2] &= \mathbb{Z}_2 \text{ SSB phase for } S_3 \text{ symmetry} \end{aligned} \tag{VI.78}$$

Note that in the phase $[\mathfrak{T}^{\mathcal{S}}_2]$ the $\mathbb{Z}_3$ subgroup of $S_3$ remains spontaneously unbroken in both the vacua. See [2] for more details.

Another equivalent way to deduce these $S_3$-symmetric gapped phases is to apply the results of section V E. Applying them to the theory $\mathfrak{T}^{\mathcal{S}'} = \mathfrak{T}^{\mathcal{S}'}_2$ appearing in (VI.45), we learn that the KT transformed theory $\mathfrak{T}^{\mathcal{S}} = \mathfrak{T}^{\mathcal{S}}_2$ takes the form

$$\mathfrak{T}^{\mathcal{S}}_2 \;=\; \mathbb{Z}_3 \underset{\underset{\mathbb{Z}_2}{\overbrace{\phantom{xxxxx}}}}{\overset{\curvearrowright}{\text{Trivial}_0 \oplus \text{Trivial}_1}} \overset{\curvearrowleft}{} \mathbb{Z}_3 \tag{VI.79}$$

with the $\mathbb{Z}_3$ subgroup of $S_3$ generated by IR line

$$a \equiv 1_{00} \oplus 1_{11} \tag{VI.80}$$

and any $\mathbb{Z}_2$ subgroup of $S_3$ generated by IR line

$$a^i b \equiv 1_{01} \oplus 1_{10} \,. \tag{VI.81}$$

Thus $\mathfrak{T}^{\mathcal{S}}_2$ is a 2d TQFT with two vacua on which the $\mathbb{Z}_3$ subgroup of $S_3$ acts trivially, and all $\mathbb{Z}_2$ subgroups of $S_3$ exchange the two vacua. This is precisely the $\mathbb{Z}_2$ SSB phase for $S_3$ symmetry.

Now let us consider the KT transformation of the theory $\mathfrak{T}^{\mathcal{S}'} = \mathfrak{T}^{\mathcal{S}'}_1$ appearing in (VI.41). We know that $\mathfrak{T}^{\mathcal{S}'}$ comprises of three vacua, and since $\mathfrak{T}^{\mathcal{S}}$ comprises of two universes of $\mathfrak{T}^{\mathcal{S}'}$, the 2d TQFT $\mathfrak{T}^{\mathcal{S}}$ comprises of six vacua. The only irreducible $S_3$-symmetric (1+1)d gapped phase with six vacua is the $S_3$ SSB phase. The reader can easily check that the $S_3$ symmetry indeed acts as on the $S_3$ SSB phase.

The KT transformation of the 3-state Potts CFT $\mathfrak{T}^{\mathcal{S}'}_C$ is thus an $S_3$-symmetric CFT $\mathfrak{T}^{\mathcal{S}}_C$ providing a transition

between the $S_3$ and $\mathbb{Z}_2$ SSB phases for $S_3$ symmetry, which can be expressed as

$$\mathfrak{T}^{\mathcal{S}}_2 \;=\; \mathbb{Z}_3 \underset{\underset{\mathbb{Z}_2}{\overbrace{\phantom{xxxxx}}}}{\overset{\curvearrowright}{\text{3-Potts}_0 \oplus \text{3-Potts}_1}} \overset{\curvearrowleft}{} \mathbb{Z}_3^{-1} \tag{VI.82}$$

where 3-Potts$_i$ is a copy of the 3-state Potts CFT. One may say that $\mathfrak{T}^{\mathcal{S}}_C$ is a gapless theory with two universes, such that each universe comprises of a copy of 3-state Potts CFT. Note that the relative Euler term between the two Potts universes is trivial.

The $\mathbb{Z}_3$ subgroup of $S_3$ symmetry is realized as

$$a = \eta_{00} \oplus \eta^2_{11} \,, \tag{VI.83}$$

where $\eta_{ii}$ is the generator of $\mathbb{Z}_3$ symmetry of 3-Potts$_i$. The $\mathbb{Z}_2$ subgroup of $S_3$ symmetry generated by $b$ is realized as

$$b = 1_{01} \oplus 1_{10} \,. \tag{VI.84}$$

The relevant operator responsible for the $S_3$-symmetric transition is $\epsilon_0 + \epsilon_1$. Indeed, one side of the deformation sends 3-Potts$_i$ to the trivial phase Trivial$_i$ for $\mathbb{Z}_3$ symmetry $\eta_{ii}$, thus in total realizing the $\mathbb{Z}_2$ SSB phase $[\mathfrak{T}^{\mathcal{S}}_2]$ for $S_3$ symmetry; while the other side of the deformation sends 3-Potts$_i$ to the $\mathbb{Z}_3$ SSB phase for $\mathbb{Z}_3$ symmetry $\eta_{ii}$, thus in total realizing the $S_3$ SSB phase $[\mathfrak{T}^{\mathcal{S}}_1]$ for $S_3$ symmetry.

The order parameters for the $S_3$ SSB phase $[\mathfrak{T}^{\mathcal{S}}_1]$ involve a local operator in representation $1_-$ of $S_3$ and two distinct multiplets of local operators in representation $E$ of $S_3$. Applying KT transformations (V.63), (V.64) and (V.65) respectively to $1$, $\sigma$ and $\sigma^*$ operators, we deduce that these order parameters are

$$\mathcal{O}_{1_-} = \mathrm{id}_0 - \mathrm{id}_1, \quad \begin{array}{ll} \mathcal{O}^{(1)}_{E,1} = \sigma_0, & \mathcal{O}^{(1)}_{E,2} = \sigma_1 \\ \mathcal{O}^{(2)}_{E,1} = \sigma^*_0, & \mathcal{O}^{(2)}_{E,2} = \sigma^*_1 \,, \end{array} \tag{VI.85}$$

where $\mathrm{id}_i$ is the identity local operator of 3-Potts$_i$. Here $\{\mathcal{O}^{(k)}_{E,1}, \mathcal{O}^{(k)}_{E,2}\}$ for $k = 1, 2$ are the two multiplets transforming in $E$ representation and $\mathcal{O}_{1_-}$ transforms in $1_-$ representation. After the RG flow, these operators are realized in the IR as

$$\begin{aligned} \mathrm{id}_0 &\equiv v_0 + v_1 + v_2, & \mathrm{id}_1 &\equiv v_3 + v_4 + v_5 \\ \sigma_0 &\equiv v_0 + \omega^2 v_1 + \omega v_2, & \sigma_1 &\equiv v_3 + \omega^2 v_4 + \omega v_5 \\ \sigma^*_0 &\equiv v_0 + \omega v_1 + \omega^2 v_2, & \sigma^*_1 &\equiv v_3 + \omega v_4 + \omega^2 v_5 \,, \end{aligned} \tag{VI.86}$$

where $v_i$ are the vacua or in other words the identity local operators of the IR theories Trivial$_i$ comprising the $S_3$ SSB phase. This follows from (VI.42). From this, we learn that the IR images of the above order parameters

are

$$\mathcal{O}_{E,1}^{(1)} \equiv v_0 + \omega^2 v_1 + \omega v_2$$
$$\mathcal{O}_{E,2}^{(1)} \equiv v_0 + \omega v_1 + \omega^2 v_2$$
$$\mathcal{O}_{1_-} \equiv v_0 + v_1 + v_2 - v_3 - v_4 - v_5 \qquad \text{(VI.87)}$$
$$\mathcal{O}_{E,1}^{(2)} \equiv v_3 + \omega^2 v_4 + \omega v_5$$
$$\mathcal{O}_{E,2}^{(2)} \equiv v_3 + \omega v_4 + \omega^2 v_5 \, .$$

Similarly, the order parameters for the $\mathbb{Z}_2$ SSB phase $[\mathfrak{T}_2^{\mathcal{S}}]$ involve a local operator in representation $1_-$ of $S_3$ and two distinct multiplets of local operators in $a_1$-multiplet, each comprising of two operators in $a, a^2$-twisted sectors charged trivially under $\mathbb{Z}_3$ subgroup of $S_3$. Applying KT transformations (V.63), (V.66) and (V.67) respectively to $1$, $\mu$ and $\mu^*$ operators, we deduce that these order parameters are

$$\mathcal{O}_{1_-} = \mathrm{id}_0 - \mathrm{id}_1, \quad \begin{array}{ll} \mathcal{O}_{a,1}^{(1)} = \mu_0, & \mathcal{O}_{a^2,1}^{(1)} = \mu_1 \\ \mathcal{O}_{a,1}^{(2)} = \mu_0^*, & \mathcal{O}_{a^2,1}^{(2)} = \mu_1^* \, . \end{array} \qquad \text{(VI.88)}$$

Here $\{\mathcal{O}_{a,1}^{(k)}, \mathcal{O}_{a^2,1}^{(k)}\}$ for $k = 1, 2$ are the two $a_1$-multiplets and $\mathcal{O}_{1_-}$ transforms in $1_-$ representation. After the RG flow, these operators are realized in the IR as

$$\mathrm{id}_0 \equiv v_0, \qquad \mathrm{id}_1 \equiv v_1$$
$$\mu_0 \equiv v_0, \qquad \mu_1 \equiv v_1 \qquad \text{(VI.89)}$$
$$\mu_0^* \equiv v_0, \qquad \mu_1^* \equiv v_1 \, .$$

where $v_i$ are the vacua or in other words the identity local operators of the IR theories Trivial$_i$ comprising the $\mathbb{Z}_2$ SSB phase. From this, we learn that the IR images of the above order parameters are

$$\mathcal{O}_{1_-} = v_0 - v_1, \quad \begin{array}{ll} \mathcal{O}_{a,1}^{(1)} = v_0, & \mathcal{O}_{a^2,1}^{(1)} = v_1 \\ \mathcal{O}_{a,1}^{(2)} = v_0, & \mathcal{O}_{a^2,1}^{(2)} = v_1 \, . \end{array} \qquad \text{(VI.90)}$$

### F. Phase Transition between $S_3$ SSB and $\mathbb{Z}_3$ SSB Phases for $S_3$ Symmetry

Now consider the KT transformation studied in section V F. This KT transformation converts $\mathbb{Z}_2$ symmetry to $S_3$ symmetry. Thus, we can input the $\mathbb{Z}_2$-symmetric phase transition provided by the 2d Ising CFT to obtain an $S_3$-symmetric phase transition.

The $S_3$-symmetric gapped phases lying on the two sides of the transition are obtained as KT transformations of the SSB and Trivial phases for $\mathbb{Z}_2$ symmetry discussed in section VI B 1. These can be quickly deduced by composing Lagrangian algebras as described in (V.15). This amounts to applying the map (V.71) to Lagrangian algebras $\mathcal{L}_{\mathfrak{T}_{1,2}^{\mathcal{S}'}}^{\mathrm{phys}}$. We find Lagrangian algebras associated

to the boundaries $\mathfrak{B}_{\mathfrak{T}_{1,2}^{\mathcal{S}}}^{\mathrm{phys}}$ to be

$$\mathcal{L}_{\mathfrak{T}_1^{\mathcal{S}}}^{\mathrm{phys}} = 1 \oplus 1_- \oplus 2E$$
$$\mathcal{L}_{\mathfrak{T}_2^{\mathcal{S}}}^{\mathrm{phys}} = 1 \oplus E \oplus b_+ \, . \qquad \text{(VI.91)}$$

Performing the $S_3$ sandwich construction with these physical boundaries, we can identify the KT transformed gapped phases as

$$[\mathfrak{T}_1^{\mathcal{S}}] = S_3 \text{ SSB phase for } S_3 \text{ symmetry}$$
$$[\mathfrak{T}_2^{\mathcal{S}}] = \mathbb{Z}_3 \text{ SSB phase for } S_3 \text{ symmetry} \qquad \text{(VI.92)}$$

Note that in the phase $[\mathfrak{T}_2^{\mathcal{S}}]$ a $\mathbb{Z}_2$ subgroup of $S_3$ remains spontaneously unbroken in each of the three vacua. See [2] for more details.

Another equivalent way to deduce these $S_3$-symmetric gapped phases is to apply the results of section V F. Applying them to the theory $\mathfrak{T}^{\mathcal{S}'} = \mathfrak{T}_2^{\mathcal{S}'}$ appearing in (VI.27), we learn that the KT transformed theory $\mathfrak{T}^{\mathcal{S}} = \mathfrak{T}_2^{\mathcal{S}}$ takes the form

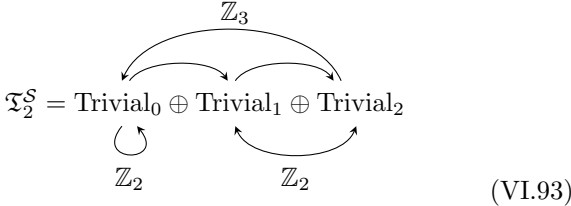

$$\text{(VI.93)}$$

with the $\mathbb{Z}_3$ subgroup of $S_3$ generated by IR line

$$a \equiv 1_{01} \oplus 1_{12} \oplus 1_{20} \qquad \text{(VI.94)}$$

and the $\mathbb{Z}_2$ subgroup of $S_3$ associated to $b$ generated by IR line

$$b \equiv 1_{00} \oplus 1_{12} \oplus 1_{21} \, . \qquad \text{(VI.95)}$$

Thus $\mathfrak{T}_2^{\mathcal{S}}$ is a 2d TQFT with three vacua on which the $\mathbb{Z}_3$ subgroup of $S_3$ acts by cyclic permutations, and the $\mathbb{Z}_2$ subgroup of $S_3$ generated by $b$ preserves a vacuum while exchanging the other two. This is precisely the $\mathbb{Z}_3$ SSB phase for $S_3$ symmetry.

Now let us consider the KT transformation of the theory $\mathfrak{T}^{\mathcal{S}'} = \mathfrak{T}_1^{\mathcal{S}'}$ appearing in (VI.23). We know that $\mathfrak{T}^{\mathcal{S}'}$ comprises of two vacua, and since $\mathfrak{T}^{\mathcal{S}}$ comprises of three universes of $\mathfrak{T}^{\mathcal{S}'}$, the 2d TQFT $\mathfrak{T}^{\mathcal{S}}$ comprises of six vacua. The only irreducible $S_3$-symmetric (1+1)d gapped phase with six vacua is the $S_3$ SSB phase. The reader can easily check that the $S_3$ symmetry indeed acts as on the $S_3$ SSB phase.

The KT transformation of the Ising CFT $\mathfrak{T}_C^{\mathcal{S}'}$ is thus an $S_3$-symmetric CFT $\mathfrak{T}_C^{\mathcal{S}}$ providing a transition between the $S_3$ and $\mathbb{Z}_3$ SSB phases for $S_3$ symmetry, which can

be expressed as

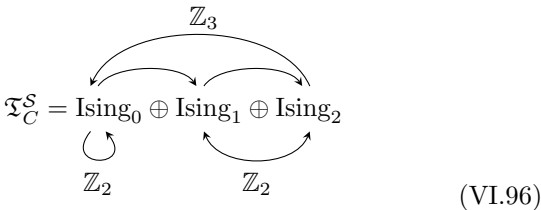

$$\mathfrak{T}_C^{\mathcal{S}} = \text{Ising}_0 \oplus \text{Ising}_1 \oplus \text{Ising}_2 \qquad (\text{VI}.96)$$

where $\text{Ising}_i$ is a copy of the Ising CFT. One may say that $\mathfrak{T}_C^{\mathcal{S}}$ is a gapless theory with three universes, such that each universe comprises of a copy of Ising CFT. Note that the relative Euler term between the three Ising universes is trivial.

The relevant topological line operators of $\mathfrak{T}_C^{\mathcal{S}}$ are

$$1_{ij}, \quad \eta_{ij}, \qquad (\text{VI}.97)$$

where $1_{ii}$ is the identity line of $\text{Ising}_i$, $\eta_{ii}$ is the $\mathbb{Z}_2$ spin flip symmetry of $\text{Ising}_i$, the lines $1_{ij}$ for $i \neq j$ transform $\text{Ising}_i$ into $\text{Ising}_j$, and the lines $\eta_{ij}$ are obtained as

$$\eta_{ij} := 1_{ij} \otimes \eta_{jj} = \eta_{ii} \otimes 1_{ij}, \qquad (\text{VI}.98)$$

which also transform $\text{Ising}_i$ into $\text{Ising}_j$. The lines generating $S_3$ symmetry are realized as

$$\begin{aligned} a &= 1_{01} \oplus 1_{12} \oplus 1_{20} \\ b &= \eta_{00} \oplus \eta_{12} \oplus \eta_{21}. \end{aligned} \qquad (\text{VI}.99)$$

The action of the $S_3$ symmetry on the local operators living in $\mathfrak{T}_C^{\mathcal{S}}$ is

$$a: \begin{array}{ccc} \sigma_0 \to \sigma_1, & \sigma_1 \to \sigma_2 & \sigma_2 \to \sigma_0 \\ \mu_0 \to \mu_1, & \mu_1 \to \mu_2 & \mu_2 \to \mu_0 \\ \epsilon_0 \to \epsilon_1, & \epsilon_1 \to \epsilon_2 & \epsilon_2 \to \epsilon_0 \end{array} \qquad (\text{VI}.100)$$

and

$$b: \begin{array}{ccc} \sigma_0 \to -\sigma_0, & \sigma_1 \to -\sigma_2 & \sigma_2 \to -\sigma_1 \\ \mu_0 \to \mu_0, & \mu_1 \to \mu_2 & \mu_2 \to \mu_1 \\ \epsilon_0 \to \epsilon_0, & \epsilon_1 \to \epsilon_2 & \epsilon_2 \to \epsilon_1. \end{array} \qquad (\text{VI}.101)$$

The relevant operator responsible for the $S_3$-symmetric transition is the KT transformation of $\epsilon$ which according to (V.73) is $\epsilon_0 + \epsilon_1 + \epsilon_2$. Indeed, one side of the deformation sends $\text{Ising}_i$ to the trivial phase $\text{Trivial}_i$ for $\mathbb{Z}_2$ symmetry $\eta_{ii}$, thus in total realizing the $\mathbb{Z}_3$ SSB phase $[\mathfrak{T}_2^{\mathcal{S}}]$ for $S_3$ symmetry; while the other side of the deformation sends $\text{Ising}_i$ to the $\mathbb{Z}_2$ SSB phase for $\mathbb{Z}_2$ symmetry $\eta_{ii}$, thus in total realizing the $S_3$ SSB phase $[\mathfrak{T}_1^{\mathcal{S}}]$ for $S_3$ symmetry.

The order parameters for the $S_3$ SSB phase $[\mathfrak{T}_1^{\mathcal{S}}]$ involve a local operator in representation $1_-$ of $S_3$ and two distinct multiplets of local operators in representation $E$ of $S_3$. Applying KT transformations (V.73) and (V.74) respectively to the identity operator $1$ and the spin operator $\sigma$ of the Ising CFT, we deduce that these order

parameters are

$$\begin{aligned} \mathcal{O}_{E,1}^{(1)} &= \text{id}_0 + \omega^2 \text{id}_1 + \omega \text{id}_2 \\ \mathcal{O}_{E,2}^{(1)} &= \text{id}_0 + \omega \text{id}_1 + \omega^2 \text{id}_2 \\ \mathcal{O}_{1_-} &= \sigma_0 + \sigma_1 + \sigma_2 \\ \mathcal{O}_{E,1}^{(2)} &= \sigma_0 + \omega^2 \sigma_1 + \omega \sigma_2 \\ \mathcal{O}_{E,2}^{(2)} &= -\sigma_0 - \omega \sigma_1 - \omega^2 \sigma_2 \end{aligned} \qquad (\text{VI}.102)$$

where $\text{id}_i$ is the identity local operator of $\text{Ising}_i$. Here $\{\mathcal{O}_{E,1}^{(k)}, \mathcal{O}_{E,2}^{(k)}\}$ for $k = 1, 2$ are the two multiplets transforming in $E$ representation and $\mathcal{O}_{1_-}$ transforms in $1_-$ representation. After the RG flow, these operators are realized in the IR as

$$\begin{aligned} \text{id}_0 &\equiv v_0 + v_3, & \sigma_0 &\equiv v_0 - v_3 \\ \text{id}_1 &\equiv v_1 + v_4, & \sigma_1 &\equiv v_1 - v_4 \\ \text{id}_2 &\equiv v_2 + v_5, & \sigma_2 &\equiv v_2 - v_5 \end{aligned} \qquad (\text{VI}.103)$$

where $v_i$ are the vacua or in other words the identity local operators of the IR theories $\text{Trivial}_i$ comprising the $S_3$ SSB phase. This follows from (VI.24). From this, we learn that the IR images of the above order parameters is

$$\begin{aligned} \mathcal{O}_{E,1}^{(1)} &\equiv v_0 + \omega^2 v_1 + \omega v_2 + v_3 + \omega^2 v_4 + \omega v_5 \\ \mathcal{O}_{E,2}^{(1)} &\equiv v_0 + \omega v_1 + \omega^2 v_2 + v_3 + \omega v_4 + \omega^2 v_5 \\ \mathcal{O}_{1_-} &\equiv v_0 + v_1 + v_2 - v_3 - v_4 - v_5 \\ \mathcal{O}_{E,1}^{(2)} &\equiv v_0 + \omega^2 v_1 + \omega v_2 - v_3 - \omega^2 v_4 - \omega v_5 \\ \mathcal{O}_{E,2}^{(2)} &\equiv -v_0 - \omega v_1 - \omega^2 v_2 + v_3 + \omega v_4 + \omega^2 v_5. \end{aligned} \qquad (\text{VI}.104)$$

Similarly, the order parameters for the $\mathbb{Z}_3$ SSB phase $[\mathfrak{T}_2^{\mathcal{S}}]$ involve two local operators in representation $E$ of $S_3$ and three local operators in $b_+$-multiplet of $S_3$ in which the three operators are in twisted sectors for generators of the three $\mathbb{Z}_2$ subgroups of $S_3$ respectively and are uncharged under those $\mathbb{Z}_2$ subgroups. Applying KT transformations (V.73) and (V.76) to the identity operator $1$ and the disorder operator $\mu$ of the Ising CFT, we deduce that these order parameters are

$$\begin{aligned} \mathcal{O}_{E,1}^{(1)} &= \text{id}_0 + \omega^2 \text{id}_1 + \omega \text{id}_2 \\ \mathcal{O}_{E,2}^{(1)} &= \text{id}_0 + \omega \text{id}_1 + \omega^2 \text{id}_2 \\ \mathcal{O}_{a^i b}^{(0)} &= \mu_i, \qquad i \in \{0, 1, 2\}. \end{aligned} \qquad (\text{VI}.105)$$

Here $\mathcal{O}_{E,1}^{(1)}, \mathcal{O}_{E,2}^{(1)}$ are operators transforming in $E$ representation and $\mathcal{O}_{a^i b}^{(0)}$ are operators transforming as a $b_+$-multiplet. After the RG flow, these operators are realized in the IR as

$$\text{id}_i \equiv v_i, \qquad \mu_i \equiv v_i, \qquad (\text{VI}.106)$$

where $v_i$ are the vacua of the IR theories $\text{Trivial}_i$. From this, we learn that the IR images of the above order pa-

rameters is

$$\mathcal{O}_{E,1}^{(1)} \equiv v_0 + \omega^2 v_1 + \omega v_2$$
$$\mathcal{O}_{E,2}^{(1)} \equiv v_0 + \omega v_1 + \omega^2 v_2 \qquad \text{(VI.107)}$$
$$\mathcal{O}_{a^i b}^{(0)} \equiv v_i, \qquad i \in \{0,1,2\},$$

where $\mathcal{O}_{a^i b}^{(0)}$ are viewed as operators in the $a^i b$ twisted sector.

## G. Phase Transition between $\mathbb{Z}_2$ SSB and Trivial Phases for $S_3$ Symmetry

Now consider the KT transformation studied in section V G. This KT transformation also converts $\mathbb{Z}_2$ symmetry to $S_3$ symmetry, but this time it is done using pullback via the non-trivial homomorphism $S_3 \to \mathbb{Z}_2$. We again use the Ising CFT to obtain an $S_3$-symmetric phase transition described below.

This KT transformation simply amounts to regarding the generator of $\mathbb{Z}_2$ symmetry as the generators for $\mathbb{Z}_2$ subgroups inside $S_3$, and the $\mathbb{Z}_3$ subgroup of $S_3$ acts trivially. Thus the $\mathbb{Z}_2$ SSB phase for $\mathbb{Z}_2$ symmetry becomes the $\mathbb{Z}_2$ SSB phase for $S_3$ symmetry. On the other hand, the trivial phase for $\mathbb{Z}_2$ becomes the trivial phase for $S_3$. That is the gapped phases on the two sides of the $S_3$-symmetric transition are

$$[\mathfrak{T}_1^{\mathcal{S}}] = \mathbb{Z}_2 \text{ SSB phase for } S_3 \text{ symmetry}$$
$$[\mathfrak{T}_2^{\mathcal{S}}] = \text{Trivial phase for } S_3 \text{ symmetry} \qquad \text{(VI.108)}$$

This can also be seen by using the map (V.80) to deduce

$$\mathcal{L}_{\mathfrak{T}_1^{\mathcal{S}}}^{\text{phys}} = 1 \oplus 1_- \oplus 2a_1$$
$$\mathcal{L}_{\mathfrak{T}_2^{\mathcal{S}}}^{\text{phys}} = 1 \oplus a_1 \oplus b_+ , \qquad \text{(VI.109)}$$

which precisely correspond to the above quoted gapped phases.

The $S_3$-symmetric phase transition is simply the Ising CFT regarded as an $S_3$ symmetric theory

$$\mathfrak{T}_C^{\mathcal{S}} = \text{Ising} \curvearrowleft S_3 \qquad \text{(VI.110)}$$

with the generators of $\mathbb{Z}_2$ subgroups of $S_3$ being realized by the $\eta$ line, while all the other elements being realized by the trivial line

$$b = ab = a^2 b = \eta . \qquad \text{(VI.111)}$$

The relevant operator responsible for the $S_3$-symmetric transition is $\epsilon$. The order parameters for the $Z_2$ SSB phase $[\mathfrak{T}_1^{\mathcal{S}}]$ involve a local operator in representation $1_-$ of $S_3$ and two distinct $a_1$-multiplets of local operators, each comprising of two operators in $a, a^2$-twisted sectors charged trivially under $\mathbb{Z}_3$ subgroup of $S_3$. Applying

KT transformations (V.81) and (V.82) respectively to the identity operator id and the spin operator $\sigma$ of the Ising CFT, we deduce that these order parameters are

$$\mathcal{O}_{a^i}^{(1)} = \text{id}, \quad \mathcal{O}_{1_-} = \sigma, \quad \mathcal{O}_{a^i}^{(2)} = (-1)^i \sigma \qquad \text{(VI.112)}$$

for $i = 1, 2$. Here $\mathcal{O}_{a^i}^{(k)}$ for $k = 1, 2$ are the two $a_1$-multiplets and $\mathcal{O}_{1_-}$ is the operator in $1_-$ representation of $S_3$. Using the IR image (VI.24) of $\sigma$, the IR images of above operators are

$$\mathcal{O}_{a^i}^{(1)} \equiv v_0 + v_1, \quad \mathcal{O}_{1_-} \equiv v_0 - v_1, \quad \mathcal{O}_{a^i}^{(2)} \equiv (-1)^i (v_0 - v_1). \qquad \text{(VI.113)}$$

Similarly, the order parameters for the trivial phase $[\mathfrak{T}_2^{\mathcal{S}}]$ involve an $a_1$-multiplet and a $b_+$-multiplet. Applying KT transformations (V.81) and (V.83) respectively to the identity operator id and the disorder operator $\mu$ of the Ising CFT, we deduce that these order parameters are

$$\mathcal{O}_{a^i}^{(1)} = \text{id}, \quad \mathcal{O}_{a^k b}^{(0)} = \mu \qquad \text{(VI.114)}$$

for $i = 1, 2$ parametrizing operators in $a_1$ multiplet and $k = 0, 1, 2$ parametrizing operators in $b_+$ multiplet. The IR images of the above operators are

$$\mathcal{O}_{a^i}^{(1)} \equiv 1, \quad \mathcal{O}_{a^i b}^{(0)} \equiv 1, \qquad \text{(VI.115)}$$

where 1 is the identity operator of the trivial phase.

## H. Phase Transition between $\text{Rep}(S_3)/\mathbb{Z}_2$ SSB and Trivial Phases for $\text{Rep}(S_3)$ Symmetry

Now consider the KT transformation studied in section V H. This KT transformation converts $\mathbb{Z}_3$ symmetry to $\text{Rep}(S_3)$ symmetry. We use the 3-state Potts CFT to obtain a $\text{Rep}(S_3)$-symmetric phase transition described below.

Using the map (V.87), we can quickly deduce that the $\text{Rep}(S_3)$-symmetric gapped phases obtained after applying KT transformation on $\mathbb{Z}_3$ SSB and trivial phases for $\mathbb{Z}_3$ symmetry are described respectively by the physical Lagrangian algebras

$$\mathcal{L}_{\mathfrak{T}_1^{\mathcal{S}}}^{\text{phys}} = 1 \oplus 1_- \oplus 2a_1$$
$$\mathcal{L}_{\mathfrak{T}_2^{\mathcal{S}}}^{\text{phys}} = 1 \oplus 1_- \oplus 2E , \qquad \text{(VI.116)}$$

which correspond to the following $\text{Rep}(S_3)$-symmetric gapped phases respectively

$$[\mathfrak{T}_1^{\mathcal{S}}] = \text{Rep}(S_3)/\mathbb{Z}_2 \text{ SSB phase for Rep}(S_3) \text{ symmetry}$$
$$[\mathfrak{T}_2^{\mathcal{S}}] = \text{Trivial phase for Rep}(S_3) \text{ symmetry} \qquad \text{(VI.117)}$$

For more details on these gapped phases, we refer the reader to [2].

Another equivalent way to deduce these $\text{Rep}(S_3)$-symmetric gapped phases is to apply the results of section

V H. Applying them to the theory $\mathfrak{T}^{\mathcal{S}'} = \mathfrak{T}_2^{\mathcal{S}'}$ appearing in (VI.45), we learn that the KT transformed theory $\mathfrak{T}^{\mathcal{S}} = \mathfrak{T}_2^{\mathcal{S}}$ takes the form

$$\mathfrak{T}^{\mathcal{S}} = \text{Trivial} \,\overset{\curvearrowleft}{} \text{Rep}(S_3) \tag{VI.118}$$

which means that $\mathfrak{T}_2^{\mathcal{S}}$ is an SPT phase for $\text{Rep}(S_3)$ symmetry since it involves a single vacuum, but there is only one such SPT phase that is often referred to as the trivial $\text{Rep}(S_3)$-symmetric phase.

Now let us consider the KT transformation of the theory $\mathfrak{T}^{\mathcal{S}'} = \mathfrak{T}_1^{\mathcal{S}'}$ appearing in (VI.41). We know that $\mathfrak{T}^{\mathcal{S}'}$ comprises of three vacua, and hence $[\mathfrak{T}_1^{\mathcal{S}}]$ contains $\text{Rep}(S_3)$ symmetry realized on a $(1+1)$d gapped phase with three vacua. There are two possible options for such a phase, namely $\text{Rep}(S_3)/\mathbb{Z}_2$ SSB and $\text{Rep}(S_3)$ SSB phases. Since the three vacua of $\mathfrak{T}_1^{\mathcal{S}}$ are identified as the three vacua of $\mathfrak{T}_1^{\mathcal{S}'}$, we learn that the relative Euler terms between the three vacua of $\mathfrak{T}_1^{\mathcal{S}}$ must all be trivial. This fixes $\mathfrak{T}_1^{\mathcal{S}}$ to be the $\text{Rep}(S_3)/\mathbb{Z}_2$ SSB phase. Using (IV.146), we can identify the lines implementing $\text{Rep}(S_3)$ symmetry of $\mathfrak{T}_1^{\mathcal{S}}$ as

$$\begin{aligned} 1_- &\equiv 1_{00} \oplus 1_{11} \oplus 1_{22} \\ E &\equiv 1_{01} \oplus 1_{12} \oplus 1_{20} \oplus 1_{02} \oplus 1_{10} \oplus 1_{21}\,, \end{aligned} \tag{VI.119}$$

which matches the results of [2].

The $\text{Rep}(S_3)$-symmetric phase transition is simply the 3-state Potts CFT regarded as a $\text{Rep}(S_3)$ symmetric theory

$$\mathfrak{T}_C^{\mathcal{S}} = \text{3-Potts} \,\overset{\curvearrowleft}{} \text{Rep}(S_3) \tag{VI.120}$$

with the generators of $\text{Rep}(S_3)$ being realized as

$$1_- = 1, \qquad E = \eta \oplus \eta^2\,. \tag{VI.121}$$

The relevant operator responsible for the $\text{Rep}(S_3)$-symmetric transition is $\epsilon$.

The order parameters for the $\text{Rep}(S_3)/Z_2$ SSB phase $[\mathfrak{T}_1^{\mathcal{S}}]$ involve a $1_-$-multiplet and two distinct $a_1$-multiplets of $\text{Rep}(S_3)$, whose detailed form was discussed above (V.88) and (V.89) respectively. Applying KT transformations (V.88) and (V.89) to the operators id, $\sigma$ and $\sigma^*$ of the 3-state Potts CFT, we deduce that these order parameters are

$$\mathcal{O}_{1_-} = \text{id}, \;\; \mathcal{O}_{a_1,u}^{(1)} = \sigma, \;\; \mathcal{O}_{a_1,t}^{(1)} = \sigma, \;\; \mathcal{O}_{a_1,u}^{(2)} = \sigma^*, \;\; \mathcal{O}_{a_1,t}^{(2)} = \sigma^* \tag{VI.122}$$

where $\mathcal{O}_{1_-}$ forms the $1_-$-multiplet and $\{\mathcal{O}_{a_1,u}^{(k)}, \mathcal{O}_{a_1,t}^{(k)}\}$ for $k = 1, 2$ are the two $a_1$-multiplets. The IR images of above operators are

$$\begin{aligned} \mathcal{O}_{1_-} &= v_0 + v_1 + v_2 \\ \mathcal{O}_{a_1,u}^{(1)} &= v_0 + \omega^2 v_1 + \omega v_2 \\ \mathcal{O}_{a_1,t}^{(1)} &= v_0 + \omega^2 v_1 + \omega v_2 \\ \mathcal{O}_{a_1,u}^{(2)} &= v_0 + \omega v_1 + \omega^2 v_2 \\ \mathcal{O}_{a_1,t}^{(2)} &= v_0 + \omega v_1 + \omega^2 v_2 \end{aligned} \tag{VI.123}$$

The order parameters for the trivial phase $[\mathfrak{T}_2^{\mathcal{S}}]$ involve a $1_-$-multiplet and two distinct $E$-multiplets of $\text{Rep}(S_3)$, whose detailed form was discussed above (V.88) and (V.90) respectively. Applying KT transformations (V.88) and (V.90) to the operators id, $\mu$ and $\mu^*$ of the 3-state Potts CFT, we deduce that these order parameters are are

$$\begin{aligned} \mathcal{O}_{E,+}^{(1)} &= \mu, \quad \mathcal{O}_{E,-}^{(1)} = \mu, \quad \mathcal{O}_{1_-} = \text{id}, \quad \mathcal{O}_{E,+}^{(2)} = \mu^* \\ \mathcal{O}_{E,-}^{(2)} &= \mu^*\,. \end{aligned} \tag{VI.124}$$

Here $\mathcal{O}_{E,\pm}^{(k)}$ for $k = 1, 2$ form two $E$-multiplets and $\mathcal{O}_{1_-}$ forms a $1_-$-multiplet of $\text{Rep}(S_3)$. The IR images of above operators are

$$\begin{aligned} \mathcal{O}_{E,+}^{(1)} &\equiv 1, \quad \mathcal{O}_{E,-}^{(1)} \equiv 1, \quad \mathcal{O}_{1_-} \equiv 1, \quad \mathcal{O}_{E,+}^{(2)} \equiv 1 \\ \mathcal{O}_{E,-}^{(2)} &\equiv 1\,. \end{aligned} \tag{VI.125}$$

That is, the above operators are simply the identity operator viewed in different fashions as transitioning between different line operators associated to the $\text{Rep}(S_3)$ symmetry. This result is not surprising as identity operator is the only operator available in the trivial phase.

## I. Phase Transition between $\mathbb{Z}_2$ SSB and Trivial Phases for $\text{Rep}(S_3)$ Symmetry

Now consider the KT transformation studied in section VI. This KT transformation converts $\mathbb{Z}_2$ symmetry to $\text{Rep}(S_3)$ symmetry. We use the Ising CFT to obtain a $\text{Rep}(S_3)$-symmetric phase transition described below.

Using the map (V.95), we can quickly deduce that the $\text{Rep}(S_3)$-symmetric gapped phases obtained after applying KT transformation on $\mathbb{Z}_2$ SSB and trivial phases for $\mathbb{Z}_2$ symmetry are described respectively by the physical Lagrangian algebras

$$\begin{aligned} \mathcal{L}_{\mathfrak{T}_1^{\mathcal{S}}}^{\text{phys}} &= 1 \oplus E \oplus b_+ \\ \mathcal{L}_{\mathfrak{T}_2^{\mathcal{S}}}^{\text{phys}} &= 1 \oplus 1_- \oplus 2E\,, \end{aligned} \tag{VI.126}$$

which correspond to the following $\text{Rep}(S_3)$-symmetric gapped phases respectively

$$\begin{aligned} [\mathfrak{T}_1^{\mathcal{S}}] &= \mathbb{Z}_2 \text{ SSB phase for } \text{Rep}(S_3) \text{ symmetry} \\ [\mathfrak{T}_2^{\mathcal{S}}] &= \text{Trivial phase for } \text{Rep}(S_3) \text{ symmetry} \end{aligned} \tag{VI.127}$$

For more details on these gapped phases, we refer the reader to [2].

Another equivalent way to deduce these $\text{Rep}(S_3)$-symmetric gapped phases is to apply the results of section VI. Applying them to the theory $\mathfrak{T}^{\mathcal{S}'} = \mathfrak{T}_2^{\mathcal{S}'}$ appearing in (VI.27), we learn that the KT transformed theory $\mathfrak{T}^{\mathcal{S}} = \mathfrak{T}_2^{\mathcal{S}}$ takes the form

$$\mathfrak{T}^{\mathcal{S}} = \text{Trivial} \,\overset{\curvearrowleft}{} \text{Rep}(S_3) \tag{VI.128}$$

which means that $\mathfrak{T}_2^{\mathcal{S}}$ is the trivial $\mathsf{Rep}(S_3)$-symmetric phase.

Now let us consider the KT transformation of the theory $\mathfrak{T}^{\mathcal{S}'} = \mathfrak{T}_1^{\mathcal{S}'}$ appearing in (VI.23). We know that $\mathfrak{T}^{\mathcal{S}'}$ comprises of two vacua, and hence $\mathfrak{T}_1^{\mathcal{S}}$ contains $\mathsf{Rep}(S_3)$ symmetry realized on a 2d TQFT with two vacua, which fixes $[\mathfrak{T}_1^{\mathcal{S}}]$ $\mathbb{Z}_2$ SSB phase for $\mathsf{Rep}(S_3)$ symmetry. Using (IV.138), we can identify the lines implementing $\mathsf{Rep}(S_3)$ symmetry of $\mathfrak{T}_1^{\mathcal{S}}$ as

$$1_- \equiv 1_{01} \oplus 1_{10}, \quad E \equiv 1_{00} \oplus 1_{11} \oplus 1_{01} \oplus 1_{10}, \quad \text{(VI.129)}$$

which matches the results of [2].

The $\mathsf{Rep}(S_3)$-symmetric phase transition is simply the Ising CFT regarded as a $\mathsf{Rep}(S_3)$ symmetric theory

$$\mathfrak{T}_C^{\mathcal{S}} = \text{Ising} \,\curvearrowleft\!\!\supset \mathsf{Rep}(S_3) \tag{VI.130}$$

with the generators of $\mathsf{Rep}(S_3)$ being realized as

$$1_- = \eta, \qquad E = 1 \oplus \eta. \tag{VI.131}$$

The relevant operator responsible for the $\mathsf{Rep}(S_3)$-symmetric transition is $\epsilon$.

The order parameters for the $Z_2$ SSB phase $[\mathfrak{T}_1^{\mathcal{S}}]$ involve an $E$-multiplet and a $b_+$-multiplet of $\mathsf{Rep}(S_3)$, whose detailed form was discussed above (V.90) and (V.98) respectively. Applying KT transformations (V.96) and (V.98) respectively to the identity operator id and the spin operator $\sigma$ of the Ising CFT, we deduce that these order parameters are

$$\mathcal{O}_{E,+}^{(0)} = \text{id}, \quad \mathcal{O}_{E,-}^{(0)} = \text{id}_\eta, \quad \mathcal{O}_{b_+} = \sigma, \quad \mathcal{O}_{b_+,+} = \sigma$$
$$\mathcal{O}_{b_+,-} = \sigma \otimes \text{id}_\eta\,, \tag{VI.132}$$

where $\text{id}_\eta$ is the identity local operator on the $\eta$ line. Here $\mathcal{O}_{E,\pm}^{(0)}$ form an $E$-multiplet and $\mathcal{O}_{b_+}, \mathcal{O}_{b_+,\pm}$ form a $b_+$-multiplet of $\mathsf{Rep}(S_3)$. The IR images of above operators are

$$\mathcal{O}_{E,+}^{(0)} \equiv v_0 + v_1, \quad \mathcal{O}_{E,-}^{(0)} \equiv \text{id}_{01} + \text{id}_{10}, \quad \mathcal{O}_{b_+} \equiv v_0 - v_1$$
$$\mathcal{O}_{b_+,+} \equiv v_0 - v_1, \quad \mathcal{O}_{b_+,-} \equiv \text{id}_{01} - \text{id}_{10}\,, \tag{VI.133}$$

where $\text{id}_{ij}$ is the identity local operator on the $1_{ij}$ vacua changing line.

The order parameters for the trivial phase $[\mathfrak{T}_2^{\mathcal{S}}]$ involve a $1_-$-multiplet and two distinct $E$-multiplets of $\mathsf{Rep}(S_3)$, whose detailed form was discussed above (V.88) and (V.90) respectively. Applying KT transformations (V.96) and (V.97) respectively to the identity operator id and the disorder operator $\mu$ of the Ising CFT, we deduce that these order parameters are

$$\mathcal{O}_{E,+}^{(0)} = \text{id}, \quad \mathcal{O}_{E,-}^{(0)} = \text{id}_P, \quad \mathcal{O}_{1_-} = \mu, \quad \mathcal{O}_{E,+}^{(1)} = \mu$$
$$\mathcal{O}_{E,-}^{(1)} = \mu\,. \tag{VI.134}$$

Here $\mathcal{O}_{E,\pm}^{(k)}$ for $k = 0, 1$ form two $E$-multiplets and $\mathcal{O}_{1_-}$ forms a $1_-$-multiplet of $\mathsf{Rep}(S_3)$. The IR images of above operators are

$$\mathcal{O}_{E,+}^{(0)} \equiv 1, \quad \mathcal{O}_{E,-}^{(0)} \equiv 1, \quad \mathcal{O}_{1_-} \equiv 1, \quad \mathcal{O}_{E,+}^{(1)} \equiv 1$$
$$\mathcal{O}_{E,-}^{(1)} \equiv 1\,. \tag{VI.135}$$

That is, the above operators are simply the identity operator viewed in different fashions as transitioning between different line operators associated to the $\mathsf{Rep}(S_3)$ symmetry. This result is not surprising as identity operator is the only operator available in the trivial phase.

## J. Phase Transition between $\mathsf{Rep}(S_3)/\mathbb{Z}_2$ SSB and $\mathsf{Rep}(S_3)$ SSB Phases for $\mathsf{Rep}(S_3)$ Symmetry

Now consider the KT transformation studied in section V J. This KT transformation converts $\mathbb{Z}_2$ symmetry to $\mathsf{Rep}(S_3)$ symmetry. We again use the Ising CFT to obtain a $\mathsf{Rep}(S_3)$-symmetric phase transition described below.

Using the map (V.107), we can quickly deduce that the $\mathsf{Rep}(S_3)$-symmetric gapped phases obtained after applying KT transformation on $\mathbb{Z}_2$ SSB and trivial phases for $\mathbb{Z}_2$ symmetry are described respectively by the physical Lagrangian algebras

$$\mathcal{L}_{\mathfrak{T}_1^{\mathcal{S}}}^{\text{phys}} = 1 \oplus 1_- \oplus 2a_1$$
$$\mathcal{L}_{\mathfrak{T}_2^{\mathcal{S}}}^{\text{phys}} = 1 \oplus a_1 \oplus b_+ \tag{VI.136}$$

which correspond to the following $\mathsf{Rep}(S_3)$-symmetric gapped phases respectively

$$[\mathfrak{T}_1^{\mathcal{S}}] = \mathsf{Rep}(S_3)/\mathbb{Z}_2 \text{ SSB phase for } \mathsf{Rep}(S_3) \text{ symmetry}$$
$$[\mathfrak{T}_2^{\mathcal{S}}] = \mathsf{Rep}(S_3) \text{ SSB phase for } \mathsf{Rep}(S_3) \text{ symmetry} \tag{VI.137}$$

For more details on these gapped phases, we refer the reader to [2].

Another equivalent way to deduce these $\mathsf{Rep}(S_3)$-symmetric gapped phases is to apply the results of section V J. Applying them to the theory $\mathfrak{T}^{\mathcal{S}'} = \mathfrak{T}_1^{\mathcal{S}'}$ appearing in (VI.23), we learn that the universes involved in the KT transformed theory $\mathfrak{T}^{\mathcal{S}} = \mathfrak{T}_1^{\mathcal{S}}$ are

$$(\mathfrak{T}_1^{\mathcal{S}})_e = \mathbb{Z}_2 \text{ SSB} = \text{Trivial}_0 \oplus \text{Trivial}_1$$
$$(\mathfrak{T}_1^{\mathcal{S}})_m = \widehat{\mathbb{Z}}_2 \text{ Trivial} = \text{Trivial}_2\,, \tag{VI.138}$$

where $\widehat{\mathbb{Z}}_2$ denotes the dual symmetry obtained after gauging the original $\mathbb{Z}_2$ symmetry of $\mathfrak{T}_1^{\mathcal{S}'}$. Thus in total, $[\mathfrak{T}_1^{\mathcal{S}}]$ is a $\mathsf{Rep}(S_3)$-symmetric (1+1)d gapped phase with three vacua. Now the question arises which one it is, because there are two possible $\mathsf{Rep}(S_3)$-symmetric gapped phases having three vacua, namely the $\mathsf{Rep}(S_3)/\mathbb{Z}_2$ SSB and $\mathsf{Rep}(S_3)$ SSB phases (see [2] for more details). In order to answer this, we need to understand the precise form of the line operators implementing the $\mathsf{Rep}(S_3)$ symmetry

on $\mathfrak{T}_1^{\mathcal{S}}$, which follows from (IV.158). The identification of the lines appearing there is

$$1_{ee} \equiv 1_{00} \oplus 1_{11}, \quad 1_{mm} \equiv 1_{22} \quad S_{em} \equiv 1_{02} \oplus 1_{12},$$
$$P_{ee} \equiv 1_{01} \oplus 1_{10}, \quad P_{mm} \equiv 1_{22} \quad S_{me} \equiv 1_{20} \oplus 1_{21}. \tag{VI.139}$$

This implies that the $\mathsf{Rep}(S_3)$ symmetry generators are identified as

$$1_- \equiv 1_{00} \oplus 1_{11} \oplus 1_{22}$$
$$E \equiv 1_{02} \oplus 1_{12} \oplus 1_{20} \oplus 1_{21} \oplus 1_{01} \oplus 1_{10}. \tag{VI.140}$$

Note that there are no relative Euler terms between the three vacua since the linking action (V.104) becomes

$$S_{em}: v_0 + v_1 \to 2v_2 \qquad S_{me}: v_2 \to v_0 + v_1, \tag{VI.141}$$

which means that the linking actions of the vacua changing lines are

$$1_{02}: v_0 \to v_2, \qquad 1_{20}: v_2 \to v_0$$
$$1_{12}: v_1 \to v_2, \qquad 1_{21}: v_2 \to v_1 \tag{VI.142}$$

implying that there are no relative Euler terms between the three vacua. All these properties imply that $[\mathfrak{T}_1^{\mathcal{S}}]$ is the $\mathsf{Rep}(S_3)/\mathbb{Z}_2$ SSB phase. In particular, note that the $\mathbb{Z}_2$ subsymmetry of $\mathsf{Rep}(S_3)$ remains spontaneously unbroken as we have found in (VI.140) that $1_-$ is realized by the identity line operator. The three vacua are thus mixed into each other solely by the action of the non-invertible symmetry $E$.

Applying the results of section V J to the theory $\mathfrak{T}^{\mathcal{S}'} = \mathfrak{T}_2^{\mathcal{S}'}$ appearing in (VI.27), we learn that the universes involved in the KT transformed theory $\mathfrak{T}^{\mathcal{S}} = \mathfrak{T}_2^{\mathcal{S}}$ are

$$(\mathfrak{T}_2^{\mathcal{S}})_e = \mathbb{Z}_2 \text{ Trivial} = \text{Trivial}_0$$
$$(\mathfrak{T}_2^{\mathcal{S}})_m = \widehat{\mathbb{Z}}_2 \text{ SSB} = \text{Trivial}_1 \oplus \text{Trivial}_2 \tag{VI.143}$$

Thus in total, $[\mathfrak{T}_2^{\mathcal{S}}]$ is a $\mathsf{Rep}(S_3)$-symmetric (1+1)d gapped phase with three vacua. Again the question arises whether it is the $\mathsf{Rep}(S_3)/\mathbb{Z}_2$ SSB phase or the $\mathsf{Rep}(S_3)$ SSB phase. The identification of the lines appearing in (IV.158) is now

$$1_{ee} \equiv 1_{00}, \quad 1_{mm} \equiv 1_{11} \oplus 1_{22} \quad S_{em} \equiv 1_{01} \oplus 1_{02},$$
$$P_{ee} \equiv 1_{00}, \quad P_{mm} \equiv 1_{12} \oplus 1_{21} \quad S_{me} \equiv 1_{10} \oplus 1_{20}. \tag{VI.144}$$

This implies that the $\mathsf{Rep}(S_3)$ symmetry generators are identified as

$$1_- \equiv 1_{00} \oplus 1_{12} \oplus 1_{21}$$
$$E \equiv 1_{01} \oplus 1_{02} \oplus 1_{10} \oplus 1_{20} \oplus 1_{00}. \tag{VI.145}$$

Note that there are **non-trivial relative Euler terms** between the three vacua now. To see this, note that the linking action (V.104) becomes

$$S_{em}: v_0 \to 2v_1 + 2v_2 \qquad S_{me}: v_1 + v_2 \to v_0, \tag{VI.146}$$

which means that the linking actions of the vacua changing lines are

$$1_{01}: v_0 \to 2v_1, \qquad 1_{10}: v_1 \to v_0/2$$
$$1_{02}: v_0 \to 2v_2, \qquad 1_{20}: v_2 \to v_0/2 \tag{VI.147}$$

implying that there are non-trivial relative Euler terms between vacua 0 and 1, and vacua 0 and 2, but trivial relative Euler terms between vacua 1 and 2. All these properties imply that $[\mathfrak{T}_2^{\mathcal{S}}]$ is the $\mathsf{Rep}(S_3)$ SSB phase in which both $1_-$ and $E$ are spontaneously broken.

The $\mathsf{Rep}(S_3)$-symmetric phase transition $\mathfrak{T}_C^{\mathcal{S}}$ obtained after applying the KT transformation to the Ising phase transition can be expressed as follows. It comprises of two universes

$$(\mathfrak{T}_C^{\mathcal{S}})_e = \text{Ising}_e, \quad (\mathfrak{T}_C^{\mathcal{S}})_m = (\text{Ising}/\mathbb{Z}_2)_{\sqrt{2}} = (\text{Ising}_m)_{\sqrt{2}}, \tag{VI.148}$$

where we have used the well-known isomorphism of $\text{Ising}/\mathbb{Z}_2$ with Ising to express it as a copy $\text{Ising}_m$ of Ising CFT. Note that there is relative Euler term between the two Ising universes with the linking action of universe changing lines

$$1_{em}: \text{id}_e \to \sqrt{2}\text{id}_m \qquad 1_{me}: \text{id}_m \to \text{id}_e/\sqrt{2} \tag{VI.149}$$

on the identity local operators $\text{id}_e$ and $\text{id}_m$ of $\text{Ising}_e$ and $\text{Ising}_m$ respectively. These linking actions reproduce (V.104) for lines $S_{em}$ and $S_{me}$ as these lines can be expressed as

$$S_{em} = S_{ee} \otimes 1_{em} = 1_{em} \otimes S_{mm}$$
$$S_{me} = S_{mm} \otimes 1_{me} = 1_{me} \otimes S_{ee}, \tag{VI.150}$$

where $S_{ee}$ and $S_{mm}$ are Kramers-Wannier duality defects of $\text{Ising}_e$ and $\text{Ising}_m$ respectively, whose quantum dimensions are $\sqrt{2}$, i.e.

$$S_{ee}: \text{id}_e \to \sqrt{2}\text{id}_e, \qquad S_{mm}: \text{id}_m \to \sqrt{2}\text{id}_m. \tag{VI.151}$$

The schematic form of $\mathfrak{T}_C^{\mathcal{S}}$ is thus

$$\mathfrak{T}_C^{\mathcal{S}} = E \underset{E}{\underbrace{\overset{\displaystyle\curvearrowright}{\text{Ising}_e \oplus (\text{Ising}_m)_{\sqrt{2}}}}} 1_- \tag{VI.152}$$

The $\mathsf{Rep}(S_3)$ symmetry is realized as

$$1_- = 1_{ee} \oplus \eta_{mm}, \quad E = S_{em} \oplus S_{me} \oplus \eta_{ee}, \tag{VI.153}$$

where $\eta_{ee}$ and $\eta_{mm}$ are the $\mathbb{Z}_2$ symmetries of $\text{Ising}_e$ and $\text{Ising}_m$ respectively.

The relevant operator responsible for the transition is $\epsilon_e - \epsilon_m$. Adding this operator with positive sign sends $\text{Ising}_e$ to $\mathbb{Z}_2$ SSB phase and $\text{Ising}_m$ to $\widehat{\mathbb{Z}}_2$ trivial phase, and hence we land on the $\mathsf{Rep}(S_3)/\mathbb{Z}_2$ SSB phase discussed above. On the other hand, adding this operator with negative sign sends $\text{Ising}_e$ to $\mathbb{Z}_2$ trivial phase

and Ising$_m$ to $\widehat{\mathbb{Z}}_2$ SSB phase, and hence we land on the Rep($S_3$) SSB phase discussed above.

The order parameters for the Rep($S_3$)/$Z_2$ SSB phase $[\mathfrak{T}_1^{\mathcal{S}}]$ involve a $1_-$-multiplet and two distinct $a_1$-multiplets of Rep($S_3$). Applying KT transformations (V.108) and (V.109) respectively to the identity operator id and the spin operator $\sigma$ of the Ising CFT, we deduce that these order parameters are

$$\mathcal{O}_{a_1,u}^{(0)} = \mathrm{id}_e - 2\mathrm{id}_m$$
$$\mathcal{O}_{a_1,t}^{(0)} = -\frac{3\mathrm{id}_e}{1 + 2\omega}$$
$$\mathcal{O}_{1_-} = \sigma_e + \mu_m \qquad \text{(VI.154)}$$
$$\mathcal{O}_{a_1,u}^{(1)} = -\frac{3\sigma_e}{1 + 2\omega}$$
$$\mathcal{O}_{a_1,t}^{(1)} = \sigma_e - 2\mu_m \,,$$

where $\{\mathcal{O}_{a_1,u}^{(k)}, \mathcal{O}_{a_1,t}^{(k)}\}$ for $k = 0, 1$ are the two $a_1$-multiplets and the operator $\mathcal{O}_{1_-}$ comprises the $1_-$-multiplet. The IR images of above operators are

$$\mathcal{O}_{a_1,u}^{(0)} \equiv v_0 + v_1 - 2v_2$$
$$\mathcal{O}_{a_1,t}^{(0)} \equiv -\frac{3(v_0 + v_1)}{1 + 2\omega}$$
$$\mathcal{O}_{1_-} \equiv v_0 - v_1 + v_2 \qquad \text{(VI.155)}$$
$$\mathcal{O}_{a_1,u}^{(1)} \equiv -\frac{3(v_0 - v_1)}{1 + 2\omega}$$
$$\mathcal{O}_{a_1,t}^{(1)} \equiv v_0 - v_1 - 2v_2 \,.$$

The order parameters for the Rep($S_3$) SSB phase $[\mathfrak{T}_2^{\mathcal{S}}]$ involve an $a_1$-multiplet and a $b_+$-multiplet of Rep($S_3$). Applying KT transformations (V.108) and (V.110) respectively to the identity operator id and the disorder operator $\mu$ of the Ising CFT, we deduce that these order parameters are

$$\mathcal{O}_{a_1,u}^{(0)} = \mathrm{id}_e - 2\mathrm{id}_m$$
$$\mathcal{O}_{a_1,t}^{(0)} = -\frac{3\mathrm{id}_e}{1 + 2\omega}$$
$$\mathcal{O}_{b_+} = \sigma_m \qquad \text{(VI.156)}$$
$$\mathcal{O}_{b_+,+} = \mu_e$$
$$\mathcal{O}_{b_+,-} = \mu_e \,,$$

where $\{\mathcal{O}_{a_1,u}^{(0)}, \mathcal{O}_{a_1,t}^{(0)}\}$ is the $a_1$-multiplet and $\{\mathcal{O}_{b_+}, \mathcal{O}_{b_+,+}, \mathcal{O}_{b_+,-}\}$ the $b_+$-multiplet. The IR images of above operators are

$$\mathcal{O}_{a_1,u}^{(0)} \equiv v_0 - 2v_1 - 2v_2$$
$$\mathcal{O}_{a_1,t}^{(0)} \equiv -\frac{3v_0}{1 + 2\omega}$$
$$\mathcal{O}_{b_+} \equiv v_1 - v_2 \qquad \text{(VI.157)}$$
$$\mathcal{O}_{b_+,+} \equiv v_0$$
$$\mathcal{O}_{b_+,-} \equiv v_0 \,.$$

## K. Phase Transition between two Ising SSB Phases for Ising Symmetry

Now consider the KT transformation studied in section V K. This KT transformation converts $\mathbb{Z}_2$ symmetry to Ising $=$ TY($\mathbb{Z}_2$) symmetry. We use the Ising CFT to now obtain an Ising-symmetric phase transition described below.

Using the map (V.119), we can quickly deduce that the Ising-symmetric gapped phases obtained after applying KT transformation on $\mathbb{Z}_2$ SSB and trivial phases for $\mathbb{Z}_2$ symmetry are described respectively by the physical Lagrangian algebras

$$\mathcal{L}_{\mathfrak{T}_1^{\mathcal{S}}}^{\mathrm{phys}} = 1 \oplus P\overline{P} \oplus S\overline{S}$$
$$\mathcal{L}_{\mathfrak{T}_2^{\mathcal{S}}}^{\mathrm{phys}} = 1 \oplus P\overline{P} \oplus S\overline{S}. \qquad \text{(VI.158)}$$

Thus, we have the same Ising-symmetric gapped phase on both sides of the transition, which is the only possible irreducible Ising-symmetric gapped phase that we refer to as the Ising SSB phase

$$[\mathfrak{T}_1^{\mathcal{S}}] = \text{Ising SSB phase for Ising symmetry}$$
$$[\mathfrak{T}_2^{\mathcal{S}}] = \text{Ising SSB phase for Ising symmetry} \qquad \text{(VI.159)}$$

For more details on this gapped phase, we refer the reader to [2].

Another equivalent way to deduce this Ising-symmetric gapped phase is to apply the results of section V K. Applying them to the theory $\mathfrak{T}^{\mathcal{S}'} = \mathfrak{T}_1^{\mathcal{S}'}$ appearing in (VI.23), we learn that the universes involved in the KT transformed theory $\mathfrak{T}^{\mathcal{S}} = \mathfrak{T}_1^{\mathcal{S}}$ are

$$(\mathfrak{T}_1^{\mathcal{S}})_e = \mathbb{Z}_2 \text{ SSB} = \text{Trivial}_0 \oplus \text{Trivial}_1$$
$$(\mathfrak{T}_1^{\mathcal{S}})_m = \widehat{\mathbb{Z}}_2 \text{ Trivial} = \text{Trivial}_2 \,, \qquad \text{(VI.160)}$$

where $\widehat{\mathbb{Z}}_2$ denotes the dual symmetry obtained after gauging the original $\mathbb{Z}_2$ symmetry of $\mathfrak{T}_1^{\mathcal{S}'}$. Thus in total, $[\mathfrak{T}_1^{\mathcal{S}}]$ is an Ising-symmetric (1+1)d gapped phase with three vacua. The identification of the lines appearing in section V K is

$$1_{ee} \equiv 1_{00} \oplus 1_{11}, \quad 1_{mm} \equiv 1_{22} \quad S_{em} \equiv 1_{02} \oplus 1_{12},$$
$$P_{ee} \equiv 1_{01} \oplus 1_{10}, \quad P_{mm} \equiv 1_{22} \quad S_{me} \equiv 1_{20} \oplus 1_{21} . \qquad \text{(VI.161)}$$

This implies that the Ising symmetry generators are identified as

$$P \equiv 1_{01} \oplus 1_{10} \oplus 1_{22}$$
$$S \equiv 1_{02} \oplus 1_{12} \oplus 1_{20} \oplus 1_{21} \,, \qquad \text{(VI.162)}$$

where we have used (IV.174). Note that there are **nontrivial relative Euler terms** between the three vacua since the linking action (V.116) becomes

$$S_{em} : \ v_0 + v_1 \to \sqrt{2}v_2 \qquad S_{me} : \ v_2 \to \sqrt{2}(v_0 + v_1) \,, \qquad \text{(VI.163)}$$

which means that the linking actions of the vacua changing lines are

$$1_{02} : \ v_0 \to v_2/\sqrt{2} \,, \qquad 1_{20} : \ v_2 \to \sqrt{2}v_0$$
$$1_{12} : \ v_1 \to v_2/\sqrt{2} \,, \qquad 1_{21} : \ v_2 \to \sqrt{2}v_1 \,, \qquad \text{(VI.164)}$$

implying non-trivial relative Euler terms between vacua 0 and 2, and vacua 1 and 2, but trivial relative Euler terms between vacua 0 and 1. All these properties imply that $[\mathfrak{T}_1^{\mathcal{S}}]$ is the Ising SSB phase in which both $P$ and $S$ are spontaneously broken.

Applying the results of section V K to the theory $\mathfrak{T}^{\mathcal{S}'} = \mathfrak{T}_2^{\mathcal{S}'}$ appearing in (VI.27) leads to a similar analysis.

The Ising-symmetric phase transition $\mathfrak{T}_C^{\mathcal{S}}$ obtained after applying the KT transformation to the $\mathbb{Z}_2$-symmetric Ising phase transition can be expressed as follows. It comprises of two universes

$$(\mathfrak{T}_C^{\mathcal{S}})_e = \text{Ising}_e \,, \quad (\mathfrak{T}_C^{\mathcal{S}})_m = \text{Ising}/\mathbb{Z}_2 = \text{Ising}_m \,, \qquad \text{(VI.165)}$$

where we have used the well-known isomorphism of $\text{Ising}/\mathbb{Z}_2$ with Ising to express it as a copy $\text{Ising}_m$ of Ising CFT. Note that there is no relative Euler term between the two Ising universes as the linking action of universe changing lines is

$$1_{em} : \ \text{id}_e \to \text{id}_m \qquad 1_{me} : \ \text{id}_m \to \text{id}_e \qquad \text{(VI.166)}$$

on the identity local operators $\text{id}_e$ and $\text{id}_m$ of $\text{Ising}_e$ and $\text{Ising}_m$ respectively. These linking actions reproduce (V.116) for lines $S_{em}$ and $S_{me}$ as these lines can be expressed as

$$S_{em} = S_{ee} \otimes 1_{em} = 1_{em} \otimes S_{mm}$$
$$S_{me} = S_{mm} \otimes 1_{me} = 1_{me} \otimes S_{ee} \,, \qquad \text{(VI.167)}$$

where $S_{ee}$ and $S_{mm}$ are Kramers-Wannier duality defects of $\text{Ising}_e$ and $\text{Ising}_m$ respectively, whose quantum dimensions are $\sqrt{2}$, i.e.

$$S_{ee} : \ \text{id}_e \to \sqrt{2}\text{id}_e \,, \qquad S_{mm} : \ \text{id}_m \to \sqrt{2}\text{id}_m \,. \qquad \text{(VI.168)}$$

The schematic form of $\mathfrak{T}_C^{\mathcal{S}}$ is thus

$$\mathfrak{T}_C^{\mathcal{S}} \ = \ P \ \overset{\curvearrowright}{\phantom{x}} \ \text{Ising}_e \oplus \text{Ising}_m \ \overset{\curvearrowleft}{\phantom{x}} \ P$$
$$\underbrace{\phantom{xxxxxxxxxxxx}}_{S} \qquad \text{(VI.169)}$$

The Ising symmetry is realized as

$$P = \eta_{ee} \oplus \eta_{mm} \,, \quad S = S_{em} \oplus S_{me} \,, \qquad \text{(VI.170)}$$

where $\eta_{ee}$ and $\eta_{mm}$ are the $\mathbb{Z}_2$ symmetries of $\text{Ising}_e$ and $\text{Ising}_m$ respectively.

The relevant operator responsible for the transition is $\epsilon_e - \epsilon_m$. Adding this operator with positive sign sends $\text{Ising}_e$ to $\mathbb{Z}_2$ SSB phase and $\text{Ising}_m$ to $\widehat{\mathbb{Z}}_2$ trivial phase, and hence we land on the Ising SSB phase. On the other hand, adding this operator with negative sign sends $\text{Ising}_e$ to $\mathbb{Z}_2$ trivial phase and $\text{Ising}_m$ to $\widehat{\mathbb{Z}}_2$ SSB phase, and hence we again land on the Ising SSB phase.

The order parameters for the Ising SSB phase $[\mathfrak{T}_1^{\mathcal{S}}]$ involves an operator with generalized charge $P\overline{P}$ and two operators forming an $S\overline{S}$-multiplet of Ising symmetry. Applying KT transformations (V.120) and (V.121) respectively to the identity operator id and the spin operator $\sigma$ of the Ising CFT, we deduce that these order parameters are

$$\mathcal{O}_{P\overline{P}} = \text{id}_e - \text{id}_m \,, \quad \mathcal{O}_{S\overline{S},u}^{(e)} = \sigma_e \,, \quad \mathcal{O}_{S\overline{S},t}^{(e)} = \mu_m \,, \qquad \text{(VI.171)}$$

where $\mathcal{O}_{P\overline{P}}$ is the operator transforming in generalized charge $P\overline{P}$ and the operators $\{\mathcal{O}_{S\overline{S},u}^{(e)}, \mathcal{O}_{S\overline{S},t}^{(e)}\}$ forming the $S\overline{S}$-multiplet. The IR images of above operators are

$$\mathcal{O}_{P\overline{P}} \equiv v_0 + v_1 - v_2 \,, \quad \mathcal{O}_{S\overline{S},u}^{(e)} \equiv v_0 - v_1 \,, \quad \mathcal{O}_{S\overline{S},t}^{(e)} = v_2 \,. \qquad \text{(VI.172)}$$

The order parameters for the Ising SSB phase $[\mathfrak{T}_2^{\mathcal{S}}]$ also involve an operator with generalized charge $P\overline{P}$ and two operators forming an $S\overline{S}$-multiplet of Ising symmetry. Applying KT transformations (V.120) and (V.122) respectively to the identity operator id and the disorder operator $\mu$ of the Ising CFT, we deduce that these order parameters are

$$\mathcal{O}_{P\overline{P}} = \text{id}_e - \text{id}_m \,, \quad \mathcal{O}_{S\overline{S},u}^{(m)} = \sigma_m \,, \quad \mathcal{O}_{S\overline{S},t}^{(m)} = \mu_e \,, \qquad \text{(VI.173)}$$

where $\mathcal{O}_{P\overline{P}}$ is the operator transforming in generalized charge $P\overline{P}$ and the operators $\{\mathcal{O}_{S\overline{S},u}^{(m)}, \mathcal{O}_{S\overline{S},t}^{(m)}\}$ forming the $S\overline{S}$-multiplet. The IR images of above operators are

$$\mathcal{O}_{P\overline{P}} \equiv v_0 - v_1 - v_2 \,, \quad \mathcal{O}_{S\overline{S},u}^{(e)} \equiv v_1 - v_2 \,, \quad \mathcal{O}_{S\overline{S},t}^{(e)} \equiv v_0 \,. \qquad \text{(VI.174)}$$

### L. Phase Transition between $(\mathbb{Z}_2, \mathbb{Z}_2)$ SSB and $(\mathbb{Z}_1, \mathbb{Z}_4)$ SSB Phases for $\text{TY}(\mathbb{Z}_4)$ Symmetry

Now consider the KT transformation studied in section V L. This KT transformation converts $\mathbb{Z}_2$ symmetry to $\text{TY}(\mathbb{Z}_4)$ symmetry. We use the Ising CFT to obtain a $\text{TY}(\mathbb{Z}_4)$-symmetric phase transition described below.

Using the map (V.130), we can quickly deduce that the $\text{TY}(\mathbb{Z}_4)$-symmetric gapped phases obtained after applying KT transformation on $\mathbb{Z}_2$ SSB and trivial phases for $\mathbb{Z}_2$ symmetry are described respectively by the physical Lagrangian algebras

$$\mathcal{L}_{\mathfrak{T}_1^{\mathcal{S}}}^{\text{phys}} = L_0^+ + L_0^- + L_2^+ + L_2^- + 2L_{2,0}$$
$$\mathcal{L}_{\mathfrak{T}_2^{\mathcal{S}}}^{\text{phys}} = L_0^+ + L_0^- + L_{1,0} + L_{2,0} + L_{3,0} \,, \qquad \text{(VI.175)}$$

which correspond to the following $\text{TY}(\mathbb{Z}_4)$-symmetric

gapped phases respectively

$$[\mathfrak{T}_1^{\mathcal{S}}] = (\mathbb{Z}_2, \mathbb{Z}_2) \text{ SSB phase for } \mathsf{TY}(\mathbb{Z}_4) \text{ symmetry}$$

$$[\mathfrak{T}_2^{\mathcal{S}}] = (\mathbb{Z}_1, \mathbb{Z}_4) \text{ SSB phase for } \mathsf{TY}(\mathbb{Z}_4) \text{ symmetry}. \quad \text{(VI.176)}$$

A $(\mathbb{Z}_p, \mathbb{Z}_q)$ SSB phase for $\mathsf{TY}(\mathbb{Z}_N)$ symmetry with $pq = N$ has $p + q$ vacua, where the first $p$ vacua comprise a $\mathbb{Z}_p$ SSB phase for the $\mathbb{Z}_N$ subsymmetry, and the last $q$ vacua comprise a $\mathbb{Z}_q$ SSB phase for the $\mathbb{Z}_N$ subsymmetry, with these $\mathbb{Z}_p$ and $\mathbb{Z}_q$ SSB phases exchanged by the action of non-invertible duality defect $S \in \mathsf{TY}(\mathbb{Z}_N)$. For more details on these gapped phases, we refer the reader to [2].

Another equivalent way to deduce these $\mathsf{TY}(Z_4)$-symmetric gapped phases is to apply the results of section V L. Applying them to the theory $\mathfrak{T}^{\mathcal{S}'} = \mathfrak{T}_1^{\mathcal{S}'}$ appearing in (VI.23), we learn that the universes involved in the KT transformed theory $\mathfrak{T}^{\mathcal{S}} = \mathfrak{T}_1^{\mathcal{S}}$ are

$$(\mathfrak{T}_1^{\mathcal{S}})_0^e = \mathbb{Z}_2 \text{ SSB} = \text{Trivial}_0 \oplus \text{Trivial}_1$$

$$(\mathfrak{T}_1^{\mathcal{S}})_1^m = \widehat{\mathbb{Z}}_2 \text{ Trivial} = \text{Trivial}_2 \quad \text{(VI.177)}$$

$$(\mathfrak{T}_1^{\mathcal{S}})_2^m = \widehat{\mathbb{Z}}_2 \text{ Trivial} = \text{Trivial}_3$$

where $\widehat{\mathbb{Z}}_2$ denotes the dual symmetry obtained after gauging the original $\mathbb{Z}_2$ symmetry of $\mathfrak{T}_1^{\mathcal{S}'}$. Thus in total, $[\mathfrak{T}_1^{\mathcal{S}}]$ is a $\mathsf{TY}(\mathbb{Z}_4)$-symmetric (1+1)d gapped phase with four vacua. This is enough to fix it to $(\mathbb{Z}_2, \mathbb{Z}_2)$ SSB phase. We can also identify the $\mathsf{TY}(\mathbb{Z}_4)$ lines following (IV.194) as

$$A \equiv 1_{01} \oplus 1_{10} \oplus 1_{23} \oplus 1_{32}$$
$$S \equiv 1_{02} \oplus 1_{12} \oplus 1_{03} \oplus 1_{13} \oplus 1_{20} \oplus 1_{21} \oplus 1_{30} \oplus 1_{31}, \quad \text{(VI.178)}$$

which is precisely the realization of $\mathsf{TY}(\mathbb{Z}_4)$ symmetry in the $(\mathbb{Z}_2, \mathbb{Z}_2)$ SSB phase. Note that (IV.198) is satisfied by linking action

$$1_{02}: \ v_0 \to v_2, \quad \text{(VI.179)}$$

which implies that there are no relative Euler terms between any of the four vacua, reproducing again a property expected of the $(\mathbb{Z}_2, \mathbb{Z}_2)$ SSB phase as described in [2].

Applying the results of section V L to the theory $\mathfrak{T}^{\mathcal{S}'} = \mathfrak{T}_2^{\mathcal{S}'}$ appearing in (VI.27), we learn that the universes involved in the KT transformed theory $\mathfrak{T}^{\mathcal{S}} = \mathfrak{T}_2^{\mathcal{S}}$ are

$$(\mathfrak{T}_2^{\mathcal{S}})_0^e = \mathbb{Z}_2 \text{ Trivial} = \text{Trivial}_0$$

$$(\mathfrak{T}_2^{\mathcal{S}})_1^m = \widehat{\mathbb{Z}}_2 \text{ SSB} = \text{Trivial}_1 \oplus \text{Trivial}_2 \quad \text{(VI.180)}$$

$$(\mathfrak{T}_2^{\mathcal{S}})_2^m = \widehat{\mathbb{Z}}_2 \text{ SSB} = \text{Trivial}_3 \oplus \text{Trivial}_4$$

where $\widehat{\mathbb{Z}}_2$ denotes the dual symmetry obtained after gauging the original $\mathbb{Z}_2$ symmetry of $\mathfrak{T}_2^{\mathcal{S}'}$. Thus in total, $[\mathfrak{T}_2^{\mathcal{S}}]$ is a $\mathsf{TY}(\mathbb{Z}_4)$-symmetric (1+1)d gapped phase with five vacua. This is enough to fix it to $(\mathbb{Z}_1, \mathbb{Z}_4)$ SSB phase. We can also identify the $\mathsf{TY}(\mathbb{Z}_4)$ lines following (IV.194) as

$$A \equiv 1_{00} \oplus 1_{13} \oplus 1_{24} \oplus 1_{32} \oplus 1_{41}$$
$$S \equiv 1_{01} \oplus 1_{02} \oplus 1_{03} \oplus 1_{04} \oplus 1_{10} \oplus 1_{20} \oplus 1_{30} \oplus 1_{40}, \quad \text{(VI.181)}$$

which is precisely the realization of $\mathsf{TY}(\mathbb{Z}_4)$ symmetry in the $(\mathbb{Z}_1, \mathbb{Z}_4)$ SSB phase. Note that (IV.198) is satisfied by linking action

$$1_{01}: \ v_0 \to 2v_1, \quad \text{(VI.182)}$$

which implies that there is a **non-trivial relative Euler term** between vacua 0 and $i \in \{1,2,3,4\}$ but trivial relative Euler terms between vacua $i \in \{1,2,3,4\}$ and $j \in \{1,2,3,4\}$, reproducing again a property expected of the $(\mathbb{Z}_1, \mathbb{Z}_4)$ SSB phase as described in [2].

The $\mathsf{TY}(\mathbb{Z}_4)$-symmetric phase transition $\mathfrak{T}_C^{\mathcal{S}}$ obtained after applying the KT transformation to the Ising phase transition can be expressed as follows. It comprises of three universes

$$(\mathfrak{T}_C^{\mathcal{S}})_0^e = \text{Ising}_0^e, \quad \begin{aligned} (\mathfrak{T}_C^{\mathcal{S}})_1^m &= (\text{Ising}/\mathbb{Z}_2)_{\sqrt{2}} = (\text{Ising}_1^m)_{\sqrt{2}} \\ (\mathfrak{T}_C^{\mathcal{S}})_2^m &= (\text{Ising}/\mathbb{Z}_2)_{\sqrt{2}} = (\text{Ising}_2^m)_{\sqrt{2}}. \end{aligned} \quad \text{(VI.183)}$$

The schematic form of $\mathfrak{T}_C^{\mathcal{S}}$ is thus

$$\mathfrak{T}_C^{\mathcal{S}} = \text{Ising}_0^e \oplus \text{Ising}_1^m \oplus \text{Ising}_2^m \quad \text{(VI.184)}$$

The $\mathsf{TY}(\mathbb{Z}_4)$ symmetry is realized as

$$A = \eta_{00} \oplus 1_{12} \oplus \eta_{21}, \quad S = S_{01} \oplus S_{02} \oplus S_{10} \oplus S_{20}. \quad \text{(VI.185)}$$

The relevant operator responsible for the transition is $\epsilon_0^e - \epsilon_1^m - \epsilon_2^m$. Adding this operator with positive sign sends $\text{Ising}_0^e$ to $\mathbb{Z}_2$ SSB phase and $\text{Ising}_i^m$ to $\widehat{\mathbb{Z}}_2$ trivial phase, and hence we land on the $(\mathbb{Z}_2, \mathbb{Z}_2)$ SSB phase. On the other hand, adding this operator with negative sign sends $\text{Ising}_0^e$ to $\mathbb{Z}_2$ trivial phase and $\text{Ising}_i^m$ to $\widehat{\mathbb{Z}}_2$ SSB phase, and hence we land on the $(\mathbb{Z}_1, \mathbb{Z}_4)$ SSB phase.

The order parameters for the $(\mathbb{Z}_2, \mathbb{Z}_2)$ SSB phase $[\mathfrak{T}_1^{\mathcal{S}}]$ involve a copy each of $L_0^-, L_2^\pm$ multiplets and two distinct $L_{2,0}$ multiplets of $\mathsf{TY}(\mathbb{Z}_4)$ symmetry. Applying KT transformations (V.132) and (V.134) respectively to the identity operator id and the spin operator $\sigma$ of the Ising CFT, we deduce that these order parameters are

$$\mathcal{O}_0^- = \text{id}_0^e - \text{id}_1^m - \text{id}_2^m$$
$$\mathcal{O}_{2,0}^{u,(1)} = \text{id}_1^m - \text{id}_2^m$$
$$\mathcal{O}_{2,0}^{t,(1)} = \text{id}_0^e$$
$$\mathcal{O}_2^+ = \sigma_0^e + \mu_1^m - \mu_2^m \quad \text{(VI.186)}$$
$$\mathcal{O}_2^- = \sigma_0^e - \mu_1^m + \mu_2^m$$
$$\mathcal{O}_{2,0}^{u,(1)} = \sigma_0^e$$
$$\mathcal{O}_{2,0}^{t,(1)} = \mu_1^m + \mu_2^m,$$

where $\mathcal{O}_e^\pm$ form $L_e^\pm$ multiplets and $\{\mathcal{O}_{2,0}^{u,(k)}, \mathcal{O}_{2,0}^{t,(k)}\}$ for $k = 1, 2$ form the two $L_{2,0}$-multiplets of $\mathsf{TY}(\mathbb{Z}_4)$ symmetry.

The IR images of above operators are

$$
\begin{aligned}
\mathcal{O}_0^- &\equiv v_0 + v_1 - v_2 - v_3 \\
\mathcal{O}_{2,0}^{u,(1)} &\equiv v_2 - v_3 \\
\mathcal{O}_{2,0}^{t,(1)} &\equiv v_0 + v_1 \\
\mathcal{O}_2^+ &\equiv v_0 - v_1 + v_2 - v_3 \\
\mathcal{O}_2^- &\equiv v_0 - v_1 - v_2 + v_3 \\
\mathcal{O}_{2,0}^{u,(1)} &\equiv v_0 - v_1 \\
\mathcal{O}_{2,0}^{t,(1)} &\equiv v_2 + v_3 \,.
\end{aligned}
\tag{VI.187}
$$

The order parameters for the $(\mathbb{Z}_1, \mathbb{Z}_4)$ SSB phase $[\mathfrak{T}_2^{\mathcal{S}}]$ involve a copy each of $L_0^-, L_{1,0}, L_{2,0}, L_{3,0}$ multiplets of $\mathsf{TY}(\mathbb{Z}_4)$ symmetry. Applying KT transformations (V.132) and (V.136) respectively to the identity operator id and the disorder operator $\mu$ of the Ising CFT, we deduce that these order parameters are

$$
\begin{aligned}
\mathcal{O}_0^- &= \mathrm{id}_0^e - \mathrm{id}_1^m - \mathrm{id}_2^m \\
\mathcal{O}_{2,0}^{u,(1)} &= \mathrm{id}_1^m - \mathrm{id}_2^m \\
\mathcal{O}_{2,0}^{t,(1)} &= \mathrm{id}_0^e \\
\mathcal{O}_{1,0}^u &= \sigma_1^m - i\sigma_2^m \\
\mathcal{O}_{1,0}^t &= \mu_0^e \\
\mathcal{O}_{3,0}^u &= \sigma_1^m + i\sigma_2^m \\
\mathcal{O}_{3,0}^t &= \mu_0^e \,,
\end{aligned}
\tag{VI.188}
$$

where $\mathcal{O}_0^-$ forms $L_0^-$ multiplet, $\{\mathcal{O}_{2,0}^{u,(1)}, \mathcal{O}_{2,0}^{t,(1)}\}$ forms $L_{2,0}$-multiplet and $\{\mathcal{O}_{e,0}^u, \mathcal{O}_{e,0}^t\}$ form $L_{e,0}$-multiplets for $e = 1, 3$ of $\mathsf{TY}(\mathbb{Z}_4)$ symmetry. The IR images of above operators are

$$
\begin{aligned}
\mathcal{O}_0^- &\equiv v_0 - v_1 - v_2 - v_3 - v_4 \\
\mathcal{O}_{2,0}^{u,(1)} &\equiv v_1 + v_2 - v_3 - v_4 \\
\mathcal{O}_{2,0}^{t,(1)} &\equiv v_0 \\
\mathcal{O}_{1,0}^u &\equiv v_1 - v_2 - iv_3 + iv_4 \\
\mathcal{O}_{1,0}^t &\equiv v_0 \\
\mathcal{O}_{3,0}^u &\equiv v_1 - v_2 + iv_3 - iv_4 \\
\mathcal{O}_{3,0}^t &\equiv v_0 \,.
\end{aligned}
\tag{VI.189}
$$

## M. Extensions: Higher-Order Phase Transitions and Multi-Critical Points

We can also apply KT transformations to known higher-order phase transitions and multi-critical points to obtain new higher-order phase transitions and multi-critical points, including the ones involving non-invertible symmetries.

For example, it is well-known that the tri-critical Ising CFT admits a relevant deformation preserving the Ising symmetry that leads to the gapless Ising CFT on one side of the deformation and to the gapped Ising SSB phase on the other side of the deformation. See e.g. [133] for more details.

We can now apply KT transformations to the tricritical Ising CFT with $\mathcal{S}' = \mathsf{Ising}$, and obtain $\mathcal{S}$-symmetric CFTs (for some larger symmetry $\mathcal{S}$) that act as phase transitions between $\mathcal{S}$-symmetric gapless phases on one side, and $\mathcal{S}$-symmetric gapped phases on the other side.

Such considerations are natural extensions of our work, which we leave to future works. Note however that the conceptual part of these extensions involves KT transformations whose general theory has already been discussed here.

**Acknowledgements.** We thank Yunqin Zheng for discussions. SSN thanks King's College London for hospitality during the completion of this work. LB is funded as a Royal Society University Research Fellow through grant URF\R1\231467. The work of SSN is supported by the UKRI Frontier Research Grant, underwriting the ERC Advanced Grant "Generalized Symmetries in Quantum Field Theory and Quantum Gravity" and the Simons Foundation Collaboration on "Special Holonomy in Geometry, Analysis, and Physics", Award ID: 724073, Schafer-Nameki.

## Appendix A: Gapped (Boundary) Phases as Pivotal Higher-Functors

### 1.   General Systems with Non-Invertible Symmetries

Let $\mathcal{S}$ be a pivotal [134] fusion $(d-1)$-category characterizing symmetries of $d$-dimensional systems. Consider a $d$-dimensional system $\mathfrak{T}$, which may be a bulk or boundary system, or even a defect inside a higher-dimensional system. The $\mathcal{S}$ symmetry is realized on $\mathfrak{T}$ by specifying a pivotal tensor $(d-1)$-functor

$$\phi: \ \mathcal{S} \to \mathcal{C}_{d-1}(\mathfrak{T}), \qquad (A.1)$$

where $\mathcal{C}_{d-1}(\mathfrak{T})$ is the pivotal monoidal $(d-1)$-category formed by topological defects of the system $\mathfrak{T}$. Different functors provide different realizations of the symmetry $\mathcal{S}$ on $\mathfrak{T}$. This includes different couplings to background fields for $\mathcal{S}$ symmetry, including discrete torsion. The non-existence of such a functor means that $\mathfrak{T}$ cannot be made $\mathcal{S}$-symmetric.

The system $\mathfrak{T}$ may naturally lie in a family $\mathcal{F}$ of $d$-dimensional systems satisfying certain desired properties. Then, we have a pivotal $d$-category $\mathcal{C}_d(\mathcal{F})$ associated to $\mathcal{F}$ in which objects are the systems in the family and morphisms are topological defects that may transition between different systems inside the family. We may now be interested in possible ways of realizing the symmetry $\mathcal{S}$ on some system in the family $\mathcal{F}$. Such ways are classified by pivotal $d$-functors

$$\Phi: \ B\mathcal{S} \to \mathcal{C}_d(\mathcal{F}), \qquad (A.2)$$

where $B\mathcal{S}$ is the pivotal $d$-category obtained by delooping the pivotal fusion $(d-1)$-category $\mathcal{S}$ [135]. The image under $\Phi$ of the sole object in $B\mathcal{S}$ is the system $\mathfrak{T} \in \mathcal{F}$ on which the symmetry $\mathcal{S}$ is being realized. The rest of the information of $\Phi$ descends to a pivotal tensor $(d-1)$-functor

$$\phi: \ \mathcal{S} \to \mathrm{End}_{\mathcal{C}_d(\mathcal{F})}(\mathfrak{T}), \qquad (A.3)$$

where $\mathrm{End}_{\mathcal{C}_d(\mathcal{F})}(\mathfrak{T})$ is the pivotal monoidal $(d-1)$-category formed by endomorphisms of the object $\mathfrak{T}$ in $\mathcal{C}_d(\mathcal{F})$, which are precisely the topological defects living in the system $\mathfrak{T}$, and hence we have $\mathrm{End}_{\mathcal{C}_d(\mathcal{F})}(\mathfrak{T}) = \mathcal{C}_{d-1}(\mathfrak{T})$, recovering the description (A.1).

See also the recent review [82] which also takes this perspective on $\mathcal{S}$-symmetric systems.

### 2.   Gapped Phases with Non-Invertible Symmetries

Taking $\mathcal{F}$ to be the family $\mathcal{F}^{\mathrm{top}}$ formed by $d$-dimensional (unitary, oriented, fully extended) TQFTs leads to the classification of $\mathcal{S}$-symmetric $d$-dimensional TQFTs as the classification of pivotal $d$-functors

$$\Phi: \ B\mathcal{S} \to \mathcal{C}_d\left(\mathcal{F}^{\mathrm{top}}\right). \qquad (A.4)$$

The non-symmetric $d$-dimensional TQFT underlying an $\mathcal{S}$-symmetric $d$-dimensional TQFT is obtained as the image under $\Phi$ of the sole object of $B\mathcal{S}$. The $\mathcal{S}$-symmetric $d$-dimensional gapped phases are then obtained as deformation classes of such $d$-functors.

In the special case $d = 2$, the irreducible 2d TQFTs are $\mathfrak{Z}_\lambda$ parametrized by $\lambda \in \mathbb{R}$. There is a single irreducible interface $\mathcal{I}_{\lambda_1, \lambda_2}$ from $\mathfrak{Z}_{\lambda_1}$ to $\mathfrak{Z}_{\lambda_2}$ whose linking action on the identity operator $1_{\lambda_1}$ of $\mathfrak{Z}_{\lambda_1}$ is

$$\mathcal{I}_{\lambda_1, \lambda_2}: \ 1_{\lambda_1} \to e^{-(\lambda_2 - \lambda_1)} 1_{\lambda_2}, \qquad (A.5)$$

where $1_{\lambda_2}$ is the identity operator of $\mathfrak{Z}_{\lambda_2}$. The corresponding pivotal 2-category is denoted as

$$\mathcal{C}_2\left(\mathcal{F}^{\mathrm{top}}\right) = 2\text{-}\mathsf{Vec}^{\odot} \qquad (A.6)$$

and understood as the Euler completion of the 2-category of 2-vector spaces. See [136] for more details on the operation of Euler completion. Thus, $\mathcal{S}$-symmetric (where $\mathcal{S}$ is a fusion category) $(1+1)$d gapped phases are obtained as deformation classes of pivotal 2-functors of the form

$$\Phi: \ B\mathcal{S} \to 2\text{-}\mathsf{Vec}^{\odot}. \qquad (A.7)$$

The complete information of such a 2-functor $\Phi$ can be extracted from the SymTFT sandwich construction, as discussed in detail in [2].

As a special case of the above, consider the case when the underlying 2d TQFT is trivial, i.e. the image of $\Phi$ picks out the object $\mathfrak{Z}_0$ in 2-$\mathsf{Vec}^{\odot}$. In this case, we have

$$\mathrm{End}_{2\text{-}\mathsf{Vec}^{\odot}}(\mathfrak{Z}_0) = \mathsf{Vec}, \qquad (A.8)$$

where $\mathsf{Vec}$ denotes the category formed by finite dimensional vector spaces. The possible $\mathcal{S}$-symmetric TQFTs whose underlying TQFT is trivial are known as symmetry protected topological (SPT) phases for $\mathcal{S}$-symmetry. Applying (A.1), we recover the well-known fact [137] that 2d SPT phases for $\mathcal{S}$-symmetry are classified by fiber functors from $\mathcal{S}$, i.e. functors of the form

$$\phi: \ \mathcal{S} \to \mathsf{Vec}. \qquad (A.9)$$

In fact, by what we discussed above, we also have

$$\mathrm{End}_{2\text{-}\mathsf{Vec}^{\odot}}(\mathfrak{Z}_\lambda) = \mathsf{Vec} \qquad (A.10)$$

for any $\lambda$. Thus, we learn that any $\mathcal{S}$-symmetric 2d TQFT whose underlying non-symmetric 2d TQFT is invertible can be obtained by stacking the underlying invertible 2d TQFT with an SPT phase.

### 3.   Gapped Boundary Phases with Non-Invertible Symmetries

As another instance relevant for this paper, take $\mathcal{F}$ to be the family $\mathcal{F}^{\mathrm{top}}_{\mathfrak{Z}_{d+1}}$ formed by all topological boundary conditions of a $(d+1)$-dimensional TQFT $\mathfrak{Z}_{d+1}$. Then

$\mathcal{C}_d\left(\mathcal{F}^{\text{top}}_{\mathfrak{Z}_{d+1}}\right)$ is the pivotal $d$-category $\mathcal{B}(\mathfrak{Z}_{d+1})$ formed by topological boundary conditions of the TQFT $\mathfrak{Z}_{d+1}$. $\mathcal{S}$-symmetric topological boundary conditions of the TQFT $\mathfrak{Z}_{d+1}$ are then classified by pivotal $d$-functors

$$\Phi: \ B\mathcal{S} \to \mathcal{B}(\mathfrak{Z}_{d+1}). \tag{A.11}$$

Moving forward, let's restrict to $d = 2$ and $\mathfrak{Z}_{d+1}$ to be an irreducible 3d TQFT $\mathfrak{Z}$ admitting gapped boundary conditions. As discussed in the main text, such a 3d TQFT can be identified as the SymTFT associated to a unitary fusion category $\mathcal{C}$

$$\mathfrak{Z} \cong \mathfrak{Z}(\mathcal{C}). \tag{A.12}$$

Note that in general there are multiple choices of $\mathcal{C}$ satisfying the above relation, all of which are related by Morita equivalences.

The 2-category $\mathcal{B}(\mathfrak{Z})$ is completely determined in terms of $\mathcal{C}$, whose Drinfeld center $\mathcal{Z}(\mathcal{C})$ is equivalent to the MTC $\mathcal{Z}$

$$\mathcal{Z}(\mathcal{C}) \cong \mathcal{Z}. \tag{A.13}$$

Let us describe the information of $\mathcal{B}(\mathfrak{Z})$ in terms of the information of $\mathcal{C}$. The simple objects of $\mathcal{B}(\mathfrak{Z})$ are irreducible topological boundary conditions of $\mathfrak{Z}$ and they are parametrized by pairs $(\mathcal{M}, \lambda)$ where $\mathcal{M}$ is an indecomposable module category for $\mathcal{C}$ and $\lambda \in \mathbb{R}$. We label the corresponding irreducible topological boundary condition, or the corresponding simple object, as $\mathfrak{B}_\lambda(\mathcal{M})$.

The morphisms

$$\text{Hom}\big(\mathfrak{B}_\lambda(\mathcal{M}), \mathfrak{B}_{\lambda'}(\mathcal{M}')\big) \tag{A.14}$$

from $\mathfrak{B}_\lambda(\mathcal{M})$ to $\mathfrak{B}_{\lambda'}(\mathcal{M}')$ are the given by the category $\mathcal{N}$ having the property

$$\mathcal{M} \boxtimes_{\mathcal{C}^*_\mathcal{M}} \mathcal{N} = \mathcal{M}', \tag{A.15}$$

where $\boxtimes_{\mathcal{C}^*_\mathcal{M}}$ is the Deligne product taken relative to the fusion category $\mathcal{C}^*_\mathcal{M}$ obtained as the dual of $\mathcal{C}$ with respect to $\mathcal{M}$. The category $\mathcal{N}$ describes the topological lines acting as interfaces from $\mathfrak{B}_\lambda(\mathcal{M})$ to $\mathfrak{B}_{\lambda'}(\mathcal{M}')$. In particular, we have

$$\text{Hom}\big(\mathfrak{B}_\lambda(\mathcal{M}), \mathfrak{B}_{\lambda'}(\mathcal{M})\big) = \mathcal{C}^*_\mathcal{M} \tag{A.16}$$

and

$$\text{Hom}\big(\mathfrak{B}_\lambda(\mathcal{C}), \mathfrak{B}_{\lambda'}(\mathcal{C})\big) = \mathcal{C} \tag{A.17}$$

for the special case of a regular module $\mathcal{M} = \mathcal{C}$. Thus, the topological line operators on a boundary $\mathfrak{B}_\lambda(\mathcal{M})$ form the fusion category $\mathcal{C}^*_\mathcal{M}$, and for the special case $\mathcal{M} = \mathcal{C}$, we have that the topological line operators on $\mathfrak{B}_\lambda(\mathcal{C})$ form the fusion category $\mathcal{C}$.

Let us now describe some properties of the pivotal structure. Consider a simple line operator $L$ in $\mathcal{N} =$

$\text{Hom}\big(\mathfrak{B}_{\lambda_1}(\mathcal{M}_1), \mathfrak{B}_{\lambda_2}(\mathcal{M}_2)\big)$. It has some quantum dimension $d_{\lambda_1, \lambda_2}(L)$, i.e. its linking action on the identity operator $1_{\mathfrak{B}_{\lambda_1}(\mathcal{M}_1)}$ of the boundary $\mathfrak{B}_{\lambda_1}(\mathcal{M}_1)$ is

$$L: \ 1_{\mathfrak{B}_{\lambda_1}(\mathcal{M}_1)} \to d_{\lambda_1, \lambda_2}(L) 1_{\mathfrak{B}_{\lambda_2}(\mathcal{M}_2)}, \tag{A.18}$$

where $1_{\mathfrak{B}_{\lambda_2}(\mathcal{M}_2)}$ is the identity operator on the boundary $\mathfrak{B}_{\lambda_2}(\mathcal{M}_2)$. Then the quantum dimension of the same operator $L$ in $\mathcal{N} = \text{Hom}\big(\mathfrak{B}_{\lambda'_1}(\mathcal{M}_1), \mathfrak{B}_{\lambda'_2}(\mathcal{M}_2)\big)$ is

$$d_{\lambda'_1, \lambda'_2}(L) = \frac{e^{-(\lambda'_2 - \lambda_2)}}{e^{-(\lambda'_1 - \lambda_1)}} d_{\lambda_1, \lambda_2}(L). \tag{A.19}$$

**Toric Code Example.** As an example, take $\mathfrak{Z}$ to be the Toric Code. In this case, the only choice for $\mathcal{C}$ is

$$\mathcal{C} = \text{Vec}_{\mathbb{Z}_2}. \tag{A.20}$$

There are two module categories $\mathcal{M} = \text{Vec}_{\mathbb{Z}_2}$ and $\mathcal{M} = \text{Vec}$. We thus have irreducible topological boundaries

$$\mathfrak{B}_\lambda(\text{Vec}_{\mathbb{Z}_2}), \qquad \mathfrak{B}_\lambda(\text{Vec}) \tag{A.21}$$

with

$$\begin{aligned} \text{Hom}\big(\mathfrak{B}_\lambda(\text{Vec}_{\mathbb{Z}_2}), \mathfrak{B}_{\lambda'}(\text{Vec}_{\mathbb{Z}_2})\big) &= \text{Vec}_{\mathbb{Z}_2} = \{1^e_{\lambda,\lambda'}, \ P^e_{\lambda,\lambda'}\} \\ \text{Hom}\big(\mathfrak{B}_\lambda(\text{Vec}), \mathfrak{B}_{\lambda'}(\text{Vec})\big) &= \text{Vec}_{\mathbb{Z}_2} = \{1^m_{\lambda,\lambda'}, \ P^m_{\lambda,\lambda'}\} \\ \text{Hom}\big(\mathfrak{B}_\lambda(\text{Vec}_{\mathbb{Z}_2}), \mathfrak{B}_{\lambda'}(\text{Vec})\big) &= \text{Vec} = \{S^{em}_{\lambda,\lambda'}\} \\ \text{Hom}\big(\mathfrak{B}_\lambda(\text{Vec}), \mathfrak{B}_{\lambda'}(\text{Vec}_{\mathbb{Z}_2})\big) &= \text{Vec} = \{S^{me}_{\lambda,\lambda'}\}, \end{aligned} \tag{A.22}$$

where we have displayed the simple lines as well.

The non-zero fusion rules are

$$\begin{aligned} 1^i_{\lambda_1, \lambda_2} \otimes 1^i_{\lambda_2, \lambda_3} &= 1^i_{\lambda_1, \lambda_3} \\ 1^i_{\lambda_1, \lambda_2} \otimes P^i_{\lambda_2, \lambda_3} = P^i_{\lambda_1, \lambda_2} \otimes 1^i_{\lambda_2, \lambda_3} &= P^i_{\lambda_1, \lambda_3} \\ P^i_{\lambda_1, \lambda_2} \otimes P^i_{\lambda_2, \lambda_3} &= 1^i_{\lambda_1, \lambda_3} \\ 1^i_{\lambda_1, \lambda_2} \otimes S^{ij}_{\lambda_2, \lambda_3} = S^{ij}_{\lambda_1, \lambda_2} \otimes 1^j_{\lambda_2, \lambda_3} &= S^{ij}_{\lambda_1, \lambda_3} \\ P^i_{\lambda_1, \lambda_2} \otimes S^{ij}_{\lambda_2, \lambda_3} = S^{ij}_{\lambda_1, \lambda_2} \otimes P^j_{\lambda_2, \lambda_3} &= S^{ij}_{\lambda_1, \lambda_3} \\ S^{ij}_{\lambda_1, \lambda_2} \otimes S^{ji}_{\lambda_2, \lambda_3} &= 1^i_{\lambda_1, \lambda_3} \oplus P^i_{\lambda_1, \lambda_3}, \end{aligned} \tag{A.23}$$

where $i, j \in \{e, m\}$, and the quantum dimensions are

$$\begin{aligned} d\left(1^i_{\lambda_1, \lambda_2}\right) &= e^{-(\lambda_2 - \lambda_1)} \\ d\left(P^i_{\lambda_1, \lambda_2}\right) &= e^{-(\lambda_2 - \lambda_1)} \\ d\left(S^{ij}_{\lambda_1, \lambda_2}\right) &= \sqrt{2}\, e^{-(\lambda_2 - \lambda_1)} \end{aligned} \tag{A.24}$$

## Appendix B: Condensable Algebras

The conditions for a condensable algebra $\mathcal{A} = \oplus_a n^a a$ in $\mathcal{Z}$ are [121]

(CA1) $n^{\mathbf{1}} = 1, \quad n^a \in \mathbb{N}, \quad n^a = n^{\bar{a}}$

(CA2) $s_a = 0$ for $a \in \mathcal{A}$

(CA3) $\frac{\sum_{b \in \mathcal{Z}} S^{ab} n^b}{\sum_{b \in \mathcal{Z}} S^{1b} n^b} =$ cyclotomic integer for all $a \in \mathcal{Z}$

(CA4) $n^a \leq d_a - \delta(d_a)$

(CA5) $n^a n^b \leq \sum_c N_c^{ab} n^c - \delta_{a,\bar{b}} \delta(d_a)$

(CA6) $n^a = \sum_b S^{ab} n^b$ if $\mathcal{A}$ is Lagrangian.

Here $S$ is the S-matrix and $s_a$ the spin. These conditions however are not sufficient. A sufficient condition is C15 in [138], which however requires knowledge of the reduced topological order. We proceed by identifying algebras that satisfy the above sufficient conditions and then check that there is a consistent condensation.

**Anyon Condensation.** We now explain how to determine the reduced topological order

$$\mathcal{Z}' = \mathcal{Z}/\mathcal{A} \qquad (B.1)$$

following [139, 140]. The first step is to obtain from $\mathcal{Z}$ a consistent fusion algebra $\mathcal{F}$

$$\mathcal{Z} \to \mathcal{F}. \qquad (B.2)$$

To do this, we notice that the algebra $\mathcal{A}$ is to be identified with the vacuum of the condensed theory $\mathcal{F}$, which implies we have the following identity

$$\text{Hom}_{\mathcal{F}}(a, b) = \text{Hom}_{\mathcal{Z}}(a, b \otimes \mathcal{A}), \qquad (B.3)$$

where $a, b$ denote two generic simple objects of $\mathcal{Z}$. In practice, this implies generally that the simple objects of $\mathcal{Z}$ are not going to be simple anymore in $\mathcal{F}$, which means the they will undergo a splitting procedure. It can also happen that two distinct anyons in $\mathcal{Z}$ get identified in $\mathcal{F}$. We can then write a restriction map

$$z \to \bigoplus_{f \in \mathcal{F}} n_z^f f, \qquad (B.4)$$

which expresses an anyon $z \in \mathcal{Z}$ as a superposition of anyons $f \in \mathcal{F}$. For example, all the anyons $a \in \mathcal{A}$ should contain the trivial anyon 1 of $\mathcal{F}$ in their restriction. We can also define the inverse map, called the lift

$$f \to \bigoplus_{z \in \mathcal{Z}} n_z^f z. \qquad (B.5)$$

For example, the vacuum 1 of the new theory should lift to $\mathcal{A}$. We also assume that the fusion commutes with the restriction map, so that we have

$$a \otimes b = N_{ab}^c c \to \left(\oplus_f n_a^f f\right) \otimes \left(\oplus_g n_b^g g\right) = \oplus_{c,t} N_{ab}^c n_c^t t. \qquad (B.6)$$

This allows us to determine all the simple objects of $\mathcal{F}$ and their fusion rules.

However, $\mathcal{F}$ still contains anyons that braid non-trivially with the condensate. To obtain a consistent MTC, these lines have to be removed (they are confined to the gapped interface). An anyon $f \in \mathcal{F}$ is deconfined, i.e. it survives as a bulk excitation after condensation, if all the anyons appearing in its lift have the same spin

$$\theta_z = \theta_{z'} \quad \forall z, z' : n_z^f, n_{z'}^f \neq 0. \qquad (B.7)$$

After this step, we finally obtain the final consensed theory

$$\mathcal{F} \to \mathcal{Z}'. \qquad (B.8)$$

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

arXiv:2005.09072 [cond-mat.str-el].

arXiv:2308.09670 [math.QA].

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
