# Peer review of "The Club Sandwich: Gapless Phases and Phase Transitions with Non-Invertible Symmetries"

_SciPost Physics_

## Round 2 · Referee Report · Anonymous (Referee 1) · 2024-11-22

Strengths

The idea of generalizing KT transformation using the club sandwich and applying to phase transitions seems novel. Some phase transitions under a "larger" (possibly non-invertible) symmetry may be understood in terms of a phase transition under a "smaller" symmetry, using the framework proposed in this work.

Weaknesses

The treatment of condensable/Lagrangian algebra is too naive.

The rigorous mathematical framework of algebras in generic tensor categories, including their representation theory, has been established for years. Even the review article [128] in the context of anyon condensation was published ten years ago. However, it seems that the authors are even unclear about the standard definition of an algebra in a tensor category. What they are playing with is merely Lagrangian algebras truncated to the object level (See for example https://link.springer.com/article/10.1007/s11005-015-0766-x on how limited such methodology would be). There are many imprecise/incorrect statements made regarding condensable algebras, e.g.: 1. The condition C15 in [138] quoted in Appendix B, which the authors claim to be sufficient, are in fact still necessary conditions at the object level; 2. (B.7) is only a necessary condition for an anyon to survive condensation. The overall careless treatment of condensable algebra puts the technical validity of this work in danger. It is even morally worse that instead of learning and further developing the existing mathematical theory of Lagrangian algebras, this manuscript is trying to reinvent the wheels. It should not be encouraged.

Report

The examples of new phase transitions may be of physical significance. However, I cannot recommend publication in its current form due to the technical weakness on condensable algebras.

Requested changes

The authors should either 1) carefully review the mathematical theory of condensable algebras and try to improve their treatment, or 2) be honest about the limitations and potential risks of their current truncated treatment.

Recommendation

Ask for major revision

---

## Round 2 · Referee Report · Anonymous (Referee 2) · 2024-11-28

Strengths

  1. Develops a general framework for constructing and analyzing phase transitions with categorical symmetry.

  2. Contains many detailed examples.

  3. The paper is clearly written.

Weaknesses

The treatment of condensable algebra is not standard and in many places inaccurate. See requested changes below.

The idea that non-Lagrangian condensation corresponds to gapless boundary is not new. See e.g. arxiv:2205.06244.

Report

This paper applies the SymTFT framework to study phases transitions enriched by general categorical symmetries, focusing on 1+1D systems with a 2+1D bulk SymTFT.

The authors develop a club sandwich and a club quiche construction to describe the coupling of a critical theory to an enlarged symmetry. The club sandwich can be viewed as defining a categorical KT transformation, which couples a critical theory to a symmetry larger than the one acting faithfully on it. The interface domain wall in the club sandwich is described by a non-Lagrangian condensable algebra, and the symmetry acting faithfully on the critical theory is described by the reduced topological order after the condensation. The paper also shows how to obtain the local and non-local order parameters of phase transitions from SymTFT .

This work generalizes the group-like cases studied in earlier works to general fusion category symmetries in 1+1D, establishing a general framework for constructing and analyzing categorical symmetry-enriched phases transitions. Some examples studied in this paper are novel. As such, the referee would recommend this paper for publication.

Requested changes

  1. P.2 right column second paragraph: what is "a map between them"? Do you mean a monoidal functor?

  2. P.2 right column last paragraph: "a phase transition between two gapped phases, which are defined by the Lagrangian algebras $\mathcal{L}_1$ and $\mathcal{L}_2$ respectively, is characterized by an algebra $\mathcal{A}_{1,2}=\mathcal{L}_1\cap \mathcal{L}_2$ ". Does this statement hold for abelian topological orders only? In general taking intersection does not define a new condensable algebra. E.g. consider two different condensable algebras whose underlying object agree(which happens in $\mathcal{Z}(Vec_G)$ for certain $|G|=128$), what is the algebra structure on the intersection? Maximal common condensable subalgebra may be what you want.

  3. The discussion of condensable algebra in appendix B. is misleading. A condensable algebra should be a connected separable commutative algebra. Please follow the standard treatment in e.g. section 3 of arxiv: 1008.2117. It is $\textbf{impossible}$ to define condensable algebra purely at the object level, since there are examples where two distinct condensable algebras have the same underlying object. Also what is the "fusion algebra" $\mathcal{F}$? The reduced topological order should be described by the braided fusion category of local modules. See e.g. arXiv:1307.8244.

  4. P.5 right column first paragraph. Not all gapped phases correspond to TQFTs. Only those that are "liquid-like" correspond to TQFTs in the continuum. E.g. fractons do not correspond to TQFTs. Also fermionic gapped quantum liquids correspond to spin TQFTs.

  5. Eq.(II.10): a condensable algebra is more that its underlying object.

  6. Eq.(IV.25): ${1,P,P^2,P^3}$ is only the list of simple objects of the symmetry category.

  7. P.20 right column: "In order to determine the condensable algebras among these, we check whether there is a consistent reduced topological order. This shows $\mathcal{A}^{(2)}_2$ for instance is not consistent, while $\mathcal{A}_2$, $\mathcal{A}_{2,3}$ and $\mathcal{A}_{1,2}$ are. " This derivation is not shown. The referee finds this method of identifying condensable algebras very mysterious.

  8. P.11 Eq. (IV.32): This "algebra" is not a condensable algebra, because a condensable algebra should be connected, meaning that $\mathsf{Hom}(I,\mathcal{A})=\mathbb{C}$. The same problem appears in other places where "reducible gapped boundary" is discussed. Perhaps etale algebra is what is meant here.

Recommendation

Ask for minor revision

---

## Editorial Decision

resubmitted